# Theoretical Analysis of Contrastive Learning under Imbalanced Data: From Training Dynamics to a Pruning Solution

**Haixu Liao**
New Jersey Institute of Technology
hl534@njit.edu

**Yating Zhou**
Cornell University
yz3554@cornell.edu

**Songyang Zhang**
University of Louisiana at Lafayette
songyang.zhang@louisiana.edu

**Meng Wang**
Rensselaer Polytechnic Institute
wangm7@rpi.edu

**Shuai Zhang**
New Jersey Institute of Technology
sz457@njit.edu

## Abstract

Contrastive learning has emerged as a powerful framework for learning generalizable representations, yet its theoretical understanding remains limited, particularly under imbalanced data distributions that are prevalent in real-world applications. Such an imbalance can degrade representation quality and induce biased model behavior, yet a rigorous characterization of these effects is lacking. In this work, we develop a theoretical framework to analyze the training dynamics of contrastive learning with Transformer-based encoders under imbalanced data. Our results reveal that neuron weights evolve through three distinct stages of training, with different dynamics for majority features, minority features, and noise. We further show that minority features reduce representational capacity, increase the need for more complex architectures, and hinder the separation of ground-truth features from noise. Inspired by these neuron-level behaviors, we show that pruning restores performance degraded by imbalance and enhances feature separation, offering both conceptual insights and practical guidance. Major theoretical findings are validated through numerical experiments.

## 1 Introduction

Contrastive learning has emerged as a powerful paradigm in representation learning, effectively leveraging unlabeled data without relying on labels. Within this framework, samples with similar semantic meaning are treated as positive pairs, while those with different semantics are considered negative pairs. By pulling positive pairs closer together and pushing negative pairs farther apart in the representation space, contrastive learning enables models to capture rich and discriminative features. Compared with supervised learning, the resulting representations are often more robust and less sensitive to noise (Xue et al., 2022; Ghosh & Lan, 2021; Zhong et al., 2022a; Jiang et al., 2020; Yang & Xu, 2020; Kang et al., 2020). This approach has demonstrated remarkable success across a wide range of applications (Zhong et al., 2022b; Zhang et al., 2022; Jiang et al., 2023; Luo et al., 2023) and has been particularly influential in multi-modal learning (Nakada et al., 2023; Khan et al., 2025), driving major advances in the early development of vision-language models (Radford et al., 2021; Li et al., 2022; 2023).

Despite its strengths, contrastive learning struggles with class imbalance in real-world datasets Jiang et al. (2021), where majority classes dominate pair formation and minority classes are underrepresented. This imbalance hinders the capture of discriminative features for minority classes and degrades representation quality. Conventional approaches to class imbalance in supervised learning

typically rely on re-weighting and re-sampling, and these ideas have inspired analogous methods in contrastive learning. Re-weighting strategies adjust the contribution of pairs or instances to reduce the dominance of majority classes (Cui et al., 2019; Huang et al., 2016), while resampling methods construct more balanced training batches by oversampling minority samples or undersampling majority ones (Drummond & Holte, 2003; He & Garcia, 2009; Peng et al., 2020). Although these approaches have shown effectiveness in certain cases, their application in contrastive settings remains challenging, as they often rely on accurate class labels that are unavailable in self-supervised learning. To address this limitation, an alternative line of research has proposed pruning-based methods, which have been empirically validated to enhance the representation of underrepresented classes (Jiang et al., 2021; Qian et al., 2022).

Despite the progress made by these approaches, most efforts have been largely empirical, relying on heuristic methods to alleviate the imbalance problem. While these techniques often provide performance gains in practice, they do not explain why or how imbalance undermines the quality of learned representations. Recent work has begun to develop theoretical understandings of contrastive learning, primarily addressing questions such as its superiority over traditional generative approaches like GANs (Ji et al., 2023), the necessity of data augmentation for effective representation learning (Wen & Li, 2021), and its ability to produce representations that reduce the sample complexity of downstream tasks (Garg & Liang, 2020). Nonetheless, these studies have not considered the implications of imbalanced data distributions.

In this work, we provide a theoretical analysis of how neurons learn feature representations through contrastive training. We study a simplified but representative setting: a Transformer-MLP framework with a single-head attention mechanism followed by an MLP with bilateral ReLU activations. To make the analysis clear, we use a structured data model where each input includes majority and minority features with different frequencies. This setup highlights the key role of feature frequencies and helps us describe their impact on training dynamics and how neurons learn features. In turn, the model allows us to formalize how contrastive learning enhances majority features and drives neurons to learn purer feature representations. Overall, our paper makes three main **contributions**:

**First, we develop a theoretical framework to characterize the training dynamics of contrastive learning under Transformer-based encoders with an imbalanced data distribution.** We show that learning proceeds in three stages: first, neuron weights grow in feature directions while non-feature components are suppressed; second, Lucky neurons then specialize in single features, and ordinary neurons learn a mix of features; finally, each neuron converges in a way that guarantees a small training loss, becoming strongly aligned with one or more features, weakly aligned with other features, and remaining small in non-feature directions. See Figure 1 for reference.

**Second, we quantitatively characterize how the presence of minority features influences neurons' learning capacity and, consequently, representation learning.** Our analysis reveals that imbalance degrades representation performance in multiple ways: it slows the learning of minority features, decreases the number of neurons that specialize in a single feature, and produces a chain effect that necessitates a more complex model to adequately capture all features.

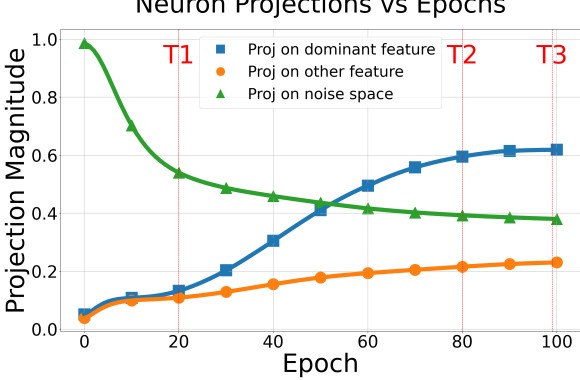

Figure 1: Neuron projection dynamics over training epochs. The blue curve shows the growth of a neuron's projection onto its dominant feature, the orange curve shows the projection onto a non-dominant feature, and the green curve shows the projection onto the noise space direction (which remains larger than the projections onto other features). In the first stage, the neuron grows mainly along feature directions while suppressing noise. In the second stage, the projection onto the dominant feature grows faster than all other features, creating clear separation. In the third stage, as training approaches $T_3$, the neuron converges, and its final representation is dominated by the learned feature, with negligible components in other directions.

**Third, magnitude-based pruning can enhance the learning of minority features.** Our results reveal that magnitude-based pruning enhances updates along minority feature directions, encouraging more neurons to specialize in pure minority features and thereby yielding more robust and balanced representations. Intuitively, neurons with small magnitudes are more sensitive to samples containing minority features, which implicitly allows pruning to amplify their contribution.

## 1.1 RELATED WORK

**Data Imbalance in Self-Supervised Learning:** Data imbalance or long-tail data has been a longstanding challenge since the early development of supervised learning (Chu et al., 2020; Liu et al., 2020; Yang et al., 2022; Chawla et al., 2002). At a high level, tackling data imbalance follows a simple principle: balancing the influence of different groups of data during weight updates, typically through re-sampling (Buda et al., 2018; Choi et al., 2018), which alters the data distribution, or reweighting (Mahajan et al., 2018), which adjusts loss contributions across classes. These methods all require label information (Cui et al., 2021; Zhu et al., 2022). However, without label information, as in self-supervised learning (SSL), these strategies are far more difficult to apply, and only a few works have addressed the imbalance. Beyond re-weighting and re-sampling (Lin et al., 2017; Shrivastava et al., 2016; Shang et al., 2025; Shen et al., 2016), other alternative approaches have been proposed: optimization-based regularization for rare samples (Liu et al.), mixup for implicit rebalancing (Li & Jia, 2025), and pruning as an implicit means of detecting long-tail data (Jiang et al., 2021; Qian et al., 2022).

**Convergence and Generalization Analysis of Contrastive Learning:** Despite its empirical success, contrastive learning lacks a mature theoretical understanding, largely due to the complexity of its loss function. Early research investigates why augmentation is essential for the success of contrastive learning, showing that such an alignment between augmented positive pairs facilitates learning useful representations (Saunshi et al., 2022; Tian et al., 2020; Saunshi et al., 2019; Wen & Li, 2021). Tian et al. (2021); Wang et al. (2023) establishes a connection between the gradients of contrastive learning and graph neural networks, highlighting interpretability through a graph-theoretic perspective. HaoChen et al. (2021) also explores the connections between contrastive learning and graph theory, proposing a new loss function linked to graph spectral clustering to help explain its success. Wen & Li (2021) emphasizes the necessity of data augmentation for breaking dependencies on spurious noise. None of these works has explored how imbalanced data influences the training dynamics of contrastive learning.

**Feature Learning Paradigm:** The mathematical framework in this paper is closely related to the feature learning paradigm. Specifically, we assume the data follow a sparse coding model, which is a mixture of latent features, and study the training dynamics of model weights to examine how they align with these features. Most prior works focus on supervised learning (Allen-Zhu & Li, 2022; Zhang et al., 2023; Li et al., 2025; Cao et al., 2022; Chowdhury et al., 2023; Shandirasegaran et al., 2025), where features are tied to ground-truth labels; however, such settings cannot be directly extended to contrastive learning. Because of the complexity of analyzing fine-grained training dynamics, existing studies are typically limited to simple one-hidden-layer neural networks, with some recent efforts exploring Transformers but still restricted to a single layer (Huang et al., 2024; Oymak et al., 2023; Li et al., 2024), even under supervised settings. The most relevant works are Wen & Li (2021); Sun et al. (2025), which analyze the training dynamics of contrastive learning with one-hidden-layer feedforward networks. In contrast, our paper studies Transformer architectures under a different data model, and further incorporates data imbalance, providing a comprehensive analysis of how it influences the model's ability to decouple features, rather than being only a direct extension through feature magnitude changes.

## 2 PROBLEM FORMULATION AND ALGORITHM

**Contrastive Learning Framework.** Let $\boldsymbol{X} = [\boldsymbol{x}^{(1)}, \dots, \boldsymbol{x}^{(L)}] \in \mathbb{R}^{d_1 \times L}$ or $\boldsymbol{Y} \in \mathbb{R}^{d_1 \times L}$ be an input sequence with $L$ tokens. The goal of contrastive learning is to learn a mapping $\boldsymbol{f}(\cdot) : \mathbb{R}^{d_1 \times L} \to \mathbb{R}^m$ that outputs a meaningful embedding from the input sequence.

Let $(\boldsymbol{X}_n, \boldsymbol{Y}_n)$ denote a *positive pair* (e.g., derived from the same objective or sharing semantic meaning), and let $\mathfrak{N}$ denote a set of corresponding *negative samples* (e.g., random samples). The

InfoNCE loss with temperature parameter $\tau > 0$ is defined as:

$$\ell(\boldsymbol{f_\theta}, \boldsymbol{X}_n, \boldsymbol{Y}_n, \mathfrak{N}) := -\log\left(\frac{e^{\text{sim}_{\boldsymbol{f_\theta}}(\boldsymbol{X}_n, \boldsymbol{Y}_n)/\tau}}{\sum_{\boldsymbol{X} \in \{\boldsymbol{Y}_n\} \cup \mathfrak{N}} e^{\text{sim}_{\boldsymbol{f_\theta}}(\boldsymbol{X}_n, \boldsymbol{X})/\tau}}\right), \tag{1}$$

where the similarity function is given by

$$\text{sim}_{\boldsymbol{f_\theta}}(\boldsymbol{X}_n, \boldsymbol{Y}_n) := \langle \boldsymbol{f_\theta}(\boldsymbol{X}_n), \text{StopGrad}(\boldsymbol{f_\theta}(\boldsymbol{Y}_n)) \rangle, \tag{2}$$

and $\text{StopGrad}(\cdot)$ acts as the identity in forward pass while blocking gradients in backpropagation.

Then, the learning objective is to minimize an empirical risk with $l_2$-regularizer, i.e.,

$$\widehat{L}_{\text{aug}}(\boldsymbol{f_\theta}) = \widehat{L}(\boldsymbol{f_\theta}) + \frac{\lambda}{2}\|\boldsymbol{\theta}\|_F^2 = \frac{1}{K}\sum_{k=1}^{K}\ell(\boldsymbol{f_\theta}, \boldsymbol{X}_k, \boldsymbol{Y}_k, \mathfrak{N}_k) + \frac{\lambda}{2}\|\boldsymbol{\theta}\|_F^2, \tag{3}$$

where $\boldsymbol{\theta}$ denotes the neural network parameters and $K = \text{poly}(d_1)$.

**Model Architecture: Transformer-MLP.** We employ a simplified single-head self-attention mechanism on top of an MLP layer. Each input sequence is passed through the attention layer, where every token serves as a query. Then, it is followed by a bilateral ReLU (BReLU) activation in the MLP layer, where $\text{BReLU}_b(s) = \text{ReLU}(s - b) - \text{ReLU}(-s - b)$. Specifically, the embedding function $f$ is expressed as

$$\boldsymbol{f_\theta}(\boldsymbol{X}_n) = \big(h_1(\boldsymbol{X}_n), \ldots, h_m(\boldsymbol{X}_n)\big)^\top \in \mathbb{R}^m,$$

$$\text{with} \quad h_i(\boldsymbol{X}_n) = \sum_{r=1}^{L}\text{BReLU}_{b_i^{(t)}}\Big(\langle \boldsymbol{w}_i^{(t)}, \text{Attention}(\boldsymbol{W_Q}\boldsymbol{x}_n^{(r)}, \boldsymbol{W_K}\boldsymbol{X}_n, \boldsymbol{W_V}\boldsymbol{X}_n)\rangle\Big). \tag{4}$$

**Pruning Algorithm.** To address the issue of data imbalance, we revisit (Jiang et al., 2021; Qian et al., 2022) and propose a pruning algorithm that dynamically removes small-magnitude neuron weights during the forward pass, while retaining all parameters as trainable in the backward pass [1]. Specifically, we initialize the MLP layer weights with Gaussian distributions and the attention weights as identity matrices. The binary mask is initially set to all ones, meaning no neurons are pruned at the start. At each epoch, a fraction $\alpha$ of the neurons with the smallest magnitudes are pruned, and the corresponding binary mask is updated. During the forward pass, the masked parameters $\theta_{\text{mk}}^{(t)}$ are used to encode the inputs. In the backward pass, gradients are computed with respect to the pruned model but applied to the full parameter set, namely, the gradient is calculated as

$$g(\boldsymbol{\theta}_t^{(t)}, \boldsymbol{M}^{(t)}) := \frac{1}{K}\sum_{k=1}^{K}\Big[(\ell'_{p,\boldsymbol{\theta}_{\text{mk}}^{(t)}} - 1)h_i(\boldsymbol{Y}_k)\nabla_{\boldsymbol{\theta}}h_i(\boldsymbol{X}_k) + \sum_{\boldsymbol{X}_{n,s} \in \mathfrak{N}_k}\ell'_{s,\boldsymbol{\theta}_{\text{mk}}^{(t)}}h_i(\boldsymbol{X}_{n,s})\nabla_{\boldsymbol{\theta}}h_i(\boldsymbol{X}_k)\Big], \quad (5)$$

where $\ell'_{p,\cdot} := \dfrac{\exp\big(\text{Sim}_{\boldsymbol{f}.}(\boldsymbol{X}_k, \boldsymbol{Y}_k)/\tau\big)}{\sum_{\boldsymbol{X} \in \{\boldsymbol{Y}_k\} \cup \mathfrak{N}_k}\exp\big(\text{Sim}_{\boldsymbol{f}.}(\boldsymbol{X}_k, \boldsymbol{X})/\tau\big)}$ is the positive logit and $\ell'_{s,\cdot} :=$

$\dfrac{\exp\big(\text{Sim}_{\boldsymbol{f}.}(\boldsymbol{X}_k, \boldsymbol{X}_{n,s})/\tau\big)}{\sum_{\boldsymbol{X} \in \{\boldsymbol{Y}_k\} \cup \mathfrak{N}_k}\exp\big(\text{Sim}_{\boldsymbol{f}.}(\boldsymbol{X}_k, \boldsymbol{X})/\tau\big)}$ is negative logit with respect to the native sample $\boldsymbol{X}_{n,s}$.

Note that this procedure does not permanently eliminate any neurons for efficiency purposes, even though a reduction in computation cost can be observed. The pruning mask acts as a temporary filter by automatically removing small-magnitude neurons. As shown in Theorem 3.2, these neurons are associated with minority features. Consequently, samples containing such features incur a higher loss, which in turn encourages the model to allocate greater attention to them during training.

## 3 THEORETICAL ANALYSIS

### 3.1 KEY INSIGHTS OF THE FINDINGS

We first give a summary of the key insights from our analysis before turning to the data model and the formal theoretical results. Our findings show how neurons gradually learn feature representations across different stages of training. In particular, we have

---

[1] We do not introduce a new algorithm; instead, we adapt established approaches to our theoretical setting.

---

**Algorithm 1** Forward Magnitude Pruning with Backward Unmasked Update

---

**Require:** Training dataset $\{(\boldsymbol{X}_k, \boldsymbol{Y}_k, \mathfrak{N}_k)\}_{k=1}^K$ (positive pairs $(\boldsymbol{X}_k, \boldsymbol{Y}_k)$ and negative set $\mathfrak{N}_k$)
**Require:** Pruning ratio $\alpha$
**Require:** Training epochs $T$, weight decay parameter $\lambda$, temperature $\tau$
1: Initialize network parameters $\boldsymbol{w}_i^{(0)} \sim \mathcal{N}(0, \sigma_0^2 \boldsymbol{I}_{\boldsymbol{d_1}})$, $\boldsymbol{W}_K^{(0)} = \boldsymbol{W}_Q^{(0)} = \boldsymbol{I}$.
2: Set the initial pruning mask $\boldsymbol{M}^{(0)} \leftarrow \boldsymbol{1}$ with the same shape as $\boldsymbol{\theta}^{(0)}$.
3: **for** $t = 0$ to $T - 1$ **do**
4:     **Magnitude based pruning:** At each iteration $t$, prune $\alpha$ fraction of neurons with the smallest magnitude in $\boldsymbol{\theta}^{(t)}$ by creating the corresponding binary mask $\boldsymbol{M}^{(t)}$.
5:     **Forward (masked):** Apply the mask to obtain $\boldsymbol{\theta}_{\text{mk}}^{(t)} \leftarrow \boldsymbol{\theta}^{(t)} \odot \boldsymbol{M}^{(t)}$, then encode $\boldsymbol{X}_k$, $\boldsymbol{Y}_k$, and negatives $\mathfrak{N}_k$ using $\boldsymbol{f}_{\boldsymbol{\theta}_{\text{mk}}^{(t)}}$.
6:     **Compute loss:** $\widehat{L}_{\text{aug}}(\boldsymbol{f}_{\boldsymbol{\theta}_{\text{mk}}^{(t)}}) = \frac{1}{K} \sum_{k=1}^K \ell(\boldsymbol{f}_{\boldsymbol{\theta}_{\text{mk}}^{(t)}}, \boldsymbol{X}_k, \boldsymbol{Y}_k, \mathfrak{N}_k; \tau) + \frac{\lambda}{2} \|\boldsymbol{\theta}_{\text{mk}}^{(t)}\|_F^2$.
7:     **Backward and update:** Release the mask $\boldsymbol{M}^{(t)}$ on the masked parameters and update the full parameters by
$$\boldsymbol{\theta}^{(t+1)} \leftarrow (1 - \eta\lambda)\boldsymbol{\theta}^{(t)} - \eta \cdot g(\theta_t^{(t)}, \boldsymbol{M}^{(t)})$$
8: **end for**
9: **return** $\boldsymbol{\theta}^{(T)}$

---

**(K1). Training dynamics of contrastive learning based on the Transformer-MLP framework.** The theory divides the learning process into three stages. In Stage 1 (Lemma 3.1), neuron weights grow in feature directions at rates determined by the feature frequencies $\epsilon_j$, while their components in non-feature directions are suppressed. In Stage 2 (Lemma 3.2), lucky neurons in $\mathcal{M}_j^\star$ strengthen their alignment with the feature direction $\boldsymbol{M}_j$, and ordinary neurons in $\mathcal{M}_j$ remain bounded by these lucky neurons, so that the learned features become purer and non-feature components remain suppressed. In the final stage, each neuron aligns with a specific set of features $\mathcal{N}_i$, becoming strongly aligned with some features, weakly with others, and remaining small in non-feature directions.

**(K2). Feature frequency ratio controls neuron specialization.** At convergence, each neuron is dominated by features in $\mathcal{N}_i$, with negligible contribution from other directions. First, the neuron magnitude in $\mathcal{N}_i$, denoted $\alpha_{i,j}$, scales as $\frac{\varepsilon_j}{\varepsilon_{\max}}$, so rarer features are learned more weakly. Second, the size of $\mathcal{N}_i$ scales as $d^{1-(\varepsilon_{\min}/\varepsilon_{\max})^2}$: smaller ratios enlarge $\mathcal{N}_i$ and cause feature mixing, while larger ratios shrink it and yield purer alignment. Third, the number of neurons specializing in purified features scales as $d^{-(\varepsilon_{\max}/\varepsilon_{\min})^2}$, which decreases as the gap between $\varepsilon_{\max}$ and $\varepsilon_{\min}$ grows. Since contrastive learning works best when neurons specialize in purified features, imbalance introduces three interrelated obstacles: minority features are learned with smaller magnitude, neurons mix multiple features instead of staying pure, and the overall number of specialized neurons decreases. Together, these effects weaken representation quality and require larger models to learn all features.

**(K3). Pruning enhances minority feature learning.** With pruning ratio $\alpha$, neurons aligned with minority features gain stronger updates of order $\frac{\alpha}{d}$, while those aligned with non-minority features grow only weakly, with updates of order $\frac{\alpha}{d^2}$. At convergence, the coefficient of neurons learning a minority feature can reach the same order as that of majority features, so the performance downgrade from imbalance is alleviated. Intuitively, minority neurons are pruned more often because their magnitudes are smaller, which in turn amplifies the contribution of samples containing the minority feature in gradient updates. As a result, pruning strengthens the minority feature, makes it clearly distinguished from other contributions, and drives more neurons to specialize in it, leading to more robust representation learning.

## 3.2 ASSUMPTIONS

**Data Model.** Our data assumption is adopted from the widely used sparse coding model, which constitutes a common foundation for theoretical analyses of deep learning (Allen-Zhu & Li, 2022; Wen & Li, 2021). Moreover, sparse coding provides a conceptual framework for modeling real-world data across diverse domains, including CV (Protter & Elad, 2008; Yang et al., 2009; Mairal et al., 2014; Liao et al., 2025), NLP (Arora et al., 2018), compressed sensing (Candes & Recht,

Table 1: Summary of main notations

| $\eta$ | Learning rate | $\lambda$ | Regularization parameter |
|---|---|---|---|
| $\tau$ | Temperature coefficient | $K$ | Batch size |
| $\mathfrak{N}$ | Set of negative samples | $\mathfrak{B}$ | The set of $\boldsymbol{Y}_n$ and negative samples |
| $\epsilon_{\min}$ | frequency of minority feature | $\epsilon_{\max}$ | frequency of majority feature |
| $\epsilon_j$ | Feature frequency for feature $j$ | $\mathcal{N}_i$ | Set of dominate features for neuron $i$ |
| $\mathcal{M}_j$ | Set of ordinary neurons for feature $j$ | $\mathcal{M}_j^\star$ | Set of lucky neurons for feature $j$ |

2012; Candès & Tao, 2010), and neuroscience (Vinje & Gallant, 2000; Olshausen & Field, 1997; 2004; Foldiak, 2003).

Assumption 3.1 states that each token within a sample can be expressed as a weighted sum of a subset of features from the dictionary matrix $\boldsymbol{M}$, corrupted by additive noise $\boldsymbol{\xi}$. Here, $\boldsymbol{M}$ denotes the dictionary matrix, $\boldsymbol{z}$ represents the latent signal, and $\boldsymbol{\xi}$ corresponds to spurious noise. Importantly, in the presence of noise, particularly when the noise level is comparable to or even exceeds the signal magnitude, no linear mapping can recover the latent signal directly from the input. This makes the model simple in form yet intrinsically challenging, thereby providing a favorable abstraction for theoretical analyses of nonlinear neural networks.

**Assumption 3.1** (Sparse Coding Model). *For a paired data $(\boldsymbol{X}_n, \boldsymbol{Y}_n)$, the data structure is:*

$$\boldsymbol{X}_n = \left[ \boldsymbol{M}\boldsymbol{z}_n^{(1)} + \boldsymbol{\xi}_n^{(1)}, \ \boldsymbol{M}\boldsymbol{z}_n^{(2)} + \boldsymbol{\xi}_n^{(2)}, \ \ldots, \ \boldsymbol{M}\boldsymbol{z}_n^{(L)} + \boldsymbol{\xi}_n^{(L)} \right]$$
$$\boldsymbol{Y}_n = \left[ \boldsymbol{M}\boldsymbol{z}_n^{+(1)} + \boldsymbol{\xi}_n^{+(1)}, \ \boldsymbol{M}\boldsymbol{z}_n^{+(2)} + \boldsymbol{\xi}_n^{+(2)}, \ \ldots, \ \boldsymbol{M}\boldsymbol{z}_n^{+(L)} + \boldsymbol{\xi}_n^{+(L)} \right]$$

(6)

*Here, each $\boldsymbol{z}_n^{(i)} \in \mathbb{R}^d$ represents the latent signal at the $\ell$-th token, and $\boldsymbol{\xi}_n^{(i)}$ denotes the additive noise. $\boldsymbol{M} = [\boldsymbol{M}_1, \ldots, \boldsymbol{M}_d] \in \mathbb{R}^{d_1 \times d}$ is the dictionary matrix, which is a column-orthonormal matrix and satisfies $\|\boldsymbol{M}_j\|_\infty \leq \widetilde{O}\big(\frac{1}{\sqrt{d_1}}\big), \quad \forall j \in [d]$. We also assume $d_1 = poly(d)$.*

Assumption 3.2 requires that the latent signal be both bounded and sparse. Sparsity is a standard assumption, introduced primarily to facilitate the theoretical analysis, yet it also agrees with empirical observations that real-world data typically activate only a small subset of latent factors rather than spreading energy across all coordinates. Moreover, the assumption enforces sign consistency across tokens within the same sample, meaning that whenever a particular coordinate is active, its sign remains identical across all tokens. This ensures that different parts of the same sample contribute coherently to the underlying latent feature instead of producing conflicting activations.

**Assumption 3.2** (Latent Signal). *We have assumptions on the latent signal $\{\boldsymbol{z}^{(i)}\}_{i=1}^L$ with $\boldsymbol{z}^{(i)} = (z_1^{(i)}, \ldots, z_j^{(i)}, \ldots, z_d^{(i)})^\top$: (i) all $z_j^{(i)}$ are bounded and symmetric around zero over all samples. Moreover, we have $\Pr(|z_{n,j}^{(i)}| \neq 0) = \Theta\left(\frac{\log\log d}{d}\right)$; (ii) $z_j^{(i)}$ share the same sign across all $i \in [L]$.*

Assumption 3.3 states that noise follows Gaussian distributions. This is a mild condition, as no strong restriction is imposed on its variance. In particular, the noise magnitude can exceed that of the sparse signal when $d_1 \gg d$. The assumption is adopted for analytical purposes and demonstrates that contrastive learning can recover meaningful latent representations even in regimes where the signal is dominated by noise.

**Assumption 3.3** (Noise). *Here each noise term $\boldsymbol{\xi}_n^{(\ell)}$ and $\boldsymbol{\xi}_n^{+(\ell)}$ for $\ell \in [L]$ is independently drawn from the same distribution $\boldsymbol{\xi}_n^{(\ell)} \sim \mathcal{N}(0, \sigma_\xi^2 \boldsymbol{I}_{d_1})$, with variance $\sigma_\xi^2 = \Theta\left(\frac{\sqrt{\log d}}{d}\right)$.*

Assumption 3.4 states that a pair of positive samples shares the same set of features when aggregated over all tokens within the sample. Intuitively, this means that the two samples encode the same semantic structure, even though their individual token-level representations may differ. In contrast, a negative pair is formed by two random samples whose latent signals are completely independent.

**Assumption 3.4** (Positive and Negative Pairs). *A pair of samples $\boldsymbol{X}_n$ and $\boldsymbol{Y}_n$ form a positive pair if and only if $\text{supp}\big(\sum_{\ell=1}^L \boldsymbol{z}_n^{(\ell)}\big) = \text{supp}\big(\sum_{\ell=1}^L \boldsymbol{z}_n^{+(\ell)}\big), \quad \text{sign}\big(\sum_{\ell=1}^L \boldsymbol{z}_n^{(\ell)}\big) = \text{sign}\big(\sum_{\ell=1}^L \boldsymbol{z}_n^{+(\ell)}\big)$. By contrast, negative pairs are defined such that the corresponding latent signals are independent.*

Definition 3.1 states that each feature is controlled by $\epsilon_j$. Intuitively, $\epsilon_j$ characterizes how often feature $j$ appears across the data. When $\epsilon_j$ is small, feature $j$ is regarded as a minority feature.

**Definition 3.1** (Majority and minority features). *For each feature index $j \in [d]$, and for all $i \in [L]$ and all samples, the activation probability of the sparse signal satisfies:* $\Pr\left(\left|z_j^{(i)}\right| \neq 0\right) = \Theta\left(\epsilon_j \frac{\log\log d}{d}\right)$ . *We define the* majority features *as those associated with $\epsilon_{\max} = \max_{j \in [d]} \epsilon_j$, and the* minority features *as those associated with $\epsilon_{\min} = \min_{j \in [d]} \epsilon_j$.*

### 3.3 FORMAL THEORETICAL RESULTS

Theorem 3.1 analyzes the vanilla contrastive learning algorithm without pruning, showing how data imbalance affects performance. Lemmas 3.1 and 3.2 provide intermediate steps toward its proof and reveal how training dynamics evolve, despite the algorithm appearing to follow a consistent gradient-based procedure. Theorem 3.2 then gives the results with pruning, showing how pruning improves performance under imbalance.

#### 3.3.1 VANILLA CONTRASTIVE LEARNING

Lemma 3.1 shows two main effects of contrastive learning in the first training stage: (a) neuron weights grow in feature directions but are suppressed in non-feature directions, and (b) the growth rate in a feature direction $M_j$ depends on its frequency $\epsilon_j$, with larger $\epsilon_j$ leading to faster growth and smaller $\epsilon_j$ making the feature harder to capture early in training. We can find the Proof of Lemma 3.1 in Appendix C.4.

**Lemma 3.1** (Stage 1). *During the first training stage, the update of neuron weights $w_i^{(t)}$ can be bounded for all $t \in [0, T_1]$ as follows, where $C_z$ denotes positive constants and $T_1 = \Theta\left(\frac{d_1 \log d}{\eta \log\log d}\right)$.*

$$|\langle w_i^{(t+1)}, M_j \rangle| \geq |\langle w_i^{(t)}, M_j \rangle|(1 - \eta\lambda + \epsilon_j \frac{\eta C_z \log\log d}{d}) - \widetilde{O}\left(\frac{\eta\|w_i^{(t)}\|_2}{\text{poly}(d_1)}\right), \qquad (7)$$

$$and \quad |\langle w_i^{(t+1)}, M_j^{\perp} \rangle| \leq (1 - \eta\lambda)|\langle w_i^{(t)}, M_j^{\perp} \rangle| + \widetilde{O}\left(\frac{\eta\|w_i^{(t)}\|_2}{\text{poly}(d_1)}\right). \qquad (8)$$

Before presenting the theoretical results in Stage 2, we first categorize neurons into two groups. The *ordinary neurons* $\mathcal{M}_j$ strongly align with a certain direction, while the *lucky neurons* $\mathcal{M}_j^{\star}$ form a special subset that aligns with only one feature direction (see Appendix B for the formal definition). In Stage 2: (a) lucky neurons in $\mathcal{M}_j^{\star}$ grow significantly in alignment with $M_j$, controlled by $\epsilon_j$, though their number remains small; (b) ordinary neurons in $\mathcal{M}_j$ are bounded by the feature components of lucky neurons up to a constant factor. We can find the Proof of Lemma 3.2 in Appendix D.4.

**Lemma 3.2** (Stage 2). *During the second training stage, the update of neuron weights $w_i^{(t)}$ can be bounded for all $t \in [T_1, T_2]$ as follows, where $T_2 = T_1 + \Theta\left(\frac{d\tau \log d}{\epsilon_{\max}\eta \log\log d}\right)$.*

*(a) For each $j \in [d]$, if $i \in \mathcal{M}_j^{\star}$, then:*

$$|\langle w_i^{(T_2)}, M_j \rangle|^2 \geq 2 \cdot \frac{\varepsilon_j}{\varepsilon_{\max}} \cdot \|w_i^{(T_1)}\|_2^2, \quad with \quad |\mathcal{M}_j^{\star}| \geq m \cdot d^{-\left(\frac{\varepsilon_{\max}}{\varepsilon_{\min}}\right)^2}. \qquad (9)$$

*(b) For each $j \in [d]$, if $i' \in \mathcal{M}_j$ and $i \in \mathcal{M}_j^{\star}$, then:*

$$|\langle w_{i'}^{(T_2)}, M_j \rangle| \leq O(|\langle w_i^{(T_2)}, M_j \rangle|). \qquad (10)$$

Theorem 3.1 establishes the convergence of the algorithm. In particular, (11) shows that the algorithm converges with bounded training error. Moreover, (12) characterizes the structure of the learned neuron weights: upon convergence, they become strongly aligned with a subset of features within $\mathcal{N}_j$, weakly aligned with the remaining features, and remain small in the non-feature directions. The size of $\mathcal{N}_j$ is bounded as in (14), and only a limited number of neurons specialize in learning a single feature. We can find the Proof of Theorem 3.1 in Appendix E.4.

**Theorem 3.1** (**Stage 3: Convergence**). *Let* $m = d^{C_m}$ *be the number of neurons and* $\tau = \text{polylog}(d)$, *where* $C_m$ *denotes positive constants and* $\Xi_2 = d^{C_m - \left(\frac{\epsilon_{\min}}{\epsilon_{\max}}\right)^2}$. *Suppose we train the neural net* $f_{\boldsymbol{\theta}}$ *via contrastive learning, and consider iterations* $T \in [T_3, T_4]$ *with* $T_3 = \frac{d^{1.01}}{\eta}$ *and* $T_4 = \frac{d^{1.99}}{\eta}$. *Then the following guarantees hold:*

$$\frac{1}{T}\sum_{t\in[T]} L_{\text{aug}}(f_{\boldsymbol{\theta}^{(t)}}) \leq o(1) \tag{11}$$

*Moreover, for each neuron* $i \in [m]$ *and* $t \in [T_3, T_4]$, *the weight will learn the following set of features:*

$$\boldsymbol{w}_i^{(t)} = \sum_{j\in\mathcal{N}_i} \alpha_{i,j}\boldsymbol{M}_j + \sum_{j\notin\mathcal{N}_i} \alpha'_{i,j}\boldsymbol{M}_j + \sum_{j\in[d_1]\setminus[d]} \beta_{i,j}\boldsymbol{M}_j^{\perp}, \tag{12}$$

*where*

$$\alpha_{i,j} \in \left[\frac{\epsilon_j}{\epsilon_{\max}}\frac{\tau}{\Xi_2}, \frac{\epsilon_j}{\epsilon_{\max}}\tau\right], \alpha'_{i,j} \leq o\left(\frac{\epsilon_j}{\epsilon_{\max}}\frac{1}{\sqrt{d}}\right)\|\boldsymbol{w}_i^{(t)}\|_2, |\beta_{i,j}| \leq o\left(\frac{1}{\sqrt{d_1}}\right)\|\boldsymbol{w}_i^{(t)}\|_2. \tag{13}$$

*Furthermore, the size of* $\mathcal{N}_i$ *is bounded as*

$$|\mathcal{N}_i| = O\left(d^{1-\left(\frac{\epsilon_{\min}}{\epsilon_{\max}}\right)^2}\right). \tag{14}$$

*Finally, for each* $\boldsymbol{M}_j$, *there are at least* $\Omega(m \cdot d^{-\left(\frac{\epsilon_{\max}}{\epsilon_{\min}}\right)^2})$ *neurons* $i \in [m]$ *such that* $\mathcal{N}_i = \{j\}$.

**Remark 1**: For a neuron $\boldsymbol{w}_i$, its convergent weights are aligned with a subset of features $\mathcal{N}_i$. In contrast, all other feature directions are smaller by an order of $\frac{1}{\sqrt{d}}$. Hence, we can say that neuron $\boldsymbol{w}_i$ is dominated by the features in $\mathcal{N}_i$. Moreover, the neurons associated with learning feature $j$ are influenced by the frequency of that feature, which intuitively explains how imbalance shapes the distribution of neuron weights.

**Remark 2:** We emphasize that the success of contrastive learning relies on neurons that specialize in a single feature, referred to as lucky neurons, i.e., $\cup_j \mathcal{M}_j^{\star}$. In contrast, neurons that learn mixtures of features are useful only for a limited subset of downstream tasks. The number of lucky neurons for each feature is lower bounded by $m \cdot d^{-\left(\frac{\epsilon_{\max}}{\epsilon_{\min}}\right)^2}$, as derived from (9). Consequently, beyond the reduced neuron magnitude in minority feature directions, imbalance also decreases the number of neurons that learn purified features. This, in turn, requires a more complex model with a larger number of neurons to capture all features, leading to higher computational cost. Moreover, the upper bound of $|\mathcal{N}_i|$ increases as the ratio $\frac{\varepsilon_{\min}}{\varepsilon_{\max}}$ decreases, which is undesirable because it indicates that more neurons learn mixtures of features rather than pure ones.

**Remark 3:** Theorem 3.1 shows that each underlying semantic feature is captured cleanly by a subset of lucky neurons. When upstream contrastive learning produces a representation in which all semantic features are encoded in pure and separable directions, the resulting feature space becomes highly structured: it contains explicit axes corresponding to every true feature. If a downstream task relies on any subset of these features, a linear probe (or any simple classifier) can easily extract them because the corresponding feature directions are directly represented by the lucky neurons. In this sense, stronger neuron specialization leads to better linear separability and, consequently, improved downstream generalization.

### 3.3.2 CONTRASTIVE LEARNING WITH PRUNING

Theorem 3.2 describes the training dynamics in the pruning setting, serving as the counterpart to the earlier result obtained without pruning. To highlight the effect more clearly, we focus on stage 3. In particular, pruning amplifies the learning of minority features: (a) for lucky neurons aligned with minority directions, the neuron weights increase in that direction at the order of $\frac{\alpha}{d}$, where $\alpha$ is the pruning ratio. (b) In contrast, neurons associated with non-minority features exhibit much smaller growth, with updates in those directions on the order of $\frac{\alpha}{d^2}$ per iteration. (c) Most importantly, when training converges, the coefficients $\alpha_{i,j^{\star}}$, projecting neuron weights onto the minority feature $\boldsymbol{M}_{j^{\star}}$, become dominant and independent of the ratio $\frac{\varepsilon_{\min}}{\varepsilon_{\max}}$. We can find the Proof of Theorem 3.2 in Appendix F.4.

**Theorem 3.2** (**Pruning: Reinforcing Minority Feature Learning**). *With pruning ratio $\alpha$, the following statements hold:*

*(a) When $i^\star \in \mathcal{M}_{j^\star}^\star$, we have*

$$\langle \boldsymbol{w}_{i^\star}^{(t+1)}, \boldsymbol{M}_{j^\star} \rangle \geq \left( 1 - \eta\lambda + \Omega\left( \eta\epsilon_{j^\star}^2 \alpha \, \frac{C_z \log\log d}{d} \right) \right) \langle \boldsymbol{w}_{i^\star}^{(t)}, \boldsymbol{M}_{j^\star} \rangle. \tag{15}$$

*(b) When $\forall i$ and $j \neq j^\star$, we have*

$$\langle \boldsymbol{w}_i^{(t+1)}, \boldsymbol{M}_j \rangle \leq \left( 1 + O\left( \eta\epsilon_{j^\star}^3 \alpha \, \frac{C_z \log\log d}{d^2} \right) \right) \langle \boldsymbol{w}_i^{(t)}, \boldsymbol{M}_j \rangle. \tag{16}$$

*(c) For neuron $i \in \mathcal{M}_{j^\star}^\star$ and $t = T_5$, contrastive learning learns the following decomposition:*

$$\boldsymbol{w}_i^{(t)} = \alpha_{i,j^\star} \boldsymbol{M}_{j^\star} + \sum_{j \notin \mathcal{N}_i} \alpha_{i,j}' \boldsymbol{M}_j + \sum_{j \in [d_1] \setminus [d]} \beta_{i,j} \boldsymbol{M}_j^\perp, \tag{17}$$

*where*

$$\alpha_{i,j^\star} \in \left[ \frac{\tau}{\Xi_2}, \tau \right], \quad \alpha_{i,j}' \leq o\left( \left( 1 + \frac{1}{d} \right) \cdot \frac{1}{\sqrt{d}} \right) \|\boldsymbol{w}_i^{(t)}\|_2, \quad |\beta_{i,j}| \leq o\left( \frac{1}{\sqrt{d_1}} \right) \|\boldsymbol{w}_i^{(t)}\|_2. \tag{18}$$

*Finally, for feature $\boldsymbol{M}_{j^\star}$, there are at least $\Omega(m \cdot d^{-1})$ neurons $i \in [m]$ such that $\mathcal{N}_i = \{j^\star\}$.*

**Remark 1:** We would like to clarify two implicit assumptions underlying the results. First, the pruning ratio is implicitly upper bounded by $|\mathcal{M}_{j^\star}|$, so that under magnitude-based pruning, we can guarantee that all pruned neurons are those aligned with the minority feature $\boldsymbol{M}_{j^\star}$. In practice, however, the pruning ratio can be extended to include any neurons that have learned minority features, i.e., any $i$ with $j \in \mathcal{N}_i$. Second, we assume that the magnitude of all non-minority features is comparable. Intuitively, in the general case, neurons associated with the minority feature grow until their magnitude reaches the level of the second-smallest feature. At that point, both the original minority feature and the second-smallest feature effectively become the new minority features, and the process continues inductively across features. A detailed analysis of this extension is omitted for simplicity, so that we can prove and present the pruning benefits in a clear manner.

**Remark 2:** The difference between neurons learning minority features and those learning majority features arises from their sensitivity to pruning. As shown in Theorem 3.1, the magnitude of a neuron is determined by its dominant feature and the frequency of that feature. For neurons in $\mathcal{M}_{j^\star}$ that specialize in purified minority features, their magnitudes are significantly smaller than those of other neurons and are therefore more likely to be pruned. This pruning effect results in relatively smaller positive logits and larger negative logits on samples containing the minority feature (see (5)), thereby increasing the influence of these samples on the gradient updates. Since features are assumed to be independent across the data, such samples have a low probability of simultaneously containing other features, resulting in a difference on the order of $1/d$ in the growth dynamics of these neurons.

**Remark 3:** Unlike in the vanilla learning paradigm, the magnitude of $\alpha_{i,j^\star}$ no longer depends on the ratio $\frac{\varepsilon_{\min}}{\varepsilon_{\max}}$, which suggests that the representation of the minority feature is not suppressed by data imbalance. Although the coefficients $\alpha_{i,j}'$ for other features may grow slightly due to the extended number of iterations required for convergence, their increase remains only on the order of $1/d$. Consequently, $\alpha_{i,j^\star} \gg \alpha_{i,j}'$, which suggests that the minority feature is strongly amplified and clearly distinguished from other contributions. This, in turn, drives more neurons to specialize in the purified minority feature, leading to more robust and effective representation learning.

## 4 NUMERICAL EXPERIMENTS

**Experiments on CIFAR10-LT, CIFAR100-LT, and ImageNet-LT.** Table 2 reports the results of linear probe evaluation on CIFAR10-LT, CIFAR100-LT, and ImageNet-LT under long-tailed settings, comparing vanilla contrastive learning (w/o pruning) against our proposed approach (w/ pruning). Following the setup in (Jiang et al., 2021; Kang et al., 2020; Chen et al., 2020), models are

first pretrained and then evaluated using a linear probe, where a linear classifier is trained on frozen representations. The imbalance ratio, $\rho$, is defined as the ratio between the number of samples in the majority and minority classes, with larger values indicating more severe imbalance. Two evaluation metrics are considered: overall classification accuracy (%) and the accuracy gap ($\Delta_{20}$) between the top 20% head classes and the bottom 20% tail classes. The results show that pruning consistently improves accuracy across all datasets, with improvements becoming more substantial as $\rho$ increases. Furthermore, pruning generally reduces $\Delta_{20}$, indicating better balance between head and tail classes. These results indicate that pruning not only enhances overall downstream task performance but also reduces the performance gap between head and tail classes. We also provide additional synthetic data experiments to support our theoretical insights; due to space limitations, these results are deferred to Appendix A.2.

Table 2: Linear probe accuracy (%) on CIFAR10-LT, CIFAR100-LT, and ImageNet-LT. $\Delta_{20}$ denotes the accuracy gap between the top 20% head classes and bottom 20% tail classes.

| Dataset | $\rho$ | Accuracy | | $\Delta_{20}$ | |
|---|---|---|---|---|---|
| | | w/o pruning | w/ pruning | w/o pruning | w/ pruning |
| CIFAR10-LT | 1 | 90.93 | 91.52 | 1.54 | 1.28 |
| | 10 | $79.25 \pm 1.03$ | $84.92 \pm 0.67$ | $3.42 \pm 1.02$ | $2.99 \pm 0.92$ |
| | 50 | $75.58 \pm 0.84$ | $83.60 \pm 1.02$ | $3.92 \pm 1.21$ | $3.35 \pm 0.76$ |
| | 100 | $74.24 \pm 0.82$ | $81.31 \pm 0.94$ | $5.69 \pm 1.35$ | $5.62 \pm 0.99$ |
| CIFAR100-LT | 10 | $51.21 \pm 1.21$ | $56.33 \pm 1.51$ | $2.45 \pm 0.57$ | $1.37 \pm 0.46$ |
| | 50 | $49.32 \pm 0.45$ | $56.12 \pm 0.32$ | $4.95 \pm 1.02$ | $2.57 \pm 0.92$ |
| | 100 | $47.12 \pm 0.51$ | $54.93 \pm 0.50$ | $7.11 \pm 0.45$ | $4.38 \pm 0.22$ |
| ImageNet-LT | 256 | 63.21 | 65.12 | 8.47 | 7.21 |

## 5 LIMITATION

Our work has two main limitations. The first concerns studying the pruning ratio and pruning scheme in magnitude-based pruning. Providing a fully precise characterization of how performance varies across different ratios and schemes is highly nontrivial, and doing so would require making more precise assumptions about the data distribution. This will be part of our future work. Furthermore, existing theoretical results in our feature learning framework focus on a single, simplified architectural setting. Extending the analysis to more complex or realistic models will be another direction for future work, and may require fundamentally different derivations and analytical tools.

## 6 CONCLUSION

This work provides a theoretical analysis of the training dynamics of a Transformer-MLP model in learning feature representations through contrastive learning under imbalanced data settings. Specifically, we quantitatively characterize how the presence of minority features reduces the number of neurons that capture those features, as well as the number of "lucky neurons" that specialize in a single feature. This reduction, in turn, harms the overall representation learning ability of the model. Motivated by this theoretical characterization, we revisit the magnitude-based pruning approach to address data imbalance. In particular, we theoretically demonstrate that pruning can enhance gradient updates along the minority feature direction. This encourages more neurons to specialize in pure minority features, thereby yielding more robust and balanced representations. Looking ahead, promising directions include exploring alternative strategies beyond pruning that could further promote minority-feature learning.

## ACKNOWLEDGMENTS

This work was supported in part by the National Science Foundation (NSF) under Grants #2349879, #2349878, #2425811, and #2430223. Part of Yating's work was completed while she was a Ph.D.

student at Rensselaer Polytechnic Institute (RPI) and was supported in part by the Army Research Office (ARO) under Grant W911NF-25-1-0020, as well as by the Rensselaer–IBM Future of Computing Research Collaboration (`http://airc.rpi.edu`). We also thank the anonymous reviewers for their constructive and insightful comments.

## LLM USAGE DISCLOSURE

We used large-language models (ChatGPT) to aid in polishing the writing of this paper. For numerical experiments, we employed AI-assisted coding tools (GitHub Copilot and ChatGPT) to support code development.

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

# A    OVERVIEW OF THE APPENDIX AND PROOF SKETCH

The appendices are organized systematically to provide supporting materials for the main text. Appendix B introduces key notations and definitions, along with basic lemmas describing properties at initialization. Appendices C, D, and E present the proofs of the training dynamics of vanilla contrastive learning (without pruning) under the imbalanced data setting. Specifically, Appendix C contains the proof of Stage 1, corresponding to Lemma 3.1 in the main text; Appendix D contains the proof of Stage 2, corresponding to Lemma 3.2; and Appendix E contains the proof of Stage 3, corresponding to Theorem 3.1, which concludes the analysis with the final convergence results. Appendix F then provides the proof of our proposed algorithm (with pruning), corresponding to Theorem 3.2 in the main text. We recommend that readers first consult the proof sketch before examining the detailed lemmas and proofs in the appendices.

In addition, Appendices G-K collect the proofs of the lemmas referenced throughout the earlier appendices. To maintain readability, some of these lemma proofs are included only in the supplementary material. While these details are not essential for following the main arguments, we provide them in full for completeness.

## A.1    PROOF SKETCH

In Stage 1, we analyze how neurons learn the features. Each neuron gradually learns the relevant feature directions while hardly learning the non-feature directions. Concretely, the projection of a neuron weights onto the feature subspace, though small at the beginning, grows rapidly during training and becomes significant, reaching the order of $\Omega(\|\boldsymbol{w}_i^{(T_1)}\|_2^2)$ (see Appendix D, Theorem C.1), while the projection onto the non-feature subspace stays nearly unchanged. The reason why the neuron weights grow toward the feature subspace is that the latent variable $z_{n,j}^{(i)}$ and $z_{n,j}^{+(i)}$ are dependent. This dependence produces an incremental term of order: $\epsilon_j \frac{\eta C_z \log \log d}{d}$, which accumulates during training and drives the neuron weights further into the feature space. In contrast, because the feature are orthogonal to the non-feature directions, and the latent variable $z_{n,j}^{(i)}$ is independent of the noise, the weights in the non-feature subspace remain essentially unchanged. The only variation that appears there is a negligible increment of size about $\frac{1}{\text{poly}(d_1)}$. (see Appendix C, Lemma C.1).

In Stage 2, the lucky neurons with large projection on a feature direction become activated and align clearly with that feature. If a neuron does not belong to $\mathcal{M}_j$, its projection on feature $j$ remains small, so it cannot be activated and has only weak alignment. The projection on non-feature directions stays very small, so neurons do not learn the non-feature components (Appendix D, Lemma D.1). As a result, if neuron $i$ is lucky for feature $j$, the projection of $\boldsymbol{w}_i^{(T_2)}$ onto $\boldsymbol{M}_j$ is on the order of the $\Omega(1)\|\boldsymbol{w}_i^{(T_2)}\|_2$, meaning the neuron has already focused on $\boldsymbol{M}_j$ (see Appendix D, Theorem D.1).

In Stage 3, neurons in $\mathcal{M}_j^\star$ continue to strengthen their projection on the corresponding feature $j$, and this projection remains the dominant part of their weight. Neurons not in $\mathcal{M}_j$ keep only a small projection on feature $j$, so they cannot be activated. The projections on non-feature directions stay negligible throughout. Overall, the growth of neurons continues along the same directions established earlier, and the network starts to converge around $T_3$. At this point, each neuron weight vector $\boldsymbol{w}_i$ eventually aligns with a set of features $\mathcal{N}_i$, which corresponds to the features that already had some degree of alignment with $\boldsymbol{w}_i$ at initialization.

In pruning stage, we rigorously show that pruning the neurons which have learned minority features enhances the learning of those features. After pruning, the gradients in backpropagation for neurons aligned with minority features become significantly stronger, which forces these neurons to further learn the minority features. To some extent, this reinforcement compensates for their lower frequency $\epsilon_{j^\star}$ compared to majority features. In contrast, for neurons associated with majority features, pruning does not change their gradients, so they continue to update in the same speed and direction as before. As a result, the decomposition of neurons aligned with minority features becomes concentrated on those features, while contributions from other features and from non-feature directions remain suppressed and negligible.

### A.2 SYNTHETIC EXPERIMENTAL SETTINGS

In this subsection, we provide the detailed settings of our synthetic experiments. We follow the standard sparse coding model to generate synthetic data, consistent with our main paper. Each generated data sample is passed into a Transformer to obtain a token embeddings, which is then processed by an MLP trained with a contrastive objective. After training, we evaluate the alignment of the learned neurons to the minority feature. Specifically, we report: (i) the number of neurons with alignment above a threshold (Figure 2); (ii) the maximum alignment value (Figure 3); (iii) the mean cosine similarity between positive pairs on the test set (Figure 4); and (iv) the regression test mean squared error (MSE) (Figure 5).

**Experiment 1–2 (Alignment with the minority feature).** We evaluate how well the learned neurons align with the minority feature. Specifically, for each $w_i$, we compute its normalized projection onto the minority feature. Figure 2 reports the number of neurons with projection larger than 0.3, while Figure 3 shows the maximum projection value across neurons. We vary $\varepsilon_{\min}$ from 0.1 to 1.0, and consider different noise-to-signal ratio (NSR) levels, where NSR $= \sigma^2 d_1$ with $\sigma^2 \in \{(1/100)^2, (3/100)^2, (5/100)^2\}$ and $d_1 = 500$. Each experiment is independently repeated 100 times, and we report the mean results. The results demonstrate that as $\varepsilon_{\min}$ increases, both the number of aligned neurons and the maximum alignment consistently grow, providing direct empirical support for our theoretical results. The detailed hyperparameter settings can be found in the code.

**Experiment 3 (Average cosine similarity on the test set).** We evaluate performance on the test set using the average cosine similarity between positive pairs. At test time, we keep the feature space identical. For each configuration, we generate 5000 test pairs with a fixed test seed and report the mean cosine similarity. We vary $\varepsilon_{\min}$ from 0.05 to 0.5 in increments of 0.05, and use $\sigma^2 \in (5/100)^2, (7.5/100)^2, (10/100)^2$ to compute the corresponding NSR levels. Each configuration is independently repeated 100 times, and the averaged results are reported. The results in Figure 4 show that the average test cosine similarity consistently increases as $\varepsilon_{\min}$ grows, indicating a stronger ability to learn the minority feature. Consequently, the quality of the learned features on the test set is enhanced, the model generalizes better, and the test performance becomes stronger, which provides further empirical support for our theoretical results. Detailed hyperparameter settings can be found in the code.

**Experiment 4 (Test MSE on the downstream regression task).** We evaluate the performance of the downstream regression task on the test set, measured by Test MSE. Both the downstream training stage and the test stage use a unified feature space. A linear regression head is trained on the representations obtained from upstream learning, using 1000 training pairs, and then evaluated on 5000 test pairs with a fixed test seed. In the setup, we vary $\varepsilon_{\min}$ from 0.05 to 0.5 with a step size of 0.05, and use $\sigma^2 \in \{(3/100)^2, (5/100)^2, (7.5/100)^2\}$ to compute the corresponding NSR levels. Each configuration is independently repeated 100 times, and the averaged results are reported (Figure 5). The results show that as $\varepsilon_{\min}$ increases, the test MSE consistently decreases, indicating a stronger ability to learn the minority feature. Consequently, the model achieves better overall learning and stronger generalization in downstream tasks, which is consistent with our theoretical analysis. Detailed hyperparameter settings can be found in the code.

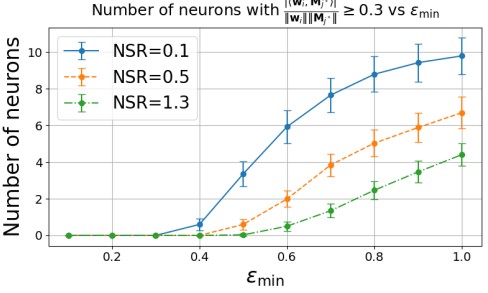

Figure 2: Number of neurons with $\frac{|\langle w_i, M_j \rangle|}{\|w_i\|\|M_j\|} \geq 0.3$ vs $\varepsilon_{\min}$ for different NSR values.

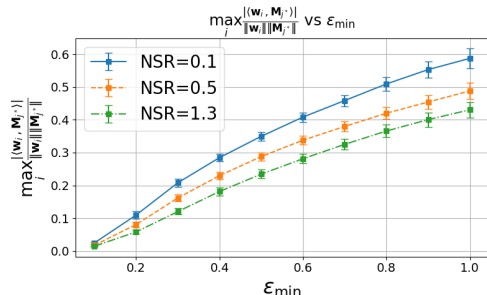

Figure 3: Maximum $\frac{|\langle w_i, M_j \rangle|}{\|w_i\|\|M_j\|}$ vs $\varepsilon_{\min}$ for different NSR values.

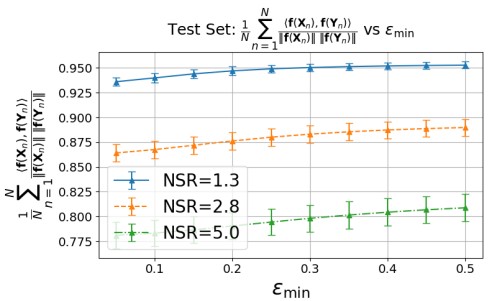 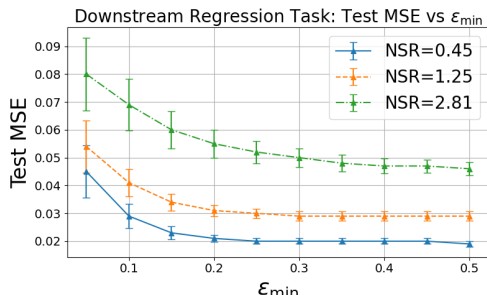

Figure 4: $\frac{1}{N}\sum_{n=1}^{N}\frac{\langle f(X_n),f(Y_n)\rangle}{\|f(X_n)\|\|f(Y_n)\|}$ vs $\varepsilon_{\min}$ for different NSR values.

Figure 5: Downstream regression task: Test MSE vs $\varepsilon_{\min}$ for different NSR values.

**Experiment 5 (lucky vs. mixed-feature neurons).**

The following heatmap visualizes the alignment values of 24 neurons (indexed 0-23) across 9 features (indexed 0-8), where the first five features are majority features and the last four are minority features.

Example of a lucky neuron: In our experiment, $d_1 = 500$, so the expected alignment from random initialization is approximately 0.002. After training, Neuron 0 exhibits a strong alignment with Feature 4 (around 0.3), while its alignment with the remaining eight features is negligible. Thus, Neuron 0 can be viewed as a lucky neuron for Feature 4.

Example of a mixed-feature neuron: Neuron 10, in contrast, does not exhibit a dominant alignment with any single feature. Instead, it shows moderate alignment with multiple features, specifically, its values on Features 0, 2, and 7 are 0.13, 0.18, and 0.15 respectively, while remaining negligible on all other features. This behavior corresponds to a mixed-feature neuron.

Additional examples: Further instances observed in our experiments include, but are not limited to. Lucky neurons: Neuron 6 and Neuron 17 for Feature 0; Neuron 3 and Neuron 18 for Feature 1; Neuron 4 for Feature 2; Neuron 13 for Feature 3. Mixed-feature neurons: Neuron 1 (Features 1 and 2), Neuron 2 (Features 1 and 3), Neuron 21 (Features 3 and 8).

These empirical patterns closely reflect the specialization and superposition behaviors predicted by our theoretical analysis.

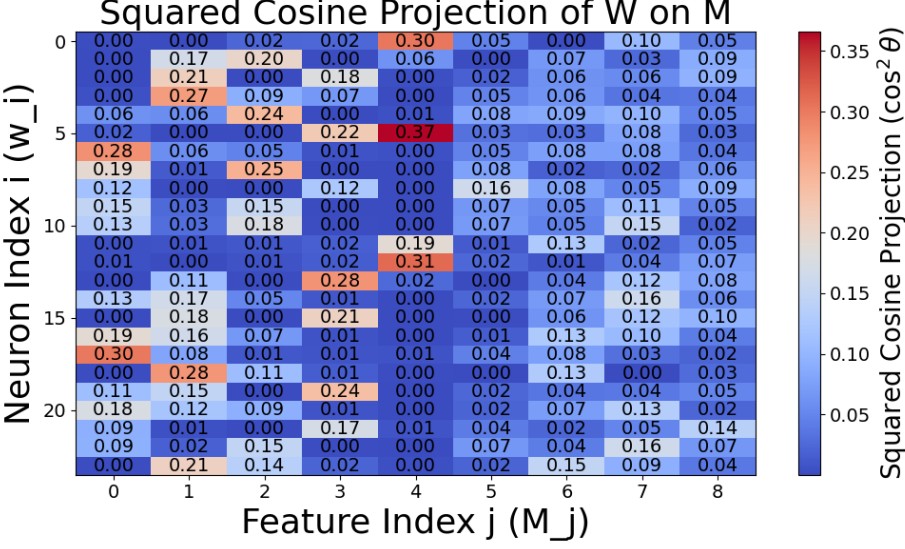

Figure 6: Squared cosine alignment heatmap between 24 neurons and 9 features, illustrating lucky neurons (single strong alignment) and mixed-feature neurons (multiple features alignment).

# B  NOTATIONS AND LEMMAS

To streamline the presentations, we begin by introducing the key notations and outlining key fundamental derivations that will serve as the basis for the subsequent analysis.

**Notations.**  First, we introduce the notations that will appear in the appendix.

Let $\boldsymbol{z}_{\boldsymbol{X}}^{(r)}$ denote the representation of the $r$-th token of data sample $\boldsymbol{X}_n$ after passing through the transformer. Similarly, $\boldsymbol{z}_{\boldsymbol{Y}}^{(s)}$ denotes the $s$-th token of data sample $\boldsymbol{Y}_n$ after the transformer.

**Empirical Gradient.**  To facilitate the calculation of the gradient of the loss function $\ell\big(\boldsymbol{f_\theta}, \boldsymbol{X}_k, \mathfrak{B}_k\big)$ with respect to the weights $\{\boldsymbol{w}_i^{(t)}\}_{i\in[m]}$, we introduce the following notation. We denote the positive logit by $\ell'_{p,t}(\boldsymbol{X}_n, \mathfrak{B})$ and the negative logits by $\ell'_{s,t}(\boldsymbol{X}_n, \mathfrak{B})$.

$$\ell'_{p,t}(\boldsymbol{X}_n, \mathfrak{B}) := \frac{\exp\big(\mathrm{Sim}_{f_t}(\boldsymbol{X}_n, \boldsymbol{Y}_n)/\tau\big)}{\sum_{\boldsymbol{X}\in\mathfrak{B}} \exp\big(\mathrm{Sim}_{f_t}(\boldsymbol{X}_n, \boldsymbol{X})/\tau\big)}, \tag{19}$$

$$\ell'_{s,t}(\boldsymbol{X}_n, \mathfrak{B}) := \frac{\exp\big(\mathrm{Sim}_{f_t}(\boldsymbol{X}_n, \boldsymbol{X}_{n,s})/\tau\big)}{\sum_{\boldsymbol{X}\in\mathfrak{B}} \exp\big(\mathrm{Sim}_{f_t}(\boldsymbol{X}_n, \boldsymbol{X})/\tau\big)}. \tag{20}$$

For convenience, we simplify the *positive logit* $\ell'_{p,\boldsymbol{\theta}^{(t)}}(\boldsymbol{X}_n, \boldsymbol{Y}_n, \mathfrak{N}_n)$ as $\ell'_{p,t}$, and the *negative logit* $\ell'_{s,\boldsymbol{\theta}^{(t)}}(\boldsymbol{X}_n, \boldsymbol{Y}_n, \boldsymbol{X}_{n,s}, \mathfrak{N}_n)$ as $\ell'_{s,t}$. For clarity of exposition, we suppress the dependence on $(\boldsymbol{X}_n, \mathcal{B})$ when it can be inferred from the context.

Then, the gradient of the empirical risk function $\widehat{L}(f_t)$ with respect to the weight $\boldsymbol{w}_i^{(t)}$ at iteration $t$ is given by:

$$\nabla_{\boldsymbol{w}_i}\widehat{L}(f_t) = \frac{1}{K}\sum_{n=1}^{K}\Big[(\ell'_{p,t}-1)h_i(\boldsymbol{Y}_n)\sum_{r=1}^{L}\mathbf{1}_{|\langle\boldsymbol{w}_i,\boldsymbol{z}_{\boldsymbol{X}}^{(r)}\rangle|\geq b_i}\boldsymbol{z}_{\boldsymbol{X}}^{(r)} \\ + \sum_{\boldsymbol{X}_{n,s}\in\mathfrak{N}}\ell'_{s,t}h_i(\boldsymbol{X}_{n,s})\sum_{r=1}^{L}\mathbf{1}_{|\langle\boldsymbol{w}_i,\boldsymbol{z}_{\boldsymbol{X}}^{(r)}\rangle|\geq b_i}\boldsymbol{z}_{\boldsymbol{X}}^{(r)}\Big]. \tag{21}$$

**Population Gradient.**  Similar to the empirical gradient, the gradient of the population risk function $L(f_t)$ with respect to the weight $\boldsymbol{w}_i^{(t)}$ at iteration $t$ is given by:

$$\nabla_{\boldsymbol{w}_i}L(f_t) = \mathbb{E}[(\ell'_{p,t}-1)h_i(\boldsymbol{Y}_n)\sum_{r=1}^{L}\mathbf{1}_{|\langle\boldsymbol{w}_i,\boldsymbol{z}_{\boldsymbol{X}}^{(r)}\rangle|\geq b_i}\boldsymbol{z}_{\boldsymbol{X}}^{(r)} \\ + \sum_{\boldsymbol{X}_{n,s}\in\mathfrak{N}}\ell'_{s,t}h_i(\boldsymbol{X}_{n,s})\sum_{r=1}^{L}\mathbf{1}_{|\langle\boldsymbol{w}_i,\boldsymbol{z}_{\boldsymbol{X}}^{(r)}\rangle|\geq b_i}\boldsymbol{z}_{\boldsymbol{X}}^{(r)}], \tag{22}$$

where $L$ is the population risk function as

$$L(f_t) \;=\; \mathbb{E}\big[\,\ell(\boldsymbol{f_\theta}, \boldsymbol{X}_n, \boldsymbol{Y}_n, \mathfrak{N})\,\big]. \tag{23}$$

**Stop Gradient.**  Note that the similarity measure explicitly uses the `StopGrad` operation to block gradient flow through the second input. The similarity is computed as

$$\mathrm{Sim}_{\boldsymbol{f}_t}(\boldsymbol{X_1}, \boldsymbol{X_2}) = \langle\boldsymbol{f}_t(\boldsymbol{X_1}), \texttt{StopGrad}(\boldsymbol{f}_t(\boldsymbol{X_2}))\rangle. \tag{24}$$

**Concentration Bound.**  The following lemma shows that, given a sufficiently large number of samples, the approximation error between the empirical gradient and the population gradient remains bounded with high probability. Building on this principle, we will first analyze the training dynamics under the population gradient, and subsequently account for the deviation arising from the empirical gradient. The proof of Lemma B.1 follows standard techniques based on sub-Gaussian tail bounds and is therefore omitted.

**Lemma B.1** (Approximation of empirical gradients by population gradients). *Suppose that* $\|\boldsymbol{W}^{(t)}\|_F^2 \leq \text{poly}(d)$. *Then there exists some* $K = \text{poly}(d_1)$ *such that, with high probability, the difference between the empirical gradients and the population gradients is bounded for every iteration* $t$:

$$\left\|\nabla_{\boldsymbol{w}_i}\widehat{L}_{\text{aug}}(f_t) - \nabla_{\boldsymbol{w}_i}L_{\text{aug}}(f_t)\right\|_2 \leq \frac{\|\boldsymbol{w}_i^{(t)}\|_2}{\text{poly}(d_1)}, \quad \forall i \in [m]. \tag{25}$$

This Definition B.1 divides neurons into two categories, ordinary neurons and lucky neurons, based on their initial alignment with feature vectors $\boldsymbol{M}_j$. These sets will serve as the foundation for our later analysis.

**Definition B.1** (Characterization of Neurons). *We define the following sets of neurons, which will be useful for analyzing the stochastic gradient descent trajectory in later sections:*

*(a) For each $j \in [d]$, we define the set of* ordinary neurons $\mathcal{M}_j \subseteq [m]$ *as:*

$$\mathcal{M}_j := \left\{i \in [m] : \langle \boldsymbol{w}_i^{(0)}, \boldsymbol{M}_j\rangle^2 \geq \frac{c_2 \log d}{d}\|\boldsymbol{M}\boldsymbol{M}^\top\boldsymbol{w}_i^{(0)}\|_2^2\right\}, \quad \forall j \in [d] \tag{26}$$

*(b) For each $j \in [d]$, we define the set of* lucky neurons $\mathcal{M}_j^\star \subseteq [m]$ *as:*

$$\mathcal{M}_j^\star := \left\{\begin{array}{l} i \in [m] : \langle \boldsymbol{w}_i^{(0)}, \boldsymbol{M}_j\rangle^2 \geq \frac{c_1 \log d}{d}\|\boldsymbol{M}\boldsymbol{M}^\top\boldsymbol{w}_i^{(0)}\|_2^2, \\ \quad\quad \langle \boldsymbol{w}_i^{(0)}, \boldsymbol{M}_{j'}\rangle^2 \leq \frac{c_2 \log d}{d}\|\boldsymbol{M}\boldsymbol{M}^\top\boldsymbol{w}_i^{(0)}\|_2^2, \quad \forall j' \in [d], j' \neq j \end{array}\right\}, \tag{27}$$

*where*

$$c_1 = \left(\frac{\epsilon_{\max}}{\epsilon_{\min}}\right)^2 \cdot 2(1+\gamma), \quad c_2 = \left(\frac{\epsilon_{\min}}{\epsilon_{\max}}\right)^2 \cdot 2(1-\gamma), \quad \gamma \text{ is a small constant.} \tag{28}$$

**Properties at initialization:** At initialization ($t = 0$), we note key facts about the neurons for later analysis of the SGD trajectory.

Before presenting Lemma B.2, we outline its essential idea: (a) Each $\boldsymbol{w}_i^{(0)}$ has magnitude in the order of $\sigma_0^2 d_1$; (b) Each $\boldsymbol{w}_i^{(0)}$ has a projection onto the feature subspace in the order of $\sigma_0^2 d$; (c) For each feature, the numbers of lucky and ordinary neurons are influenced by the frequencies of the majority and minority features; (d) For each neuron, the number of aligned features forms only a limited subset, typically of size smaller than $d$. We defer the proof of Lemma B.2 to Appendix G for the clarification of presentation.

**Lemma B.2.** *At initialization ($t = 0$), the following properties hold:*

*(a) With high probability, for every $i \in [m]$,*

$$\|\boldsymbol{w}_i^{(0)}\|_2^2 \in \left[\sigma_0^2 d_1\left(1 - \widetilde{O}\left(\tfrac{1}{\sqrt{d_1}}\right)\right), \ \sigma_0^2 d_1\left(1 + \widetilde{O}\left(\tfrac{1}{\sqrt{d_1}}\right)\right)\right]. \tag{29}$$

*(b) With high probability, for every $i \in [m]$,*

$$\|\boldsymbol{M}\boldsymbol{M}^\top\boldsymbol{w}_i^{(0)}\|_2^2 \in \left[\sigma_0^2 d\left(1 - \widetilde{O}\left(\tfrac{1}{\sqrt{d}}\right)\right), \ \sigma_0^2 d\left(1 + \widetilde{O}\left(\tfrac{1}{\sqrt{d}}\right)\right)\right]. \tag{30}$$

*(c) Let $m = d^{C_m}$ be the number of neurons. With probability at least $1 - o\left(\tfrac{1}{d^4}\right)$, for each $j \in [d]$,*

$$|\mathcal{M}_j^\star| \geq \Omega(d^{\omega_1}) =: \Xi_1, \qquad |\mathcal{M}_j| \leq O(d^{\omega_2}) =: \Xi_2. \tag{31}$$

*where*

$$\omega_1 = C_m - \left(\frac{\epsilon_{\max}}{\epsilon_{\min}}\right)^2 (1+\gamma), \qquad \omega_2 = C_m - \left(\frac{\epsilon_{\min}}{\epsilon_{\max}}\right)^2 (1-\gamma). \tag{32}$$

*(d) For each $i \in [m]$, there are at most $O\left(d^{1-\left(\frac{\epsilon_{\min}}{\epsilon_{\max}}\right)^2 \cdot (1-\gamma)}\right)$ indices $j \in [d]$ such that $i \in \mathcal{M}_j$.*

## C    THEOREM C.1

In this section we analyze the training process at the initial stage. Here we define the stage transition time

$$T_1 = \Theta\left(\frac{d_1 \log d}{\eta \log \log d}\right) \tag{33}$$

to be the iteration when

$$\|MM^\top w_i^{(t)}\|_2^2 \geq \tfrac{1}{2}\|w_i^{(t)}\|_2^2, \tag{34}$$

where the neuron weights are more concentrated in the feature space.

### C.1    THEOREM C.1

Before stating Theorem C.1, we give a short description of its parts: (a) For all neurons, most of the weights lie in the feature subspace; (b) Lucky neurons are strongly aligned with their associated feature directions; (c) Neurons not in the set $\mathcal{M}_j$ have only weak alignment with feature $j$; (d) Each neuron can have strong alignment with only a limited number of features; and (e) All neuron weights have only small components in non-feature directions.

**Theorem C.1** (Initial feature decoupling). *At iteration $t = T_1$, we have the following results:*

*(a) For all $i \in [m]$,*

$$\|MM^\top w_i^{(T_1)}\|_2^2 \geq \tfrac{1}{2}\|w_i^{(T_1)}\|_2^2. \tag{35}$$

*(b) For each $j \in [d]$, and each $i \in \mathcal{M}_j^\star$,*

$$|\langle w_i^{(T_1)}, M_j\rangle| \geq \sqrt{1+\gamma}\,\frac{\sqrt{2\log d}}{\sqrt{d}}\|w_i^{(T_1)}\|_2. \tag{36}$$

*(c) For each $j \in [d]$, and each $i \notin \mathcal{M}_j$,*

$$|\langle w_i^{(T_1)}, M_j\rangle| \leq \sqrt{1-\gamma}\,\frac{\sqrt{2\log d}}{\sqrt{d}}\|w_i^{(T_1)}\|_2. \tag{37}$$

*(d) For each $i \in [m]$,*

$$|\langle w_i^{(T_1)}, M_j\rangle| \geq \frac{\log^{1/4} d}{\sqrt{d}}\|w_i^{(T_1)}\|_2, \quad \textit{for at most } \mathcal{O}\left(\frac{d}{2^{\sqrt{\log d}}}\right) \textit{ indices } j \in [d]. \tag{38}$$

*(e) For each $i \in [m]$ and $j \in [d_1] \setminus [d]$,*

$$|\langle w_i^{(T_1)}, M_j^\perp\rangle| \leq \mathcal{O}\left(\sqrt{\tfrac{\log d}{d_1}}\right)\|w_i^{(T_1)}\|_2. \tag{39}$$

### C.2    USEFUL LEMMAS

In Lemma C.1, we show that for each neuron $i \in [m]$, the weight vector $w_i$ largely disregards the non-feature components $M^\perp$ and instead focuses on the relevant features $M$.

We first describe Lemma C.1: (a) The projection of $w_i^{(t)}$ onto the feature subspace, though initially small, grows rapidly during training and reaches the order of $d_1$ relative to its initialization. (b) The component of $w_i^{(t)}$ in the non-feature subspace remains essentially unchanged, up to negligible variation.

**Lemma C.1.** *For all $t \leq T_1$, the following properties hold:*

*(a)*

$$\left\|MM^\top w_i^{(t)}\right\|_2^2 \leq \left\|MM^\top w_i^{(0)}\right\|_2^2\left(1 + \epsilon_{\max}\frac{\eta C_z \log \log d}{d}\right)^{2t} + O\left(\tfrac{1}{d}\right)\left\|MM^\top w_i^{(0)}\right\|_2^2,$$
$$\textit{moreover, } \left\|MM^\top w_i^{(t)}\right\|_2^2 \leq O\left(\left\|w_i^{(0)}\right\|_2^2\right). \tag{40}$$

*(b)*
$$\left\|\boldsymbol{M}\boldsymbol{M}^\top \boldsymbol{w}_i^{(t)}\right\|_2^2 \geq \left\|\boldsymbol{M}\boldsymbol{M}^\top \boldsymbol{w}_i^{(0)}\right\|_2^2 \left(1 - \eta\lambda + \epsilon_{\min}\tfrac{\eta C_z \log\log d}{d}\right)^{2t} - O\left(\tfrac{1}{d}\right)\left\|\boldsymbol{M}\boldsymbol{M}^\top \boldsymbol{w}_i^{(0)}\right\|_2^2. \tag{41}$$

*(c)*
$$\left\|\boldsymbol{M}^\perp(\boldsymbol{M}^\perp)^\top \boldsymbol{w}_i^{(t)}\right\|_2^2 \leq \left(1 + O\left(\tfrac{1}{\mathrm{poly}(d)}\right)\right)\left\|\boldsymbol{M}^\perp(\boldsymbol{M}^\perp)^\top \boldsymbol{w}_i^{(0)}\right\|_2^2. \tag{42}$$

**Lemma C.2.** *For each $i \in [m]$, there are at most $O(2^{-\sqrt{\log d}}d)$ indices $j \in [d]$ such that*
$$|\langle \boldsymbol{w}_i^{(0)}, \boldsymbol{M}_j\rangle| \geq \Omega(\sigma_0 \log^{1/4} d). \tag{43}$$

## C.3 Proof of Theorem C.1

*Proof of Theorem C.1(a):* The result (a) can be derived from Lemma C.1 (c). We have,
$$\begin{aligned}
\|\boldsymbol{M}\boldsymbol{M}^\top \boldsymbol{w}_i^{(T_1)}\|_2^2 &= \|\boldsymbol{w}_i^{(T_1)}\|_2^2 - \|\boldsymbol{M}^\perp(\boldsymbol{M}^\perp)^\top \boldsymbol{w}_i^{(T_1)}\|_2^2 \\
&\geq \|\boldsymbol{w}_i^{(T_1)}\|_2^2 - \left(1 + \frac{1}{\mathrm{poly}(d)}\right)\|\boldsymbol{M}^\perp(\boldsymbol{M}^\perp)^\top \boldsymbol{w}_i^{(0)}\|_2^2 \\
&\geq \|\boldsymbol{w}_i^{(T_1)}\|_2^2 - \|\boldsymbol{w}_i^{(0)}\|_2^2 \\
&\geq \|\boldsymbol{w}_i^{(T_1)}\|_2^2 - \frac{\|\boldsymbol{w}_i^{(T_1)}\|_2^2}{(1 + \epsilon_{\min}C_z \log d)} \\
&\geq \frac{1}{2}\|\boldsymbol{w}_i^{(T_1)}\|_2^2.
\end{aligned} \tag{44}$$
$\square$

*Proof of Theorem C.1(b):* Note that from similar gradient calculations to those in the proof of Lemma C.1 (b), we have, for $j \in [d]$ and $i \in \mathcal{M}_j^\star$:
$$\begin{aligned}
&|\langle \boldsymbol{w}_i^{(T_1)}, \boldsymbol{M}_j\rangle| \\
&= |\langle \boldsymbol{w}_i^{(T_1-1)}, \boldsymbol{M}_j\rangle - \eta\langle\nabla_{\boldsymbol{w}_i} L_{\mathrm{aug}}(f_{T_1-1}), \boldsymbol{M}_j\rangle \ \pm \frac{\|\boldsymbol{w}_i^{((T_1-1))}\|_2}{\mathrm{poly}(d_1)}| \\
&\geq |\langle \boldsymbol{w}_i^{(T_1-1)}, \boldsymbol{M}_j\rangle|\left(1 - \eta\lambda + \epsilon_j\frac{\eta C_z \log\log d}{d}\right) - \widetilde{O}\left(\frac{\eta\|\boldsymbol{w}_i^{((T_1-1))}\|_2}{\mathrm{poly}(d_1)}\right) \\
&\geq |\langle \boldsymbol{w}_i^{(0)}, \boldsymbol{M}_j\rangle|\left(1 - \eta\lambda + \epsilon_j\frac{\eta C_z \log\log d}{d}\right)^{T_1} - \widetilde{O}\left(\frac{\eta T_1\|\boldsymbol{w}_i^{(T_1)}\|_2}{\mathrm{poly}(d_1)}\right).
\end{aligned} \tag{45}$$

These gradient descent steps above can be derived from the last few inequalities in the proof of Lemma C.1(b).
$$\begin{aligned}
&|\langle \boldsymbol{w}_i^{(T_1)}, \boldsymbol{M}_j\rangle| \\
&\overset{①}{\geq} \frac{\sqrt{c_1 \log d}}{\sqrt{d}}\|\boldsymbol{M}\boldsymbol{M}^\top \boldsymbol{w}_i^{(0)}\|_2\left(1 - \eta\lambda + \epsilon_j\frac{\eta C_z \log\log d}{d}\right)^{T_1} - \widetilde{O}\left(\frac{\eta T_1\|\boldsymbol{w}_i^{(T_1)}\|_2}{\mathrm{poly}(d_1)}\right) \\
&\overset{②}{\geq} \frac{\sqrt{c_1 \log d}}{\sqrt{d}}\|\boldsymbol{M}\boldsymbol{M}^\top \boldsymbol{w}_i^{(0)}\|_2\left(1 - \eta\lambda + \epsilon_j\frac{\eta C_z \log\log d}{d}\right)^{T_1} - \widetilde{O}\left(\frac{\|\boldsymbol{w}_i^{(T_1)}\|_2}{\mathrm{poly}(d)}\right) \\
&\overset{③}{\geq} \frac{\sqrt{c_1 \log d}}{\sqrt{d}}\|\boldsymbol{M}\boldsymbol{M}^\top \boldsymbol{w}_i^{(0)}\|_2\left(1 - \eta\lambda + \epsilon_j\frac{\eta C_z \log\log d}{d}\right)^{T_1} - \widetilde{O}\left(\frac{\left\|\boldsymbol{w}_i^{(0)}\right\|_2}{\mathrm{poly}(d)}\right) \\
&\overset{④}{\geq} \frac{\sqrt{c_1 \log d}}{\sqrt{d}}\|\boldsymbol{M}\boldsymbol{M}^\top \boldsymbol{w}_i^{(0)}\|_2\left(1 - \eta\lambda + \epsilon_j\frac{\eta C_z \log\log d}{d}\right)^{T_1} - \widetilde{O}\left(\frac{\sqrt{\frac{d_1}{d}}\|\boldsymbol{M}\boldsymbol{M}^\top \boldsymbol{w}_i^{(0)}\|_2}{\mathrm{poly}(d)}\right) \\
&\geq \frac{\sqrt{c_1 \log d}}{\sqrt{d}}\|\boldsymbol{M}\boldsymbol{M}^\top \boldsymbol{w}_i^{(0)}\|_2\left(1 - \eta\lambda + \epsilon_j\frac{\eta C_z \log\log d}{d}\right)^{T_1} - \frac{\|\boldsymbol{M}\boldsymbol{M}^\top \boldsymbol{w}_i^{(0)}\|_2}{\mathrm{poly}(d)} \\
&\overset{⑤}{\geq} \sqrt{1+\gamma}\frac{\sqrt{\log d}}{\sqrt{d}}\|\boldsymbol{w}_i^{(T_1)}\|_2.
\end{aligned} \tag{46}$$

① is because Definition B.1 (b). ② is because $\frac{\eta T_1}{\text{poly}(d_1)} \leq \frac{1}{\text{poly}(d)}$. ③ is because $\left\|\boldsymbol{w}_i^{(t)}\right\|_2^2 \leq$ $O(1)\left\|\boldsymbol{w}_i^{(0)}\right\|_2^2$ (equation 258). ④ is because Lemma B.2 (a) (b). ⑤ holds because the following equation is valid:

$$
\begin{aligned}
&|\langle \boldsymbol{w}_i^{(T_1)}, \boldsymbol{M}_j \rangle| \\
&\geq \frac{\sqrt{c_1 \log d}}{\sqrt{d}} \|\boldsymbol{M}\boldsymbol{M}^\top \boldsymbol{w}_i^{(0)}\|_2 \left(1 - \eta\lambda + \epsilon_j \frac{\eta C_z \log\log d}{d}\right)^{T_1} - \frac{\|\boldsymbol{M}\boldsymbol{M}^\top \boldsymbol{w}_i^{(0)}\|_2}{\text{poly}(d)} \\
&= \frac{\sqrt{c_1 \log d}}{\sqrt{d}} \|\boldsymbol{M}\boldsymbol{M}^\top \boldsymbol{w}_i^{(0)}\|_2 \left(1 - \eta\lambda + \epsilon_j \frac{d_1}{d} C_z \log d\right) - \frac{\|\boldsymbol{M}\boldsymbol{M}^\top \boldsymbol{w}_i^{(0)}\|_2}{\text{poly}(d)} \\
&\geq (\frac{\epsilon_j}{\epsilon_{\max}}) \frac{\sqrt{c_1 \log d}}{\sqrt{d}} \|\boldsymbol{M}\boldsymbol{M}^\top \boldsymbol{w}_i^{(0)}\|_2 \left(1 - \eta\lambda + \epsilon_{\max} \frac{d_1}{d} C_z \log d\right) - \frac{\|\boldsymbol{M}\boldsymbol{M}^\top \boldsymbol{w}_i^{(0)}\|_2}{\text{poly}(d)} \\
&\geq (\frac{\epsilon_j}{\epsilon_{\max}}) \frac{\sqrt{c_1 \log d}}{\sqrt{d}} \|\boldsymbol{M}\boldsymbol{M}^\top \boldsymbol{w}_i^{(T_1)}\|_2 \\
&\overset{⑥}{\geq} \frac{1}{\sqrt{2}} (\frac{\epsilon_j}{\epsilon_{\max}}) \frac{\sqrt{c_1 \log d}}{\sqrt{d}} \|\boldsymbol{w}_i^{(T_1)}\|_2 \\
&\geq \frac{1}{\sqrt{2}} (\frac{\epsilon_{\min}}{\epsilon_{\max}}) \frac{\sqrt{c_1 \log d}}{\sqrt{d}} \|\boldsymbol{w}_i^{(T_1)}\|_2 \\
&\geq \sqrt{1+\gamma} \frac{\sqrt{\log d}}{\sqrt{d}} \|\boldsymbol{w}_i^{(T_1)}\|_2.
\end{aligned}
\tag{47}
$$

⑥ holds because of the conclusion of Theorem C.1(a).

$\square$

*Proof of Theorem C.1(c):* Theorem C.1(c) can be verified using Definition B.1 (b), Lemma B.2 (a) (b) together with the proof of Lemma C.1(a).

$$
\begin{aligned}
&|\langle \boldsymbol{w}_i^{(T_1)}, \boldsymbol{M}_j \rangle| \\
&\leq |\langle \boldsymbol{w}_i^{(0)}, \boldsymbol{M}_j \rangle| \left(1 + \epsilon_j \frac{\eta C_z \log\log d}{d} + \widetilde{O}\left(\frac{\eta}{d^2}\right)\right)^{T_1} + \widetilde{O}\left(\frac{\eta T_1 \|\boldsymbol{w}_i^{(T_1)}\|_2}{\text{poly}(d_1)}\right).
\end{aligned}
\tag{48}
$$

The above equation can be obtained from the first inequality in the proof of Lemma C.1(a).

$$
\begin{aligned}
&|\langle \boldsymbol{w}_i^{(T_1)}, \boldsymbol{M}_j \rangle| \\
&\leq |\langle \boldsymbol{w}_i^{(0)}, \boldsymbol{M}_j \rangle| \left(1 + \epsilon_j \frac{\eta C_z \log\log d}{d} + \widetilde{O}\left(\frac{\eta}{d^2}\right)\right)^{T_1} + \frac{\|\boldsymbol{M}(\boldsymbol{M})^\top \boldsymbol{w}_i^{(0)}\|_2}{\text{poly}(d)} \\
&\leq \sqrt{\frac{c_2 \log d}{d}} \left\|\boldsymbol{M}\boldsymbol{M}^\top \boldsymbol{w}_i^{(0)}\right\|_2 \left(1 + \epsilon_j \frac{\eta C_z \log\log d}{d} + \widetilde{O}\left(\frac{\eta}{d^2}\right)\right)^{T_1} + \frac{\|\boldsymbol{M}(\boldsymbol{M})^\top \boldsymbol{w}_i^{(0)}\|_2^2}{\text{poly}(d)} \\
&\leq \frac{\epsilon_j}{\epsilon_{\min}} \sqrt{\frac{c_2 \log d}{d}} \left\|\boldsymbol{M}\boldsymbol{M}^\top \boldsymbol{w}_i^{(T_1)}\right\|_2 + O\left(\frac{\|\boldsymbol{M}(\boldsymbol{M})^\top \boldsymbol{w}_i^{(0)}\|_2^2}{\text{poly}(d)}\right) \\
&\leq \frac{\epsilon_j}{\epsilon_{\min}} \sqrt{\frac{c_2 \log d}{d}} \|\boldsymbol{w}_i^{(T_1)}\|_2 + O\left(\frac{\|\boldsymbol{w}_i^{(T_1)}\|_2}{\text{poly}(d)}\right) \\
&\leq \frac{\epsilon_{\max}}{\epsilon_{\min}} \sqrt{\frac{c_2 \log d}{d}} \|\boldsymbol{w}_i^{(T_1)}\|_2 + O\left(\frac{\|\boldsymbol{w}_i^{(T_1)}\|_2}{\text{poly}(d)}\right) \\
&\leq \sqrt{1-\gamma} \frac{\sqrt{2 \log d}}{\sqrt{d}} \|\boldsymbol{w}_i^{(T_1)}\|_2.
\end{aligned}
\tag{49}
$$

$\square$

*Proof of Theorem C.1(d):* First, by Lemma C.2 we obtain that for each $i \in [m]$, there are at most $O(2^{-\sqrt{\log d}} d)$ indices $j \in [d]$ such that:

$$|\langle \boldsymbol{w}_i^{(0)}, \boldsymbol{M}_j \rangle| \geq \Omega(\sigma_0 \log^{1/4} d). \tag{50}$$

Next, we proceed to the formal calculation:

$$
\begin{aligned}
|\langle \boldsymbol{w}_i^{(T_1)}, \boldsymbol{M}_j \rangle| &\geq |\langle \boldsymbol{w}_i^{(0)}, \boldsymbol{M}_j \rangle| \left( 1 - \eta\lambda + \epsilon_j \frac{\eta C_z \log\log d}{d} \right)^{T_1} \\
&\geq \Omega(\sigma_0 \log^{1/4} d) \left( 1 - \eta\lambda + \epsilon_j \frac{\eta C_z \log\log d}{d} \right)^{T_1} \\
&\geq \Omega\left( \frac{\|\boldsymbol{w}_i^{(0)}\|_2}{\sqrt{d_1}} \log^{1/4} d \right) \Theta\left( \frac{d_1}{d} \right) \\
&\geq \frac{\log^{1/4} d}{\sqrt{d}} \|\boldsymbol{w}_i^{(T_1)}\|_2.
\end{aligned}
\tag{51}
$$

$\square$

*Proof of Theorem C.1(e).* At initialization we have

$$\boldsymbol{w}_i^{(0)} \sim \mathcal{N}(0, \sigma_0^2 I_{d_1}). \tag{52}$$

Hence for any unit vector $\boldsymbol{M}_j^\perp$, the projection satisfies

$$\langle \boldsymbol{w}_i^{(0)}, \boldsymbol{M}_j^\perp \rangle \sim \mathcal{N}(0, \sigma_0^2). \tag{53}$$

By the standard Gaussian tail bound (sub-Gaussian with parameter $\sigma_0$),

$$\mathbb{P}\left( |\langle \boldsymbol{w}_i^{(0)}, \boldsymbol{M}_j^\perp \rangle| > \sigma_0 \sqrt{2\log d} \right) \leq 2\exp\left( -\frac{t^2}{2\sigma_0^2} \right) = \frac{2}{d}. \tag{54}$$

Therefore, with high probability,

$$\left| \langle \boldsymbol{w}_i^{(0)}, \boldsymbol{M}_j^\perp \rangle \right| \leq \sigma_0 \cdot O(\sqrt{\log d}). \tag{55}$$

Moreover, since $\|\boldsymbol{w}_i^{(0)}\|_2 = \Theta(\sigma_0 \sqrt{d_1})$ with high probability, the above bound is equivalently

$$\left| \langle \boldsymbol{w}_i^{(0)}, \boldsymbol{M}_j^\perp \rangle \right| \leq O\left( \sqrt{\frac{\log d}{d_1}} \right) \cdot \|\boldsymbol{w}_i^{(0)}\|_2. \tag{56}$$

We have

$$
\begin{aligned}
&\langle \boldsymbol{w}_i^{(T_1)}, \boldsymbol{M}_j^\perp \rangle \\
&= (1 - \eta\lambda)\langle \boldsymbol{w}_i^{(T_1-1)}, \boldsymbol{M}_j^\perp \rangle \pm \widetilde{O}\left( \frac{\eta \sum_{i \in [m]} \left\| \boldsymbol{w}_i^{(t)} \right\|_2^2 \cdot \left\| \boldsymbol{w}_i^{(t)} \right\|_2}{\tau d} \right) \\
&\leq (1 - \eta\lambda)\langle \boldsymbol{w}_i^{(T_1-1)}, \boldsymbol{M}_j^\perp \rangle + \widetilde{O}\left( \frac{\eta\|\boldsymbol{w}_i^{(t)}\|_2}{\text{poly}(d_1)} \right) \\
&\leq |\langle \boldsymbol{w}_i^{(0)}, \boldsymbol{M}_j^\perp \rangle| + O(T_1\eta) \cdot \max_{t \leq T_1} \widetilde{O}\left( \frac{\|\boldsymbol{w}_i^{(t)}\|_2}{\text{poly}(d_1)} \right) \\
&\leq O\left( \sqrt{\frac{\log d}{d_1}} \right) \cdot \|\boldsymbol{w}_i^{(0)}\|_2 + O(T_1\eta) \cdot \max_{t \leq T_1} \widetilde{O}\left( \frac{\|\boldsymbol{w}_i^{(t)}\|_2}{\text{poly}(d_1)} \right) \\
&\leq O\left( \sqrt{\frac{\log d}{d_1}} \right) \cdot \|\boldsymbol{w}_i^{(T_1)}\|_2 + O(T_1\eta) \cdot \max_{t \leq T_1} \widetilde{O}\left( \frac{\|\boldsymbol{w}_i^{(t)}\|_2}{\text{poly}(d_1)} \right) \\
&\overset{\textcircled{7}}{\leq} O\left( \sqrt{\frac{\log d}{d_1}} \right) \|\boldsymbol{w}_i^{(T_1)}\|_2.
\end{aligned}
\tag{57}
$$

⑦ is because $\frac{T_1\eta}{\text{poly}(d_1)} \ll \sqrt{\frac{\log d}{d_1}}$ $\qquad\qquad\qquad\qquad\qquad\qquad\qquad\qquad\qquad\qquad\qquad$ $\square$

Lemma 3.1 can be viewed as an informal version of Theorem C.1. In particular, part (a) of Lemma 3.1 corresponds to the first inequality in the proof of Theorem C.1(b), while part (b) of Lemma 3.1 corresponds to the first inequality in the proof of Theorem C.1(e). Hence, Lemma 3.1 is essentially a simplified restatement of the more general Theorem C.1.

## C.4 PROOF OF LEMMA 3.1

*Proof of Lemma 3.1.* For $j \in [d]$ and $i \in [m]$, the following bounds hold for all $t \in [0, T_1]$:

(a) Lower bound:

$$|\langle \boldsymbol{w}_i^{(t+1)}, \boldsymbol{M}_j\rangle| \geq |\langle \boldsymbol{w}_i^{(t)}, \boldsymbol{M}_j\rangle|(1 - \eta\lambda + \epsilon_j \frac{\eta C_z \log\log d}{d}) - \widetilde{O}\Big(\frac{\eta\|\boldsymbol{w}_i^{(t)}\|_2}{\text{poly}(d_1)}\Big). \tag{58}$$

(b) Orthogonal component:

$$|\langle \boldsymbol{w}_i^{(t+1)}, \boldsymbol{M}_j^{\perp}\rangle| \leq (1 - \eta\lambda)|\langle \boldsymbol{w}_i^{(t)}, \boldsymbol{M}_j^{\perp}\rangle| + \widetilde{O}\Big(\frac{\eta\|\boldsymbol{w}_i^{(t)}\|_2}{\text{poly}(d_1)}\Big). \tag{59}$$

$\qquad\qquad\qquad\qquad\qquad\qquad\qquad\qquad\qquad\qquad\qquad\qquad\qquad\qquad\qquad\qquad$ $\square$

# D THEOREM D.1

The second stage is defined as the iterations $t \geq T_1$ but $t \leq T_2$, where

$$T_2 = T_1 + \Theta\left(\frac{d\tau \log d}{\epsilon_{\max}\eta \log\log d}\right) \tag{60}$$

is defined as the iteration when one of the neuron $i \in [m]$ satisfies

$$\left\|\boldsymbol{w}_i^{(T_2)}\right\|_2^2 \geq d\left\|\boldsymbol{w}_i^{(T_1)}\right\|_2^2. \tag{61}$$

## D.1 THEOREM D.1

We first provide an explanation of Theorem D.1: (a) If a neuron $i$ is a lucky neuron for feature $j$, then the projection of $\boldsymbol{w}_i^{(T_2)}$ onto $\boldsymbol{M}_j$ is very large, on the order of the full neuron weight $\|\boldsymbol{w}_i^{(T_2)}\|_2$. In other words, such neurons have already "focused" on $\boldsymbol{M}_j$. (b) The bias term $\boldsymbol{b}_i^{(T_2)}$ grows proportionally with the neuron weight $\|\boldsymbol{w}_i^{(T_2)}\|_2$, and at iteration $T_2$ it reaches at least $\frac{\text{polylog}(d)}{\sqrt{d}}\|\boldsymbol{w}_i^{(T_2)}\|_2$. In other words, the continuously increasing bias effectively controls the activation of the neuron $\boldsymbol{w}_i^{(T_2)}$. (c) Among the lucky neurons in $\mathcal{M}_j^{\star}$, there exists one neuron $\boldsymbol{w}_i^{(T_2)}$ whose projection onto $\boldsymbol{M}_j$ is the largest, and this neuron has a larger projection than all the other neurons in $\mathcal{M}_j$.

**Theorem D.1** (Emergence of singletons). *For each neuron $i \in [m]$, the following conditions hold at iteration $t = T_2$:*

*(a) For each $j \in [d]$, if $i \in \mathcal{M}_j^{\star}$, then*

$$\left|\langle \boldsymbol{w}_i^{(T_2)}, \boldsymbol{M}_j\rangle\right| \geq \Omega(\frac{\varepsilon_{\min}}{\varepsilon_{\max}})\|\boldsymbol{w}_i^{(T_2)}\|_2. \tag{62}$$

*(b)*

$$\boldsymbol{b}_i^{(T_2)} \geq \frac{\text{polylog}(d)}{\sqrt{d}}\|\boldsymbol{w}_i^{(T_2)}\|_2. \tag{63}$$

*(c) Let*

$$\alpha_j^{\star} = \max_{i \in \mathcal{M}_j^{\star}}\left|\langle \boldsymbol{w}_i^{(T_2)}, \boldsymbol{M}_j\rangle\right|, \tag{64}$$

*then there exists a constant $C_j = \Theta(1)$ such that*

$$\left|\langle \boldsymbol{w}_i^{(t)}, \boldsymbol{M}_j\rangle\right| \leq C_j\alpha_j^{\star}, \quad \forall i \in \mathcal{M}_j. \tag{65}$$

## D.2 Useful Lemmas

Next, we discuss Lemma D.1. For example, the first item illustrates how each feature $M_j$ can be captured by certain subsets of neurons, a process influenced by the stochastic nature of initialization. We elaborate on the full content of Lemma D.1 below.

(a) Lucky neurons have large projection on their feature direction, which means they can be activated and are clearly aligned with that feature. (b) If a neuron does not belong to $\mathcal{M}_j$, then its projection on feature $j$ stays small, which means it cannot be activated and has only weak alignment. (c) A neuron can only be well aligned with a small number of features, not with many at the same time. (d) The projection of a neuron weight on non-feature directions is very small, which means the neuron does not learn the non-feature directions. (e) The size of each neuron weight is controlled by its bias, so the weight does not grow without limit.

**Lemma D.1.** *For all iterations $t \in (T_1, T_2]$, the neurons $i \in [m]$ satisfy the following properties:*

*(a) For $j \in [d]$, if $i \in \mathcal{M}_j^\star$, then*

$$\left|\langle \boldsymbol{w}_i^{(t)}, \boldsymbol{M}_j\rangle\right| \geq \sqrt{1+\gamma}\, \boldsymbol{b}_i^{(t)}. \tag{66}$$

*(b) For $j \in [d]$, if $i \notin \mathcal{M}_j$, then*

$$\left|\langle \boldsymbol{w}_i^{(t)}, \boldsymbol{M}_j\rangle\right| \leq \sqrt{1-\gamma}\, \boldsymbol{b}_i^{(t)}, \tag{67}$$

*and furthermore,*

$$\left|\langle \boldsymbol{w}_i^{(t)}, \boldsymbol{M}_j\rangle\right| \leq \widetilde{\mathcal{O}}\left(\frac{\|\boldsymbol{w}_i^{(t)}\|_2}{\sqrt{d}}\right). \tag{68}$$

*(c) For each $i \in [m]$, there are at most $\mathcal{O}(2^{-\sqrt{\log d}}d)$ many $j \in [d]$ such that*

$$\langle \boldsymbol{w}_i^{(t)}, \boldsymbol{M}_j\rangle^2 \geq \frac{(\boldsymbol{b}_i^{(t)})^2}{\sqrt{\log d}}. \tag{69}$$

*(d) For each $i \in [m]$, and for all $j \in [d_1] \setminus [d]$,*

$$\left|\langle \boldsymbol{w}_i^{(t)}, \boldsymbol{M}_j^\perp\rangle\right| \leq \widetilde{\mathcal{O}}\left(\frac{\|\boldsymbol{w}_i^{(t)}\|_2}{\sqrt{d_1}}\right). \tag{70}$$

*(e) For all $i \in [m]$,*

$$\|\boldsymbol{w}_i^{(t)}\|_2^2 \leq \frac{d(\boldsymbol{b}_i^{(t)})^2}{\log d}. \tag{71}$$

**Lemma D.2.** *For each $i \in [m]$, define*

$$\Lambda_i := \left\{j \in [d] : |\langle \boldsymbol{w}_i^{(0)}, \boldsymbol{M}_j\rangle| \leq \frac{\sigma_0}{d}\right\} \subseteq [d]. \tag{72}$$

*Then*

$$|\Lambda_i| = O\left(\frac{d}{\text{polylog}(d)}\right). \tag{73}$$

## D.3 Proof of Theorem D.1

*Proof of Theorem D.1:* We follow similar analysis as in the proof of Lemma D.1. In order to prove (a)-(c), we have to discuss the two substages of the learning process below.

When all $\|\boldsymbol{w}_i^{(t)}\|_2 \leq (1 + \frac{\varepsilon_{\min}}{\varepsilon_{\max}})\|\boldsymbol{w}_i^{(T_1)}\|_2$: From similar analysis in the proof of Lemma D.1, the iteration complexity for a neuron $i \in [m]$ to reach $\|\boldsymbol{w}_i^{(t)}\|_2 \geq (1 + \frac{\varepsilon_{\min}}{\varepsilon_{\max}})\|\boldsymbol{w}_i^{(T_1)}\|_2$ is no smaller than

$$T_{i,1}' := \max\left\{T_1 + \Omega\left(\frac{d\log d}{\epsilon_{\max}\eta \log\log d}\right), T_2\right\}. \tag{74}$$

When some $\|\boldsymbol{w}_i^{(t)}\|_2 \geq (1 + \frac{\varepsilon_j}{\varepsilon_{\max}})\|\boldsymbol{w}_i^{(T_1)}\|_2$.

We first prove Theorem D.1(a). In the first stage, for $j \notin \mathcal{N}_i$, we have

$$
\sum_{j \in [d], j \notin \mathcal{N}_i} \langle \boldsymbol{w}_i^{(T'_{i,1})}, \boldsymbol{M}_j \rangle^2 \leq \sum_{j \in [d], j \notin \mathcal{N}_i} \langle \boldsymbol{w}_i^{(T_1)}, \boldsymbol{M}_j \rangle^2 \left( 1 + \epsilon_j \frac{O(\eta)}{d \operatorname{polylog}(d)} \right)^{T_2} + \widetilde{O} \left( \frac{\|\boldsymbol{w}_i^{(T_1)}\|_2^2}{d^{3/2}} \right)
$$

$$
\leq (1 + o(1)(\frac{\epsilon_j}{\epsilon_{\max}}) + o(1)(\frac{\epsilon_j}{\epsilon_{\max}})^2) \left\| \boldsymbol{M}\boldsymbol{M}^\top \boldsymbol{w}_i^{(T_1)} \right\|_2^2,
$$

(75)

where we used the fact that $\|\boldsymbol{w}_i^{(T_1)}\|_2 \lesssim \|\boldsymbol{M}\boldsymbol{M}^\top \boldsymbol{w}_i^{(T_1)}\|_2$.

For $j \in [d_1] \setminus [d]$ we have

$$
\sum_{j \in [d_1] \setminus [d]} \langle \boldsymbol{w}_i^{(T'_{i,1})}, \boldsymbol{M}_j^\perp \rangle^2
$$

$$
\leq \sum_{j \in [d_1] \setminus [d]} \langle \boldsymbol{w}_i^{(T_1)}, \boldsymbol{M}_j^\perp \rangle^2 + O\left( \frac{\eta(T'_{i,1} - T_1)}{d} \right) e^{-\Omega(\log^{1/4} d)} \max_{t' \in [T_1, T'_{i,1}]} \|\boldsymbol{w}_i^{(t')}\|_2^2
$$

(76)

$$
\leq \left( 1 + o\big(\frac{\epsilon_j}{\epsilon_{\max}}\big) \right) \left\| \boldsymbol{M}^\perp (\boldsymbol{M}^\perp)^\top \boldsymbol{w}_i^{(T_1)} \right\|_2^2.
$$

Typically, if $i \in \mathcal{M}_j^\star$, there exists $t \leq T_2$ such that $\|\boldsymbol{w}_i^{(t)}\|_2 \geq (1 + \frac{\varepsilon_j}{\varepsilon_{\max}})\|\boldsymbol{w}_i^{(T_2)}\|_2$, as we have argued in the proof of Lemma D.1. Thus, we have

$$
|\langle \boldsymbol{w}_i^{(T'_{i,1})}, \boldsymbol{M}_j \rangle|^2 \geq \|\boldsymbol{w}_i^{(T'_{i,1})}\|_2^2 - \sum_{j \in [d], j \notin \mathcal{N}_i} \langle \boldsymbol{w}_i^{(T'_{i,1})}, \boldsymbol{M}_j \rangle^2 - \sum_{j \in [d_1] \setminus [d]} \langle \boldsymbol{w}_i^{(T'_{i,1})}, \boldsymbol{M}_j^\perp \rangle^2
$$

$$
\geq (1 + \frac{\varepsilon_j}{\varepsilon_{\max}})^2 \|\boldsymbol{w}_i^{(T_1)}\|_2^2 - (1 + o(1)(\frac{\epsilon_j}{\epsilon_{\max}}) + o(1)(\frac{\epsilon_j}{\epsilon_{\max}})^2)\|\boldsymbol{w}_i^{(T_1)}\|_2^2
$$

(77)

$$
\geq \frac{\varepsilon_j}{\varepsilon_{\max}} \cdot (2 - o(1))\|\boldsymbol{w}_i^{(T_1)}\|_2^2,
$$

which proves the claim.

In the second stage, if $i \in \mathcal{M}_j^\star$, then from similar calculations as above, we can prove by induction that starting from $t = T'_{i,1}$, it holds:

$$
|\langle \boldsymbol{w}_i^{(t+1)}, \boldsymbol{M}_j \rangle| \geq |\langle \boldsymbol{w}_i^{(t)}, \boldsymbol{M}_j \rangle| \left( 1 + \Omega \left( \epsilon_j \frac{\eta \log\log d}{d} \right) \right)
$$

$$
\geq \|\boldsymbol{w}_i^{(t)}\|_2 \left( 1 + \Omega \left( \epsilon_j \frac{\eta \log\log d}{d} \right) \right)
$$

$$
\sum_{j' \in [d], j' \neq j} \langle \boldsymbol{w}_i^{(t+1)}, \boldsymbol{M}_{j'} \rangle^2 \leq \sum_{j' \in [d], j' \neq j} \langle \boldsymbol{w}_i^{(t)}, \boldsymbol{M}_{j'} \rangle^2 \left( 1 + \epsilon_j \frac{O(\eta)}{d \operatorname{polylog}(d)} \right)^2
$$

(78)

$$
\sum_{j \in [d_1] \setminus [d]} \langle \boldsymbol{w}_i^{(t+1)}, \boldsymbol{M}_j^\perp \rangle^2 \leq \sum_{j \in [d_1] \setminus [d]} \langle \boldsymbol{w}_i^{(t)}, \boldsymbol{M}_j^\perp \rangle^2 \left( 1 + \frac{O(\eta)}{d \operatorname{polylog}(d)} \right)^2,
$$

which implies

$$
|\langle \boldsymbol{w}_i^{(t+1)}, \boldsymbol{M}_j \rangle| \geq |\langle \boldsymbol{w}_i^{(t)}, \boldsymbol{M}_j \rangle| \cdot \frac{\|\boldsymbol{w}_i^{(t+1)}\|_2}{\|\boldsymbol{w}_i^{(t)}\|_2} \geq (1 - o(1))\|\boldsymbol{w}_i^{(t+1)}\|_2.
$$

(79)

Next, we prove Theorem D.1(b). In the first stage, the bias growth is large, i.e.,

$$
\boldsymbol{b}_i^{(T'_{i,1})} \geq \boldsymbol{b}_i^{(T_1)}(1 + \frac{\eta}{d})^{T'_{i,1} - T_1} \geq \boldsymbol{b}_i^{(T_1)} \cdot \operatorname{polylog}(d)
$$

$$
\geq \frac{\operatorname{polylog}(d)}{\sqrt{d}}\|\boldsymbol{w}_i^{(T_1)}\|_2 \geq \frac{\operatorname{polylog}(d)}{\sqrt{d}}\|\boldsymbol{w}_i^{(T'_{i,1})}\|_2.
$$

(80)

In the second stage, the bias is large consistently, i.e.,

$$
\boldsymbol{b}_i^{(t+1)} \geq \boldsymbol{b}_i^{(t)} \cdot \frac{\|\boldsymbol{w}_i^{(t+1)}\|_2}{\|\boldsymbol{w}_i^{(t)}\|_2} \geq \frac{\mathrm{polylog}(d)}{\sqrt{d}} \|\boldsymbol{w}_i^{(t+1)}\|_2 \geq \frac{1}{4} \|\boldsymbol{w}_i^{(T'_{i,1})}\|_2. \tag{81}
$$

Finally, we prove Theorem D.1(c): Assuming $\langle \boldsymbol{w}_i^{(t)}, \boldsymbol{M}_j \rangle > 0$ (the opposite case is similar), from $t = T_1$, for $i \in \mathcal{M}_j^\star$, we have

$$
\begin{aligned}
\langle \boldsymbol{w}_i^{(t+1)}, \boldsymbol{M}_j \rangle &= \left( \langle \boldsymbol{w}_i^{(t)}, \boldsymbol{M}_j \rangle - \boldsymbol{b}_i^{(t)} \right) \left( 1 + \epsilon_j \frac{\eta C_z \log \log d}{d} \right) \pm O\left( \frac{\eta |\langle \boldsymbol{w}_i^{(t)}, \boldsymbol{M}_j \rangle|}{d\,\mathrm{polylog}(d)} \right) \\
&\geq \Omega(1) \langle \boldsymbol{w}_i^{(t)}, \boldsymbol{M}_j \rangle \left( 1 + \epsilon_j \frac{\eta C_z \log \log d}{d} \left( 1 - \frac{1}{\mathrm{polylog}(d)} \right) \right) \\
&\geq \Omega(1) \langle \boldsymbol{w}_i^{(T_1)}, \boldsymbol{M}_j \rangle \left( 1 + \epsilon_j \frac{\eta C_z \log \log d}{d} \left( 1 - \frac{1}{\mathrm{polylog}(d)} \right) \right)^{t+1-T_1}.
\end{aligned} \tag{82}
$$

which implies that after certain iteration $t = T_1 + T'$, where $T' = \Theta\left(\frac{d}{\eta}\right)$, we shall have

$$
|\langle \boldsymbol{w}_i^{(T_1+T')}, \boldsymbol{M}_j \rangle| \geq \log \log d \cdot |\langle \boldsymbol{w}_i^{(T_1)}, \boldsymbol{M}_j \rangle| \geq \boldsymbol{b}_i^{(T_1)} \cdot \log \log d. \tag{83}
$$

However, at iteration $t = T_1 + \Theta\left(\frac{d}{\eta}\right)$, we can see from previous analysis that $\|\boldsymbol{w}_i^{(t)}\|_2 \leq (1 + o(1))\|\boldsymbol{w}_i^{(T_1)}\|_2$, so the bias growth can be bounded as

$$
\begin{aligned}
\boldsymbol{b}_i^{(t)} &\leq \boldsymbol{b}_i^{(T_1)} \left( 1 + \frac{\eta}{d} \right)^{\Theta(\frac{d}{\eta})} \cdot \max \left\{ \frac{\|\boldsymbol{w}_i^{(t)}\|_2}{\|\boldsymbol{w}_i^{(T_1)}\|_2}, 1 \right\} \\
&\leq \boldsymbol{b}_i^{(T_1)} \left( 1 + \frac{\eta}{d} \cdot \Theta(\frac{d}{\eta}) \right) \cdot \max \left\{ (1 + o(1)), 1 \right\} \\
&\leq O(\boldsymbol{b}_i^{(T_1)}).
\end{aligned} \tag{84}
$$

Now from our initialization properties in Lemma D.2, we have that $\langle \boldsymbol{w}_{i'}^{(0)}, \boldsymbol{M}_j \rangle^2 \leq O(\sigma_0^2 \log d)$ for all $i \in [m]$. Thus via similar arguments, we also have

$$
|\langle \boldsymbol{w}_{i'}^{(t)}, \boldsymbol{M}_j \rangle| \leq |\langle \boldsymbol{w}_{i'}^{(0)}, \boldsymbol{M}_j \rangle| \left( 1 + \epsilon_j \frac{\eta C_z \log \log d}{d} \left( 1 \pm \frac{1}{\mathrm{polylog}(d)} \right) \right)^t. \tag{85}
$$

holds for all $i' \in [m]$. Now it is easy to see that for $t \leq T_2 = T_1 + \Theta\left(\frac{d\tau \log d}{\eta \log \log d}\right)$, we have

$$
\begin{aligned}
\frac{|\langle \boldsymbol{w}_i^{(t)}, \boldsymbol{M}_j \rangle|}{|\langle \boldsymbol{w}_{i'}^{(t)}, \boldsymbol{M}_j \rangle|} &\geq \Omega(1) \cdot \frac{|\langle \boldsymbol{w}_i^{(T_1)}, \boldsymbol{M}_j \rangle| \left( 1 + \epsilon_j \frac{\eta C_z \log \log d}{d} \left( 1 - \frac{\eta}{\mathrm{polylog}(d)} \right) \right)^{t-T_1}}{|\langle \boldsymbol{w}_{i'}^{(T_1)}, \boldsymbol{M}_j \rangle| \left( 1 + \epsilon_j \frac{\eta C_z \log \log d}{d} + \frac{\eta}{d\,\mathrm{polylog}(d)} \right)^{t-T_1}} \\
&\geq \left( 1 - O\left( \epsilon_j \frac{\eta \log \log d}{d\,\mathrm{polylog}(d))} \right) \right)^{t-T_1} \geq \Omega(1).
\end{aligned} \tag{86}
$$

Thus, the last claim is proved. $\qquad \square$

## D.4    PROOF OF LEMMA 3.2

Lemma 3.2 can be viewed as an informal version of Theorem D.1. In particular, part (a) of Lemma 3.2 corresponds to (77) and Lemma B.2 (c), while part (b) of Lemma 3.2 corresponds to another formulation of Theorem D.1 (c).

# E  THEOREM E.1

## E.1  THEOREM E.1

At the final stage, we show that sparse activation of neurons naturally leads to convergence toward sparse solutions, thereby guaranteeing sparse representations. For all $t \geq T_2$:

**Theorem E.1.** *For all iterations $t$, the neurons $i \in [m]$ satisfy the following properties:*

*(a) For $j \in [d]$, if $i \in \mathcal{M}_j^\star$, then*

$$\left|\langle \boldsymbol{w}_i^{(t)}, \boldsymbol{M}_j \rangle\right| \geq \Omega(1) \, \|\boldsymbol{w}_i^{(t)}\|_2. \tag{87}$$

*(b) For $i \in [m]$, we have*

$$\|\boldsymbol{w}_i^{(t)}\|_2 \leq O(1). \tag{88}$$

*(c) For each $j \in [d]$,*

$$\mathfrak{F}_j^{(t)} := \sum_{i \in \mathcal{M}_j} \langle \boldsymbol{w}_i^{(t)}, \boldsymbol{M}_j \rangle^2 = \Theta\left(\left(\frac{\epsilon_j}{\epsilon_{\max}}\right)^2 \tau \log^3 d\right). \tag{89}$$

*(d) Let $j \in [d]$ and $i \in \mathcal{M}_j^\star$, then there exists $C = \Theta(1)$ such that*

$$\left|\langle \boldsymbol{w}_i^{(t)}, \boldsymbol{M}_j \rangle\right| \geq C \max_{i' \in \mathcal{M}_j} \left|\langle w_{i'}^{(t)}, \boldsymbol{M}_j \rangle\right|. \tag{90}$$

*(e) For $i \notin \mathcal{M}_j$, it holds*

$$\left|\langle \boldsymbol{w}_i^{(t)}, \boldsymbol{M}_j \rangle\right| \leq O\left(\frac{\epsilon_j}{\epsilon_{\max}} \frac{1}{\sqrt{d}\, \Xi_2^5}\right) \|\boldsymbol{w}_i^{(t)}\|_2. \tag{91}$$

*(f) For any $i \in [m]$ and any $j \in [d_1] \setminus [d]$, it holds*

$$\left|\langle \boldsymbol{w}_i^{(t)}, \boldsymbol{M}_j^\perp \rangle\right| \leq O\left(\frac{1}{\sqrt{d_1}\, \Xi_2^5}\right) \|\boldsymbol{w}_i^{(t)}\|_2. \tag{92}$$

*(g) For all $i \in [m]$, the bias satisfies*

$$\boldsymbol{b}_i^{(t)} \geq \frac{\mathrm{polylog}(d)}{\sqrt{d}} \|\boldsymbol{w}_i^{(t)}\|_2. \tag{93}$$

## E.2  USEFUL LEMMAS

When all the conditions in Theorem E.1 hold for some iteration $t \geq T_2$, we have the following fact, which is a simple corollary of Lemma E.9.

**Lemma E.1.** *For any $i \in [m]$, we denote $\mathcal{N}_i = \{j \in [d] : i \in \mathcal{M}_j\}$. Suppose Theorem E.1 holds at iteration $t \geq T_2$, then with high probability over $x \in \mathcal{D}_x$:*

$$\max_{x \in \{\boldsymbol{X}_n, \boldsymbol{Y}_n\}} \mathbf{1}_{h_{i,t}(x) \neq 0} \leq \sum_{j \in \mathcal{N}_i} \mathbf{1}_{|\hat{z}_{p,j}| \neq 0}, \tag{94}$$

*which implies that*

$$\max_{x \in \{\boldsymbol{X}_n, \boldsymbol{Y}_n\} \cup \mathfrak{N}} \Pr\left(h_{i,t}(x) \neq 0\right) \leq O\left(\frac{\log \log d}{d}\right). \tag{95}$$

Now for the simplicity of calculations, we define the following notations which are used through out this section

**Definition E.1** (Expansion of gradient). *For each $i \in [m]$, $j \in [d]$, we expand $\langle \nabla_{\boldsymbol{w}_i} L(f_t), \boldsymbol{M}_j \rangle$ as*

$$\langle \nabla_{\boldsymbol{w}_i} L(f_t), \boldsymbol{M}_j \rangle$$
$$= \mathbb{E}\left[ \left( (\ell'_{p,t} - 1) h_{i,t}(\boldsymbol{Y}_n) + \sum_{\boldsymbol{X}_{n,s} \in \mathfrak{N}} \ell'_{s,t} h_{i,t}(\boldsymbol{X}_{n,s}) \right) \sum_{r=1}^{L} \mathbf{1}_{|\langle \boldsymbol{w}_i, \boldsymbol{z}_{\boldsymbol{X}}^{(r)} \rangle| \geq \boldsymbol{b}_i} \langle \boldsymbol{z}_{\boldsymbol{X}}^{(r)}, \boldsymbol{M}_j \rangle \right], \quad (96)$$

*and*

$$\langle \nabla_{\boldsymbol{w}_i} L(f_t), \boldsymbol{M}_j \rangle = \Psi_{i,j}^{(t)} + \Phi_{i,j}^{(t)} + \mathcal{E}_{i,j}^{(t)}, \quad (97)$$

*where the $\Psi^{(t)}$, $\Phi^{(t)}$, $\mathcal{E}^{(t)}$ are defined as follows. For each*

$$\boldsymbol{z}_{\boldsymbol{X}} = \frac{1}{L}\left( \sum_j \boldsymbol{M}_j \tilde{\boldsymbol{z}}_{n,j} + \tilde{\xi}_n \right) \sim \mathcal{D}_{\boldsymbol{z}_{\boldsymbol{X}}}, \quad \boldsymbol{z}_{\boldsymbol{Y}} = \frac{1}{L}\left( \sum_j \boldsymbol{M}_j \tilde{\boldsymbol{z}}_{n,j}^+ + \tilde{\xi}_n^+ \right) \sim \mathcal{D}_{\boldsymbol{z}_{\boldsymbol{Y}}}, \quad (98)$$

*we write*

$$\psi_{i,j}^{(t)}(\boldsymbol{Y}_n) = \sum_{s=1}^{L} \Big[ \left( \frac{1}{L}\langle \boldsymbol{w}_i^{(t)}, \boldsymbol{M}_j \rangle \tilde{\boldsymbol{z}}_{n,j}^{+(s)} - \boldsymbol{b}_i^{(t)} \right) \mathbf{1}_{\langle \boldsymbol{w}_i^{(t)}, \boldsymbol{z}_{\boldsymbol{Y}}^{(s)} \rangle > \boldsymbol{b}_i^{(t)}}$$
$$- \left( \frac{1}{L}\langle \boldsymbol{w}_i^{(t)}, \boldsymbol{M}_j \rangle \tilde{\boldsymbol{z}}_{n,j}^{+(s)} + \boldsymbol{b}_i^{(t)} \right) \mathbf{1}_{\langle \boldsymbol{w}_i^{(t)}, \boldsymbol{z}_{\boldsymbol{Y}}^{(s)} \rangle < -\boldsymbol{b}_i^{(t)}} \Big], \quad (99)$$

$$\phi_{i,j}^{(t)}(\boldsymbol{Y}_n) = \sum_{s=1}^{L} \langle \boldsymbol{w}_i^{(t)}, \boldsymbol{z}_{Y}^{(s)\backslash j} \rangle \mathbf{1}_{\langle \boldsymbol{w}_i^{(t)}, \boldsymbol{z}_{\boldsymbol{Y}}^{(s)} \rangle > \boldsymbol{b}_i^{(t)}} - \langle \boldsymbol{w}_i^{(t)}, \boldsymbol{z}_{Y}^{(s)\backslash j} \rangle \mathbf{1}_{\langle \boldsymbol{w}_i^{(t)}, \boldsymbol{z}_{\boldsymbol{Y}}^{(s)} \rangle < -\boldsymbol{b}_i^{(t)}}. \quad (100)$$

*Now we define*

$$\Psi_{i,j}^{(t)} := \mathbb{E}\left[ \left( (\ell'_{p,t} - 1) \cdot \psi_{i,j}^{(t)}(\boldsymbol{Y}_n) + \sum_{\boldsymbol{X}_{n,s} \in \mathfrak{N}} \ell'_{s,t} \cdot \psi_{i,j}^{(t)}(\boldsymbol{X}_{n,s}) \right) \sum_{r=1}^{L} \mathbf{1}_{|\langle \boldsymbol{w}_i, \boldsymbol{z}_{\boldsymbol{X}}^{(r)} \rangle| \geq \boldsymbol{b}_i} \tilde{\boldsymbol{z}}_{n,j}^{(r)} \right], \quad (101)$$

$$\Phi_{i,j}^{(t)} := \mathbb{E}\left[ \left( (\ell'_{p,t} - 1) \cdot \phi_{i,j}^{(t)}(\boldsymbol{Y}_n) + \sum_{\boldsymbol{X}_{n,s} \in \mathfrak{N}} \ell'_{s,t} \cdot \phi_{i,j}^{(t)}(\boldsymbol{X}_{n,s}) \right) \sum_{r=1}^{L} \mathbf{1}_{|\langle \boldsymbol{w}_i, \boldsymbol{z}_{\boldsymbol{X}}^{(r)} \rangle| \geq \boldsymbol{b}_i} \tilde{\boldsymbol{z}}_{n,j}^{(r)} \right], \quad (102)$$

$$\mathcal{E}_{i,j}^{(t)} := \mathbb{E}\left[ \left( (\ell'_{p,t} - 1) \cdot h_{i,t}(\boldsymbol{Y}_n) + \sum_{\boldsymbol{X}_{n,s} \in \mathfrak{N}} \ell'_{s,t} \cdot h_{i,t}(\boldsymbol{X}_{n,s}) \right) \sum_{r=1}^{L} \mathbf{1}_{|\langle \boldsymbol{w}_i, \boldsymbol{z}_{\boldsymbol{X}}^{(r)} \rangle| \geq \boldsymbol{b}_i} \langle \boldsymbol{M}_j, \tilde{\xi}_n^{(r)} \rangle \right]. \quad (103)$$

*Moreover, for $j \in [d_1] \setminus [d]$, we can similarly define*

$$\Psi_{i,j}^{(t)}, \ \Phi_{i,j}^{(t)} \equiv 0, \quad (104)$$

$$\mathcal{E}_{i,j}^{(t)} := \mathbb{E}\left[ \left( (\ell'_{p,t} - 1) \cdot h_{i,t}(\boldsymbol{Y}_n) + \sum_{\boldsymbol{X}_{n,s} \in \mathfrak{N}} \ell'_{s,t} \cdot h_{i,t}(\boldsymbol{X}_{n,s}) \right) \sum_{r=1}^{L} \mathbf{1}_{|\langle \boldsymbol{w}_i, \boldsymbol{z}_{\boldsymbol{X}}^{(r)} \rangle| \geq \boldsymbol{b}_i} \langle \boldsymbol{M}_j^{\perp}, \tilde{\xi}_n^{(r)} \rangle \right]. \quad (105)$$

Equipped with the above definition, we are ready to characterize the training process at the final stage.

**Lemma E.2** (Lower bound for $\Psi_1^{(t)}$). *Suppose Theorem E.1 holds at iteration $t$. For $j \in [d]$ and $i \in \mathcal{M}_j^\star$, there exists $G_1 = \Theta(1)$ such that if*

$$\mathfrak{F}_j^{(t)} := \sum_{i' \in \mathcal{M}_j} \langle \boldsymbol{w}_{i'}^{(t)}, \boldsymbol{M}_j \rangle^2 \left( \sum_{r=1}^{L} \tilde{\boldsymbol{z}}_{n,j}^{(r)} \right)^2 \leq \left( \frac{\epsilon_j}{\epsilon_{\max}} \right)^2 G_1 \tau \log d, \quad (106)$$

*then we have*

$$
\Psi_{i,j}^{(t)} \cdot \mathrm{sign}\left( \sum_{s=1}^{L} \langle \boldsymbol{w}_i^{(t)}, \boldsymbol{M}_j \rangle \tilde{\boldsymbol{z}}_{n,j}^{+(s)} \right)
$$

$$
\geq \frac{\mathbb{E}\left[ \sum_{r=1}^{L} |\tilde{\boldsymbol{z}}_{n,j}^{(r)}| \right]}{\mathrm{polylog}(d)} \left( 1 - O\left( \frac{1}{\Xi_2^3} \right) \right) \left( \sum_{s=1}^{L} |\langle \boldsymbol{w}_i^{(t)}, \boldsymbol{M}_j \rangle \tilde{\boldsymbol{z}}_{n,j}^{+(s)} - \boldsymbol{b}_i^{(t)}| \right). \tag{107}
$$

**Lemma E.3** (Upper bound for $\Psi_{i,j}^{(t)}$). *Let $j \in [d]$ and $i \in \mathcal{M}_j^\star$. Suppose Theorem E.1 holds at iteration $t$, then there exists a constant $G_2 = \Theta(1)$ such that if*

$$
\mathfrak{F}_j^{(t)} := \sum_{j:\, i \in \mathcal{M}_j} \langle \boldsymbol{w}_i^{(t)}, \boldsymbol{M}_j \rangle^2 \left( \sum_{s=1}^{L} \tilde{\boldsymbol{z}}_{n,j}^{+(s)} \right)^2 \geq \left( \frac{\epsilon_j}{\epsilon_{\max}} \right)^2 G_2 \tau \log d, \tag{108}
$$

*we have*

$$
\Psi_{i,j}^{(t)} \leq \frac{1}{\mathrm{poly}(d)} \sum_{s=1}^{L} \left| \langle \boldsymbol{w}_i^{(t)}, \boldsymbol{M}_j \rangle \tilde{\boldsymbol{z}}_{n,j}^{+(s)} \right|. \tag{109}
$$

*Similarly, for $i \in \mathcal{M}_j$, we have*

$$
\Psi_{i,j}^{(t)} \leq \frac{1}{\mathrm{poly}(d)} \sum_{s=1}^{L} \left| \langle \boldsymbol{w}_i^{(t)}, \boldsymbol{M}_j \rangle \tilde{\boldsymbol{z}}_{n,j}^{+(s)} \right| + O\left( \frac{1}{d^2} \right) \boldsymbol{b}_i^{(t)}. \tag{110}
$$

**Lemma E.4.** *At iteration $t \geq T_2$, let $j \in [d]$ and $i \in [m]$. Suppose Theorem E.1 holds at $t$. Then for each $j \in [d_1]$, we have*

$$
\left| \mathcal{E}_{i,j}^{(t)} \right| \leq O\left( \frac{\Xi_2^2 \|\boldsymbol{w}_i^{(t)}\|_2}{d^2 \tau} \right) \cdot \max_{i' \in [m]} \left( \left| \langle \boldsymbol{w}_{i'}^{(t)}, \boldsymbol{M}_j \rangle \right| \right). \tag{111}
$$

**Lemma E.5** (Reduction of $\Phi^{(t)}$ to the bounds of $\Psi^{(t)}$). *Let $j \in [d]$ and $i \in \mathcal{M}_j$. Suppose Theorem E.1 holds for all iterations before $t \in \left[ \frac{d^{1.01}}{\eta}, \frac{d^{1.99}}{\eta} \right]$ and after $T_2$. Also suppose that for all $l \in [d]$, we have*

$$
\mathfrak{F}_l^{(t')} = \Omega\left( \left( \frac{\epsilon_j}{\epsilon_{\max}} \right)^2 \tau \log d \right) \quad \text{at some } t' = \Theta(T_2). \tag{112}
$$

*Then the following bounds hold:*

*For iteration $t \in \left[ \frac{d^{1.01}}{\eta}, \frac{d^{1.495}}{\eta} \right]$,*

$$
\Phi_{i,j}^{(t)} \leq \widetilde{O}\left( \frac{\epsilon_j}{\epsilon_{\max}} \cdot \frac{\Xi_2^2}{d^{3/2}} \right) \|\boldsymbol{w}_i^{(t)}\|_2. \tag{113}
$$

*For iteration $t \in \left[ \frac{d^{1.495}}{\eta}, \frac{d^{1.99}}{\eta} \right]$,*

$$
\Phi_{i,j}^{(t)} \leq \widetilde{O}\left( \frac{1}{d^{1.98}} \right) \|\boldsymbol{w}_i^{(t)}\|_2. \tag{114}
$$

**Definition E.2** (Optimal Learner). *We define a learner network that we deem as the optimal feature map for this task. Let $\kappa > 0$, we define $\theta^\star := \{\theta_i^\star\}_{i \in [m]}$ as follows:*

$$
\theta_i^\star = \begin{cases} \dfrac{\sqrt{\tau}\,\kappa}{|\mathcal{M}_j^\star|} \boldsymbol{M}_j \cdot \mathrm{sign}(\langle \boldsymbol{w}_i^{(T_2)}, \boldsymbol{M}_j \rangle), & \text{if } i \in \mathcal{M}_j^\star, \\[2mm] 0, & \text{if } i \notin \bigcup_{j \in [d]} \mathcal{M}_j^\star. \end{cases} \tag{115}
$$

Furthermore, we define the optimal feature map $f_t^\star$ as follows. For $i \in [m]$, the $i$-th neuron of $f_{t,\theta}$ given weight $\theta_i \in \mathbb{R}^{d_1}$ is

$$f_{t,\theta,i}(\boldsymbol{X}_n) = \sum_{r=1}^{L} \left[ \left( \langle \theta_i, \boldsymbol{z}_{\boldsymbol{X}}^{(r)} \rangle - \boldsymbol{b}_i \right) \mathbf{1}_{\langle \boldsymbol{w}_i^{(t)}, \boldsymbol{z}_{\boldsymbol{X}}^{(r)} \rangle \geq \boldsymbol{b}_i} - \left( -\langle \theta_i, \boldsymbol{z}_{\boldsymbol{X}}^{(r)} \rangle - \boldsymbol{b}_i \right) \mathbf{1}_{-\langle \boldsymbol{w}_i^{(t)}, \boldsymbol{z}_{\boldsymbol{X}}^{(r)} \rangle \geq \boldsymbol{b}_i} \right].$$

(116)

Finally, we write $f_{t,\theta}$ as the concatenation

$$f_{t,\theta}(\cdot) = \left( f_{t,\theta,1}(\cdot), \dots, f_{t,\theta,m}(\cdot) \right)^\top.$$

(117)

**Lemma E.6** (Optimality). *Let $\{\theta_i^\star\}_{i \in [m]}$ and $f_{t,\theta}$ be defined as in Definition E.1. When Theorem E.1, define the pseudo loss function*

$$\widetilde{L}(f_{t,\theta^\star}, f_t) := \mathbb{E}\left[ -\tau \log\left( \frac{e^{\langle f_{t,\theta^\star}(\boldsymbol{X}_n), f_t(\boldsymbol{Y}_n) \rangle / \tau}}{\sum_{\boldsymbol{X} \in \mathfrak{B}} e^{\langle f_{t,\theta^\star}(\boldsymbol{X}_n), f_t(\boldsymbol{X}) \rangle / \tau}} \right) \right].$$

(118)

*Then by choosing $\kappa = \Theta(\Xi_2)$, and assuming*

$$\sum_{i \in \mathcal{M}_j^\star} |\langle \boldsymbol{w}_i^{(t)}, \boldsymbol{M}_j \rangle| \geq \Omega\left( \frac{\sqrt{\tau}}{\Xi_2} \right),$$

(119)

*we obtain the following loss guarantee:*

$$\widetilde{L}(f_{t,\theta^\star}, f_t) \leq O\left( \frac{1}{\log d} \right).$$

(120)

**Lemma E.7** (Pre-activation size I). *Let $\boldsymbol{z}_{\boldsymbol{X}}^{(r)} = \frac{1}{L}\left( \boldsymbol{M}\tilde{\boldsymbol{z}}_n^{(r)} + \tilde{\xi}_n^{(r)} \right) \sim \mathcal{D}_{\boldsymbol{z}_{\boldsymbol{X}}}, \quad \boldsymbol{w}_i \in \mathbb{R}^{d_1}$. Define $\boldsymbol{z}_X^{(r)\backslash j} = \frac{1}{L}\left( \sum_{j' \neq j,\, j' \in [d]} \boldsymbol{M}_{j'}\tilde{\boldsymbol{z}}_{n,j'}^{(r)} + \tilde{\xi}_n^{(r)} \right)$. Then the following results hold:*

*(a) Naive Chebyshev bound: For any $\lambda > 0$,*

$$\Pr_{\tilde{\boldsymbol{z}}_n^{(r)\backslash j},\, \tilde{\xi}_n^{(r)}}\left( \left( \langle \boldsymbol{w}_i, \boldsymbol{z}_X^{(r)\backslash j} \rangle + \frac{1}{L}\langle \boldsymbol{w}_i, \boldsymbol{M}_j \rangle \tilde{\boldsymbol{z}}_{n,j}^{(r)} \right)^2 > \frac{\lambda \|\boldsymbol{w}_i\|_2^2 \sqrt{\log d}}{d} \right) \leq O\left( \frac{1}{\lambda} \right).$$

(121)

*The same tail bound applies to $\langle \boldsymbol{w}_i, \boldsymbol{z}_{\boldsymbol{X}}^{(r)} \rangle$, $\langle \boldsymbol{w}_i, \frac{\boldsymbol{z}_{\boldsymbol{Y}}^{(s)} - \boldsymbol{z}_{\boldsymbol{X}}^{(r)}}{2} \rangle$, and $\langle \boldsymbol{w}_i, \tilde{\xi}_n^{(r)} \rangle$.*

*(b) High probability bound for sparse signal:*

$$\Pr\left( \langle \boldsymbol{w}_i, \boldsymbol{M}\tilde{\boldsymbol{z}}_n^{(r)} \rangle^2 > \|\boldsymbol{w}_i\|_2^2 \cdot \max_{j \in [d]} \|\boldsymbol{M}_j\|_\infty^2 \log^4 d \right) \lesssim e^{-\Omega(\log^2 d)}.$$

(122)

*(c) High probability bound for dense signal: Let $Z = \langle \boldsymbol{w}_i, \tilde{\xi}_n^{(r)} \rangle$. Then*

$$\Pr\left( \boldsymbol{z}^2 \geq \frac{\|\boldsymbol{w}_i\|_2^2 \log^4 d}{d} \right) \lesssim e^{-\Omega(\log^2 d)}.$$

(123)

**Lemma E.8** (Pre-activation size II). *Suppose the following conditions hold:*

$$\langle \boldsymbol{w}_i^{(t)}, \boldsymbol{M}_j \rangle^2 \geq \Omega\left( (\boldsymbol{b}_i^{(t)})^2 \right) \quad \text{for at most } O(1) \text{ indices } j \in [d],$$

(124)

$$\langle \boldsymbol{w}_i^{(t)}, \boldsymbol{M}_j \rangle^2 \geq \Omega\left( \frac{(\boldsymbol{b}_i^{(t)})^2}{\sqrt{\log d}} \right) \quad \text{for at most } O\left( e^{-\Omega(\sqrt{\log d})} d \right) \text{ indices } j \in [d],$$

(125)

$$\|\boldsymbol{w}_i^{(t)}\|_2^2 \leq O\left( \frac{d(\boldsymbol{b}_i^{(t)})^2}{\log d} \right).$$

(126)

*Then, for any $\lambda \geq 0.0001$,*

$$\Pr\left( |\langle \boldsymbol{w}_i^{(t)}, \boldsymbol{z}_{\boldsymbol{X}}^{(r)} \rangle| \geq \lambda \boldsymbol{b}_i^{(t)} \right) \lesssim e^{-\Omega(\log^{1/4} d)},$$

(127)

*and*

$$\Pr\left( \left| \langle \boldsymbol{w}_i^{(t)}, \frac{\boldsymbol{z}_{\boldsymbol{X}}^{(r)} + \boldsymbol{z}_{\boldsymbol{X}}^{(s)}}{2} \rangle \right| \geq \lambda \boldsymbol{b}_i^{(t)} \right) \lesssim e^{-\Omega(\log^{1/4} d)}.$$

(128)

**Lemma E.9** (Pre-activation size III). *Let $i \in [m]$. Suppose there exists a set $\mathcal{N}_i \subseteq [d]$ with $|\mathcal{N}_i| = O(1)$ such that*

$$\langle \boldsymbol{w}_i^{(t)}, \boldsymbol{M}_j \rangle^2 \leq O\left( \frac{(\boldsymbol{b}_i^{(t)})^2}{\text{polylog}(d)} \right), \quad \forall j \notin \mathcal{N}_i, \tag{129}$$

*and*

$$\|\boldsymbol{w}_i^{(t)}\|_2^2 \leq O\left( \frac{d(\boldsymbol{b}_i^{(t)})^2}{\text{polylog}(d)} \right). \tag{130}$$

*Then, for any $\lambda \in [0.01, 0.99]$,*

$$\Pr\left[ \left| \sum_{j \notin \mathcal{N}_i} \langle \boldsymbol{w}_i^{(t)}, \boldsymbol{M}_j \rangle \tilde{z}_{n,j}^{(r)} + \langle \boldsymbol{w}_i, \tilde{\xi}_n^{(r)} \rangle \right| \geq \lambda \boldsymbol{b}_i^{(t)} \right] \lesssim e^{-\Omega(\log^2 d)}. \tag{131}$$

**Lemma E.10** (Gradient for sparse features). *Suppose D.1 holds at iteration $t \geq 0$. For $j \in [d]$, we denote events*

$$\begin{aligned}
A_1 &:= \left\{ S_{i,t}^{\backslash j} \geq \boldsymbol{b}_i^{(t)} - \alpha_{i,j}^{(t)} C_{\tilde{\boldsymbol{z}}} \right\}, \\
A_2 &:= \left\{ \bar{S}_{i,t}^{\backslash j} \geq \boldsymbol{b}_i^{(t)} - \bar{\alpha}_{i,j}^{(t)} C_{\tilde{\boldsymbol{z}}} \right\}, \\
A_3 &:= \left\{ \left| \bar{S}_{i,t}^{\backslash j} + \bar{\alpha}_{i,j}^{(t)} C_{\tilde{\boldsymbol{z}}} \right| \geq \tfrac{1}{2} \left( \alpha_{i,j}^{(t)} C_{\tilde{\boldsymbol{z}}} - \boldsymbol{b}_i^{(t)} \right) \right\}, \\
A_4 &:= \left\{ S_{i,t}^{\backslash j} \geq \tfrac{1}{2} \left( \alpha_{i,j}^{(t)} C_{\tilde{\boldsymbol{z}}} - \boldsymbol{b}_i^{(t)} \right) \right\};
\end{aligned} \tag{132}$$

*and quantities $L_1, L_2, L_3, L_4$ as*

$$\begin{aligned}
L_1 &:= \sqrt{ \frac{\mathbb{E}[|\bar{S}_{i,t}^{\backslash j}|^2 (\mathbf{1}_{A_1} + \mathbf{1}_{A_2})]}{\mathbb{E}[\langle \boldsymbol{w}_i^{(t)}, \tilde{\xi} \rangle^2]} }, \quad L_2 := \Pr(A_1), \\
L_3 &:= \sqrt{ \frac{\mathbb{E}[|\bar{S}_{i,t}^{\backslash j}|^2 (\mathbf{1}_{A_3} + \mathbf{1}_{A_4})]}{\mathbb{E}[\langle \boldsymbol{w}_i^{(t)}, \tilde{\xi} \rangle^2]} }, \quad L_4 := \Pr(A_3).
\end{aligned} \tag{133}$$

*Then we have the following results:*

*(a) (all features) For all $i \in [m]$, if $\alpha_{i,j}^{(t)} \geq 0$, we have (when $\alpha_{i,j}^{(t)} \leq 0$ the opposite inequality holds)*

$$\begin{aligned}
&\mathbb{E}\left[ h_i(\boldsymbol{Y}_n) \sum_{r=1}^{L} \mathbf{1}_{|\langle \boldsymbol{w}_i, \boldsymbol{z}_{\boldsymbol{X}}^{(r)} \rangle| \geq \boldsymbol{b}_i} \tilde{z}_{n,j}^{(r)} \right] \\
&\leq \frac{1}{L} \alpha_{i,j}^{(t)} \cdot \mathbb{E}\left[ \sum_{s=1}^{L} \tilde{z}_{n,j}^{+(s)} \sum_{r=1}^{L} \tilde{z}_{n,j}^{(r)} \mathbf{1}_{|\langle \boldsymbol{w}_i^{(t)}, \frac{\boldsymbol{z}_{\boldsymbol{X}} + \boldsymbol{z}_{\boldsymbol{Y}}}{2} \rangle| \geq \boldsymbol{b}_i + |\langle \boldsymbol{w}_i^{(t)}, \boldsymbol{z}_{\boldsymbol{X}} - \frac{\boldsymbol{z}_{\boldsymbol{X}} + \boldsymbol{z}_{\boldsymbol{Y}}}{2} \rangle|} \right] \\
&\pm \left( \alpha_{i,j}^{(t)} + O\left( \sqrt{\mathbb{E}|\bar{\alpha}_{i,j}^{(t)}|^2} \right) \right) \cdot \mathbb{E}\left[ \sum_{s=1}^{L} \sum_{r=1}^{L} \left| \frac{\tilde{z}_{n,j}^{(r)} + \tilde{z}_{n,j}^{+(s)}}{2} \right| |\tilde{z}_{n,j}^{(r)}| \right] \cdot O(L_1 + L_2).
\end{aligned} \tag{134}$$

*(b) (lucky features) If $\alpha_{i,j}^{(t)} > \boldsymbol{b}_i^{(t)}$, we have*

$$\begin{aligned}
&\mathbb{E}\left[ h_i(\boldsymbol{Y}_n) \sum_{r=1}^{L} \mathbf{1}_{|\langle \boldsymbol{w}_i, \boldsymbol{z}_{\boldsymbol{X}}^{(r)} \rangle| \geq \boldsymbol{b}_i} \tilde{\boldsymbol{z}}_{n,j}^{(r)} \right] \\
&\leq \frac{1}{L} \left( \alpha_{i,j}^{(t)} - \boldsymbol{b}_i^{(t)} \right) \cdot \mathbb{E}\left[ \sum_{s=1}^{L} \tilde{z}_{n,j}^{+(s)} \sum_{r=1}^{L} \tilde{z}_{n,j}^{(r)} \mathbf{1}_{|\langle \boldsymbol{w}_i^{(t)}, \frac{\boldsymbol{z}_{\boldsymbol{X}} + \boldsymbol{z}_{\boldsymbol{Y}}}{2} \rangle| \geq \boldsymbol{b}_i + |\langle \boldsymbol{w}_i^{(t)}, \boldsymbol{z}_{\boldsymbol{X}} - \frac{\boldsymbol{z}_{\boldsymbol{X}} + \boldsymbol{z}_{\boldsymbol{Y}}}{2} \rangle|} \right] \\
&\pm \left( \alpha_{i,j}^{(t)} + O\left( \sqrt{\mathbb{E}|\bar{\alpha}_{i,j}^{(t)}|^2} \right) \right) \cdot \mathbb{E}\left[ \sum_{s=1}^{L} \sum_{r=1}^{L} \left| \frac{\tilde{z}_{n,j}^{(r)} + \tilde{z}_{n,j}^{+(s)}}{2} \right| |\tilde{z}_{n,j}^{(r)}| \right] \cdot O(L_3 + L_4).
\end{aligned} \tag{135}$$

If $\alpha_{i,j}^{(t)} < -b_i^{(t)}$, then the opposite inequality holds with $(\alpha_{i,j}^{(t)} - b_i^{(t)})$ replaced by $(\alpha_{i,j}^{(t)} + b_i^{(t)})$.

**Lemma E.11** (Gradient from dense signals)**.** *Let $i \in [m]$ and $j \in [d]$. Suppose D.1 holds for the current iteration $t$. Then*

$$\left| \mathbb{E}\left[ h_i(\boldsymbol{Y}_n) \sum_{r=1}^L \mathbf{1}_{|\langle w_i^{(t)}, \boldsymbol{z}_{\boldsymbol{X}}^{(r)}\rangle| \geq b_i^{(t)}} \langle \tilde{\xi}_n^{(r)}, \boldsymbol{M}_j \rangle \right] \right| \leq \tilde{\mathcal{O}}\left( \frac{\|w_i^{(t)}\|_2}{d^2} \right) \cdot \Pr\left( h_{i,t}(\boldsymbol{Y}_n) \neq 0 \right). \quad (136)$$

*For dense features $\boldsymbol{M}_j^\perp$, $j \in [d_1] \setminus [d]$, we have a similar result:*

$$\left| \mathbb{E}\left[ h_i(\boldsymbol{Y}_n) \sum_{r=1}^L \mathbf{1}_{|\langle w_i^{(t)}, \boldsymbol{z}_{\boldsymbol{X}}^{(r)}\rangle| \geq b_i^{(t)}} \langle \tilde{\xi}_n^{(r)}, \boldsymbol{M}_j^\perp \rangle \right] \right| \leq \tilde{\mathcal{O}}\left( \frac{\|w_i^{(t)}\|_2}{d\sqrt{d_1}} \right) \cdot \Pr\left( h_{i,t}(\boldsymbol{Y}_n) \neq 0 \right). \quad (137)$$

### E.3 PROOF OF THEOREM E.1

*Proof of Theorem E.1:* First we need to prove all the Theorem E.1 hold for $t = T_2$. Indeed, (1), (4), (5), (6), (7) is valid at $T_2$ from Lemma E.9. and Theorem D.1; (2) and (3) holds at $T_2$ obviously.

Now suppose it hold for some $t \geq T_2$, we will prove that it still hold for $t+1$. We first deal with the case where $j \in [d]$ and $i \notin \mathcal{M}_j$, where it holds that

$$\langle w_i^{(t+1)}, \boldsymbol{M}_j \rangle = \langle w_i^{(t)}, \boldsymbol{M}_j \rangle (1 - \eta\lambda) + \eta \mathbb{E}[h_{i,t}(\boldsymbol{Y}_n) \sum_{r=1}^L \mathbf{1}_{|\langle w_i, \boldsymbol{z}_{\boldsymbol{X}}^{(r)}\rangle| \geq b_i} \langle \boldsymbol{z}_{\boldsymbol{X}}^{(r)}, \boldsymbol{M}_j \rangle]$$
$$- \eta \mathbb{E}\left[ \sum_{\boldsymbol{X}_{n,s} \in \mathfrak{N}} \ell_{s,t}' \cdot h_{i,t}(\boldsymbol{X}_{n,s}) \sum_{r=1}^L \mathbf{1}_{|\langle w_i, \boldsymbol{z}_{\boldsymbol{X}}^{(r)}\rangle| \geq b_i} \langle \boldsymbol{z}_{\boldsymbol{X}}^{(r)}, \boldsymbol{M}_j \rangle \right] \pm \frac{\eta}{\text{poly}(d_1)}. \quad (138)$$

In this case, to calculate the expectation, we need to use Lemma E.10, Lemma E.4. First we compute the probability of events $A_1 - A_4$ by using Lemma E.7, Lemma E.8, Lemma E.9 and our Theorem E.1 to obtain

$$\Pr(A_1), \Pr(A_2) \leq \frac{1}{\text{poly}(d)^{\Omega(\log d)}}, \quad (139)$$

which implies

$$L_1, L_2 \leq \frac{1}{\text{poly}(d)^{\Omega(\log d)}}. \quad (140)$$

Furthermore, from Fact E.1, we also have

$$\mathbb{E}\left[ \sum_{s=1}^L \tilde{z}_{n,j}^{+(s)} \sum_{r=1}^L \tilde{z}_{n,j}^{(r)} \mathbf{1}_{|\langle w_i, \frac{\boldsymbol{z}_{\boldsymbol{X}}^{(r)} + \boldsymbol{z}_{\boldsymbol{Y}}^{(s)}}{2}\rangle| \geq b_i + |\langle w_i, \boldsymbol{z}_{\boldsymbol{X}}^{(r)} - \frac{\boldsymbol{z}_{\boldsymbol{X}}^{(r)} + \boldsymbol{z}_{\boldsymbol{Y}}^{(s)}}{2}\rangle|} \right] \leq \epsilon_j \frac{1}{\text{poly}(d)^{\Omega(\log d)}}. \quad (141)$$

Now we further take into considerations Lemma E.11, Lemma E.4. We can obtain

$$|\langle w_i^{(t+1)}, \boldsymbol{M}_j \rangle| \leq \langle w_i^{(t)}, \boldsymbol{M}_j \rangle (1 - \eta\lambda) + \tilde{\mathcal{O}}\left( \frac{\eta \Xi_2^2 \|w_i^{(t)}\|_2}{d^2} \right) \pm \frac{\eta}{\text{poly}(d_1)}. \quad (142)$$

Indeed, since we have chosen learning rate $\eta = \frac{1}{\text{poly}(d)}$ and $\lambda \in \left[ \frac{1}{d^{1.01}}, \frac{1}{d^{1.49}} \right]$, it is easy to prove (5) as follows:

• For $i \notin \mathcal{M}_j$, $|\langle w_i^{(t)}, \boldsymbol{M}_j \rangle| \leq O\left( \frac{\epsilon_j}{\epsilon_{\max}} \frac{\|w_i^{(t)}\|_2}{\sqrt{d}\Xi_2^5} \right)$: This is easy since by using Lemma E.10, Lemma E.4, we can prove the following inequality by contradiction

$$|\langle w_i^{(t)}, \boldsymbol{M}_j \rangle| \leq |\langle w_i^{(t-1)}, \boldsymbol{M}_j \rangle| (1 + \epsilon_j \frac{\eta}{d^2} - \eta\lambda) + \tilde{\mathcal{O}}\left( \frac{\eta \Xi_2^2}{d^2} \right) \|w_i^{(t)}\|_2$$
$$\leq \cdots \leq O\left( \frac{\epsilon_j}{\epsilon_{\max}} \frac{\|w_i^{(t)}\|_2}{\sqrt{d}\Xi_2^5} \right). \quad (143)$$

Now we begin to prove (6). For all $i \in [m]$, we have $\max_{j \in [d_1] \setminus [d]} |\langle \boldsymbol{w}_i^{(t)}, \boldsymbol{M}_j^\perp \rangle| \leq O\left(\frac{\|\boldsymbol{w}_i^{(t)}\|_2}{\sqrt{d_1} \Xi_2^5}\right)$

at iteration $t = T_2$. Now, by expanding the gradient updates of $\langle \boldsymbol{w}_i^{(t)}, \boldsymbol{M}_j^\perp \rangle$, we can see that

$$
\begin{aligned}
|\langle \boldsymbol{w}_i^{(t+1)}, \boldsymbol{M}_j^\perp \rangle| &\leq |\langle \boldsymbol{w}_i^{(t)}, \boldsymbol{M}_j^\perp \rangle|(1 - \eta\lambda) + |\Psi_{i,j}^{(t)}| + |\Phi_{i,j}^{(t)}| + |\mathcal{E}_{i,j}^{(t)}| \\
&\leq |\langle \boldsymbol{w}_i^{(t)}, \boldsymbol{M}_j^\perp \rangle|(1 - \eta\lambda) + \tilde{\mathcal{O}}\left(\frac{\Xi_2^5}{\tau\sqrt{d_1}d^2}\right)\|\boldsymbol{w}_i^{(t)}\|_2.
\end{aligned}
\tag{144}
$$

where the last inequality are obtained as follows: From Lemma E.4 we have

$$
\begin{aligned}
|\mathcal{E}_{i,j}^{(t)}| &\leq O\left(\frac{\|\boldsymbol{w}_i^{(t)}\|_2 \Xi_2^2}{d^2\tau}\right) \cdot \max_{i' \in [m]}\left(|\langle w_{i'}^{(t)}, \boldsymbol{M}_j^\perp \rangle|\right) \\
&\leq O\left(\frac{\|\boldsymbol{w}_i^{(t)}\|_2 \Xi_2^2}{d^2\tau}\right) \cdot \tilde{\mathcal{O}}\left(\frac{1}{\sqrt{d_1}}\right) \quad (since \max_{i' \in [m]} |\langle w_{i'}^{(t)}, \boldsymbol{M}_j^\perp \rangle| \leq \tilde{\mathcal{O}}\left(\frac{1}{\sqrt{d_1}\Xi_2^5}\right) \\
&\leq \tilde{\mathcal{O}}\left(\frac{\Xi_2^5}{\tau\sqrt{d_1}d^2}\right)\|\boldsymbol{w}_i^{(t)}\|_2.
\end{aligned}
\tag{145}
$$

After (5) and (6) are proven, it is easy to observe (1) is true at t. Below we shall prove (2), (3) and (4), after which (7) can be also trivially proven.

Indeed, (2) is a corollary of (3) and (4), since if $\mathfrak{F}_j^{(t)} \leq O(\tau \log^3 d)$ and (4) holds, we simply have

$$
\begin{aligned}
\|\boldsymbol{w}_i^{(t)}\|_2^2 &= \sum_{j \in \mathcal{N}_i} \langle \boldsymbol{w}_i^{(t)}, \boldsymbol{M}_j \rangle^2 + \sum_{j \notin \mathcal{N}_i, j \in [d]} \langle \boldsymbol{w}_i^{(t)}, \boldsymbol{M}_j \rangle^2 + \sum_{j \in [d_1] \setminus [d]} \langle \boldsymbol{w}_i^{(t)}, \boldsymbol{M}_j^\perp \rangle^2 \\
&\leq \sum_{j \in \mathcal{N}_i} \langle \boldsymbol{w}_i^{(t)}, \boldsymbol{M}_j \rangle^2 + O(d) \cdot O\left((\frac{\epsilon_j}{\epsilon_{\max}})^2 \frac{\|\boldsymbol{w}_i^{(t)}\|_2^2}{d\Xi_2^{10}}\right) + O(d_1) \cdot O\left(\frac{\|\boldsymbol{w}_i^{(t)}\|_2^2}{d_1\Xi_2^{10}}\right) \\
&\leq \sum_{j \in \mathcal{N}_i} \langle \boldsymbol{w}_i^{(t)}, \boldsymbol{M}_j \rangle^2 + o\left((\frac{\epsilon_j}{\epsilon_{\max}})^2 \frac{1}{\Xi_2^{10}}\|\boldsymbol{w}_i^{(t)}\|_2^2\right),
\end{aligned}
\tag{146}
$$

which implies (2).

$$
\begin{aligned}
\|\boldsymbol{w}_i^{(t)}\|_2^2 &\leq \sum_{j \in \mathcal{N}_i} \langle \boldsymbol{w}_i^{(t)}, \boldsymbol{M}_j \rangle^2 + o\left((\frac{\epsilon_j}{\epsilon_{\max}})^2 \frac{1}{\Xi_2^{10}}\|\boldsymbol{w}_i^{(t)}\|_2^2\right) \\
&\leq \sum_{j \in \mathcal{N}_i} \langle \boldsymbol{w}_i^{(t)}, \boldsymbol{M}_j \rangle^2 \\
&\leq O(1)O(\frac{\text{polylod}(d)}{d^c}) \\
&\leq O(1).
\end{aligned}
\tag{147}
$$

Thus we only need to prove (3) and (4). Indeed, for (3), letting $i \in \mathcal{M}_j$, we proceed as follows: we first write the updates of $\langle \boldsymbol{w}_i^{(t)}, \boldsymbol{M}_j \rangle$ as

$$
\begin{aligned}
\langle \boldsymbol{w}_i^{(t+1)}, \boldsymbol{M}_j \rangle &= \langle \boldsymbol{w}_i^{(t)}, \boldsymbol{M}_j \rangle(1 - \eta\lambda) + \Psi_{i,j}^{(t)} + \Phi_{i,j}^{(t)} + \mathcal{E}_{i,j}^{(t)} \\
&= \langle \boldsymbol{w}_i^{(t)}, \boldsymbol{M}_j \rangle(1 - \eta\lambda) + \Psi_{i,j}^{(t)} + \tilde{\mathcal{O}}\left(\frac{\Xi_2^2}{d^2}\right)\|\boldsymbol{w}_i^{(t)}\|_2.
\end{aligned}
\tag{148}
$$

where the last inequality comes again from Lemma E.4. Now suppose for some t we have $\mathfrak{F}_j^{(t)} \geq \Omega((\frac{\epsilon_j}{\epsilon_{\max}})^2 \tau \log^3 d)$, by Lemma E.3, we have

$$
\begin{aligned}
\langle \boldsymbol{w}_i^{(t+1)}, \boldsymbol{M}_j \rangle &= \langle \boldsymbol{w}_i^{(t)}, \boldsymbol{M}_j \rangle\left(1 + \epsilon_j \frac{1}{\text{poly}(d)} - \eta\lambda\right) + \tilde{\mathcal{O}}\left(\frac{\Xi_2^2}{d^2}\right)\|\boldsymbol{w}_i^{(t)}\|_2 \\
&\leq \langle \boldsymbol{w}_i^{(t)}, \boldsymbol{M}_j \rangle\left(1 + \epsilon_j \frac{1}{\text{poly}(d)} - \frac{\eta\lambda}{2}\right).
\end{aligned}
\tag{149}
$$

which means that $\langle \boldsymbol{w}_i^{(t+1)}, \boldsymbol{M}_j \rangle \leq \langle \boldsymbol{w}_i^{(t)}, \boldsymbol{M}_j \rangle$. This in fact gives $\mathfrak{F}_j^{(t+1)} \leq \mathfrak{F}_j^{(t)}$, so that (3) is proven.

Now for (4), we need to induct as follows: for $t \leq T_j' := \frac{d \log d}{\eta \log \log d}$ which is the specific iteration when $\mathfrak{F}_j^{(t)} \geq G_2 \tau \log d$, where $G_2$ is defined in Lemma E.3. The induction of (4) follows from similar proof in Theorem D.1. After $T_j'$, we discuss as follows

- When $t \in \left[ T_j', \frac{d^{1.49}}{\eta} \right]$, from above calculations, for each $i' \in \mathcal{M}_j$, we have

$$\frac{|\langle \boldsymbol{w}_i^{(t+1)}, \boldsymbol{M}_j \rangle|}{|\langle \boldsymbol{w}_{i'}^{(t+1)}, \boldsymbol{M}_j \rangle|} = \frac{|\langle \boldsymbol{w}_i^{(t)}, \boldsymbol{M}_j \rangle|(1 - \eta\lambda) + \eta\Psi_{i,j}^{(t)} \pm \mathcal{O}\left(\frac{\sqrt{\Xi_2}}{t\sqrt{d}}\right)\|\boldsymbol{w}_i^{(t)}\|_2}{|\langle \boldsymbol{w}_{i'}^{(t)}, \boldsymbol{M}_j \rangle|(1 - \eta\lambda) + \eta\Psi_{i',j}^{(t)} \pm \mathcal{O}\left(\frac{\sqrt{\Xi_2}}{t\sqrt{d}}\right)\|\boldsymbol{w}_{i'}^{(t)}\|_2}. \tag{150}$$

On one hand, for those $i' \in \mathcal{M}_j$ such that $|\langle \boldsymbol{w}_{i'}^{(t)}, \boldsymbol{M}_j \rangle| \leq \boldsymbol{b}_i^{(t)}\Xi_2^2 \leq \mathcal{O}\left(\frac{\Xi_2^2}{\sqrt{d}}\|\boldsymbol{w}_i^{(t)}\|_2\right)$, we can safely get $\left|\langle \boldsymbol{w}_i^{(t+1)}, \boldsymbol{M}_j \rangle\right| \gg \left|\langle \boldsymbol{w}_{i'}^{(t+1)}, \boldsymbol{M}_j \rangle\right|$. On the other hand, if $\left|\langle \boldsymbol{w}_{i'}^{(t)}, \boldsymbol{M}_j \rangle\right| \geq \boldsymbol{b}_i^{(t)}\Xi_2^2$, then we have

$$\left| \frac{\Psi_{i,j}^{(t)}}{\langle \boldsymbol{w}_i^{(t)}, \boldsymbol{M}_j \rangle} - \frac{\Psi_{i',j}^{(t)}}{\langle \boldsymbol{w}_i^{(t)}, \boldsymbol{M}_j \rangle} \right| = \frac{O(\frac{\boldsymbol{b}_i^{(t)}}{d^2})}{\langle \boldsymbol{w}_{i'}^{(t)}, \boldsymbol{M}_j \rangle} \leq O(\frac{1}{d^2\Xi_2^2}) \leq O\left(\frac{\Xi_2}{t\sqrt{d}\eta}\boldsymbol{b}_i^{(t)}\right). \tag{151}$$

Thus by letting $\widetilde{\Psi}_j := \frac{\Psi_{i,j}^{(t)}}{\langle \boldsymbol{w}_i^{(t)}, \boldsymbol{M}_j \rangle}$, then

$$\frac{\left|\langle \boldsymbol{w}_i^{(t+1)}, \boldsymbol{M}_j \rangle\right|}{\left|\langle \boldsymbol{w}_{i'}^{(t+1)}, \boldsymbol{M}_j \rangle\right|} = \frac{\left|\langle \boldsymbol{w}_i^{(t)}, \boldsymbol{M}_j \rangle\right|(1 + \eta\widetilde{\Psi}_j^{(t)} - \eta\lambda) \pm O\left(\frac{\Xi_2}{t\sqrt{d}}\right)\|\boldsymbol{w}_i^{(t)}\|_2}{\left|\langle \boldsymbol{w}_{i'}^{(t)}, \boldsymbol{M}_j \rangle\right|(1 + \eta\widetilde{\Psi}_j^{(t)} - \eta\lambda) \pm O\left(\frac{\Xi_2}{t\sqrt{d}}\right)\|\boldsymbol{w}_{i'}^{(t)}\|_2}. \tag{152}$$

Since at iteration $t \in \left[ T_j', \frac{d^{1.49}}{\eta} \right]$, it is easy to obtain that $\left| \widetilde{\Psi}_j^{(t)} - \lambda \right| \leq O\left(\frac{\Xi_2}{\eta t}\right)$.

Thus we have

$$\begin{aligned}
&\frac{\left|\langle \boldsymbol{w}_i^{(t+1)}, \boldsymbol{M}_j \rangle\right|}{\left|\langle \boldsymbol{w}_{i'}^{(t+1)}, \boldsymbol{M}_j \rangle\right|} \\
&\geq \frac{\left|\langle \boldsymbol{w}_i^{(t)}, \boldsymbol{M}_j \rangle\right|(1 + \eta(\widetilde{\Psi}_j^{(t)} - \lambda)(1 - \frac{\Xi_2^2}{\sqrt{d}}))}{\left|\langle \boldsymbol{w}_{i'}^{(t)}, \boldsymbol{M}_j \rangle\right|(1 + \eta(\widetilde{\Psi}_j^{(t)} - \lambda)(1 + \frac{\Xi_2}{\sqrt{d}}))} \\
&\geq \left( 1 + \eta(\widetilde{\Psi}_j^{(t)} - \lambda)(1 - \frac{\Xi_2^2}{\sqrt{d}}) - \eta(\widetilde{\Psi}_j^{(t)} - \lambda)(1 + \frac{\Xi_2}{\sqrt{d}}) \right) \cdot \frac{\left|\langle \boldsymbol{w}_i^{(t)}, \boldsymbol{M}_j \rangle\right|}{\left|\langle \boldsymbol{w}_{i'}^{(t)}, \boldsymbol{M}_j \rangle\right|} \\
&\geq \left( 1 - \eta(\widetilde{\Psi}_j^{(t)} - \lambda)(\frac{\Xi_2^2}{\sqrt{d}}) \right) \cdot \frac{\left|\langle \boldsymbol{w}_i^{(t)}, \boldsymbol{M}_j \rangle\right|}{\left|\langle \boldsymbol{w}_{i'}^{(t)}, \boldsymbol{M}_j \rangle\right|} \\
&\geq \left( 1 - \frac{\Xi_2^2}{t\sqrt{d}} \right) \cdot \frac{\left|\langle \boldsymbol{w}_i^{(t)}, \boldsymbol{M}_j \rangle\right|}{\left|\langle \boldsymbol{w}_{i'}^{(t)}, \boldsymbol{M}_j \rangle\right|} \\
&\geq \prod_{t'=T_j'}^{t-1} \left( 1 - O\left(\frac{\Xi_2^2}{t'\sqrt{d}}\right) \right) \cdot \frac{\left|\langle \boldsymbol{w}_i^{(T_j')}, \boldsymbol{M}_j \rangle\right|}{\left|\langle \boldsymbol{w}_{i'}^{(T_j')}, \boldsymbol{M}_j \rangle\right|} \geq \Omega(1).
\end{aligned} \tag{153}$$

where in the last inequality we have used our Theorem E.1 at $T_j'$

- The proof for iterations $t \in \left[\frac{d^{1.49}}{\eta}, \frac{d^{1.99}}{\eta}\right]$ is largely similar to the above. The only difference here is that we rely on a slightly different comparison here: Indeed, we have

$$\frac{\left|\langle \boldsymbol{w}_i^{(t+1)}, \boldsymbol{M}_j \rangle\right|}{\left|\langle \boldsymbol{w}_{i'}^{(t+1)}, \boldsymbol{M}_j \rangle\right|} = \frac{\left|\langle \boldsymbol{w}_i^{(t)}, \boldsymbol{M}_j \rangle\right| (1 + \eta \widetilde{\Psi}_j^{(t)} - \eta\lambda) \pm O\left(\frac{\Xi_2}{d^2}\right) \|\boldsymbol{w}_i^{(t)}\|_2}{\left|\langle \boldsymbol{w}_{i'}^{(t)}, \boldsymbol{M}_j \rangle\right| (1 + \eta \widetilde{\Psi}_j^{(t)} - \eta\lambda) \pm O\left(\frac{\Xi_2}{d^2}\right) \|\boldsymbol{w}_{i'}^{(t)}\|_2}. \tag{154}$$

Here we can use similar techniques as above to require $\left|\widetilde{\Psi}_j^{(t)} - \lambda\right| \leq O\left(\frac{\Xi_2}{t\eta}\right)$ Now we also have

$$\begin{aligned}
\frac{\left|\langle \boldsymbol{w}_i^{(t+1)}, \boldsymbol{M}_j \rangle\right|}{\left|\langle \boldsymbol{w}_{i'}^{(t+1)}, \boldsymbol{M}_j \rangle\right|} &\geq \frac{\left|\langle \boldsymbol{w}_i^{(t)}, \boldsymbol{M}_j \rangle\right| (1 + \eta(\widetilde{\Psi}_j^{(t)} - \lambda)(1 - \frac{\Xi_2^2}{\sqrt{d}}))}{\left|\langle \boldsymbol{w}_{i'}^{(t)}, \boldsymbol{M}_j \rangle\right| (1 + \eta(\widetilde{\Psi}_j^{(t)} - \lambda)(1 + \frac{\Xi_2^2}{\sqrt{d}}))} \\
&\geq \left(1 - \frac{\Xi_2^2}{t\sqrt{d}}\right) \cdot \frac{\left|\langle \boldsymbol{w}_i^{(t)}, \boldsymbol{M}_j \rangle\right|}{\left|\langle \boldsymbol{w}_{i'}^{(t)}, \boldsymbol{M}_j \rangle\right|} \\
&\geq \prod_{t'=d^{1.49}/\eta}^{t-1} \left(1 - \frac{\Xi_2^2}{t' d^{0.01}}\right) \cdot \frac{\left|\langle \boldsymbol{w}_i^{(d^{1.49}/\eta)}, \boldsymbol{M}_j \rangle\right|}{\left|\langle \boldsymbol{w}_{i'}^{(d^{1.49}/\eta)}, \boldsymbol{M}_j \rangle\right|} \geq \Omega(1).
\end{aligned} \tag{155}$$

Now (4) are proven. (7) is an immediate result of our update scheme. $\qquad \square$

### E.4 PROOF OF THEOREM 3.1

The first part proves the convergence of the loss function. The second part is a further extension of Theorem E.1.

*Proof of Theorem 3.1.* We start with the proof of convergence ((11) in Theorem 3.1).

Denote $w^{(t)} = (w_1^{(t)}, \ldots, w_m^{(t)})$, since our update is

$$w^{(t+1)} = w^{(t)} - \nabla_w L_{\text{aug}}(f_t) + \frac{1}{\text{poly}(d_1)}, \tag{156}$$

we have

$$\begin{aligned}
&\eta \langle \nabla_w L_{\text{aug}}(f_t), w^{(t)} - \theta^\star \rangle \\
={}& \frac{\eta^2}{2} \|\nabla_w L_{\text{aug}}(f_t)\|_F^2 + \frac{1}{2}\|w^{(t)} - \theta^\star\|_F^2 - \frac{1}{2}\|w^{(t+1)} - \theta^\star\|_F^2 + \frac{\eta^2}{\text{poly}(d_1)} \\
\leq{}& \eta^2 \, \text{poly}(d) + \frac{1}{2}\|w^{(t)} - \theta^\star\|_F^2 - \frac{1}{2}\|w^{(t+1)} - \theta^\star\|_F^2 + \frac{\eta^2}{\text{poly}(d_1)},
\end{aligned} \tag{157}$$

where the inequality comes from

$$\|\nabla_w L_{\text{aug}}(f_t)\|_F^2 = \sum_{i=1}^m \|\nabla_{\boldsymbol{w}_i} L_{\text{aug}}(f_t)\|^2. \tag{158}$$

Each term is $O(1)$, and since $m = \text{poly}(d)$, the overall complexity is $\text{poly}(d)$.

Now we will use the tools from online learning to obtain a loss guarantee: define a pseudo objective for parameter $\theta$

$$\begin{aligned}
\widetilde{L}_{\text{aug}_t}(\theta) &:= \widetilde{L}(f_{t,\theta}, f_t) + \frac{\lambda}{2} \sum_{i \in [m]} \|\theta_i\|_2^2 \\
&= \mathbb{E}\left[-\tau \log\left(\frac{e^{\langle f_{t,\theta}(\boldsymbol{X}_n), f_t(\boldsymbol{Y}_n)\rangle/\tau}}{\sum_{\boldsymbol{X} \in \mathfrak{B}} e^{\langle f_{t,\theta}(\boldsymbol{X}_n), f_t(\boldsymbol{X})\rangle/\tau}}\right)\right] + \frac{\lambda}{2} \sum_{i \in [m]} \|\theta_i\|_2^2.
\end{aligned} \tag{159}$$

Which is a convex function over $\theta$ since it is linear in $\theta$ (for a fixed $f_t$, we can consider $\widetilde{L}(f_{t,\theta}, f_t)$ to be convex with respect to $\theta$, because $f_{t,\theta}(x)$ is linear, and softmax + log is a convex composition; the regularization term is convex).

Moreover, we have

$$\widetilde{L}_{\mathrm{aug}_t}(w^{(t)}) = L_{\mathrm{aug}}(f_t), \tag{160}$$

and

$$\nabla_{\theta_i} \widetilde{L}_{\mathrm{aug}_t}(\boldsymbol{w}_i^{(t)}) = \nabla_{\boldsymbol{w}_i} L_{\mathrm{aug}}(f_t). \tag{161}$$

Thus we have

$$
\begin{aligned}
&\eta \langle \nabla_w L_{\mathrm{aug}}(f_t), w^{(t)} - \theta^\star \rangle \\
=& \eta \langle \nabla_\theta \widetilde{L}_{\mathrm{aug}_t}(w^{(t)}), w^{(t)} - \theta^\star \rangle \\
\overset{①}{\geq}& \widetilde{L}_{\mathrm{aug}_t}(w^{(t)}) - \widetilde{L}_{\mathrm{aug}_t}(\theta^\star) \\
\geq& \widetilde{L}_{\mathrm{aug}_t}(w^{(t)}) - \mathbb{E}\left[ -\tau \log\left( \frac{e^{\langle f_{t,\theta^\star}(\boldsymbol{X}_n), f_t(\boldsymbol{Y}_n) \rangle / \tau}}{\sum_{\boldsymbol{X} \in \mathfrak{B}} e^{\langle f_{t,\theta^\star}(\boldsymbol{X}_n), f_t(\boldsymbol{X}) \rangle / \tau}} \right) \right] - \frac{\lambda}{2} \sum_{i \in [m]} \|\theta_i^\star\|_2^2 \\
\overset{②}{\geq}& \widetilde{L}_{\mathrm{aug}_t}(w^{(t)}) - O\left(\frac{1}{\log d}\right) - \sum_{i \in [m]} O(\lambda \|\theta_i^\star\|_2^2) \\
\geq& L_{\mathrm{aug}}(f_t) - O\left(\frac{1}{\log d}\right).
\end{aligned}
\tag{162}
$$

① is because the surrogate objective function $\widetilde{L}_{\mathrm{aug}_t}$ is a convex function with respect to $\theta$, so we can use a first-order convex lower bound: $f(\theta) - f(\theta') \leq \langle \nabla f(\theta), \theta - \theta' \rangle$. ② is because $\sum_{i \in [m]} \lambda \|\theta_i^\star\|_2^2 = \sum_{j \in [d]} \sum_{i \in \mathcal{M}_j^\star} \lambda \|\theta_i^\star\|_2^2 = \sum_{j \in [d]} \sum_{i \in \mathcal{M}_j^\star} \lambda \frac{\tau \kappa^2}{|\mathcal{M}_j^\star|^2} = \sum_{j \in [d]} \lambda \frac{\tau \kappa^2}{|\mathcal{M}_j^\star|} = \frac{\lambda \tau \kappa^2}{|\mathcal{M}_j^\star|}$

Now choosing $\kappa = \Theta(\Xi_2) \leq \frac{1}{\lambda d}$ (so that $\sum_{i \in [m]} \lambda \|\theta_i^\star\|_2^2 < \frac{1}{\log d}$), and by a telescoping summation, we have

$$
\begin{aligned}
\frac{1}{T} \sum_{t=T_3}^{T_3+T-1} \left( L_{\mathrm{aug}}(f_t) - O\left(\frac{1}{\log d}\right) \right) &\leq \frac{1}{T} \sum_{t=T_3}^{T_3+T-1} \eta \langle \nabla_w L_{\mathrm{aug}}(f_t), w^{(t)} - \theta^\star \rangle \\
&\leq \frac{O(\|w^{(T_3)} - \theta^\star\|_F^2)}{T\eta} \\
&= \frac{O\left(\|w^{(T_3)}\|_F^2 + \|\theta^\star\|_F^2 - 2\operatorname{Tr}((w^{(T_3)})^\top \theta^\star)\right)}{T\eta} \\
&\leq \frac{O\left(\|w^{(T_3)}\|_F^2 + \|\theta^\star\|_F^2\right)}{T\eta} \\
&\leq \frac{O\left(m\|\boldsymbol{w}_i^{(T_3)}\|_2^2\right)}{T\eta} \\
&\leq O\left(\frac{m\Xi_2}{T\eta}\right).
\end{aligned}
\tag{163}
$$

Since $T\eta \geq m\Xi_2^{10}$, this proves the claim.

For (12) in Theorem 3.1, we have

$$\boldsymbol{w}_i^{(t)} = \sum_{j \in \mathcal{N}_i, \, j \in [d]} \langle \boldsymbol{w}_i^{(t)}, \boldsymbol{M}_j \rangle \boldsymbol{M}_j + \sum_{j \notin \mathcal{N}_i, \, j \in [d]} \langle \boldsymbol{w}_i^{(t)}, \boldsymbol{M}_j \rangle \boldsymbol{M}_j + \sum_{j \in [d_1] \setminus [d]} \langle \boldsymbol{w}_i^{(t)}, \boldsymbol{M}_j^\perp \rangle \boldsymbol{M}_j^\perp$$

$$\leq \sum_{j \in \mathcal{N}_i, \, j \in [d]} \langle \boldsymbol{w}_i^{(t)}, \boldsymbol{M}_j \rangle \boldsymbol{M}_j + \sum_{j \notin \mathcal{N}_i, \, j \in [d]} O\left( \frac{\epsilon_j}{\epsilon_{\max}} \frac{\|\boldsymbol{w}_i^{(t)}\|_2}{\sqrt{d}\, \Xi_2^5} \right) \boldsymbol{M}_j + \sum_{j \in [d_1] \setminus [d]} O\left( \frac{\|\boldsymbol{w}_i^{(t)}\|_2}{\sqrt{d_1}\, \Xi_2^5} \right) \boldsymbol{M}_j^\perp$$

$$= \sum_{j \in \mathcal{N}_i, \, j \in [d]} \alpha_{i,j} \boldsymbol{M}_j + \sum_{j \notin \mathcal{N}_i, \, j \in [d]} \alpha'_{i,j} \boldsymbol{M}_j + \sum_{j \in [d_1] \setminus [d]} \beta_{i,j} \boldsymbol{M}_j^\perp.$$

$$(164)$$

From Lemma B.2(c), we know that for each $j \in [d]$, there is at least one neuron that can fully learn the feature $\boldsymbol{M}_j$, and at most $\Xi_2$ neurons can learn the feature $\boldsymbol{M}_j$. Combining this with Theorem E.1(c):

$$\sum_{i \in \mathcal{M}_j} \langle \boldsymbol{w}_i^{(t)}, \boldsymbol{M}_j \rangle^2 = \Theta\left( \left( \frac{\epsilon_j}{\epsilon_{\max}} \right)^2 \tau \log^3 d \right), \tag{165}$$

we can conclude that the range of $\langle \boldsymbol{w}_i^{(t)}, \boldsymbol{M}_j \rangle$ is $[\frac{\epsilon_j}{\epsilon_{\max}} \frac{\tau}{\Xi_2}, \frac{\epsilon_j}{\epsilon_{\max}} \tau]$, and hence the range of $\alpha_{i,j}$ is $[\frac{\epsilon_j}{\epsilon_{\max}} \frac{\tau}{\Xi_2}, \frac{\epsilon_j}{\epsilon_{\max}} \tau]$. Furthermore, from Theorem E.1 (e) and (f), we can obtain that $\alpha'_{i,j} \leq o(\frac{\epsilon_j}{\epsilon_{\max}} \frac{1}{\sqrt{d}})$ and $\beta_{i,j} \leq o(\frac{1}{\sqrt{d_1}})$ respectively.

Next, we compute the upper bound of $|\mathcal{N}_i|$. As a first step, we calculate the expectation of $|\mathcal{N}_i|$.

$$\mathbb{E}[|\mathcal{N}_i|] = \frac{1}{m} \sum_{i=1}^{m} |\mathcal{N}_i| = \frac{1}{m} \sum_{j=1}^{d} |\mathcal{M}_j| \leq \frac{1}{m} \cdot d \cdot O(d^{\omega_2})$$

$$= \frac{1}{m} \cdot O(d^{1+\omega_2}) = \frac{O(d^{1+\omega_2})}{d^{C_m}} = O\left( d^{1+\omega_2 - C_m} \right) \tag{166}$$

$$= O\left( d^{1 - \left( \frac{\epsilon_{\min}}{\epsilon_{\max}} \right)^2 \cdot (1-\gamma)} \right).$$

Fix a neuron $i$, we have: $\mu_i := \mathbb{E}[|\mathcal{N}_i|]$. By Bernstein's inequality,

$$\Pr[\,|\mathcal{N}_i| \geq \mu_i + t\,] \leq \exp\left( -\frac{t^2}{2(\mu_i + t/3)} \right), \qquad t \geq 0. \tag{167}$$

We set $t = 3\big( \sqrt{\mu_i L} + L \big)$ and plug this into the inequality above. Then we obtain

$$\Pr\Big[ |\mathcal{N}_i| \geq \mu_i + 3\big( \sqrt{\mu_i L} + L \big) \Big] \leq e^{-L}. \tag{168}$$

Hence, for any constant $c > 0$, taking $L = c \log d$ yields

$$\Pr\Big[ |\mathcal{N}_i| \leq \mu_i + 3(\sqrt{\mu_i c \log d} + c \log d) \Big] \geq 1 - d^{-c}. \tag{169}$$

Next, we apply the union bound. For the event

$$A_i := \Big\{ |\mathcal{N}_i| \leq \mu_i + 3(\sqrt{\mu_i L} + L) \Big\}, \tag{170}$$

the union bound gives

$$\Pr\left[ \bigcap_{i=1}^{m} A_i \right] \geq 1 - \sum_{i=1}^{m} \Pr(A_i^c) \geq 1 - m e^{-L}. \tag{171}$$

Taking $L = c \log(md)$, we obtain

$$\Pr\Big[ \forall i \in [m], \, |\mathcal{N}_i| \leq \mu_i + 3\big( \sqrt{\mu_i c \log(md)} + c \log(md) \big) \Big] \geq 1 - (md)^{-c}. \tag{172}$$

We know $\mu_i \gg \log(md)$, so we have

$$
\begin{aligned}
|\mathcal{N}_i| &= \mu_i \left(1 \pm O\left(\sqrt{\tfrac{\log(md)}{\mu_i}}\right)\right) \\
&= \mu_i \left(1 \pm o(1)\right) \\
&\leq O\left(d^{1-\left(\frac{\epsilon_{\min}}{\epsilon_{\max}}\right)^2 \cdot (1-\gamma)}\right) \quad \text{with probability at least } 1 - (md)^{-c}
\end{aligned}
\tag{173}
$$

Finally, for each dictionary atom $M_j$, there are at least $\Omega(d^{\omega_1})$ neurons $i \in [m]$ such that $\mathcal{N}_i = \{j\}$. From Lemma B.2 (c), we recall that $|\mathcal{M}_j^\star| \geq \Omega(d^{\omega_1})$. Moreover, if a neuron belongs to $\mathcal{M}_j^\star$, then it cannot belong to $\mathcal{M}_{j'}$.

For (12) in Theorem 3.1, our proof is complete. $\qquad\square$

# F  THEOREM F.1

From Lemma B.2(c), we know that for each $j \in [d]$, there is at least one neuron that can fully learn the minority feature $M_{j^\star}$. When we prune out the lucky neurons that learn these minority features during the forward pass, the network will force the lucky neurons to further strengthen their feature learning ability on the minority features during the backward pass.

After magnitude pruning, neurons encoding a specific minority feature are removed. Pruning these lucky neurons reduces $\text{sim}_{f_\theta}(X_n, Y_n)$ during the forward pass. The decrease in similarity reduces the positive logit $\ell'_{p,\theta^{(t)}_{\text{mask}}}$, which in turn increases the gradient of the loss function, thereby encouraging these lucky neurons to further enhance their learning ability on the minority features.

Fix one specific minority feature $M_{j^\star}$, and let $\mathcal{M}_{j^\star}^\star \subseteq [m]$ denote the subset of neurons primarily aligned with it, with $|\mathcal{M}_{j^\star}^\star| = n$. For a pruning rate $\alpha \in [1/m,\, n/m]$, the number of pruned neurons is $\alpha m \leq n$. Let $\mathcal{P} \subseteq \mathcal{M}_{j^\star}^\star$ be the pruned set with $|\mathcal{P}| = \alpha m$.

## F.1  THEOREM F.1

**Theorem F.1** (Feature Dynamics After Pruning). *Starting from the pruning stage $T_4$ with pruning ratio $\alpha$, the following statements hold.*

*(a) When $i^\star \in \mathcal{M}_{j^\star}^\star$, we have*

$$
\langle w_{i^\star}^{(t+1)}, M_{j^\star} \rangle \geq \left(1 - \eta\lambda + \eta\epsilon_{j^\star} \frac{C_z \log\log d}{d} \left(\Theta\left(\tfrac{1}{\text{polylog}(d)}\right) + \Omega\left(\tfrac{\alpha m \epsilon_{j^\star} \log\log d}{d \Xi_2^2}\right)\right)\right) \langle w_{i^\star}^{(t)}, M_{j^\star} \rangle.
\tag{174}
$$

*(b) When $i \notin \mathcal{M}_{j^\star}^\star$ and $j \neq j^\star$, we have*

$$
\begin{aligned}
\langle w_i^{(t+1)}, M_j \rangle \leq \bigg(&1 - \eta\lambda + \eta\epsilon_j \frac{C_z \log\log d}{d} \bigg(\Theta\left(\tfrac{1}{\text{polylog}(d)}\right) \\
&+ \epsilon_{j^\star} \frac{\log\log d}{d}\left(\Theta\left(\tfrac{1}{\text{polylog}(d)}\right) + O\left(\tfrac{\alpha m \epsilon_{j^\star} \log\log d}{d\Xi_2^2}\right)\right)\bigg)\bigg) \langle w_i^{(t)}, M_j \rangle.
\end{aligned}
\tag{175}
$$

*(c) For each neuron $i \in \mathcal{P}$ and $t \in [T_4, T_5]$, contrastive learning learns the following decomposition:*

$$
w_i^{(t)} = \alpha_{i,j^\star} M_{j^\star} + \sum_{j \notin \mathcal{N}_i} \alpha'_{i,j} M_j + \sum_{j \in [d_1] \setminus [d]} \beta_{i,j} M_j^\perp,
\tag{176}
$$

*where*

$$
\alpha_{i,j^\star} \in \left[\frac{\tau}{\Xi_2}, \tau\right], \quad \alpha'_{i,j} \leq o\left(\left(1 + \frac{1}{d}\right)\frac{1}{\sqrt{d}}\right) \|w_i^{(t)}\|_2, \quad |\beta_{i,j}| \leq o\left(\frac{1}{\sqrt{d_1}}\right) \|w_i^{(t)}\|_2.
\tag{177}
$$

## F.2 USEFUL LEMMAS

**Lemma F.1** (Expected values of neuron activations after $T_4$). *From $T_4$ onward, the following results hold:*

*(a) For positive pair,*

$$\mathbb{E}\left[\sum_{i \in \mathcal{P}} h_i(\boldsymbol{X}_n) h_i(\boldsymbol{Y}_n)\right] \geq \Omega\left(\alpha m \frac{\tau^2}{\Xi_2^2} \epsilon_{j^\star} \frac{\log \log d}{d}\right). \tag{178}$$

*(b) For negative pair,*

$$\mathbb{E}\left[\sum_{i \in \mathcal{P}} h_i(\boldsymbol{X}_n) h_i(\boldsymbol{X}_{n,s})\right] = 0. \tag{179}$$

*(c) For negative pair,*

$$\mathbb{E}\left[h_{i,t}(\boldsymbol{X}_{n,s}) \langle \nabla_{\boldsymbol{w}_i} h_i(\boldsymbol{X}_n), \boldsymbol{M}_{j^\star} \rangle\right] = 0. \tag{180}$$

**Lemma F.2** (Effect of Pruning on Positive Logit Weight). *At the pruning stage, for the data following distribution $D_1$, the post-pruning positive logit $\ell'_{p, \boldsymbol{\theta}^{(t)}_{\mathrm{mask}}}$ satisfies*

$$\mathbb{E}\left[1 - \ell'_{p, \boldsymbol{\theta}^{(t)}_{\mathrm{mask}}}\right] \geq \Theta\left(\frac{1}{\tau}\right) + \Omega\left(\frac{\alpha m}{\Xi_2^2} \epsilon_{j^\star} \frac{\log \log d}{d}\right). \tag{181}$$

**Lemma F.3** (Positive gradient). *Let $h_{i,t}(\cdot)$ denote the $i$-th neuron at iteration $t \leq T_1$ (so that $\boldsymbol{b}_i^{(t)} = 0$). Then the following hold:*

*(a) For each $j \in [d]$,*

$$\mathbb{E}[h_{i,t}(\boldsymbol{Y}_n) \langle \nabla_{\boldsymbol{w}_i} h_{i,t}(\boldsymbol{X}_n), \boldsymbol{M}_j \rangle] = \frac{1}{L^2} \langle \boldsymbol{w}_i^{(t)}, \boldsymbol{M}_j \rangle \mathbb{E}\left[\hat{z}_{n,j}^+ \hat{z}_{n,j}\right]. \tag{182}$$

*(b) For each $j \in [d_1] \setminus [d]$,*

$$\mathbb{E}\left[h_{i,t}(\boldsymbol{Y}_n) \langle \nabla_{\boldsymbol{w}_i} h_{i,t}(\boldsymbol{X}_n), \boldsymbol{M}_j^\perp \rangle\right] = 0. \tag{183}$$

## F.3 PROOF OF THEOREM F.1

Overview of the proof: first, the data can be divided into two parts: the samples that contain $\boldsymbol{M}_{j^\star}$ and those that do not. The former follow distribution $D_1$, while the latter follow distribution $D_2$. Next, let us examine $\ell'_{p, \boldsymbol{\theta}^{(t)}_{\mathrm{mask}}}$. The values of $\ell'_{p, \boldsymbol{\theta}^{(t)}_{\mathrm{mask}}}$ differ depending on the distribution: for samples from $D_1$, we have $\ell'_{p, \boldsymbol{\theta}^{(t)}_{\mathrm{mask}}} = 1 - \Theta(\frac{1}{\tau}) - \Omega(\frac{\alpha m}{\Xi_2^2} \epsilon_{j^\star} \frac{\log \log d}{d})$, whereas for samples from $D_2$, $\ell'_{p, \boldsymbol{\theta}^{(t)}_{\mathrm{mask}}} = 1 - \Theta(\frac{1}{\tau})$. Since the latter do not contain $\boldsymbol{M}_{j^\star}$, pruning does not affect them.

*Proof of Theorem F.1.* For any neuron $i^\star \in \mathcal{P}$ we have

$$\begin{aligned}
&\langle \boldsymbol{w}_{i^\star}^{(t+1)}, \boldsymbol{M}_{j^\star} \rangle \\
=&\langle \boldsymbol{w}_{i^\star}^{(t)}, \boldsymbol{M}_{j^\star} \rangle - \eta \langle \nabla_{w_{i^\star}} L_{\mathrm{aug}}(f_t), \boldsymbol{M}_{j^\star} \rangle \pm \frac{\|\boldsymbol{w}_{i^\star}^{(t)}\|_2}{\mathrm{poly}(d)} \\
=&(1 - \eta\lambda)\langle \boldsymbol{w}_{i^\star}^{(t)}, \boldsymbol{M}_{j^\star} \rangle \\
&+ \eta \mathbb{E}_{\boldsymbol{X}_n, \boldsymbol{Y}_n}\left[(1 - \ell'_{p, \boldsymbol{\theta}^{(t)}_{\mathrm{mask}}}(\boldsymbol{X}_n, \mathfrak{B})) \cdot h_{i^\star,t}(\boldsymbol{Y}_n) \langle \nabla_{\boldsymbol{w}_{i^\star}} h_{i^\star}(\boldsymbol{X}_n), \boldsymbol{M}_{j^\star} \rangle\right] \\
&- \eta \sum_{\boldsymbol{X}_{n,s} \in \mathfrak{N}} \mathbb{E}\left[\ell'_{s,t}(\boldsymbol{X}_n, \mathfrak{B}) h_{i^\star,t}(\boldsymbol{X}_{n,s}) \langle \nabla_{\boldsymbol{w}_{i^\star}} h_{i^\star}(\boldsymbol{X}_n), \boldsymbol{M}_{j^\star} \rangle\right] \pm \frac{\|\boldsymbol{w}_{i^\star}^{(t)}\|_2}{\mathrm{poly}(d)}
\end{aligned} \tag{184}$$

At stage $T_4$, pruning is applied. We regard $\ell'_{p, \boldsymbol{\theta}^{(t)}_{\mathrm{mask}}}$ and $\ell'_{s, \boldsymbol{\theta}^{(t)}_{\mathrm{mask}}}$ as fixed, and by combining Lemma F.1(c) with the law of total probability, we obtain

$$
\begin{aligned}
&\langle \boldsymbol{w}_{i^\star}^{(t+1)}, \boldsymbol{M}_{j^\star} \rangle \\
&= (1-\eta\lambda)\langle \boldsymbol{w}_{i^\star}^{(t)}, \boldsymbol{M}_{j^\star} \rangle \\
&+ \eta\, \mathbb{E}_{\boldsymbol{X}_n, \boldsymbol{Y}_n}\Big[(1-\ell'_{p,\boldsymbol{\theta}_{\mathrm{mask}}^{(t)}})\Big]\mathbb{E}_{\boldsymbol{X}_n, \boldsymbol{Y}_n}\Big[h_{i^\star,t}(\boldsymbol{Y}_n)\,\langle \nabla_{\boldsymbol{w}_{i^\star}} h_{i^\star}(\boldsymbol{X}_n), \boldsymbol{M}_{j^\star} \rangle\Big] \\
&= (1-\eta\lambda)\langle \boldsymbol{w}_{i^\star}^{(t)}, \boldsymbol{M}_{j^\star} \rangle \\
&+ \eta\, \mathbb{E}_{\boldsymbol{X}_n, \boldsymbol{Y}_n \sim D_1}\Big[(1-\ell'_{p,\boldsymbol{\theta}_{\mathrm{mask}}^{(t)}})\Big]\mathbb{E}_{\boldsymbol{X}_n, \boldsymbol{Y}_n \sim D_1}\Big[h_{i^\star,t}(\boldsymbol{Y}_n)\,\langle \nabla_{\boldsymbol{w}_{i^\star}} h_{i^\star}(\boldsymbol{X}_n), \boldsymbol{M}_{j^\star} \rangle\Big]\cdot \mathbb{P}_{\boldsymbol{X}_n, \boldsymbol{Y}_n \sim D_1} \\
&+ \eta\, \mathbb{E}_{\boldsymbol{X}_n, \boldsymbol{Y}_n \sim D_2}\Big[(1-\ell'_{p,\boldsymbol{\theta}_{\mathrm{mask}}^{(t)}})\Big]\mathbb{E}_{\boldsymbol{X}_n, \boldsymbol{Y}_n \sim D_2}\Big[h_{i^\star,t}(\boldsymbol{Y}_n)\,\langle \nabla_{\boldsymbol{w}_{i^\star}} h_{i^\star}(\boldsymbol{X}_n), \boldsymbol{M}_{j^\star} \rangle\Big]\cdot \mathbb{P}_{\boldsymbol{X}_n, \boldsymbol{Y}_n \sim D_2}
\end{aligned}
\tag{185}
$$

Combining Lemma F.3(a) with (181) in Lemma F.2, we obtain

$$
\begin{aligned}
&\langle \boldsymbol{w}_{i^\star}^{(t+1)}, \boldsymbol{M}_{j^\star} \rangle \\
&= (1-\eta\lambda)\langle \boldsymbol{w}_{i^\star}^{(t)}, \boldsymbol{M}_{j^\star} \rangle \\
&+ \eta\, \mathbb{E}_{\boldsymbol{X}_n, \boldsymbol{Y}_n \sim D_1}\Big[(1-\ell'_{p,\boldsymbol{\theta}_{\mathrm{mask}}^{(t)}})\Big]\mathbb{E}_{\boldsymbol{X}_n, \boldsymbol{Y}_n \sim D_1}\Big[\frac{1}{L^2}\langle \boldsymbol{w}_i^{(t)}, \boldsymbol{M}_j \rangle \,\mathbb{E}\big[\hat{\boldsymbol{z}}_{n,j^\star}^{+} \hat{\boldsymbol{z}}_{n,j^\star}\big]\rangle\Big]\cdot \mathbb{P}_{\boldsymbol{X}_n, \boldsymbol{Y}_n \sim D_1} \\
&+ \eta\, \mathbb{E}_{\boldsymbol{X}_n, \boldsymbol{Y}_n \sim D_2}\Big[(1-\ell'_{p,\boldsymbol{\theta}_{\mathrm{mask}}^{(t)}})\Big]\mathbb{E}_{\boldsymbol{X}_n, \boldsymbol{Y}_n \sim D_2}\Big[\frac{1}{L^2}\langle \boldsymbol{w}_i^{(t)}, \boldsymbol{M}_j \rangle \,\mathbb{E}\big[\hat{\boldsymbol{z}}_{n,j^\star}^{+} \hat{\boldsymbol{z}}_{n,j^\star}\big]\Big]\cdot \mathbb{P}_{\boldsymbol{X}_n, \boldsymbol{Y}_n \sim D_2} \\
&= (1-\eta\lambda)\langle \boldsymbol{w}_{i^\star}^{(t)}, \boldsymbol{M}_{j^\star} \rangle \\
&+ \eta\left(\Theta\left(\frac{1}{\tau}\right) + \Omega\left(\frac{\alpha m}{\Xi_2^2}\,\epsilon_{j^\star}\,\frac{\log\log d}{d}\right)\right)\cdot \langle \boldsymbol{w}_i^{(t)}, \boldsymbol{M}_j \rangle \cdot \epsilon_{j^\star}\,\frac{\log\log d}{d} \\
&+ \eta\cdot\Theta\left(\frac{1}{\tau}\right)\cdot 0\cdot 1\cdot \langle \boldsymbol{w}_{i^\star}^{(t)}, \boldsymbol{M}_{j^\star} \rangle \\
&= (1-\eta\lambda)\langle \boldsymbol{w}_{i^\star}^{(t)}, \boldsymbol{M}_{j^\star} \rangle + \eta\left(\Theta\left(\frac{1}{\tau}\right) + \Omega\left(\frac{\alpha m}{\Xi_2^2}\,\epsilon_{j^\star}\,\frac{\log\log d}{d}\right)\right)\cdot \langle \boldsymbol{w}_{i^\star}^{(t)}, \boldsymbol{M}_{j^\star} \rangle \cdot \epsilon_{j^\star}\,\frac{\log\log d}{d}
\end{aligned}
\tag{186}
$$

Hence, the post-pruning one-step update along $\boldsymbol{M}_{j^\star}$ is

$$
\langle \boldsymbol{w}_{i^\star}^{(t+1)}, \boldsymbol{M}_{j^\star} \rangle \geq \left(1 - \eta\lambda + \eta\epsilon_{j^\star}\,\frac{C_z \log\log d}{d}\left(\Theta\left(\frac{1}{\mathrm{polylog}(d)}\right) + \Omega\left(\frac{\alpha m \epsilon_{j^\star} \log\log d}{d\Xi_2^2}\right)\right)\right)\langle \boldsymbol{w}_{i^\star}^{(t)}, \boldsymbol{M}_{j^\star} \rangle.
\tag{187}
$$

Similarly to (186), for any neuron $i \notin \mathcal{P}$, we have:

$$
\begin{aligned}
&\langle \boldsymbol{w}_i^{(t+1)}, \boldsymbol{M}_j \rangle \\
&= (1-\eta\lambda)\langle \boldsymbol{w}_i^{(t)}, \boldsymbol{M}_j \rangle \\
&\quad + \eta\, \mathbb{E}_{\boldsymbol{X}_n, \boldsymbol{Y}_n \sim D_1}\Big[(1-\ell'_{p,\boldsymbol{\theta}_{\mathrm{mask}}^{(t)}})\Big]\mathbb{E}_{\boldsymbol{X}_n, \boldsymbol{Y}_n \sim D_1}\Big[\frac{1}{L^2}\langle \boldsymbol{w}_i^{(t)}, \boldsymbol{M}_j \rangle \,\mathbb{E}\big[\hat{\boldsymbol{z}}_{n,j}^{+} \hat{\boldsymbol{z}}_{n,j}\big]\rangle\Big]\cdot \mathbb{P}_{\boldsymbol{X}_n, \boldsymbol{Y}_n \sim D_1} \\
&\quad + \eta\, \mathbb{E}_{\boldsymbol{X}_n, \boldsymbol{Y}_n \sim D_2}\Big[(1-\ell'_{p,\boldsymbol{\theta}_{\mathrm{mask}}^{(t)}})\Big]\mathbb{E}_{\boldsymbol{X}_n, \boldsymbol{Y}_n \sim D_2}\Big[\frac{1}{L^2}\langle \boldsymbol{w}_i^{(t)}, \boldsymbol{M}_j \rangle \,\mathbb{E}\big[\hat{\boldsymbol{z}}_{n,j}^{+} \hat{\boldsymbol{z}}_{n,j}\big]\Big]\cdot \mathbb{P}_{\boldsymbol{X}_n, \boldsymbol{Y}_n \sim D_2} \\
&= (1-\eta\lambda)\langle \boldsymbol{w}_i^{(t)}, \boldsymbol{M}_j \rangle \\
&\quad + \eta\left(\Theta\left(\frac{1}{\tau}\right) + \Omega\left(\frac{\alpha m}{\Xi_2^2}\,\epsilon_{j^\star}\,\frac{\log\log d}{d}\right)\right)\cdot \langle \boldsymbol{w}_i^{(t)}, \boldsymbol{M}_j \rangle \cdot \epsilon_j\,\frac{\log\log d}{d}\epsilon_{j^\star}\,\frac{\log\log d}{d} \\
&\quad + \eta\cdot\Theta\left(\frac{1}{\tau}\right)\cdot \epsilon_j\,\frac{\log\log d}{d}\cdot 1\cdot \langle \boldsymbol{w}_i^{(t)}, \boldsymbol{M}_j \rangle
\end{aligned}
\tag{188}
$$

Hence, the post-pruning one-step update along $M_j$ is

$$
\begin{aligned}
\langle \boldsymbol{w}_i^{(t+1)}, \boldsymbol{M}_j \rangle \leq \bigg( 1 - \eta\lambda + \eta\epsilon_j \, &\frac{C_z \log\log d}{d} \Big( \Theta\Big( \tfrac{1}{\mathrm{polylog}(d)} \Big) \\
&+ \epsilon_{j^\star} \frac{\log\log d}{d} \Big( \Theta\Big( \tfrac{1}{\mathrm{polylog}(d)} \Big) + O\Big( \tfrac{\alpha m \epsilon_{j^\star} \log\log d}{d \Xi_2^2} \Big) \Big) \Big) \bigg) \langle \boldsymbol{w}_i^{(t)}, \boldsymbol{M}_j \rangle .
\end{aligned}
\tag{189}
$$

The above constitutes the proof of Theorem F.1 regarding pruning. $\qquad\square$

### F.4 PROOF OF THEOREM 3.2

Theorem 3.2 (a) and (b) can be derived as simplifications of Theorem F.1 (a) and (b). Theorem 3.2 (c) coincides with Theorem F.1 (c). By taking the elapsed time $T = \frac{((\epsilon_{\max}/\epsilon_{j^\star})-1)d}{\eta\alpha\epsilon_{j^\star}^2 C_z \log\log d}$ and simplifying (a) and (b), then substituting into the conclusion of Theorem 3.1, the proof follows.

### F.5 PROOF OF LEMMA F.1:

*Proof of Lemma F.1:* The alignment with the target minority feature $\boldsymbol{M}_{j^\star}$ is $\langle \boldsymbol{w}_i, \boldsymbol{M}_{j^\star} \rangle$, and we have $|\langle \boldsymbol{w}_i, \boldsymbol{M}_{j^\star} \rangle| \geq \Omega(\frac{\tau}{\Xi_2})$ at $T_4$ (This is the conclusion of Theorem 3.1, which can be found in the second part of the proof of Theorem 3.1. For the positive pair $(\boldsymbol{X}_n, \boldsymbol{Y}_n)$, the latent variables $z_{n,j^\star}$ and $z_{n,j^\star}^+$ are correlated through the augmentation process. For a negative sample $\boldsymbol{X}_{n,s}$, its latent variable $z_{n,s,j^\star}$ is independent of those of the positive pair $(z_{n,j^\star}, z_{n,j^\star}^+)$, so we have:

$$
(z_{n,j^\star}, z_{n,j^\star}^+) \perp\!\!\!\perp z_{n,s,j^\star} .
\tag{190}
$$

For the anchor $\boldsymbol{X}_n$ and its positive $\boldsymbol{Y}_n$, we have

$$
h_i(\boldsymbol{X}_n) = \sum_{r=1}^{L} \Big\langle \boldsymbol{w}_i, z_Y^{(r)} \Big\rangle = \frac{1}{L} \Big\langle \boldsymbol{w}_i, \, \boldsymbol{M} \sum_{r=1}^{L} \tilde{\boldsymbol{z}}_n^{(r)} + \sum_{r=1}^{L} \tilde{\xi}_n^{(r)} \Big\rangle ,
\tag{191}
$$

$$
h_i(\boldsymbol{Y}_n) = \sum_{s=1}^{L} \Big\langle \boldsymbol{w}_i, z_Y^{(s)} \Big\rangle = \frac{1}{L} \Big\langle \boldsymbol{w}_i, \, \boldsymbol{M} \sum_{s=1}^{L} \tilde{\boldsymbol{z}}_n^{+(s)} + \sum_{s=1}^{L} \tilde{\xi}_n^{+(s)} \Big\rangle ,
\tag{192}
$$

$$
\hat{\boldsymbol{z}}_n := \sum_{r=1}^{L} \tilde{\boldsymbol{z}}_n^{(r)}, \qquad \hat{\boldsymbol{z}}_n^+ := \sum_{s=1}^{L} \tilde{\boldsymbol{z}}_n^{+(s)}, \qquad \hat{\xi}_n := \sum_{r=1}^{L} \tilde{\xi}_n^{(r)}, \qquad \hat{\xi}_n^+ := \sum_{s=1}^{L} \tilde{\xi}_n^{+(s)} .
\tag{193}
$$

We can write the outputs as:

$$
h_i(\boldsymbol{X}_n) = \frac{1}{L} \langle \boldsymbol{w}_i, \, \boldsymbol{M}\hat{\boldsymbol{z}}_n + \hat{\xi}_n \rangle , \qquad h_i(\boldsymbol{Y}_n) = \frac{1}{L} \langle \boldsymbol{w}_i, \, \boldsymbol{M}\hat{\boldsymbol{z}}_n^+ + \hat{\xi}_n^+ \rangle .
\tag{194}
$$

For a negative sample $\boldsymbol{X}_{n,s}$: $\hat{\boldsymbol{z}}_{n,s} := \sum_{q=1}^{L} \tilde{\boldsymbol{z}}_{n,s}^{(q)}$, $\hat{\xi}_{n,s} := \sum_{q=1}^{L} \tilde{\xi}_{n,s}^{(q)}$, the output is:

$$
h_i(\boldsymbol{X}_{n,s}) = \frac{1}{L} \langle \boldsymbol{w}_i, \boldsymbol{M}\hat{\boldsymbol{z}}_{n,s} + \hat{\xi}_{n,s} \rangle .
\tag{195}
$$

We first establish a lower bound for $\mathbb{E}[h_i(\boldsymbol{X}_n)h_i(\boldsymbol{Y}_n)]$.

Expanding and using zero-mean and independence of latent variables and noises, we have

$$
\begin{aligned}
\mathbb{E}[h_i(\boldsymbol{X}_n)h_i(\boldsymbol{Y}_n)] &= \frac{1}{L^2} \mathbb{E}\big[ \langle \boldsymbol{w}_i, \boldsymbol{M}\hat{\boldsymbol{z}}_n \rangle \, \langle \boldsymbol{w}_i, \boldsymbol{M}\hat{\boldsymbol{z}}_n^+ \rangle \big] \\
&= \frac{1}{L^2} \sum_{j=1}^{d} \langle \boldsymbol{w}_i, \boldsymbol{M}_j \rangle^2 \, \mathbb{E}\big[ \hat{\boldsymbol{z}}_{n,j} \, \hat{\boldsymbol{z}}_{n,j}^+ \big] \\
&\geq \frac{1}{L^2} \langle \boldsymbol{w}_i, \boldsymbol{M}_{j^\star} \rangle^2 \, \mathbb{E}\big[ \hat{\boldsymbol{z}}_{n,j^\star} \, \hat{\boldsymbol{z}}_{n,j^\star}^+ \big] \\
&\geq \Omega\Big( \frac{\tau^2}{\Xi_2^2} \epsilon_{j^\star} \frac{\log\log d}{d} \Big) .
\end{aligned}
\tag{196}
$$

Therefore

$$\mathbb{E}[h_i(\boldsymbol{X}_n)h_i(\boldsymbol{Y}_n)] \geq \Omega(\frac{\tau^2}{\Xi_2^2}\epsilon_{j^\star}\frac{\log\log d}{d}). \tag{197}$$

Next, we compute the expectation of $h_i(\boldsymbol{X}_n), h_i(\boldsymbol{X}_{n,s}).h_i(\boldsymbol{X}_n) h_i(\boldsymbol{X}_{n,s})$,

$$\mathbb{E}[h_i(\boldsymbol{X}_n)h_i(\boldsymbol{X}_{n,s})] = \mathbb{E}\Big[\big(\frac{1}{L}\langle\boldsymbol{w}_i, \ \boldsymbol{M}\hat{\boldsymbol{z}}_n + \hat{\xi}_n\rangle\big)\big(\frac{1}{L}\langle\boldsymbol{w}_i, \ \boldsymbol{M}\hat{\boldsymbol{z}}_{n,s} + \hat{\xi}_{n,s}\rangle\big)\Big]. \tag{198}$$

By the assumption, the latent variables of $\boldsymbol{X}_n$ are independent of those of the negative $\boldsymbol{X}_{n,s}$, and all noises are mean-zero and independent. Therefore,

$$\mathbb{E}\big[\langle\boldsymbol{w}_i, \boldsymbol{M}\hat{\boldsymbol{z}}_n\rangle\langle\boldsymbol{w}_i, \boldsymbol{M}\hat{\boldsymbol{z}}_{n,s}\rangle\big] = 0, \qquad \mathbb{E}[\langle\boldsymbol{w}_i, \hat{\xi}_n\rangle\langle\boldsymbol{w}_i, \hat{\xi}_{n,s}\rangle] = 0 \tag{199}$$

Therefore, we conclude that

$$\mathbb{E}[h_i(\boldsymbol{X}_n)h_i(\boldsymbol{X}_{n,s})] = 0 \tag{200}$$

Let $\mathcal{P}$ be the pruned set with $|\mathcal{P}| = \alpha m$. Summing the per-neuron bounds over $i \in \mathcal{P}$, we obtain

$$\mathbb{E}\Big[\sum_{i\in\mathcal{P}} h_i(\boldsymbol{X}_n)h_i(\boldsymbol{Y}_n)\Big] \geq \Omega(\alpha m\frac{\tau^2}{\Xi_2^2}p_{j^\star}), \tag{201}$$

$$\mathbb{E}\Big[\sum_{i\in\mathcal{P}} h_i(\boldsymbol{X}_n)h_i(\boldsymbol{X}_{n,s})\Big] = 0. \tag{202}$$

This completes the proof of Lemma F.1 (a)(b).

Finally, we compute the expectation of $h_{i,t}, (\boldsymbol{X}_{n,s}), \langle\nabla_{\boldsymbol{w}_i}h_i(\boldsymbol{X}_n), \boldsymbol{M}_{j^\star}\rangle$, and we have

$$\begin{aligned}
&\mathbb{E}\Big[h_{i,t}(\boldsymbol{X}_{n,s})\langle\nabla_{\boldsymbol{w}_i}h_i(\boldsymbol{X}_n), \boldsymbol{M}_{j^\star}\rangle\Big]\\
=&\mathbb{E}\Big[\big(\frac{1}{L}\langle\boldsymbol{w}_i, \ \boldsymbol{M}\hat{\boldsymbol{z}}_{n,s} + \hat{\xi}_{n,s}\rangle\big)\big(\frac{1}{L}\langle\boldsymbol{M}\hat{\boldsymbol{z}}_n + \hat{\xi}_n, \boldsymbol{M}_{j^\star}\rangle\big)\Big]\\
=&\mathbb{E}\Big[\big(\frac{1}{L^2}\langle\boldsymbol{w}_i, \ \boldsymbol{M}\hat{\boldsymbol{z}}_{n,s} + \hat{\xi}_{n,s}\rangle\big)\big(\hat{\boldsymbol{z}}_{n,j^\star} + \langle\hat{\xi}_n, \boldsymbol{M}_{j^\star}\rangle\big)\Big]\\
=&\frac{1}{L^2}\mathbb{E}\left[\langle\boldsymbol{w}_i, \boldsymbol{M}\hat{\boldsymbol{z}}_{n,s}\rangle \cdot \hat{\boldsymbol{z}}_{n,j^\star}\right]\\
=&0.
\end{aligned} \tag{203}$$

This completes the proof of Lemma F.1(c),

$$\mathbb{E}\Big[h_{i,t}(\boldsymbol{X}_{n,s})\langle\nabla_{\boldsymbol{w}_i}h_i(\boldsymbol{X}_n), \boldsymbol{M}_{j^\star}\rangle\Big] = 0. \tag{204}$$

$\square$

## F.6 PROOF OF LEMMA F.2:

*Proof of Lemma F.2:* We link the logit to the pruning ratio and plug it into the gradient growth. Recall the softmax weights and partial derivatives

$$\ell'_p = \frac{e^{u_p/\tau}}{e^{u_p/\tau} + \sum_{s=1}^S e^{u_s/\tau}}, \qquad \ell'_s = \frac{e^{u_s/\tau}}{e^{u_p/\tau} + \sum_{s=1}^S e^{u_s/\tau}}, \qquad \sum_{s=1}^S \ell'_s = 1 - \ell'_p, \tag{205}$$

$$u_p = \text{Sim}_f(\boldsymbol{X}_n, \boldsymbol{Y}_n), \quad u_s = \text{Sim}_f(\boldsymbol{X}_n, \boldsymbol{X}_{n,s}), \tag{206}$$

$$\frac{\partial\ell'_p}{\partial u_p} = \frac{1}{\tau}\ell'_p(1 - \ell'_p), \qquad \frac{\partial\ell'_p}{\partial u_s} = -\frac{1}{\tau}\ell'_p\ell'_s. \tag{207}$$

Pruning the size $\alpha m$ changes the similarities by

$$\Delta u_p = -\sum_{i \in \mathcal{P}} h_i(\boldsymbol{X}_n) h_i(\boldsymbol{Y}_n), \qquad \Delta u_s = -\sum_{i \in \mathcal{P}} h_i(\boldsymbol{X}_n) h_i(\boldsymbol{X}_{n,s}). \tag{208}$$

Next, calculate the first order change of $\ell'_p$, we know:

$$\boldsymbol{u} = (u_p, u_1, \dots, u_S), \qquad \Delta\boldsymbol{u} = (\Delta u_p, \Delta u_1, \dots, \Delta u_S). \tag{209}$$

Using multivariate Taylor expansion up to second order with remainder:

$$\ell'_p(\boldsymbol{u} + \Delta\boldsymbol{u}) - \ell'_p(\boldsymbol{u}) = \nabla\ell'_p(\boldsymbol{u})^\top \Delta\boldsymbol{u} + \frac{1}{2}\Delta\boldsymbol{u}^\top H_p(\boldsymbol{u})\Delta\boldsymbol{u} + o(\|\Delta\boldsymbol{u}\|^2), \Delta\boldsymbol{u} \to 0. \tag{210}$$

By a first order Taylor expansion, we have

$$\begin{aligned}
\Delta\ell'_p &= \frac{\partial\ell'_p}{\partial u_p}\Delta u_p + \sum_{s=1}^{S}\frac{\partial\ell'_p}{\partial u_s}\Delta u_s + o\left(\|\Delta\boldsymbol{u}\|\right) \\
&= \frac{1}{\tau}\ell'_p(1 - \ell'_p)\Delta u_p - \frac{1}{\tau}\ell'_p\sum_{s=1}^{S}\ell'_s\Delta u_s.
\end{aligned} \tag{211}$$

We note that at $T_4$, by the convergence of the loss function, we obtain $\ell'_p = 1 - \Theta(\frac{1}{\tau})$, and both $\ell'_p$ and $\ell'_s$ take fixed values. Then, by taking expectations over $\Delta\ell'_p$ and using the relation $\sum_s \ell'_s = 1 - \ell'_p$, we obtain:

$$\mathbb{E}[\Delta\ell'_p] = -\Theta(\frac{1}{\tau^2})\Big(\mathbb{E}\big[\sum_{i \in \mathcal{P}} h_i(\boldsymbol{X}_n)h_i(\boldsymbol{Y}_n)\big] - \mathbb{E}\big[\sum_{i \in \mathcal{P}} h_i(\boldsymbol{X}_n)h_i(\boldsymbol{X}_{n,s})\big]\Big). \tag{212}$$

Also, by Lemma F.1, given that

$$\mathbb{E}\Big[\sum_{i \in \mathcal{P}} h_i(\boldsymbol{X}_n)h_i(\boldsymbol{Y}_n)\Big] - \mathbb{E}\Big[\sum_{i \in \mathcal{P}} h_i(\boldsymbol{X}_n)h_i(\boldsymbol{X}_{n,s})\Big] \geq \Omega(\alpha m \frac{\tau^2}{\Xi_2^2}\epsilon_{j^\star}\frac{\log\log d}{d}). \tag{213}$$

Hence,

$$\mathbb{E}[\Delta\ell'_p] = -\Omega(\frac{\alpha m}{\Xi_2^2}\epsilon_{j^\star}\frac{\log\log d}{d}) < 0. \tag{214}$$

Hence,

$$\begin{aligned}
\mathbb{E}[\ell'_{p,\boldsymbol{\theta}^{(t)}_{\text{mask}}}] &= \mathbb{E}[\ell'_{p,\boldsymbol{\theta}^{(t)}}] + \mathbb{E}[\Delta\ell'_p] \\
&= \mathbb{E}[\ell'_{p,\boldsymbol{\theta}^{(t)}}] - \Omega(\frac{\alpha m}{\Xi_2^2}\epsilon_{j^\star}\frac{\log\log d}{d}) \\
&= 1 - \Theta(\frac{1}{\tau}) - \Omega(\frac{\alpha m}{\Xi_2^2}\epsilon_{j^\star}\frac{\log\log d}{d}).
\end{aligned} \tag{215}$$

Now, converting to the form of $1 - \ell'$:

$$\mathbb{E}[1 - \ell'_{p,\boldsymbol{\theta}^{(t)}_{\text{mask}}}] = 1 - \mathbb{E}[\ell'_{p,\boldsymbol{\theta}^{(t)}_{\text{mask}}}]. \tag{216}$$

Substituting the previous expression gives

$$\mathbb{E}[1 - \ell'_{p,\boldsymbol{\theta}^{(t)}_{\text{mask}}}] = \Theta(\frac{1}{\tau}) + \Omega(\frac{\alpha m}{\Xi_2^2}\epsilon_{j^\star}\frac{\log\log d}{d}). \tag{217}$$

$\square$

### F.7 PROOF OF LEMMA F.3(A):

*Proof of Lemma F.3(a):*

$$
\begin{aligned}
&\mathbb{E}\left[h_i(\boldsymbol{Y}_n)\langle\nabla_{\boldsymbol{w}_i}h_i(\boldsymbol{X}_n),\boldsymbol{M}_j\rangle\right]\\
=&\mathbb{E}\left[h_i(\boldsymbol{Y}_n)\langle\sum_{r=1}^{L}\mathbf{1}^{(r)}_{\left|\langle\boldsymbol{w}_i^{(t)},\boldsymbol{z}_{\boldsymbol{X}}^{(r)}\rangle\right|\geq 0}\cdot\boldsymbol{z}_{\boldsymbol{X}}^{(r)},\boldsymbol{M}_j\rangle\right]\\
=&\mathbb{E}\left[\sum_{s=1}^{L}\langle\boldsymbol{w}_i,\boldsymbol{z}_{\boldsymbol{Y}}^{(s)}\rangle\cdot\left(\sum_{r=1}^{L}\langle\boldsymbol{z}_{\boldsymbol{X}}^{(r)},\boldsymbol{M}_j\rangle\right)\right]\\
=&\frac{1}{L^2}\mathbb{E}\left[\sum_{s=1}^{L}\langle\boldsymbol{w}_i,\boldsymbol{M}\tilde{z}_n^{+(s)}+\tilde{\xi}_n^{+(s)}\rangle\cdot\left(\sum_{r=1}^{L}\langle\boldsymbol{M}\tilde{z}_n^{(r)}+\tilde{\xi}_n^{(r)},\boldsymbol{M}_j\rangle\right)\right]\\
=&\frac{1}{L^2}\mathbb{E}\left[\sum_{s=1}^{L}\langle\boldsymbol{w}_i,\boldsymbol{M}\tilde{z}_n^{+(s)}\rangle\cdot\left(\sum_{r=1}^{L}\langle\boldsymbol{M}\tilde{z}_n^{(r)},\boldsymbol{M}_j\rangle\right)\right]\\
=&\frac{1}{L^2}\mathbb{E}\left[\langle\boldsymbol{w}_i,\boldsymbol{M}\hat{z}_n^{+}\rangle\cdot\left(\langle\boldsymbol{M}\hat{z}_n,\boldsymbol{M}_j\rangle\right)\right]\\
=&\frac{1}{L^2}\mathbb{E}\left[\langle\boldsymbol{w}_i,\boldsymbol{M}\hat{z}_n^{+}\rangle\cdot\langle\sum_{j'\in[d]}\boldsymbol{M}_{j'}\hat{z}_{n,j'},\boldsymbol{M}_j\rangle\right]\\
=&\frac{1}{L^2}\mathbb{E}\left[\langle\boldsymbol{w}_i,\boldsymbol{M}\hat{z}_n^{+}\rangle\cdot\sum_{j'\in[d]}\langle\boldsymbol{M}_{j'},\boldsymbol{M}_j\rangle\hat{z}_{n,j'}\right]\\
=&\frac{1}{L^2}\mathbb{E}\left[\langle\boldsymbol{w}_i,\boldsymbol{M}\hat{z}_n^{+}\rangle\cdot\hat{z}_{n,j}\right]\\
=&\frac{1}{L^2}\mathbb{E}\left[\sum_{j''\in[d]}\langle\boldsymbol{w}_i,\boldsymbol{M}_{j''}\rangle\hat{z}_{n,j''}^{+}\hat{z}_{n,j}\right]\\
=&\frac{1}{L^2}\sum_{j''\in[d]}\langle\boldsymbol{w}_i,\boldsymbol{M}_{j''}\rangle\mathbb{E}\left[\hat{z}_{n,j''}^{+}\hat{z}_{n,j}\right].
\end{aligned}
\tag{218}
$$

In the final step, we have

$$
\frac{1}{L^2}\sum_{j''\in[d]}\langle\boldsymbol{w}_i,\boldsymbol{M}_{j''}\rangle\mathbb{E}\left[\hat{z}_{n,j''}^{+}\hat{z}_{n,j}\right]=\frac{1}{L^2}\langle\boldsymbol{w}_i,\boldsymbol{M}_j\rangle\mathbb{E}\left[\hat{z}_{n,j}^{+}\hat{z}_{n,j}\right].
\tag{219}
$$

This completes the proof. □

## F.8 PROOF OF LEMMA F.3(B):

*Proof of Lemma F.3(b):*

$$
\begin{aligned}
&\mathbb{E}\left[h_i(\boldsymbol{Y}_n)\langle\nabla_{\boldsymbol{w}_i}h_i(\boldsymbol{X}_n),\boldsymbol{M}_j^\perp\rangle\right]\\
=&\mathbb{E}\left[h_i(\boldsymbol{Y}_n)\langle\sum_{r=1}^{L}\mathbf{1}_{\left|\langle\boldsymbol{w}_i^{(t)},\boldsymbol{z}_{\boldsymbol{X}}^{(r)}\rangle\right|\geq0}^{(r)}\cdot\boldsymbol{z}_{\boldsymbol{X}}^{(r)},\boldsymbol{M}_j^\perp\rangle\right]\\
=&\mathbb{E}\left[h_i(\boldsymbol{Y}_n)\cdot\left(\sum_{r=1}^{L}\langle\mathbf{1}_{\left|\langle\boldsymbol{w}_i^{(t)},\boldsymbol{z}_{\boldsymbol{X}}^{(r)}\rangle\right|\geq0}^{(r)}\cdot\boldsymbol{z}_{\boldsymbol{X}}^{(r)},\boldsymbol{M}_j^\perp\rangle\right)\right]\\
=&\mathbb{E}\left[\sum_{s=1}^{L}\langle\boldsymbol{w}_i,\boldsymbol{z}_{\boldsymbol{Y}}^{(s)}\rangle\mathbf{1}_{\left|\langle\boldsymbol{w}_i^{(t)},z_X^{(s)}\rangle\right|\geq0}^{(s)}\cdot\left(\sum_{r=1}^{L}\langle\mathbf{1}_{\left|\langle\boldsymbol{w}_i^{(t)},\boldsymbol{z}_{\boldsymbol{X}}^{(r)}\rangle\right|\geq0}^{(r)}\cdot\boldsymbol{z}_{\boldsymbol{X}}^{(r)},\boldsymbol{M}_j^\perp\rangle\right)\right]\\
=&\mathbb{E}\left[\sum_{s=1}^{L}\langle\boldsymbol{w}_i,\boldsymbol{z}_{\boldsymbol{Y}}^{(s)}\rangle\cdot\left(\sum_{r=1}^{L}\langle\boldsymbol{z}_{\boldsymbol{X}}^{(r)},\boldsymbol{M}_j^\perp\rangle\right)\right]\\
=&\frac{1}{L^2}\mathbb{E}\left[\sum_{s=1}^{L}\langle\boldsymbol{w}_i,\boldsymbol{M}\tilde{z}_n^{+(s)}+\tilde{\xi}_n^{+(s)}\rangle\cdot\left(\sum_{r=1}^{L}\langle\boldsymbol{M}\tilde{z}_n^{(r)}+\tilde{\xi}_n^{(r)},\boldsymbol{M}_j^\perp\rangle\right)\right]\\
=&\frac{1}{L^2}\mathbb{E}\left[\sum_{s=1}^{L}\langle\boldsymbol{w}_i,\boldsymbol{M}\tilde{z}_n^{+(s)}\rangle\cdot\left(\sum_{r=1}^{L}\langle\boldsymbol{M}\tilde{z}_n^{(r)},\boldsymbol{M}_j^\perp\rangle\right)\right]\\
=&\frac{1}{L^2}\mathbb{E}\left[\langle\boldsymbol{w}_i,\boldsymbol{M}\sum_{s=1}^{L}\tilde{z}_n^{+(s)}\rangle\cdot\left(\langle\boldsymbol{M}\sum_{r=1}^{L}\tilde{z}_n^{(r)},\boldsymbol{M}_j^\perp\rangle\right)\right]\\
=&\frac{1}{L^2}\mathbb{E}\left[\langle\boldsymbol{w}_i,\boldsymbol{M}\hat{\boldsymbol{z}}_n^+\rangle\cdot\left(\langle\boldsymbol{M}\hat{\boldsymbol{z}}_n,\boldsymbol{M}_j^\perp\rangle\right)\right]\\
=&\frac{1}{L^2}\mathbb{E}\left[\langle\boldsymbol{w}_i,\boldsymbol{M}\hat{\boldsymbol{z}}_n^+\rangle\cdot\langle\sum_{j'\in[d]}\boldsymbol{M}_{j'}\hat{\boldsymbol{z}}_{n,j'},\boldsymbol{M}_j^\perp\rangle\right]\\
=&\frac{1}{L^2}\mathbb{E}\left[\langle\boldsymbol{w}_i,\boldsymbol{M}\hat{\boldsymbol{z}}_n^+\rangle\cdot\langle\sum_{j'\in[d]}\boldsymbol{M}_{j'},\boldsymbol{M}_j^\perp\rangle\hat{\boldsymbol{z}}_{n,j'}\right]\\
=&\frac{1}{L^2}\mathbb{E}\left[\langle\boldsymbol{w}_i,\boldsymbol{M}\hat{\boldsymbol{z}}_n^+\rangle\cdot\sum_{j'\in[d]}\langle\boldsymbol{M}_{j'},\boldsymbol{M}_j^\perp\rangle\hat{\boldsymbol{z}}_{n,j'}\right]\\
=&\frac{1}{L^2}\mathbb{E}\left[\langle\boldsymbol{w}_i,\boldsymbol{M}\hat{\boldsymbol{z}}_n^+\rangle\cdot0\right]\\
=&0.
\end{aligned}
\tag{220}
$$

$\square$

# G PROOF OF LEMMAS IN APPENDIX B

## G.1 PROOF OF LEMMA B.2(A):

*Proof of Lemma B.2(a):* At initialization, the neuron weight $\boldsymbol{w}_i^{(0)}$ is a high dimensional Gaussian vector :

$$
\boldsymbol{w}_i^{(0)}\sim\mathcal{N}(0,\sigma_0^2 I_{d_1}),
\tag{221}
$$

with $\boldsymbol{w}_i^{(0)}\in\mathbb{R}^{d_1}$ and each coordinate $\boldsymbol{w}_i^{(0)}(k)\sim\mathcal{N}(0,\sigma_0^2)$, i.i.d.

$$\left\| \boldsymbol{w}_i^{(0)} \right\|_2^2 = \sum_{k=1}^{d_1} \left( \boldsymbol{w}_i^{(0)}(k) \right)^2. \tag{222}$$

We know that $\boldsymbol{w}_i^{(0)}(k) \sim \mathcal{N}(0, \sigma_0^2)$, so:

$$\frac{1}{\sigma_0^2} \left\| \boldsymbol{w}_i^{(0)} \right\|_2^2 \sim \chi^2(d_1). \tag{223}$$

According to the concentration inequality of the chi-square distribution:

If $X \sim \chi^2(d_1)$, then for any $0 < \varepsilon < 1$, we have:

$$\Pr\left[ \left| \frac{X}{d_1} - 1 \right| \geq \varepsilon \right] \leq 2\exp\left( -\frac{d_1 \varepsilon^2}{4} \right). \tag{224}$$

Therefore, we have:

$$\Pr\left[ \left| \frac{\left\| \boldsymbol{w}_i^{(0)} \right\|_2^2}{\sigma_0^2 d_1} - 1 \right| \geq \varepsilon \right] \leq 2\exp\left( -\frac{d_1 \varepsilon^2}{4} \right). \tag{225}$$

Choose a suitable $\varepsilon$ to derive the precision range and we choose: $\varepsilon = \widetilde{O}\left( \frac{1}{\sqrt{d_1}} \right)$.

At this time, the probability of deviation is:

$$\Pr\left[ \left| \left\| \boldsymbol{w}_i^{(0)} \right\|_2^2 - \sigma_0^2 d_1 \right| \leq \widetilde{O}(\sigma_0^2 \sqrt{d_1}) \right] \geq 1 - \frac{1}{\mathrm{poly}(d)}. \tag{226}$$

That is:

$$\left\| \boldsymbol{w}_i^{(0)} \right\|_2^2 \in \left[ \sigma_0^2 d_1 \left( 1 - \widetilde{O}\left( \frac{1}{\sqrt{d_1}} \right) \right), \ \sigma_0^2 d_1 \left( 1 + \widetilde{O}\left( \frac{1}{\sqrt{d_1}} \right) \right) \right]. \tag{227}$$

This holds with high probability $(1 - \frac{1}{\mathrm{poly}(d)})$. $\qquad\square$

## G.2 Proof of Lemma B.2(b):

*Proof of Lemma B.2(b):* Let:

$$\boldsymbol{Z}_i := \frac{1}{\sigma_0} \boldsymbol{w}_i^{(0)} \sim \mathcal{N}(0, \boldsymbol{I}_{d_1}). \tag{228}$$

Then we have:

$$\left\| \boldsymbol{M}\boldsymbol{M}^\top \boldsymbol{w}_i^{(0)} \right\|_2^2 = \sigma_0^2 \cdot \left\| \boldsymbol{M}\boldsymbol{M}^\top \boldsymbol{Z}_i \right\|_2^2. \tag{229}$$

We regard $\boldsymbol{M}\boldsymbol{M}^\top$ as a rank-$d$ projection matrix, projecting $\boldsymbol{Z}_i \in \mathbb{R}^{d_1}$ onto the column space of $\boldsymbol{M}$ so we can use the following property:

If $\boldsymbol{M}\boldsymbol{M}^\top$ is a fixed rank-$d$ projection matrix, and $\boldsymbol{Z}_i \sim \mathcal{N}(0, \boldsymbol{I}_{d_1})$, then:

$$\| \boldsymbol{M}\boldsymbol{M}^\top \boldsymbol{Z}_i \|_2^2 = \boldsymbol{Z}_i^\top (\boldsymbol{M}\boldsymbol{M}^\top)^\top \boldsymbol{M}\boldsymbol{M}^\top \boldsymbol{Z}_i = \boldsymbol{Z}_i^\top \boldsymbol{M}\boldsymbol{M}^\top \boldsymbol{Z}_i = \| \boldsymbol{M}^\top \boldsymbol{Z}_i \|_2^2, \tag{230}$$

$$\boldsymbol{M}^\top \boldsymbol{Z}_i \sim \mathcal{N}(0, I_d). \tag{231}$$

Therefore, we can conclude:

$$\left\| \boldsymbol{M}\boldsymbol{M}^\top \boldsymbol{Z}_i \right\|_2^2 \sim \chi^2(d) \quad \Longrightarrow \quad \mathbb{E}\left[ \left\| \boldsymbol{M}\boldsymbol{M}^\top \boldsymbol{Z}_i \right\|_2^2 \right] = d. \tag{232}$$

And it satisfies the following Chi-square concentration inequality:

$$\mathbb{P}\left(\left|\|\boldsymbol{M}\boldsymbol{M}^{\top}\boldsymbol{Z}_i\|_2^2 - d\right| \leq \varepsilon d\right) \geq 1 - 2\exp\left(-c\varepsilon^2 d\right). \tag{233}$$

Choose $\varepsilon = \tilde{\mathcal{O}}(1/\sqrt{d})$, and the result holds with high probability. We substitute back $\boldsymbol{w}_i^{(0)}$

$$\left\|\boldsymbol{M}\boldsymbol{M}^{\top}\boldsymbol{w}_i^{(0)}\right\|_2^2 = \sigma_0^2 \cdot \left\|\boldsymbol{M}\boldsymbol{M}^{\top}\boldsymbol{Z}_i\right\|_2^2 \in \left[\sigma_0^2 d\left(1 - \tilde{\mathcal{O}}\left(\frac{1}{\sqrt{d}}\right)\right), \sigma_0^2 d\left(1 + \tilde{\mathcal{O}}\left(\frac{1}{\sqrt{d}}\right)\right)\right]. \tag{234}$$

$\square$

## G.3 PROOF OF LEMMA B.2(C):

*Proof of Lemma B.2(c):* Recall if $g$ is standard Gaussian, then for every $t > 0$,

$$\frac{1}{\sqrt{2\pi}} \cdot \frac{t}{t^2 + 1} e^{-t^2/2} < \Pr_{g \sim \mathcal{N}(0,1)}[g > t] < \frac{1}{\sqrt{2\pi}} \cdot \frac{1}{t} e^{-t^2/2}. \tag{235}$$

Therefore, for every $i \in [m]$ and $j \in [d]$,

$$\begin{aligned}
p_1 &= \Pr\left[\langle \boldsymbol{w}_i^{(0)}, \boldsymbol{M}_j \rangle^2 \geq \frac{c_1 \log d}{d} \|\boldsymbol{M}\boldsymbol{M}^{\top}\boldsymbol{w}_i^{(0)}\|_2^2\right] \\
&= \Pr\left[\frac{\langle \boldsymbol{w}_i^{(0)}, \boldsymbol{M}_j \rangle}{\sigma_0} \geq \sqrt{c_1 \log d}\right] \\
&\geq \Omega\left(\frac{1}{d^{c_1/2}}\right) \\
&= \Omega\left(\frac{1}{d^{\left(\frac{\epsilon_{\max}}{\epsilon_{\min}}\right)^2 \cdot (1+\gamma)}}\right),
\end{aligned} \tag{236}$$

and

$$\begin{aligned}
p_2 &= \Pr\left[\langle \boldsymbol{w}_i^{(0)}, \boldsymbol{M}_j \rangle^2 \geq \frac{c_2 \log d}{d} \|\boldsymbol{M}\boldsymbol{M}^{\top}\boldsymbol{w}_i^{(0)}\|_2^2\right] \\
&= \Pr\left[\frac{\langle \boldsymbol{w}_i^{(0)}, \boldsymbol{M}_j \rangle}{\sigma_0} \geq \sqrt{c_2 \log d}\right] \\
&\leq O\left(\frac{1}{\sqrt{\log d}}\right) \cdot \frac{1}{d^{c_2/2}} \\
&= O\left(\frac{1}{\sqrt{\log d}}\right) \cdot \frac{1}{d^{\left(\frac{\epsilon_{\min}}{\epsilon_{\max}}\right)^2 \cdot (1-\gamma)}}.
\end{aligned} \tag{237}$$

We define the following events in definition B.1:

- $A_i$: Lucky neuron $i$ satisfies conditions 1(i.e., the response is large enough and in the correct direction)

- $B_i$: for all $j' \neq j$, lucky neuron $i$ satisfies condition 2 (i.e., small responses in other directions)

We now compute the probability of the intersection event $A_i \cap B_i$:

$$\begin{aligned}
\Pr[A_i] &= \frac{p_1}{2} = \Omega\left(d^{-\left(\frac{\epsilon_{\max}}{\epsilon_{\min}}\right)^2 \cdot (1+\gamma)}\right), \\
\Pr[B_i] &= (1 - p_2)^{d-1} = e^{-(d-1)p_2} = e^{-(d-1)d^{-a}} = e^{-d^{1-a}} = 1, \\
\Pr[A_i \cap B_i] &= \frac{p_1}{2} \cdot (1 - p_2)^{d-1} = \Omega\left(\frac{1}{\sqrt{\log d}} \cdot d^{-\left(\frac{\epsilon_{\max}}{\epsilon_{\min}}\right)^2 \cdot (1+\gamma)}\right).
\end{aligned} \tag{238}$$

(1) We now have $m = d^{C_m}$ neurons. Therefore, the expected number is:

$$
\begin{aligned}
\mathbb{E}\left[|\mathcal{M}_j^\star|\right] = m \cdot \Pr[A_i \cap B_i] &= d^{C_m} \cdot \Omega\left(d^{-\left(\frac{\epsilon_{\max}}{\epsilon_{\min}}\right)^2 \cdot (1+\gamma)}\right) \\
&= \Omega\left(d^{C_m - \left(\frac{\epsilon_{\max}}{\epsilon_{\min}}\right)^2 \cdot (1+\gamma)}\right).
\end{aligned}
\tag{239}
$$

Chernoff bound (Lower-tail form): For any $\delta \in (0, 1)$, we have:

$$
\Pr\left[\sum X_i < (1 - \delta)\mu\right] \le e^{-\frac{\delta^2}{2}\mu}.
\tag{240}
$$

Let $\delta = \frac{1}{2}$, we obtain:

$$
\begin{aligned}
\Pr\left[\sum X_i < \frac{1}{2}\mu\right] &\le e^{-\mu/8} \\
\Pr\left[|\mathcal{M}_j^\star| < O\left(d^{\omega_1}\right)\right] &\le e^{-\Omega(d^{\omega_1})} \\
\Pr\left[|\mathcal{M}_j^\star| > \Omega\left(d^{\omega_1}\right)\right] &\ge 1 - e^{-\Omega(d^{\omega_1})}.
\end{aligned}
\tag{241}
$$

(2) We now have $m = d^{C_m}$ neurons. Therefore, the expected number is:

$$
\begin{aligned}
\mathbb{E}\left[|\mathcal{M}_j|\right] = m \cdot p_2 &= d^{C_m} \cdot \mathcal{O}\left(\frac{1}{\sqrt{\log d}} \cdot d^{-\left(\frac{\epsilon_{\min}}{\epsilon_{\max}}\right)^2 \cdot (1-\gamma)}\right) \\
&= \mathcal{O}\left(\frac{1}{\sqrt{\log d}} \cdot d^{C_m - \left(\frac{\epsilon_{\min}}{\epsilon_{\max}}\right)^2 \cdot (1-\gamma)}\right).
\end{aligned}
\tag{242}
$$

Chernoff bound (upper tail) tells us that for any $0 < \delta < 1$, we have:

$$
\begin{aligned}
\Pr\left[\sum X_i > (1 + \delta)\mu\right] &\le e^{-\Omega(\delta^2 \mu)} \\
\Pr\left[|\mathcal{M}_j| > \Omega(\frac{1}{\sqrt{\log d}}d^{\omega_2})\right] &\le e^{-\Omega(\frac{1}{\sqrt{\log d}}d^{\omega_2})} = o\left(\frac{1}{d^4}\right) \\
\Pr\left[|\mathcal{M}_j| < O\left(\frac{1}{\sqrt{\log d}}d^{\omega_2}\right)\right] &\ge 1 - o\left(\frac{1}{d^4}\right) \\
\Pr\left[|\mathcal{M}_j| < O\left(d^{\omega_2}\right)\right] &\ge 1 - o\left(\frac{1}{d^4}\right).
\end{aligned}
\tag{243}
$$

$\square$

## G.4 PROOF OF LEMMA B.2(D):

*Proof of Lemma B.2(d):* We know: $|\mathcal{M}_j| \le O(d^{\omega_2})$. There are $d$ indices $j \in [d]$. Therefore, the total number of pairs $(i, j)$ such that $i \in \mathcal{M}_j$ is at most:

$$
\sum_{j=1}^{d} |\mathcal{M}_j| \le d \cdot O(d^{\omega_2}) = O(d^{1+\omega_2}).
\tag{244}
$$

On the other hand, the total number of neurons is $m = d^{C_m}$. So for any fixed $i$, we define:

$$
\mathcal{N}_i := \{j \in [d] : i \in \mathcal{M}_j\}.
\tag{245}
$$

Then,

$$
\sum_{i=1}^{m} |N_i| = \sum_{j=1}^{d} |\mathcal{M}_j| \le O(d^{1+\omega_2}).
\tag{246}
$$

Therefore,

$$
\mathbb{E}[|N_i|] = \frac{1}{m} \sum_{i=1}^{m} |N_i| \le O\left(d^{1+\omega_2 - C_m}\right) = O\left(d^{1 - \left(\frac{\epsilon_{\min}}{\epsilon_{\max}}\right)^2 \cdot (1-\gamma)}\right).
\tag{247}
$$

Then:

$$\Pr\left[\left|\langle \boldsymbol{w}_i^{(0)}, \boldsymbol{M}_j\rangle\right| \geq \Omega(\sigma_0 \log^{1/4} d)\right] \leq 2\exp\left(-\frac{t^2}{2\sigma_0^2}\right) = 2^{-\Omega(\sqrt{\log d})}. \tag{248}$$

Fix $i \in [m]$, and consider $d$ different $j$. Each has probability $2^{-\Omega(\sqrt{\log d})}$ to exceed the threshold. Therefore, the expectation is:

$$\mathbb{E}\left[\left|\left\{j \in [d] \;\middle|\; \left|\langle \boldsymbol{w}_i^{(0)}, \boldsymbol{M}_j\rangle\right| \geq \Omega\left(\sigma_0 \log^{1/4} d\right)\right\}\right|\right] = O\left(2^{-\sqrt{\log d}} \cdot d\right). \tag{249}$$

$\square$

## H    PROOF OF LEMMAS IN APPENDIX C

This section can be found in the Supplementary Material.

## I    PROOF OF LEMMAS IN APPENDIX D

This section can be found in the Supplementary Material.

## J    PROOF OF LEMMAS IN APPENDIX E

This section can be found in the Supplementary Material.

## K    PROOF OF ADDITIONAL LEMMAS

This section can be found in the Supplementary Material.

