*(b)*
$$\left\| MM^\top w_i^{(t)} \right\|_2^2 \geq \left\| MM^\top w_i^{(0)} \right\|_2^2 \left(1 - \eta\lambda + \epsilon_{\min} \frac{\eta C_z \log\log d}{d}\right)^{2t} - O\left(\frac{1}{d}\right) \left\| MM^\top w_i^{(0)} \right\|_2^2. \quad (41)$$

*(c)*
$$\left\| M^\perp (M^\perp)^\top w_i^{(t)} \right\|_2^2 \leq \left(1 + O\left(\frac{1}{\text{poly}(d)}\right)\right) \left\| M^\perp (M^\perp)^\top w_i^{(0)} \right\|_2^2. \quad (42)$$

**Lemma C.2.** *For each $i \in [m]$, there are at most $O(2^{-\sqrt{\log d}} d)$ indices $j \in [d]$ such that*
$$|\langle w_i^{(0)}, M_j \rangle| \geq \Omega(\sigma_0 \log^{1/4} d). \quad (43)$$

## C.3 PROOF OF THEOREM C.1

*Proof of Theorem C.1(a):* The result (a) can be derived from Lemma C.1 (c). We have,
$$\begin{aligned}
\|MM^\top w_i^{(T_1)}\|_2^2 &= \|w_i^{(T_1)}\|_2^2 - \|M^\perp (M^\perp)^\top w_i^{(T_1)}\|_2^2 \\
&\geq \|w_i^{(T_1)}\|_2^2 - \left(1 + \frac{1}{\text{poly}(d)}\right) \|M^\perp (M^\perp)^\top w_i^{(0)}\|_2^2 \\
&\geq \|w_i^{(T_1)}\|_2^2 - \|w_i^{(0)}\|_2^2 \\
&\geq \|w_i^{(T_1)}\|_2^2 - \frac{\|w_i^{(T_1)}\|_2^2}{(1 + \epsilon_{\min} C_z \log d)} \\
&\geq \frac{1}{2}\|w_i^{(T_1)}\|_2^2.
\end{aligned} \quad (44)$$
$\square$

*Proof of Theorem C.1(b):* Note that from similar gradient calculations to those in the proof of Lemma C.1 (b), we have, for $j \in [d]$ and $i \in \mathcal{M}_j^\star$:
$$\begin{aligned}
&|\langle w_i^{(T_1)}, M_j \rangle| \\
&= |\langle w_i^{(T_1-1)}, M_j \rangle - \eta \langle \nabla_{w_i} L_{\text{aug}}(f_{T_1-1}), M_j \rangle \ \pm \frac{\|w_i^{((T_1-1))}\|_2}{\text{poly}(d_1)}| \\

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

## H.1   USEFUL LEMMAS

**Lemma H.1** (Logits near initialization). *Let $\boldsymbol{w}_i \in \mathbb{R}^{d_1}$ for each $i \in [m]$. Suppose*

$$\sum_{i\in[m]}\left\|\boldsymbol{w}_i^{(t)}\right\|_2^2 \leq o\left(\tfrac{\tau}{d}\right). \tag{250}$$

*Then, with high probability over the randomness of $\boldsymbol{X}_n, \boldsymbol{Y}_n$, and $\mathfrak{N}$, it holds that*

$$\left|\ell'_{p,t}(\boldsymbol{X}_n, \mathfrak{B}) - \tfrac{1}{|\mathfrak{B}|}\right| \cdot \left|\ell'_{s,t}(\boldsymbol{X}_n, \mathfrak{B}) - \tfrac{1}{|\mathfrak{B}|}\right| \leq \widetilde{\mathcal{O}}\left(\frac{\sum_{i\in[m]}\left\|\boldsymbol{w}_i^{(t)}\right\|_2^2}{\tau|\mathfrak{B}|}\right). \tag{251}$$

## H.2   PROOF OF LEMMA C.1:

*Proof of Lemma C.1:* First, we must determine the precise gradient expression for each feature $\boldsymbol{M}_j$ and $\boldsymbol{M}_j^{\perp}$. The gradient descent update for the projection of $\boldsymbol{w}_i^{(t)}$ onto $\boldsymbol{M}_j$ can be written as

$$\langle \boldsymbol{w}_i^{(t+1)}, \boldsymbol{M}_j\rangle$$

$$=\langle \boldsymbol{w}_i^{(t)}, \boldsymbol{M}_j\rangle - \eta\langle \nabla_{\boldsymbol{w}_i} L_{\mathrm{aug}}(f_t), \boldsymbol{M}_j\rangle \quad \pm \frac{\|\boldsymbol{w}_i^{(t)}\|_2}{\mathrm{poly}(d_1)}$$

$$\overset{\text{\textcircled{1}}}{=}(1-\eta\lambda)\langle \boldsymbol{w}_i^{(t)}, \boldsymbol{M}_j\rangle + \eta\,\mathbb{E}_{\boldsymbol{X}_n, \boldsymbol{Y}_n}\left[(1-\ell'_{p,t}(\boldsymbol{X}_n, \mathfrak{B}))\cdot h_{i,t}(\boldsymbol{Y}_n)\langle \nabla_{\boldsymbol{w}_i} h_i(\boldsymbol{X}_n), \boldsymbol{M}_j\rangle\right] \tag{252}$$

$$-\eta \sum_{\boldsymbol{X}_{n,s}\in\mathfrak{N}}\mathbb{E}\left[\ell'_{s,t}(\boldsymbol{X}_n, \mathfrak{B})\,h_{i,t}(\boldsymbol{X}_{n,s})\langle \nabla_{\boldsymbol{w}_i} h(\boldsymbol{X}_n), \boldsymbol{M}_j\rangle\right] \quad \pm \frac{\|\boldsymbol{w}_i^{(t)}\|_2}{\mathrm{poly}(d_1)}.$$

\textcircled{1} is because $\lambda$ is the coefficient of the regularization term, as well as the gradient formula obtained earlier. $\frac{\|\boldsymbol{w}_i^{(t)}\|_2}{\mathrm{poly}(d_1)}$ is due to the approximation between population gradients and empirical gradients.

For the positive term: we can use Lemma F.3 and Lemma H.1 to obtain that:

$$\mathbb{E}\left[(1-\ell'_{p,t}(\boldsymbol{X}_n, \mathfrak{B}))\cdot h_{i,t}(\boldsymbol{Y}_n)\langle \nabla_{\boldsymbol{w}_i} h_{i,t}(\boldsymbol{X}_n), \boldsymbol{M}_j\rangle\right] = \frac{1}{L^2}\langle \boldsymbol{w}_i^{(t)}, \boldsymbol{M}_j\rangle\,\mathbb{E}\left[\hat{z}_{n,j}^+ \hat{z}_{n,j}\right]. \tag{253}$$

For the negative term: Here, the bound needs to be verified because Lemma H.1.

$$\mathbb{E}\left[\sum_{X\in\mathfrak{N}}\ell'_{s,t}\,h_{i,t}(X)\langle \nabla_{\boldsymbol{w}_i} h(\boldsymbol{X}_n), \boldsymbol{M}_j\rangle\right]$$

$$=\sum_{X\in\mathfrak{N}}\mathbb{E}\left[\left(\ell'_{s,t}-\tfrac{1}{|\mathfrak{B}|}\right)h_{i,t}(X)\langle \nabla_{\boldsymbol{w}_i} h(\boldsymbol{X}_n), \boldsymbol{M}_j\rangle\right]$$

$$\overset{\text{\textcircled{1}}}{\leq}\sum_{X\in\mathfrak{N}}\mathbb{E}\left[\left|\ell'_{s,t}-\tfrac{1}{|\mathfrak{B}|}\right|\cdot|h_{i,t}(X)|\cdot\left|\langle \nabla_{\boldsymbol{w}_i} h(\boldsymbol{X}_n), \boldsymbol{M}_j\rangle\right|\right] \tag{254}$$

$$\overset{\text{\textcircled{2}}}{\leq}\widetilde{O}\left(\frac{\sum_{i\in[m]}\|\boldsymbol{w}_i^{(t)}\|_2^2}{\tau d}\cdot\|\boldsymbol{w}_i^{(t)}\|_2\right).$$

① is because The product of each term is less than the product of their absolute values. ② is applied Lemma H.1 to $\left|\left(\ell_{s,t}' - \frac{1}{|\mathfrak{B}|}\right)\right|$, Lemma E.7 to $h_{i,t}(X)$ and $|\langle \nabla_{\boldsymbol{w}_i} h_{i,t}(\boldsymbol{X}_n), \boldsymbol{M}_j\rangle| = |\langle \boldsymbol{M}_j, \sum_{r=1}^L \boldsymbol{z}_{\boldsymbol{X}}^{(r)}\rangle|$.

Putting all the above calculations together, we have

$$
\begin{aligned}
\langle \boldsymbol{w}_i^{(t+1)}, \boldsymbol{M}_j\rangle = {}& \left(1 - \eta\lambda + \epsilon_j \frac{\eta C_z \log\log d}{d}\right) \langle \boldsymbol{w}_i^{(t)}, \boldsymbol{M}_j\rangle \\
&\pm \widetilde{O}\left(\frac{\eta \sum_{i\in[m]} \|\boldsymbol{w}_i^{(t)}\|_2^2}{\tau d} \cdot \|\boldsymbol{w}_i^{(t)}\|_2\right) \pm \widetilde{O}\left(\frac{\eta\|\boldsymbol{w}_i^{(t)}\|_2}{\mathrm{poly}(d_1)}\right).
\end{aligned}
\tag{255}
$$

Prior to the induction step, we establish, by a similar method, the stochastic gradient descent update of $\boldsymbol{w}_i$ along the dense feature direction $\boldsymbol{M}_j^\perp$. Specifically, we obtain the following update equation:

$$
\begin{aligned}
&\langle \boldsymbol{w}_i^{(t+1)}, \boldsymbol{M}_j^\perp\rangle \\
={}&\langle \boldsymbol{w}_i^{(t)}, \boldsymbol{M}_j^\perp\rangle - \eta\langle \nabla_{\boldsymbol{w}_i} L_{\mathrm{aug}}(f_t), \boldsymbol{M}_j^\perp\rangle \\
={}&(1 - \eta\lambda)\langle \boldsymbol{w}_i^{(t)}, \boldsymbol{M}_j^\perp\rangle + \eta\,\mathbb{E}\big[(1 - \ell_{p,t}')\, h_{i,t}(\boldsymbol{Y}_n)\langle \nabla_{\boldsymbol{w}_i} h_{i,t}(\boldsymbol{X}_n), \boldsymbol{M}_j^\perp\rangle\big] \\
&- \eta \sum_{\boldsymbol{X}_{n,s}\in\mathfrak{N}} \mathbb{E}\big[\ell_{s,t}'\, h_{i,t}(\boldsymbol{X}_{n,s})\langle \nabla_{\boldsymbol{w}_i} h(\boldsymbol{X}_n), \boldsymbol{M}_j^\perp\rangle\big] + \frac{\eta\|\boldsymbol{w}_i^{(t)}\|_2}{\mathrm{poly}(d_1)} \\
={}&(1 - \eta\lambda)\langle \boldsymbol{w}_i^{(t)}, \boldsymbol{M}_j^\perp\rangle \pm \widetilde{O}\left(\frac{\eta \sum_{i\in[m]} \|\boldsymbol{w}_i^{(t)}\|_2^2}{\tau d} \cdot \|\boldsymbol{w}_i^{(t)}\|_2\right) \pm \widetilde{O}\left(\eta\frac{\|\boldsymbol{w}_i^{(t)}\|_2}{\mathrm{poly}(d_1)}\right).
\end{aligned}
\tag{256}
$$

Then we can begin to perform our induction: at $t = 0$, our properties holds trivially. Now suppose before iteration $t = t_1$, the claimed properties holds, then we can easily obtain that for all $t \leq t_1$:

$$
\|\boldsymbol{w}_i^{(t)}\|_2^2 = \|\boldsymbol{M}\boldsymbol{M}^\top \boldsymbol{w}_i^{(t)}\|_2^2 + \|\boldsymbol{M}^\perp(\boldsymbol{M}^\perp)^\top \boldsymbol{w}_i^{(t)}\|_2^2 \overset{(1)}{\leq} O(1)\|\boldsymbol{w}_i^{(0)}\|_2^2 \overset{(2)}{\leq} \frac{1}{\mathrm{poly}(d_1)}.
\tag{257}
$$

(1) is Using Lemma C.1 along with the Lemma B.2.

$$
\begin{aligned}
\|\boldsymbol{w}_i^{(t)}\|_2^2 &= \|\boldsymbol{M}\boldsymbol{M}^\top \boldsymbol{w}_i^{(t)}\|_2^2 + \|\boldsymbol{M}^\perp(\boldsymbol{M}^\perp)^\top \boldsymbol{w}_i^{(t)}\|_2^2 \\
&\leq \left\|\boldsymbol{M}\boldsymbol{M}^\top \boldsymbol{w}_i^{(0)}\right\|_2^2 \left(1 + \epsilon_{\max}\frac{\eta C_z \log\log d}{d}\right)^{2t} + \left\|\boldsymbol{M}^\perp(\boldsymbol{M}^\perp)^\top \boldsymbol{w}_i^{(0)}\right\|_2^2 \\
&= \left\|\boldsymbol{M}\boldsymbol{M}^\top \boldsymbol{w}_i^{(0)}\right\|_2^2 \cdot \Theta(\frac{d_1}{d}) + \left\|\boldsymbol{M}^\perp(\boldsymbol{M}^\perp)^\top \boldsymbol{w}_i^{(0)}\right\|_2^2 \\
&= \frac{d}{d_1}\|\boldsymbol{w}_i^{(0)}\|_2^2 \cdot \Theta(\frac{d_1}{d}) + \|\boldsymbol{w}_i^{(0)}\|_2^2 \\
&= (1 + \epsilon_{\max} C_z \log d)\,\|\boldsymbol{w}_i^{(0)}\|_2^2 \\
&\leq O(1)\|\boldsymbol{w}_i^{(0)}\|_2^2.
\end{aligned}
\tag{258}
$$

(2) is because:

$$
\begin{aligned}
\|\boldsymbol{w}_i^{(0)}\|_2^2 &\leq \sigma_0^2 d_1 \left(1 + \widetilde{O}\left(\frac{1}{\sqrt{d_1}}\right)\right) \\
&\leq \Theta\left(\frac{1}{\mathrm{poly}(d_1)}\right) d_1 \left(1 + \widetilde{O}\left(\frac{1}{\sqrt{d_1}}\right)\right) \\
&= \frac{1}{\mathrm{poly}(d_1)}.
\end{aligned}
\tag{259}
$$

Thus we have $\sum_{i\in[m]} \|\boldsymbol{w}_i^{(t)}\|_2^2 \leq \frac{1}{\mathrm{poly}(d_1)}$. We now begin to verify all the properties for $t = t_1 + 1$, until $t_1$ reaches $T_1$.

We first derive an upper bound for

$$\left\| \boldsymbol{M}\boldsymbol{M}^\top \boldsymbol{w}_i^{(t+1)} \right\|_2^2, \tag{260}$$

at iterations $t \leq t_1$. For each $j \in [d]$, as long as

$$\langle \boldsymbol{w}_i^{(t)}, \boldsymbol{M}_j \rangle \geq \Omega \left( \frac{\left\| \boldsymbol{w}_i^{(t)} \right\|_2}{d\sqrt{d_1}} \right), \tag{261}$$

then

$$
\begin{aligned}
|\langle \boldsymbol{w}_i^{(t+1)}, \boldsymbol{M}_j \rangle| &= \left( 1 - \eta\lambda + \epsilon_j \frac{\eta C_z \log\log d}{d} \right) \langle \boldsymbol{w}_i^{(t)}, \boldsymbol{M}_j \rangle \\
&\quad \pm \widetilde{O} \left( \eta \frac{\sum_{i \in [m]} \|\boldsymbol{w}_i^{(t)}\|_2^2}{\tau d} \cdot \|\boldsymbol{w}_i^{(t)}\|_2 \right) \pm \widetilde{O} \left( \eta \frac{\|\boldsymbol{w}_i^{(t)}\|_2}{\text{poly}(d_1)} \right) \\
&\leq \left( 1 - \eta\lambda + \epsilon_j \frac{\eta C_z \log\log d}{d} \right) \langle \boldsymbol{w}_i^{(t)}, \boldsymbol{M}_j \rangle + \widetilde{O} \left( \frac{\eta \|\boldsymbol{w}_i^{(t)}\|_2}{\text{poly}(d_1)} \right) \\
&= \left( 1 + \epsilon_j \frac{\eta C_z \log\log d}{d} \right) \langle \boldsymbol{w}_i^{(t)}, \boldsymbol{M}_j \rangle + \widetilde{O} \left( \frac{\eta \|\boldsymbol{w}_i^{(t)}\|_2}{\text{poly}(d_1)} \right) - \eta\lambda \langle \boldsymbol{w}_i^{(t)}, \boldsymbol{M}_j \rangle \\
&\leq \left( 1 + \epsilon_j \frac{\eta C_z \log\log d}{d} \right) \langle \boldsymbol{w}_i^{(t)}, \boldsymbol{M}_j \rangle + \widetilde{O} \left( \frac{\eta \|\boldsymbol{w}_i^{(t)}\|_2}{d_1} \right) \\
&= \left( 1 + \epsilon_j \frac{\eta C_z \log\log d}{d} \right) \langle \boldsymbol{w}_i^{(t)}, \boldsymbol{M}_j \rangle + \widetilde{O} \left( \frac{\eta \|\boldsymbol{w}_i^{(t)}\|_2}{\sqrt{d_1}\sqrt{d_1}} \right) \\
&= \left( 1 + \epsilon_j \frac{\eta C_z \log\log d}{d} \right) \langle \boldsymbol{w}_i^{(t)}, \boldsymbol{M}_j \rangle + \widetilde{O} \left( \frac{\eta \|\boldsymbol{w}_i^{(t)}\|_2}{\sqrt{d^6}\sqrt{d_1}} \right) \\
&\leq \left( 1 + \epsilon_j \frac{\eta C_z \log\log d}{d} \right) \langle \boldsymbol{w}_i^{(t)}, \boldsymbol{M}_j \rangle + \widetilde{O} \left( \frac{\eta}{d^2} \right) \Omega \left( \frac{\|\boldsymbol{w}_i^{(t)}\|_2}{d\sqrt{d_1}} \right) \\
&\leq \left( 1 + \epsilon_j \frac{\eta C_z \log\log d}{d} \right) \langle \boldsymbol{w}_i^{(t)}, \boldsymbol{M}_j \rangle + \widetilde{O} \left( \frac{\eta}{d^2} \right) \langle \boldsymbol{w}_i^{(t)}, \boldsymbol{M}_j \rangle \\
&\leq \left( 1 + \epsilon_j \frac{\eta C_z \log\log d}{d} \right) |\langle \boldsymbol{w}_i^{(t)}, \boldsymbol{M}_j \rangle| + \widetilde{O} \left( \frac{\eta}{d^2} \right) |\langle \boldsymbol{w}_i^{(t)}, \boldsymbol{M}_j \rangle| \\
&= \left( 1 + \epsilon_j \frac{\eta C_z \log\log d}{d} + \widetilde{O} \left( \frac{\eta}{d^2} \right) \right) |\langle \boldsymbol{w}_i^{(t)}, \boldsymbol{M}_j \rangle|.
\end{aligned} \tag{262}
$$

Define set of features:

$$\mathcal{E}^{(t)} := \left\{ j \in [d] : \langle \boldsymbol{w}_i^{(t)}, \boldsymbol{M}_j \rangle < \widetilde{\mathcal{O}} \left( \frac{\left\| \boldsymbol{w}_i^{(t)} \right\|_2}{d\sqrt{d_1}} \right) \right\}, \tag{263}$$

note that $\mathcal{E}^{(t+1)} \subseteq \mathcal{E}^{(t)} \subseteq \Lambda_i$ (where the set $\Lambda_i$ is defined in Lemma D.2) in the sense that if $j \notin \mathcal{E}^{(t)}$, then

$$\langle \boldsymbol{w}_i^{(t)}, \boldsymbol{M}_j \rangle \geq \widetilde{\mathcal{O}} \left( \frac{\left\| \boldsymbol{w}_i^{(t)} \right\|_2}{d\sqrt{d_1}} \right) \Rightarrow \epsilon_j \frac{C_z \log\log d}{d} \left| \langle \boldsymbol{w}_i^{(t)}, \boldsymbol{M}_j \rangle \right| \geq \widetilde{\mathcal{O}} \left( \frac{\left\| \boldsymbol{w}_i^{(t)} \right\|_2}{d_1} \right), \tag{264}$$

in the above calculations. Therefore:

## H.3 PROOF OF LEMMA C.1(A):

*Proof of Lemma C.1(a):*

$$
\begin{aligned}
\|\boldsymbol{M}\boldsymbol{M}^\top \boldsymbol{w}_i^{(t+1)}\|_2^2 &= \sum_{j\in[d]} \left[ \left(1 - \eta\lambda + \epsilon_j \frac{\eta C_z \log\log d}{d}\right) \langle \boldsymbol{w}_i^{(t)}, \boldsymbol{M}_j\rangle \pm \widetilde{O}\left(\frac{\eta\|\boldsymbol{w}_i^{(t)}\|_2}{\mathrm{poly}(d_1)}\right)\right]^2 \\
&\overset{\text{\textcircled{1}}}{\leq} \sum_{j\in[d]} \langle \boldsymbol{w}_i^{(0)}, \boldsymbol{M}_j\rangle^2 \left(1 + \epsilon_j \frac{\eta C_z \log\log d}{d} + \widetilde{O}\left(\frac{\eta}{d^2}\right)\right)^{2t+2} \\
&\quad + \sum_{j\in[d]:j\in\mathcal{E}^{(0)}} \widetilde{O}\left(\frac{(t+1)^2\eta^2 \max_{t\leq t_1}\|\boldsymbol{w}_i^{(t)}\|_2^2}{\mathrm{poly}(d_1)}\right) \\
&\overset{\text{\textcircled{2}}}{\leq} \|\boldsymbol{M}\boldsymbol{M}^\top \boldsymbol{w}_i^{(0)}\|_2^2 \left(1 + \epsilon_{\max}\frac{\eta C_z \log\log d}{d} + \widetilde{O}\left(\frac{\eta}{d^2}\right)\right)^{2t+2} \\
&\quad + O(1/d)\|\boldsymbol{M}\boldsymbol{M}^\top \boldsymbol{w}_i^{(0)}\|_2^2.
\end{aligned}
\tag{265}
$$

\textcircled{1} is because:

$$
\begin{aligned}
&\|\boldsymbol{M}\boldsymbol{M}^\top \boldsymbol{w}_i^{(t+1)}\|_2^2 \\
&= \sum_{j\in[d]} \langle \boldsymbol{w}_i^{(t+1)}, \boldsymbol{M}_j\rangle^2 \\
&= \sum_{j\in[d]} \left[ \left(1 - \eta\lambda + \epsilon_j \frac{\eta C_z \log\log d}{d}\right) \langle \boldsymbol{w}_i^{(t)}, \boldsymbol{M}_j\rangle \pm \widetilde{O}\left(\frac{\eta\|\boldsymbol{w}_i^{(t)}\|_2}{\mathrm{poly}(d_1)}\right)\right]^2 \\
&= \sum_{j\in[d]:j\notin\mathcal{E}^{(t)}} \left[ \left(1 - \eta\lambda + \epsilon_j \frac{\eta C_z \log\log d}{d}\right) \langle \boldsymbol{w}_i^{(t)}, \boldsymbol{M}_j\rangle \pm \widetilde{O}\left(\frac{\eta\|\boldsymbol{w}_i^{(t)}\|_2}{\mathrm{poly}(d_1)}\right)\right]^2 \\
&\quad + \sum_{j\in[d]:j\in\mathcal{E}^{(t)}} \left[ \left(1 - \eta\lambda + \epsilon_j \frac{\eta C_z \log\log d}{d}\right) \langle \boldsymbol{w}_i^{(t)}, \boldsymbol{M}_j\rangle \pm \widetilde{O}\left(\frac{\eta\|\boldsymbol{w}_i^{(t)}\|_2}{\mathrm{poly}(d_1)}\right)\right]^2 \\
&\leq \sum_{j\in[d]:j\notin\mathcal{E}^{(t)}} \left[ \left(1 + \epsilon_j \frac{\eta C_z \log\log d}{d} + \widetilde{\mathcal{O}}\left(\frac{\eta}{d^2}\right)\right) \langle \boldsymbol{w}_i^{(t)}, \boldsymbol{M}_j\rangle\right]^2 \\
&\quad + \sum_{j\in[d]:j\in\mathcal{E}^{(t)}} \left[ \langle \boldsymbol{w}_i^{(t)}, \boldsymbol{M}_j\rangle + \widetilde{O}\left(\frac{\eta\|\boldsymbol{w}_i^{(t)}\|_2}{\mathrm{poly}(d_1)}\right)\right]^2 \\
&= \sum_{j\in[d]:j\notin\mathcal{E}^{(0)}} \langle \boldsymbol{w}_i^{(0)}, \boldsymbol{M}_j\rangle^2 \left(1 + \epsilon_j \frac{\eta C_z \log\log d}{d} + \widetilde{O}\left(\frac{\eta}{d^2}\right)\right)^{2(t+1)} \\
&\quad + \sum_{j\in[d]:j\in\mathcal{E}^{(0)}} \widetilde{O}\left(\frac{(t+1)^2\eta^2 \max_{t\leq t_1}\|\boldsymbol{w}_i^{(t)}\|_2^2}{\mathrm{poly}(d_1)}\right).
\end{aligned}
\tag{266}
$$

\textcircled{2} holds because for all $t \leq t_1 \leq T_1 = \Theta\left(\frac{d_1 \log d}{\eta \log\log d}\right)$, the final inequality follows from the calculations below, together with the bound $\|\boldsymbol{w}_i^{(t)}\|_2^2 \leq O(1)\|\boldsymbol{w}_i^{(0)}\|_2^2$ and Lemma B.2.

$$
\begin{aligned}
&\sum_{j\in[d]:j\in\mathcal{E}^{(0)}} \widetilde{O}\left(\frac{(t+1)^2\eta^2 \max_{t\leq t_1}\|\boldsymbol{w}_i^{(t)}\|_2^2}{\mathrm{poly}(d_1)}\right) \\
&\leq \widetilde{O}\left(\frac{d\cdot d_1^2 \cdot \max_{t\leq t_1}\|\boldsymbol{w}_i^{(t)}\|_2^2}{\mathrm{poly}(d_1)}\right) \ll \frac{\|\boldsymbol{w}_i^{(0)}\|_2^2}{d_1} = O(\frac{1}{d})\|\mathbf{M}\mathbf{M}^\top \boldsymbol{w}_i^{(0)}\|_2^2.
\end{aligned}
\tag{267}
$$

The proof of the upper bound actually implies two information: (1) $\left\| \boldsymbol{w}_i^{(t)} \right\|_2 / \left( d\sqrt{d_1} \right)$ serves as a threshold. At time step $t$, if the component of $\boldsymbol{w}_i$ along $\boldsymbol{M}_j$ exceeds this threshold, then this $\boldsymbol{w}_i$ along $\boldsymbol{M}_j$ significantly contributes to learning over the entire space; otherwise, its contribution to the learning of the overall space is negligible. (2) The number of features in $\mathcal{E}^{(t)}$ decreases over time. $\qquad\square$

## H.4 PROOF OF LEMMA C.1(B):

*Proof of Lemma C.1(b):*

$$
\|\boldsymbol{M}\boldsymbol{M}^\top \boldsymbol{w}_i^{(t+1)}\|_2^2
$$

$$
= \sum_{j \in [d]} \left[ \left( 1 - \eta\lambda + \epsilon_j \frac{\eta C_z \log\log d}{d} \right) \langle \boldsymbol{w}_i^{(t)}, \boldsymbol{M}_j \rangle \pm \widetilde{O}\left( \frac{\eta \|\boldsymbol{w}_i^{(t)}\|_2}{\mathrm{poly}(d_1)} \right) \right]^2
$$

$$
= \sum_{j \in [d]} \left[ \left( 1 - \eta\lambda + \epsilon_j \frac{\eta C_z \log\log d}{d} \right)^2 \langle \boldsymbol{w}_i^{(t-1)}, \boldsymbol{M}_j \rangle + 2\widetilde{O}\left( \frac{\eta \|\boldsymbol{w}_i^{(t)}\|_2}{\mathrm{poly}(d_1)} \right) \right.
$$

$$
\left. + \left( 1 - \eta\lambda + \epsilon_j \frac{\eta C_z \log\log d}{d} \right) \widetilde{O}\left( \frac{\eta \|\boldsymbol{w}_i^{(t)}\|_2}{\mathrm{poly}(d_1)} \right) \right]^2
$$

$$
\geq \sum_{j \in [d]} \left[ \left( 1 - \eta\lambda + \epsilon_j \frac{\eta C_z \log\log d}{d} \right)^2 \langle \boldsymbol{w}_i^{(t-1)}, \boldsymbol{M}_j \rangle + 2\widetilde{O}\left( \frac{\eta \|\boldsymbol{w}_i^{(t)}\|_2}{\mathrm{poly}(d_1)} \right) \right]^2 \qquad (268)
$$

$$
\geq \sum_{j \in [d]} \left[ \left( 1 - \eta\lambda + \epsilon_j \frac{\eta C_z \log\log d}{d} \right)^{t+1} \langle \boldsymbol{w}_i^{(0)}, \boldsymbol{M}_j \rangle + \widetilde{O}\left( \frac{(t+1)\eta \|\boldsymbol{w}_i^{(t)}\|_2}{\mathrm{poly}(d_1)} \right) \right]^2
$$

$$
\geq \sum_{j \in \mathcal{E}^{(0)}} \langle \boldsymbol{w}_i^{(0)}, \boldsymbol{M}_j \rangle^2 \left( 1 - \eta\lambda + \epsilon_j \frac{\eta C_z \log\log d}{d} \right)^{2t+2}
$$

$$
- \widetilde{O}\left( \frac{(t+1)^2 \eta^2 d \max_{t \leq t_1} \|\boldsymbol{w}_i^{(t)}\|_2^2}{\mathrm{poly}(d_1)} \right)
$$

$$
\geq \|\boldsymbol{M}\boldsymbol{M}^\top \boldsymbol{w}_i^{(0)}\|_2^2 \left( 1 - \eta\lambda + \epsilon_{\min} \frac{\eta C_z \log\log d}{d} \right)^{2t+2} - O(1/d)\|\boldsymbol{M}\boldsymbol{M}^\top \boldsymbol{w}_i^{(0)}\|_2^2.
$$

We aim to obtain the maximum of the lower bound. The extreme case here is when all $j$ belong to the set $\mathcal{E}^{(0)}$, meaning that the components of $\boldsymbol{w}_i$ on all feature $\boldsymbol{M}_j$ are smaller than the critical value $\left\| \boldsymbol{w}_i^{(t)} \right\|_2 / \left( d\sqrt{d_1} \right)$. where the last inequality follows from our computations of the upper bound. $\quad\square$

## H.5 PROOF OF LEMMA C.1(C):

*Proof of Lemma C.1(c):* Finally we give an upper bound of

$$
\left\| \boldsymbol{M}^\perp (\boldsymbol{M}^\perp)^\top \boldsymbol{w}_i^{(t+1)} \right\|_2^2 \qquad (269)
$$

for iterations $t \leq t_1$. We can calculate similarly, by

$$\|\boldsymbol{M}^{\perp}(\boldsymbol{M}^{\perp})^{\top}\boldsymbol{w}_i^{(t+1)}\|_2^2$$

$$= \sum_{j \in [d_1]\setminus[d]} \left[(1-\eta\lambda)\langle\boldsymbol{w}_i^{(t)}, \boldsymbol{M}_j^{\perp}\rangle \pm \widetilde{O}\left(\frac{\eta\|\boldsymbol{w}_i^{(t)}\|_2}{\text{ploy}(d_1)}\right)\right]^2$$

$$\leq \sum_{j \in [d_1]\setminus[d]} \left[(1-\eta\lambda)\langle\boldsymbol{w}_i^{(t)}, \boldsymbol{M}_j^{\perp}\rangle \pm \widetilde{O}\left(\frac{\eta\|\boldsymbol{w}_i^{(t)}\|_2}{\text{poly}(d_1)}\right)\right]^2$$

$$\leq \sum_{j \in [d_1]\setminus[d]} \left[\langle\boldsymbol{w}_i^{(t)}, \boldsymbol{M}_j^{\perp}\rangle + \widetilde{O}\left(\frac{\eta\|\boldsymbol{w}_i^{(t)}\|_2}{\text{poly}(d_1)}\right)\right]^2$$

$$\leq \sum_{j \in [d_1]\setminus[d]} \left[\langle\boldsymbol{w}_i^{(0)}, \boldsymbol{M}_j^{\perp}\rangle + \widetilde{O}\left(\frac{\eta(t+1)\max_{t\leq t_1}\|\boldsymbol{w}_i^{(t)}\|_2}{\text{poly}(d_1)}\right)\right]^2$$

$$= \sum_{j \in [d_1]\setminus[d]} \langle\boldsymbol{w}_i^{(0)}, \boldsymbol{M}_j^{\perp}\rangle^2 + \sum_{j \in [d_1]\setminus[d]} 2\langle\boldsymbol{w}_i^{(0)}, \boldsymbol{M}_j^{\perp}\rangle\widetilde{O}\left(\frac{\eta(t+1)\max_{t\leq t_1}\|\boldsymbol{w}_i^{(t)}\|_2}{\text{poly}(d_1)}\right) \qquad (270)$$

$$+ \sum_{j \in [d_1]\setminus[d]} \widetilde{O}\left(\frac{\eta(t+1)\max_{t\leq t_1}\|\boldsymbol{w}_i^{(t)}\|_2}{\text{poly}(d_1)}\right)^2$$

$$\leq \|\boldsymbol{M}^{\perp}(\boldsymbol{M}^{\perp})^{\top}\boldsymbol{w}_i^{(0)}\|_2^2 + \max_{j \in [d_1]\setminus[d]}|\langle\boldsymbol{w}_i^{(0)}, \boldsymbol{M}_j^{\perp}\rangle|\widetilde{O}\left(\max_{t\leq t_1}\|\boldsymbol{w}_i^{(t)}\|_2\right)$$

$$+ \widetilde{O}\left(\frac{\eta^2(t+1)^2 d\max_{t\leq t_1}\|\boldsymbol{w}_i^{(t)}\|_2^2}{\text{poly}(d_1)}\right)$$

$$\overset{①}{\leq} (1+\widetilde{O}(d/\sqrt{d_1}))\|\boldsymbol{M}^{\perp}(\boldsymbol{M}^{\perp})^{\top}\boldsymbol{w}_i^{(0)}\|_2^2 + O(1/\text{poly}(d_1))\|\boldsymbol{w}_i^{(0)}\|_2^2$$

$$\overset{②}{\leq} \left(1+\frac{1}{\text{poly}(d)}\right)\|\boldsymbol{M}^{\perp}(\boldsymbol{M}^{\perp})^{\top}\boldsymbol{w}_i^{(0)}\|_2^2.$$

① is because: at initialization, we have $\left|\langle\boldsymbol{w}_i^{(0)}, \boldsymbol{M}_j^{\perp}\rangle\right| \leq \widetilde{O}\left(\|\boldsymbol{w}_i^{(0)}\|/\sqrt{d_1}\right)$ with high probability; from our Lemma C.1 $\widetilde{O}\left(\max_{t\leq t_1}\|\boldsymbol{w}_i^{(t)}\|_2\right) \leq O(1)\|\boldsymbol{w}_i^{(0)}\|_2^2$; at initialization we have $\|\boldsymbol{w}_i^{(0)}\|_2^2 \leq O\left(\|\boldsymbol{M}^{\perp}(\boldsymbol{M}^{\perp})^{\top}\boldsymbol{w}_i^{(0)}\|_2^2\right)$ with high probability and $t = \Theta\left(\frac{d_1 \log d}{\eta \log\log d}\right)$

② is because: at initialization we have $\|\boldsymbol{w}_i^{(0)}\|_2^2 \leq O\left(\|\boldsymbol{M}^{\perp}(\boldsymbol{M}^{\perp})^{\top}\boldsymbol{w}_i^{(0)}\|_2^2\right)$ with high probability

Note that for each neuron $i \in [m]$, from Lemma B.2 combined with our upper bound and lower bound, we know when all the weights $\|\boldsymbol{w}_i^{(t)}\|_2^2$ reach $\Theta(1)\|\boldsymbol{w}_i^{(0)}\|_2^2$, the maximum:

$$\max_{i\in[m]} \|\boldsymbol{w}_i^{(t+1)}\|_2^2 \leq O(1)(\left\|\boldsymbol{M}\boldsymbol{M}^{\top}\boldsymbol{w}_i^{(t+1)}\right\|_2^2 + \left\|\boldsymbol{M}^{\perp}(\boldsymbol{M}^{\perp})^{\top}\boldsymbol{w}_i^{(t+1)}\right\|_2^2 = O(1)\|\boldsymbol{w}_i^{(t+1)}\|_2^2) \quad (271)$$

for all $t \leq t_1$. Thus we have obtained all the results for $t = t_1 + 1$, and are able to proceed induction. $\qquad \square$

$\qquad \square$

## H.6 PROOF OF LEMMA C.2:

*Proof of Lemma C.2:*

$$\Pr\left[\left|\langle\boldsymbol{w}_i^{(0)}, \boldsymbol{M}_j\rangle\right| \geq \Omega(\sigma_0 \log^{1/4} d)\right] \leq 2\exp\left(-\frac{t^2}{2\sigma_0^2}\right) = 2^{-\Omega(\sqrt{\log d})}. \qquad (272)$$

Fix $i \in [m]$, and consider $d$ different $j$. Each has probability $2^{-\Omega(\sqrt{\log d})}$ to exceed the threshold. Therefore, the expectation is:

$$\mathbb{E}\left[\left|\left\{j \in [d] \,\middle|\, \left|\langle \boldsymbol{w}_i^{(0)}, \boldsymbol{M}_j\rangle\right| \geq \Omega\left(\sigma_0 \log^{1/4} d\right)\right\}\right|\right] = O\left(2^{-\sqrt{\log d}} \cdot d\right). \tag{273}$$

$\square$

## I   PROOF OF LEMMAS IN APPENDIX D

### I.1   PROOF OF LEMMA D.1(A):

*Proof of Lemma D.1(a):*   At iteration $t = T_1$, we have verified all the above properties in Theorem C.1. Now suppose all the properties hold for $t < T_2$, we will verify that it still hold for $t+1$. In order to calculate the gradient $\nabla_{\boldsymbol{w}_i} L_{\text{aug}}$ along each feature $\boldsymbol{M}_j$ or $\boldsymbol{M}_j^{\perp}$, we have to apply Lemma E.10, Lemma E.11 and Lemma F.3. First we calculate parameters in Lemma E.10(a) and (b). In order to using Lemma E.10, we have the followings

$$\left|\bar{S}_{i,t}^{(r,s)\backslash j}\right|^2 = \left|\langle \boldsymbol{w}_i^{(t)}, \frac{z_Y^{(s)\backslash j} - z_X^{(r)\backslash j}}{2}\rangle\right|^2 \leq \widetilde{\mathcal{O}}\left(\frac{\|\boldsymbol{w}_i^{(t)}\|_2^2}{d}\right) \tag{274}$$

$$\Pr(A_1), \Pr(A_2) \leq e^{-\Omega(\log^{1/4} d)} \quad \text{when } |\alpha_{i,j}^{(t)}| \leq \sqrt{(1-\gamma)}b_i^{(t)} \tag{275}$$

$$\Pr(A_3), \Pr(A_4) \leq e^{-\Omega(\log^{1/4} d)} \quad \text{when } |\alpha_{i,j}^{(t)}| \geq \sqrt{(1+\gamma)}b_i^{(t)} \tag{276}$$

Which further implies that when $(|\alpha_{i,j}^{(t)}| \leq \sqrt{(1-\gamma)})$

$$\sqrt{\mathbb{E}[|\bar{S}^{(r,s)\backslash j}|^2(\mathbf{1}_{A_1} + \mathbf{1}_{A_2})]} \leq \sqrt{\widetilde{\mathcal{O}}\left(\frac{\|\boldsymbol{w}_i^{(t)}\|_2^2}{d}\right)(\Pr(A_1) + \Pr(A_2))} \overset{(1)}{\leq} \frac{\|\boldsymbol{w}_i^{(t)}\|_2}{\sqrt{d} \cdot \text{polylog}(d)} \tag{277}$$

(1) is because: $e^{-\Omega(\log^{1/4} d)} \leq \frac{1}{\text{polylog}(d)}$,

$L_1$ is because:

$$L_1 := \sqrt{\frac{\mathbb{E}[|\bar{S}_{i,t}^{(r,s)\backslash j}|^2(\mathbf{1}_{A_1} + \mathbf{1}_{A_2})]}{\mathbb{E}[\langle \boldsymbol{w}_i^{(t)}, \frac{\tilde{\xi}_n + \tilde{\xi}_n^+}{2}\rangle^2]}} \leq \sqrt{\frac{\frac{\|\boldsymbol{w}_i^{(t)}\|_2^2}{d \cdot \text{polylog}(d)}}{\Theta\left(\frac{\|\boldsymbol{w}_i^{(t)}\|_2^2 \sqrt{\log d}}{d}\right)}} \leq \frac{1}{\text{polylog}(d)} \tag{278}$$

$L_2 = \Pr(A_2)$ holds as well

$$\Rightarrow L_1, L_2 \leq \frac{1}{\text{polylog}(d)} \tag{279}$$

And similarly, we also have $L_3, L_4 \leq \frac{1}{\text{polylog}(d)}$ when $|\alpha_{i,j}^{(t)}| \geq \sqrt{(1+\gamma)}b_i^{(t)}$.

Now we separately discuss three cases:

(a) When $i \in \mathcal{M}_j^{\star}$, if $\tilde{z}_{n,j}^{+(s)}$ and $\tilde{z}_{n,j}^{(r)} \neq 0$, say $\frac{\tilde{z}_{n,j}^{+(s)} + \tilde{z}_{n,j}^{(r)}}{2} = C_{\tilde{z}}^{(r,s)}$, we simply have

$$\Pr\left(\left|\langle \boldsymbol{w}_i, \frac{z_X^{(r)} + z_Y^{(s)}}{2}\rangle\right| \geq b_i + |\langle \boldsymbol{w}_i, \boldsymbol{z}_X^{(r)} - \frac{z_X^{(r)} + z_Y^{(s)}}{2}\rangle|\right)$$

$$\geq 1 - \Pr\left(\left|\langle \boldsymbol{w}_i^{(t)}, \frac{z_X^{(r)\backslash j} + z_Y^{(s)\backslash j}}{2}\rangle\right| \geq b_i^{(t)} - \alpha_{i,j}^{(t)}\frac{\tilde{z}_{n,j}^{+(s)} + \tilde{z}_{n,j}^{(r)}}{2} + \left|\langle \boldsymbol{w}_i^{(t)}, \boldsymbol{z}_X^{(r)} - \frac{z_X^{(r)} + z_Y^{(s)}}{2}\rangle\right|\right) \tag{280}$$

From the observations that: (1) $\left|\langle \boldsymbol{w}_i^{(t)}, \frac{z_X^{(r)\backslash j} + z_Y^{(s)\backslash j}}{2}\rangle\right| \leq \gamma b_i^{(t)}$ with probability $\geq 1 - e^{-\Omega(\log^{1/4} d)}$;

(2) $\left|\langle \boldsymbol{w}_i^{(t)}, \boldsymbol{z}_{\boldsymbol{X}}^{(r)} - \frac{z_X^{(r)} + z_Y^{(s)}}{2}\rangle\right| \leq O(\|\boldsymbol{w}_i^{(t)}\|_2 \sigma_\xi)$ with prob $\geq 1 - e^{-\Omega(\log^{1/2} d)}$.

So it can be easily verified that:

$$
\begin{aligned}
&\mathbb{E}\left[\sum_{s=1}^{L} \tilde{z}_{n,j}^{+(s)} \sum_{r=1}^{L} \tilde{z}_{n,j}^{(r)} \mathbf{1}_{|\langle \boldsymbol{w}_i, \frac{z_X^{(r)} + z_Y^{(s)}}{2}\rangle| \geq b_i + |\langle \boldsymbol{w}_i, \boldsymbol{z}_{\boldsymbol{X}}^{(r)} - \frac{z_X^{(r)} + z_Y^{(s)}}{2}\rangle|}\right] \\
=&\mathbb{E}\left[\sum_{s=1}^{L} \tilde{z}_{n,j}^{+(s)} \sum_{r=1}^{L} \tilde{z}_{n,j}^{(r)}\right] \Pr\left(|\langle \boldsymbol{w}_i^{(t)}, \frac{z_X^{(r)} + z_Y^{(s)}}{2}\rangle| \geq b_i + |\langle \boldsymbol{w}_i^{(t)}, \boldsymbol{z}_{\boldsymbol{X}}^{(r)} - \frac{z_X^{(r)} + z_Y^{(s)}}{2}|\rangle\right) \quad (281) \\
=&\epsilon_j \frac{L^2 C_z \log\log d}{d}\left(1 - \frac{1}{\text{polylog}(d)}\right)
\end{aligned}
$$

Now we can compute as follows: for $\boldsymbol{M}_j$ such that $i \in \mathcal{M}_j^\star$, at iteration $t+1$:

$$
\begin{aligned}
&\langle \boldsymbol{w}_i^{(t+1)}, \boldsymbol{M}_j\rangle \\
=&\langle \boldsymbol{w}_i^{(t)}, \boldsymbol{M}_j\rangle - \eta\langle\nabla_{\boldsymbol{w}_i} L_{\text{aug}}(f_t), \boldsymbol{M}_j\rangle \pm \frac{\eta\|\boldsymbol{w}_i^{(t)}\|_2}{\text{poly}(d_1)} \\
=&\langle \boldsymbol{w}_i^{(t)}, \boldsymbol{M}_j\rangle(1 - \eta\lambda) \pm \frac{\eta\|\boldsymbol{w}_i^{(t)}\|_2}{\text{poly}(d_1)} \\
&+\eta\mathbb{E}\left[(1 - \ell_{p,t}^t)h_{i,t}(\boldsymbol{Y}_n)\sum_{r=1}^{L}\mathbf{1}_{|\langle \boldsymbol{w}_i, \boldsymbol{z}_{\boldsymbol{X}}^{(r)}\rangle| \geq b_i}\langle \boldsymbol{z}_{\boldsymbol{X}}^{(r)}, \boldsymbol{M}_j\rangle\right] \\
&-\eta\mathbb{E}\left[\sum_{X\in\mathfrak{N}}\ell_{s,t}^t h_{i,t}(X)\sum_{r=1}^{L}\mathbf{1}_{|\langle \boldsymbol{w}_i, \boldsymbol{z}_{\boldsymbol{X}}^{(r)}\rangle| \geq b_i}\langle \boldsymbol{z}_{\boldsymbol{X}}^{(r)}, \boldsymbol{M}_j\rangle\right] \\
\overset{①}{=}&\langle \boldsymbol{w}_i^{(t)}, \boldsymbol{M}_j\rangle(1 - \eta\lambda) \pm \frac{\eta\|\boldsymbol{w}_i^{(t)}\|_2}{\text{poly}(d_1)} \\
&+\eta\frac{1}{L}\mathbb{E}\left[(1 - \ell_{p,t}^t)h_{i,t}(\boldsymbol{Y}_n)\sum_{r=1}^{L}\mathbf{1}_{|\langle \boldsymbol{w}_i, \boldsymbol{z}_{\boldsymbol{X}}^{(r)}\rangle| \geq b_i}(\tilde{z}_{n,j}^{(r)} + \langle\tilde{\xi}^{(r)}, \boldsymbol{M}_j\rangle)\right] \\
&-\eta\frac{1}{L}\mathbb{E}\left[\sum_{X\in\mathfrak{N}}\ell_{s,t}^t h_{i,t}(X)\sum_{r=1}^{L}\mathbf{1}_{|\langle \boldsymbol{w}_i, \boldsymbol{z}_{\boldsymbol{X}}^{(r)}\rangle| \geq b_i}(\tilde{z}_{n,j}^{(r)} + \langle\tilde{\xi}_n^{(r)}, \boldsymbol{M}_j\rangle)\right] \\
\overset{②}{\geq}&\left(\langle \boldsymbol{w}_i^{(t)}, \boldsymbol{M}_j\rangle - \text{sign}(\langle \boldsymbol{w}_i^{(t)}, \boldsymbol{M}_j\rangle)\cdot b_i^{(t)}\right)\cdot\left(1 - \eta\lambda + \epsilon_j\frac{\eta C_z\log\log d}{d}\left(1 - \frac{1}{\text{polylog}(d)}\right)\right) \\
&-O\left(\frac{\eta|\langle \boldsymbol{w}_i^{(t)}, \boldsymbol{M}_j\rangle|}{d\cdot\text{polylog}(d)}\right) \pm O\left(\frac{\eta\sum_{i'\in[m]}\|w_{i'}^{(t)}\|_2^2\|\boldsymbol{w}_i^{(t)}\|_2}{d\tau}\right) \pm \widetilde{O}\left(\frac{\eta\|\boldsymbol{w}_i^{(t)}\|_2}{d\sqrt{d_1}}\right) \\
\overset{③}{\geq}&\left(\langle \boldsymbol{w}_i^{(t)}, \boldsymbol{M}_j\rangle - \text{sign}(\langle \boldsymbol{w}_i^{(t)}, \boldsymbol{M}_j\rangle)\cdot b_i^{(t)}\right)\cdot\left(1 + \epsilon_j\frac{\eta C_z\log\log d}{d}\left(1 - \frac{\eta}{\text{polylog}(d)}\right)\right)
\end{aligned}
$$
$$(282)$$

① is because $\langle \boldsymbol{M}\tilde{z}_n^{(r)}, \boldsymbol{M}_j\rangle = \langle\sum_{k=1}^{d}\tilde{z}_{n,k}^{(r)}\boldsymbol{M}_k, \boldsymbol{M}_j\rangle = \sum_{k=1}^{d}\tilde{z}_{n,k}^{(r)}\langle \boldsymbol{M}_k, \boldsymbol{M}_j\rangle = \tilde{z}_{n,j}^{(r)}$. ② is because Lemma E.11. ③ is because we have taken into consideration

$$
\sum_{i\in[m]}\|\boldsymbol{w}_i^{(t)}\|_2^2 \leq \sum_{i\in[m]}\|\boldsymbol{w}_i^{(T_2)}\|_2^2 \leq \sum_{i\in[m]}d\|\boldsymbol{w}_i^{(T_1)}\|_2^2 \leq \frac{1}{\text{poly}(d)} \quad (283)
$$

which follows from our definition of iteration $T_2$ and the properties at iteration $T_1$ in Theorem C.1, and also that

$$\frac{\langle \boldsymbol{w}_i^{(t)}, \boldsymbol{M}_j \rangle}{d} \geq \frac{b_i^{(t)}}{d} \geq \frac{b_i^{(T_1)}}{d} \geq \sqrt{\frac{2 \log d}{d^3}} \|\boldsymbol{w}_i^{(T_1)}\|_2 \geq \sqrt{\frac{2 \log d}{d^4}} \|\boldsymbol{w}_i^{(T_2)}\|_2 \gg \frac{\|\boldsymbol{w}_i^{(t)}\|_2}{\sqrt{d_1}} \quad (284)$$

Next we compare this growth to the growth of bias $b_i^{(t+1)}$. Since we raise our bias by

$$b_i^{(t+1)} = \max \left\{ b_i^{(t)}(1 + \frac{\eta}{d}), b_i^{(t)} \frac{\|\boldsymbol{w}_i^{(t+1)}\|_2}{\|\boldsymbol{w}_i^{(t)}\|_2} \right\}, \quad (285)$$

The above inequality $\langle \boldsymbol{w}_i^{(t+1)}, \boldsymbol{M}_j \rangle \geq \left( \langle \boldsymbol{w}_i^{(t)}, \boldsymbol{M}_j \rangle - \text{sign}(\langle \boldsymbol{w}_i^{(t)}, \boldsymbol{M}_j \rangle) \cdot b_i^{(t)} \right) \cdot \left( 1 + \frac{\eta}{d} \right)$ has already verified the case of $(1 + \frac{\eta}{d})$. as long as

$$\frac{\|\boldsymbol{w}_i^{(t+1)}\|_2}{\|\boldsymbol{w}_i^{(t)}\|_2} \leq \frac{|\langle \boldsymbol{w}_i^{(t+1)}, \boldsymbol{M}_j \rangle|}{|\langle \boldsymbol{w}_i^{(t)}, \boldsymbol{M}_j \rangle|}, \quad (286)$$

we can obtain the desired result $\left( \frac{\|\boldsymbol{w}_i^{(t+1)}\|_2}{\|\boldsymbol{w}_i^{(t)}\|_2} \leq \frac{|\langle \boldsymbol{w}_i^{(t+1)}, \boldsymbol{M}_j \rangle|}{|\langle \boldsymbol{w}_i^{(t)}, \boldsymbol{M}_j \rangle|} \right)$. This will be proved later, after we prove (d). $\qquad \square$

## I.2 PROOF OF LEMMA D.1(B):

*Proof of Lemma D.1(b):* When $i \notin \mathcal{M}_j$, we can similarly obtain that

$$\mathbb{E} \left[ \sum_{s=1}^{L} \tilde{z}_{n,j}^{+(s)} \sum_{r=1}^{L} \tilde{z}_{n,j}^{(r)} \mathbf{1}_{|\langle \boldsymbol{w}_i, \frac{z_X^{(r)} + z_Y^{(s)}}{2} \rangle| \geq b_i + |\langle \boldsymbol{w}_i, \boldsymbol{z}_{\mathbf{X}}^{(r)} - \frac{z_X^{(r)} + z_Y^{(s)}}{2} \rangle|} \right]$$
$$\leq \epsilon_j \frac{L^2 C_z \log \log d}{d} \left( \frac{1}{\text{polylog}(d)} \right) \quad (287)$$
$$= O \left( \epsilon_j \frac{L^2}{d \cdot \text{polylog}(d)} \right)$$

And similarly we can compute the gradient descent dynamics as follows:

For $j \in [d]$ such that $|\langle \boldsymbol{w}_i^{(t)}, \boldsymbol{M}_j \rangle| \geq \frac{\|\boldsymbol{w}_i^{(t)}\|_2 d}{\sqrt{d_1}}$, we have (assume here $\langle \boldsymbol{w}_i^{(t)}, \boldsymbol{M}_j \rangle > 0$, the opposite is similar)

$$\langle \boldsymbol{w}_i^{(t+1)}, \boldsymbol{M}_j \rangle$$
$$= \langle \boldsymbol{w}_i^{(t)}, \boldsymbol{M}_j \rangle - \eta \langle \nabla_{\boldsymbol{w}_i} L_{\text{aug}}(f_t), \boldsymbol{M}_j \rangle + \frac{\eta \|\boldsymbol{w}_i^{(t)}\|_2}{\text{poly}(d_1)}$$
$$\leq \langle \boldsymbol{w}_i^{(t)}, \boldsymbol{M}_j \rangle \left( 1 - \eta \lambda + \epsilon_j \frac{O(\eta)}{d \cdot \text{polylog}(d)} \right) \pm O \left( \frac{\eta \sum_{i' \in [m]} \|w_{i'}^{(t)}\|_2^2 \|\boldsymbol{w}_i^{(t)}\|_2}{d\tau} \right) \pm \widetilde{O} \left( \frac{\eta \|\boldsymbol{w}_i^{(t)}\|_2}{d^2} \right)$$
$$\leq \langle \boldsymbol{w}_i^{(t)}, \boldsymbol{M}_j \rangle \left( 1 + \epsilon_j \frac{O(\eta)}{d \cdot \text{polylog}(d)} \right) + \widetilde{O} \left( \frac{\eta \|\boldsymbol{w}_i^{(t)}\|_2}{d^2} \right)$$
$$(288)$$

Since from our update rule

$$b_i^{(t+1)} \geq b_i^{(t)} \left( 1 + \frac{\eta}{d} \right) \quad (289)$$

we know that

$$b_i^{(t+1)} \geq \left( 1 + \frac{\eta}{d} \right) b_i^{(t)} \geq \left( 1 + \frac{O(\eta)}{d \cdot \text{polylog}(d)} \right) b_i^{(t)} \geq \frac{|\langle \boldsymbol{w}_i^{(t+1)}, \boldsymbol{M}_j \rangle|}{|\langle \boldsymbol{w}_i^{(t)}, \boldsymbol{M}_j \rangle|} b_i^{(t)} \quad (290)$$

Thus, if $|\langle \boldsymbol{w}_i^{(t)}, \boldsymbol{M}_j \rangle| \leq \sqrt{1-\gamma} b_i^{(t)}$ at iteration $t$, we have

$$|\langle \boldsymbol{w}_i^{(t+1)}, \boldsymbol{M}_j \rangle| \leq \frac{b_i^{(t+1)} |\langle \boldsymbol{w}_i^{(t)}, \boldsymbol{M}_j \rangle|}{b_i^{(t)}} \leq \sqrt{1-\gamma} b_i^{(t+1)} \quad \text{if } |\langle \boldsymbol{w}_i^{(t)}, \boldsymbol{M}_j \rangle| \geq \frac{\|\boldsymbol{w}_i^{(t)}\|_2 d}{\sqrt{d_1}} \text{ at iteration } t;$$

$$|\langle \boldsymbol{w}_i^{(t+1)}, \boldsymbol{M}_j \rangle| \leq \sqrt{1-\gamma} b_i^{(t+1)} \quad \text{if } |\langle \boldsymbol{w}_i^{(t)}, \boldsymbol{M}_j \rangle| \leq \frac{\|\boldsymbol{w}_i^{(t)}\|_2 d}{\sqrt{d_1}} \text{ at iteration } t.$$

It is also worth noting that similar calculations also lead to a lower bound:

$$|\langle \boldsymbol{w}_i^{(t+1)}, \boldsymbol{M}_j \rangle| \geq |\langle \boldsymbol{w}_i^{(t)}, \boldsymbol{M}_j \rangle| (1 - \eta\lambda) - \widetilde{\mathcal{O}}\left( \eta \frac{\|\boldsymbol{w}_i^{(t)}\|_2}{d^2} \right) \tag{291}$$

We leave the part of proving $|\langle \boldsymbol{w}_i^{(t+1)}, \boldsymbol{M}_j \rangle| \leq \widetilde{\mathcal{O}}\left( \frac{\|\boldsymbol{w}_i^{(t+1)}\|_2}{\sqrt{d}} \right)$ to later. $\qquad \square$

## I.3 Proof of Lemma D.1(d):

*Proof of Lemma D.1(d):* Next we consider the learning dynamics for the dense features. We can use Lemma E.11 to calculate its dynamics by

$$
\begin{aligned}
\langle \boldsymbol{w}_i^{(t+1)}, \boldsymbol{M}_j^\perp \rangle &= \langle \boldsymbol{w}_i^{(t)}, \boldsymbol{M}_j^\perp \rangle (1 - \eta\lambda) \pm \frac{\eta \|\boldsymbol{w}_i^{(t)}\|_2}{\text{poly}(d_1)} \\
&\quad + \eta \mathbb{E}\left[ (1 - \ell'_{p,t}) h_{i,t}(\boldsymbol{Y}_n) \sum_{r=1}^L \mathbf{1}_{|\langle \boldsymbol{w}_i, \boldsymbol{z}_{\boldsymbol{X}}^{(r)} \rangle| \geq b_i} \langle \boldsymbol{z}_{\boldsymbol{X}}^{(r)}, \boldsymbol{M}_j^\perp \rangle \right] \\
&\quad - \eta \sum_{X \in \mathfrak{N}} \mathbb{E}\left[ \ell'_{s,t} h_{i,t}(X) \sum_{r=1}^L \mathbf{1}_{|\langle \boldsymbol{w}_i, \boldsymbol{z}_{\boldsymbol{X}}^{(r)} \rangle| \geq b_i} \langle \boldsymbol{z}_{\boldsymbol{X}}^{(r)}, \boldsymbol{M}_j^\perp \rangle \right] \\
&= \langle \boldsymbol{w}_i^{(t)}, \boldsymbol{M}_j^\perp \rangle (1 - \eta\lambda) \pm \frac{\eta \|\boldsymbol{w}_i^{(t)}\|_2}{\text{poly}(d_1)} \\
&\quad + \eta \mathbb{E}\left[ (1 - \ell'_{p,t}) h_{i,t}(\boldsymbol{Y}_n) \sum_{r=1}^L \mathbf{1}_{|\langle \boldsymbol{w}_i, \boldsymbol{z}_{\boldsymbol{X}}^{(r)} \rangle| \geq b_i} \langle \tilde{\xi}_n^{(r)}, \boldsymbol{M}_j^\perp \rangle \right] \\
&\quad - \eta \sum_{X \in \mathfrak{N}} \mathbb{E}\left[ \ell'_{s,t} h_{i,t}(X) \sum_{r=1}^L \mathbf{1}_{|\langle \boldsymbol{w}_i, \boldsymbol{z}_{\boldsymbol{X}}^{(r)} \rangle| \geq b_i} \langle \tilde{\xi}_n^{(r)}, \boldsymbol{M}_j^\perp \rangle \right] \\
&= \langle \boldsymbol{w}_i^{(t)}, \boldsymbol{M}_j^\perp \rangle (1 - \eta\lambda) + \widetilde{\mathcal{O}}\left( \frac{\eta \|\boldsymbol{w}_i^{(t)}\|_2}{d\sqrt{d_1}} \right) \cdot \Pr\left( h_{i,t}(\boldsymbol{Y}_n) \neq 0 \right) \\
&\leq \langle \boldsymbol{w}_i^{(t)}, \boldsymbol{M}_j^\perp \rangle + O\left( \frac{\eta \|\boldsymbol{w}_i^{(t)}\|_2}{d\sqrt{d_1}} e^{-\Omega(\log^{1/4} d)} \right)
\end{aligned}
\tag{292}
$$

$\qquad \square$

## I.4 Supplement to Lemma D.1(a):

After establishing the bounds of growth speed for each feature, we now calculate the proportions they contribute to each neuron weight $i \in [m]$. Namely, we need to prove that when Lemma D.1 holds at iteration $t \in [T_1, T_2]$, we have to prove:

$$\frac{|\langle \boldsymbol{w}_i^{(t+1)}, \boldsymbol{M}_j \rangle|}{|\langle \boldsymbol{w}_i^{(t)}, \boldsymbol{M}_j \rangle|} \geq \frac{\|\boldsymbol{w}_i^{(t+1)}\|_2}{\|\boldsymbol{w}_i^{(t)}\|_2} \quad \text{for } i \in \mathcal{M}_j^\star, \tag{293}$$

we argue as follows: from previous calculations we have:

$$
\begin{aligned}
\sum_{j' \in [d], j' \neq j} &\langle \boldsymbol{w}_i^{(t+1)}, \boldsymbol{M}_{j'} \rangle^2 + \sum_{j' \in [d_1] \setminus [d]} \langle \boldsymbol{w}_i^{(t+1)}, \boldsymbol{M}_{j'}^{\perp} \rangle^2 \\
&\leq \sum_{j' \in [d], j' \neq j} \langle \boldsymbol{w}_i^{(t)}, \boldsymbol{M}_{j'} \rangle^2 \left( 1 + \epsilon_{j'} \frac{O(\eta)}{d \operatorname{polylog}(d)} \right)^2 \\
&+ \sum_{j' \in [d_1] \setminus [d]} \langle \boldsymbol{w}_i^{(t)}, \boldsymbol{M}_{j'}^{\perp} \rangle^2 + \widetilde{\mathcal{O}} \left( \frac{\eta}{d} \right) e^{-\Omega(\log^{1/4} d)} \| \boldsymbol{w}_i^{(t)} \|_2^2
\end{aligned}
\tag{294}
$$

Therefore by adding $\langle \boldsymbol{w}_i^{(t+1)}, \boldsymbol{M}_j \rangle^2$ to the LHS we have:

$$
\begin{aligned}
\| \boldsymbol{w}_i^{(t+1)} \|_2^2 \leq \| \boldsymbol{w}_i^{(t)} \|_2^2 &\left( 1 + \epsilon_{\max} \frac{O(\eta)}{d \cdot \operatorname{polylog}(d)} \right)^2 \\
&+ \left( \frac{|\langle \boldsymbol{w}_i^{(t+1)}, \boldsymbol{M}_j \rangle|}{|\langle \boldsymbol{w}_i^{(t)}, \boldsymbol{M}_j \rangle|} - \frac{O(\eta)}{d \cdot \operatorname{polylog}(d)} \right) |\langle \boldsymbol{w}_i^{(t)}, \boldsymbol{M}_j \rangle|^2
\end{aligned}
\tag{295}
$$

which implies

$$
\frac{\| \boldsymbol{w}_i^{(t+1)} \|_2^2}{\| \boldsymbol{w}_i^{(t)} \|_2^2} \leq \left( 1 + \epsilon_{\max} \frac{O(\eta)}{d \cdot \operatorname{polylog}(d)} \right)^2 + \left( \frac{|\langle \boldsymbol{w}_i^{(t+1)}, \boldsymbol{M}_j \rangle|}{|\langle \boldsymbol{w}_i^{(t)}, \boldsymbol{M}_j \rangle|} \right) \frac{|\langle \boldsymbol{w}_i^{(t)}, \boldsymbol{M}_j \rangle|^2}{\| \boldsymbol{w}_i^{(t)} \|_2^2}
\tag{296}
$$

Therefore, $\frac{|\langle \boldsymbol{w}_i^{(t+1)}, \boldsymbol{M}_j \rangle|}{|\langle \boldsymbol{w}_i^{(t)}, \boldsymbol{M}_j \rangle|} \geq \frac{\| \boldsymbol{w}_i^{(t+1)} \|_2}{\| \boldsymbol{w}_i^{(t)} \|_2}$ is as desired.

## I.5 SUPPLEMENT TO LEMMA D.1(B):

To prove $|\langle \boldsymbol{w}_i^{(t+1)}, \boldsymbol{M}_j \rangle| \leq \widetilde{O} \left( \frac{\| \boldsymbol{w}_i^{(t+1)} \|_2}{\sqrt{d}} \right)$ if $i \notin \mathcal{M}_j$, we first use inequality to compute

$$
\begin{aligned}
\| &\boldsymbol{M} \boldsymbol{M}^\top \boldsymbol{w}_i^{(t+1)} \|_2 \\
&\geq \| \boldsymbol{M} \boldsymbol{M}^\top \boldsymbol{w}_i^{(T_1)} \|_2 (1 - \eta\lambda)^{t-T_1+1} - O \left( \frac{\eta(t - T_1 + 1) \max_{t' \in [T_1, t+1]} \| \boldsymbol{w}_i^{(t')} \|_2}{\sqrt{dd_1}} \right) \\
&\geq \| \boldsymbol{M} \boldsymbol{M}^\top \boldsymbol{w}_i^{(T_1)} \|_2 (1 - \eta\lambda)^{t-T_1+1} - O \left( \frac{\eta(t - T_1) \| \boldsymbol{w}_i^{(T_1)} \|_2 \sqrt{d}}{\sqrt{d_1}} \right) \\
&\geq \| \boldsymbol{M} \boldsymbol{M}^\top \boldsymbol{w}_i^{(T_1)} \|_2 (1 - o(1)) \quad \text{for} \quad t \leq \Theta \left( \frac{d \log d}{\eta \log \log d} \right)
\end{aligned}
\tag{297}
$$

Notice that

$$
|\langle \boldsymbol{w}_i^{(T_1)}, \boldsymbol{M}_j \rangle| \leq \sqrt{1 - \gamma} \frac{\sqrt{2 \log d}}{\sqrt{d}} \| \boldsymbol{w}_i^{(T_1)} \|_2 \leq O \left( \sqrt{\frac{\log d}{d}} \right) \left\| \boldsymbol{M} \boldsymbol{M}^\top \boldsymbol{w}_i^{(T_1)} \right\|_2
\tag{298}
$$

From Theorem C.1. Suppose it also holds for iteration t, we have

$$|\langle \boldsymbol{w}_i^{(t+1)}, \boldsymbol{M}_j\rangle|$$

$$\leq |\langle \boldsymbol{w}_i^{(t)}, \boldsymbol{M}_j\rangle| \left(1 + \epsilon_j \frac{O(\eta)}{d \operatorname{polylog}(d)}\right) + \widetilde{O}\left(\frac{\eta \|\boldsymbol{w}_i^{(t)}\|_2}{d^2}\right)$$

$$\leq |\langle \boldsymbol{w}_i^{(T_1)}, \boldsymbol{M}_j\rangle| \left(1 + \epsilon_j \frac{O(\eta)}{d \operatorname{polylog}(d)}\right)^{t-T_1} + \widetilde{O}\left(\frac{\eta(t-T_1)\|\boldsymbol{w}_i^{(T_1)}\|_2}{d^{3/2}}\right)$$

(because $\|\boldsymbol{w}_i^{(t)}\|_2 \leq \sqrt{d}\|\boldsymbol{w}_i^{(T_1)}\|_2$ by the definition of $T_2$)

$$\leq |\langle \boldsymbol{w}_i^{(T_1)}, \boldsymbol{M}_j\rangle|(1 + o(1)\frac{\epsilon_j}{\epsilon_{\max}}) + \widetilde{O}\left(\frac{\|\boldsymbol{w}_i^{(T_1)}\|_2}{d}\right) \tag{299}$$

$$\overset{①}{\leq} O\left(\sqrt{\frac{\log d}{d}}\right) \left\|\boldsymbol{M}\boldsymbol{M}^\top \boldsymbol{w}_i^{(T_1)}\right\|_2 (1 + o(1)\frac{\epsilon_j}{\epsilon_{\max}})$$

$$\overset{②}{\leq} O\left(\sqrt{\frac{\log d}{d}}\right) \left\|\boldsymbol{M}\boldsymbol{M}^\top \boldsymbol{w}_i^{(t+1)}\right\|_2 (1 + o(1)\frac{\epsilon_j}{\epsilon_{\max}})$$

$$\leq O\left(\sqrt{\frac{\log d}{d}}\right) \|\boldsymbol{w}_i^{(t+1)}\|_2(1 + o(1)\frac{\epsilon_j}{\epsilon_{\max}})$$

① and ② use the above inequality respectively.

## I.6 SUPPLEMENT TO LEMMA D.1(D):

For the dense features, we can compute as follows:

$$|\langle \boldsymbol{w}_i^{(t+1)}, \boldsymbol{M}_j^\perp\rangle|$$

$$\leq |\langle \boldsymbol{w}_i^{(t)}, \boldsymbol{M}_j^\perp\rangle| + O\left(\frac{\eta\|\boldsymbol{w}_i^{(t)}\|_2}{d\sqrt{d_1}} e^{-\Omega(\log^{1/4} d)}\right)$$

$$\leq |\langle \boldsymbol{w}_i^{(T_1)}, \boldsymbol{M}_j^\perp\rangle| + \sum_{t'=T_1}^t O\left(\frac{\eta\|\boldsymbol{w}_i^{(t')}\|_2}{d\sqrt{d_1}} e^{-\Omega(\log^{1/4} d)}\right)$$

$$\leq O\left(\sqrt{\frac{\log d}{d_1}}\right) \|\boldsymbol{w}_i^{(T_1+1)}\|_2 \left(1 + \frac{O(\eta)}{d \operatorname{polylog}(d)}\right)^2 + \sum_{t'=T_1+1}^t O\left(\frac{\eta\|\boldsymbol{w}_i^{(t')}\|_2}{d\sqrt{d_1}} e^{-\Omega(\log^{1/4} d)}\right)$$

$$\leq O\left(\sqrt{\frac{\log d}{d_1}}\right) \|\boldsymbol{w}_i^{(t+1)}\|_2 \left(1 + \frac{O(\eta)}{d \operatorname{polylog}(d)}\right)^{2(t-T_1+1)}$$

$$\leq O\left(\sqrt{\frac{\log d}{d_1}}\right) \|\boldsymbol{w}_i^{(t+1)}\|_2 \tag{300}$$

where we have used the assumption that $\|\boldsymbol{w}_i^{(t)}\|_2 \leq \|\boldsymbol{w}_i^{(t+1)}\|_2 \left(1 + \frac{O(\eta)}{d \operatorname{polylog}(d)}\right)$ for all $i \in [m]$ and all $t \leq T_2 = T_1 + \Theta\left(\frac{d\tau \log d}{\epsilon_{\max}\eta \log\log d}\right)$.

Which we prove here: First of all, from previous calculations we have

$$\left\|\boldsymbol{M}\boldsymbol{M}^\top \boldsymbol{w}_i^{(t+1)}\right\|_2 \geq \left\|\boldsymbol{M}\boldsymbol{M}^\top \boldsymbol{w}_i^{(t)}\right\|_2 (1 - \eta\lambda) - O\left(\frac{\eta\|\boldsymbol{w}_i^{(t)}\|_2}{d^{3/2}} e^{-\Omega(\log^{1/4} d)}\right) \tag{301}$$

also the trajectory of $\left\|\boldsymbol{M}^\perp(\boldsymbol{M}^\perp)^\top \boldsymbol{w}_i^{(t+1)}\right\|_2$ can be lower bounded as

$$\left\|\boldsymbol{M}^\perp(\boldsymbol{M}^\perp)^\top \boldsymbol{w}_i^{(t+1)}\right\|_2 \geq \left\|\boldsymbol{M}^\perp(\boldsymbol{M}^\perp)^\top \boldsymbol{w}_i^{(t)}\right\|_2 (1 - \eta\lambda) - O\left(\frac{\eta}{d}\right) e^{-\Omega(\log^{1/4} d)}\|\boldsymbol{w}_i^{(t)}\|_2 \tag{302}$$

thus by combining the change of $\boldsymbol{w}_i^{(t)}$ over two subspaces, we have

$$\begin{aligned}
\|\boldsymbol{w}_i^{(t+1)}\|_2^2 &\geq \|\boldsymbol{w}_i^{(t)}\|_2^2(1-\eta\lambda)^2 - O\left(\frac{\eta}{d}\right)e^{-\Omega(\log^{1/4} d)}\|\boldsymbol{w}_i^{(t)}\|_2^2 \\
&\geq \|\boldsymbol{w}_i^{(t)}\|_2^2\left(1 - \frac{O(\eta)}{d\,\mathrm{polylog}(d)}\right)
\end{aligned} \tag{303}$$

From the inequality above:

$$\|\boldsymbol{w}_i^{(t+1)}\|_2^2 \geq \|\boldsymbol{w}_i^{(t)}\|_2^2(1-\varepsilon) \quad \text{where} \quad \varepsilon = \frac{O(\eta)}{d\cdot\mathrm{polylog}(d)} \tag{304}$$

Divide both sides by $1-\varepsilon$, we get:

$$\|\boldsymbol{w}_i^{(t)}\|_2^2 \leq \|\boldsymbol{w}_i^{(t+1)}\|_2^2 \cdot \frac{1}{1-\varepsilon} \tag{305}$$

Note that when $\varepsilon \ll 1$, we have:

$$\frac{1}{1-\varepsilon} = 1 + \varepsilon + \varepsilon^2 + \cdots \leq 1 + 2\varepsilon \quad \text{(since $\varepsilon$ is very small)} \tag{306}$$

Therefore, we can write:

$$\|\boldsymbol{w}_i^{(t)}\|_2^2 \leq \|\boldsymbol{w}_i^{(t+1)}\|_2^2(1 + O(\varepsilon)) \tag{307}$$

$$\begin{aligned}
\|\boldsymbol{w}_i^{(t)}\|_2 &\leq \|\boldsymbol{w}_i^{(t+1)}\|_2\sqrt{(1+O(\varepsilon))} \leq \|\boldsymbol{w}_i^{(t+1)}\|_2(1+O(\varepsilon)) \\
&= \|\boldsymbol{w}_i^{(t+1)}\|_2\left(1 + \frac{O(\eta)}{d\cdot\mathrm{polylog}(d)}\right)
\end{aligned} \tag{308}$$

which gives the desired bound.

## I.7 SUPPLEMENT TO LEMMA D.1($T_2$):

In the proof above, we have depended on the crucial assumption that $T_2 := \min\left\{t \in \mathbb{N} : \exists i \in [m] \text{ s.t. } \|\boldsymbol{w}_i^{(t)}\|_2^2 \geq d\|\boldsymbol{w}_i^{(T_1)}\|_2^2\right\}$ is of order $T_1 + \Theta\left(\frac{d\tau\log d}{\epsilon_{\max}\eta\log\log d}\right)$. Now we verify it as follows. If $i \in \mathcal{M}_j^\star$ for some $j \in [d]$ (which also means $j' \notin \mathcal{N}_i$ for $j' \neq j$), we have

$$\begin{aligned}
|\langle \boldsymbol{w}_i^{(t)}, \boldsymbol{M}_j\rangle| &\geq |\langle \boldsymbol{w}_i^{(T_1)}, \boldsymbol{M}_j\rangle|\left(1 + \Omega\left(\epsilon_j\frac{\eta\log\log d}{d}\right)\right)^{t-T_1} \\
&\geq d\sqrt{\frac{2\log d}{d}}\|\boldsymbol{w}_i^{(T_1)}\|_2
\end{aligned} \tag{309}$$

For some $t = T_1 + O\left(\frac{d\tau\log d}{\epsilon_{\max}\eta\log\log d}\right)$ and $\epsilon_j = \epsilon_{\max}$

Thus for some $t = T_1 + O\left(\frac{d\tau\log d}{\epsilon_{\max}\eta\log\log d}\right)$, we have $|\langle \boldsymbol{w}_i^{(t)}, \boldsymbol{M}_j\rangle|^2 \geq d\|\boldsymbol{w}_i^{(T_1)}\|_2^2$, which proves that $T_2 \leq T_1 + O\left(\frac{d\tau\log d}{\epsilon_{\max}\eta\log\log d}\right)$.

These results are to verify that $\|\boldsymbol{w}_i^{(t)}\|_2^2 \geq d\|\boldsymbol{w}_i^{(T_1)}\|_2^2$ holds under the order of $T_2 = T_1 + \Theta\left(\frac{d\tau\log d}{\epsilon_{\max}\eta\log\log d}\right)$.

$$\begin{aligned}
\left(1 + \Omega\left(\epsilon_j\frac{\eta\log\log d}{d}\right)\right)^{\Theta\left(\frac{d\log d}{\epsilon_{\max}\eta\log\log d}\right)} &= \exp\left(\Omega\left(\epsilon_j\frac{\eta\log\log d}{d}\right)\cdot\Theta\left(\frac{d\log d}{\epsilon_{\max}\eta\log\log d}\right)\right) \\
&= \exp(\Omega(\frac{\epsilon_j}{\epsilon_{\max}}\log d)) \\
&= d^{\Omega(\frac{\epsilon_j}{\epsilon_{\max}})}
\end{aligned} \tag{310}$$

Conversely, we also have for all $t \leq T_1 + O\left(\frac{d\tau \log d}{\epsilon_{\max}\eta \log \log d}\right)$

$$
\begin{aligned}
\sum_{j' \in [d]: j' \neq j} &\langle \boldsymbol{w}_i^{(t)}, \boldsymbol{M}_{j'} \rangle^2 + \sum_{j' \in [d_1] \setminus [d]} \langle \boldsymbol{w}_i^{(t)}, \boldsymbol{M}_{j'}^\perp \rangle^2 \\
&\leq \|\boldsymbol{w}_i^{(T_1)}\|_2^2 \left(1 + \epsilon_{j'} \frac{O(\eta)}{d \operatorname{polylog}(d)}\right)^{t-T_1} + \max_{t' \leq t} O\left(\frac{\eta(t-T_1)}{d}\right) e^{-\Omega(\log^{1/4} d)} \|\boldsymbol{w}_i^{(t')}\|_2^2 \quad (311) \\
&\leq o\left(d \|\boldsymbol{w}_i^{(T_1)}\|_2^2\right)
\end{aligned}
$$

Except for the principal direction $\boldsymbol{M}_j$ (i.e., the alignment direction of neuron i), the total growth of squared weights along all other directions remains far below the target scale $d \cdot \left\|\boldsymbol{w}_i^{(T_1)}\right\|_2^2$.

And also

$$
\begin{aligned}
|\langle \boldsymbol{w}_i^{(t)}, \boldsymbol{M}_j \rangle| &\leq |\langle \boldsymbol{w}_i^{(T_1)}, \boldsymbol{M}_j \rangle| \left(1 + \epsilon_j \frac{C_z \eta \log \log d}{d}\left(1 - \frac{1}{\operatorname{polylog}(d)}\right)\right)^{t-T_1} \\
&\leq O\left(\sqrt{\frac{\log d}{d}} \|\boldsymbol{w}_i^{(T_1)}\|_2\right) \left(1 + \epsilon_j \frac{C_z \eta \log \log d}{d}\left(1 - \frac{1}{\operatorname{polylog}(d)}\right)\right)^{t-T_1}
\end{aligned} \quad (312)
$$

Therefore we at least need $T_1 + \frac{d \log\left(\Omega\left(\sqrt{d}\sqrt{\frac{d}{\log d}}\right)\right)}{\epsilon_{\max}\eta C_z \log \log d}(1 - o(1))$ iteration to let any neuron $i \in [m]$ reach $\|\boldsymbol{w}_i^{(t)}\|_2^2 \geq d\|\boldsymbol{w}_i^{(T_1)}\|_2$, which proves that $T_2 = T_1 + \Theta\left(\frac{d\tau \log d}{\epsilon_{\max}\eta \log \log d}\right)$.

**Definition I.1** (Notations). *For simpler presentation, we define the following notations: given $\boldsymbol{z_X} = \frac{1}{L}(\boldsymbol{M}\tilde{z}_n + \tilde{\xi}_n) \sim \mathcal{D}_{\boldsymbol{z_X}}$, $\boldsymbol{z_Y} = \frac{1}{L}(\boldsymbol{M}\tilde{z}_n^+ + \tilde{\xi}_n^+) \sim \mathcal{D}_{\boldsymbol{z_Y}}$, we let (for each $j \in [d]$):*

$$
z_X^{\setminus j} := \frac{1}{L}\left(\sum_{\substack{j' \neq j \\ j' \in [d]}} \boldsymbol{M}_{j'}\tilde{z}_{n,j'} + \tilde{\xi}_n\right), \quad z_Y^{\setminus j} := \frac{1}{L}\left(\sum_{\substack{j' \neq j \\ j' \in [d]}} \boldsymbol{M}_{j'}\tilde{z}_{n,j'}^+ + \tilde{\xi}_n^+\right) \quad (313)
$$

$$
S_{i,t}^{(r)\setminus j} := \langle \boldsymbol{w}_i^{(t)}, z_X^{(r)\setminus j} \rangle, \quad S_{i,t}^{(s)\setminus j} := \langle \boldsymbol{w}_i^{(t)}, z_Y^{(s)\setminus j} \rangle \quad (314)
$$

$$
S_{i,t}^{(r,s)\setminus j} := \tfrac{1}{2}\left(S_{i,t}^{(r)\setminus j} + S_{i,t}^{(s)\setminus j}\right), \quad \bar{S}_{i,t}^{(r,s)\setminus j} := \tfrac{1}{2}\left(S_{i,t}^{(s)\setminus j} - S_{i,t}^{(r)\setminus j}\right) \quad (315)
$$

$$
\alpha_{i,j}^{(t)} := \langle \boldsymbol{w}_i^{(t)}, \boldsymbol{M}_j \rangle, \quad \bar{\alpha}_{i,j}^{(r,s)(t)} := \left\langle \boldsymbol{w}_i^{(t)}, \frac{\tilde{z}_{n,j}^{+(s)} - \tilde{z}_{n,j}^{(r)}}{\tilde{z}_{n,j}^{(r)} + \tilde{z}_{n,j}^{+(s)}} \boldsymbol{M}_j \right\rangle \quad (316)
$$

Whenever the neuron index $i \in [m]$ is clear from the context, we drop the subscript $i$ and the time index $t$ for notational simplicity.

### I.8 Proof of Lemma D.2

*Proof of Lemma D.2:* We have $\langle \boldsymbol{w}_i^{(0)}, \boldsymbol{M}_j \rangle \sim \mathcal{N}(0, \sigma_0^2)$ and we want to control is the event:

$$
\left|\left\langle \boldsymbol{w}_i^{(0)}, \boldsymbol{M}_j \right\rangle\right| \leq \frac{\sigma_0}{d}. \quad (317)
$$

To get Lemma D.2, we can use the standard Gaussian anti-concentration property near the mean. $\qquad \square$

## J  PROOF OF LEMMAS IN APPENDIX E

### J.1  USEFUL LEMMAS

**Lemma J.1.** *For any $j' \neq j$, we have*

$$|\mathcal{M}_{j'} \cap \mathcal{M}_j| \leq O(\log d), \tag{318}$$

*with probability at least $1 - o(1/d^4)$.*

### J.2  PROOF OF LEMMA E.2:

*Proof of Lemma E.2:* We first decompose

$$\Psi_{i,j}^{(t)} = \Psi_{i,j,1}^{(t)} + \Psi_{i,j,2}^{(t)} \tag{319}$$

where

$$\Psi_{i,j,1}^{(t)} = \mathbb{E}\left[(\ell'_{p,t} - 1) \cdot \psi_{i,j}^{(t)}(\boldsymbol{Y}_n) \cdot \sum_{r=1}^{L} \mathbf{1}_{|\langle \boldsymbol{w}_i, \boldsymbol{z}_{\boldsymbol{X}}^{(r)}\rangle| \geq b_i} \tilde{z}_{n,j}^{(r)}\right] \tag{320}$$

$$\Psi_{i,j,2}^{(t)} = \sum_{\boldsymbol{X}_{n,s} \in \mathfrak{N}} \mathbb{E}\left[\ell'_{s,t} \cdot \psi_{i,j}^{(t)}(X) \cdot \sum_{r=1}^{L} \mathbf{1}_{|\langle \boldsymbol{w}_i, \boldsymbol{z}_{\boldsymbol{X}}^{(r)}\rangle| \geq b_i} \tilde{z}_{n,j}^{(r)}\right] \tag{321}$$

We first deal with $\Psi_{i,j,2}^{(t)}$. Using the notation $\boldsymbol{X}_n^{\backslash j}$: For each $z_X^{(r)\backslash j} := \frac{1}{L}\left(\sum_{j' \neq j} \boldsymbol{M}_{j'} \tilde{z}_{n,j'}^{(r)} + \xi_n^{(r)}\right)$, we can rewrite as:

$$\Psi_{i,j,2}^{(t)}$$

$$= \sum_{\boldsymbol{X}_{n,s} \in \mathfrak{N}} \mathbb{E}\left[\ell'_{s,t}(\boldsymbol{X}_n, \mathfrak{B}) \cdot \psi_{i,j}^{(t)}(\boldsymbol{X}_{n,s}) \cdot \sum_{r=1}^{L} \mathbf{1}_{|\langle \boldsymbol{w}_i, \boldsymbol{z}_{\boldsymbol{X}}^{(r)}\rangle| \geq b_i} \tilde{z}_{n,j}^{(r)}\right]$$

$$= \sum_{\boldsymbol{X}_{n,s} \in \mathfrak{N}} \mathbb{E}\left[\left(\ell'_{s,t}(\boldsymbol{X}_n, \mathfrak{B}) - \ell'_{s,t}(\boldsymbol{X}_n^{\backslash j}, \mathfrak{B})\right) \cdot \psi_{i,j}^{(t)}(\boldsymbol{X}_{n,s}) \sum_{r=1}^{L} \mathbf{1}_{|\langle \boldsymbol{w}_i, \boldsymbol{z}_{\boldsymbol{X}}^{(r)}\rangle| \geq b_i} \tilde{z}_{n,j}^{(r)}\right] \tag{322}$$

$$+ \sum_{\boldsymbol{X}_{n,s} \in \mathfrak{N}} \mathbb{E}\left[\ell'_{s,t}(\boldsymbol{X}_n^{\backslash j}, \mathfrak{B}) \cdot \psi_{i,j}^{(t)}(\boldsymbol{X}_{n,s}) \sum_{r=1}^{L} \mathbf{1}_{|\langle \boldsymbol{w}_i, \boldsymbol{z}_{\boldsymbol{X}}^{(r)}\rangle| \geq b_i} \tilde{z}_{n,j}^{(r)}\right]$$

$$= R_1 + R_2$$

We now deal with the term $R_1$. Denoting $\widehat{X}_n^{(v)}$: For each $\widehat{z}_X^{(r)}(v) = \tilde{z}_X^{(r)} + \frac{v}{L}\boldsymbol{M}_j \tilde{z}_{n,j}^{(r)}$, for $v \in [0, 1]$, by Newton-Leibniz formula and the basic fact that

$$\frac{d}{dr} \frac{e^r}{e^r + \sum_{s \neq r} e^s} = \frac{e^r}{e^r + \sum_{s \neq r} e^s}\left(1 - \frac{e^r}{e^r + \sum_{s \neq r} e^s}\right), \tag{323}$$

we can rewrite $\ell'_{s,t}(\widehat{X}_n(v), \mathfrak{B})$ and $\ell'_{p,t}(\widehat{X}_n(v), \mathfrak{B})$ as

$$\widehat{\ell}'_{s,t}(\nu) := \frac{e^{\langle f_t(\boldsymbol{X}_n^{\backslash j}) + \nu(f_t(\boldsymbol{X}_n) - f_t(\boldsymbol{X}_n^{\backslash j})), f_t(\boldsymbol{X}_{n,s})\rangle/\tau}}{\sum_{x \in \mathfrak{B}} e^{\langle f_t(\boldsymbol{X}_n^{\backslash j}) + \nu(f_t(\boldsymbol{X}_n) - f_t(X_p^{\backslash j})), f_t(x)\rangle/\tau}} \equiv \ell'_{s,t}(\widehat{X}_n(v), \mathfrak{B}) \tag{324}$$

$$\widehat{\ell}'_{p,t}(\nu) := \frac{e^{\langle f_t(\boldsymbol{X}_n^{\backslash j}) + \nu(f_t(\boldsymbol{X}_n) - f_t(\boldsymbol{X}_n^{\backslash j})), f_t(\boldsymbol{X}_n)\rangle/\tau}}{\sum_{x \in \mathfrak{B}} e^{\langle f_t(\boldsymbol{X}_n^{\backslash j}) + \nu(f_t(\boldsymbol{X}_n) - f_t(\boldsymbol{X}_n^{\backslash j})), f_t(x)\rangle/\tau}} \equiv \ell'_{p,t}(\widehat{X}_n(v), \mathfrak{B}) \tag{325}$$

and we can then proceed to calculate as follows:

$$
\begin{aligned}
R_1 &= \sum_{\boldsymbol{X}_{n,s}\in\mathfrak{N}} \mathbb{E}\left[\left(\ell'_{s,t}(\boldsymbol{X}_n,\mathfrak{B}) - \ell'_{s,t}(\boldsymbol{X}_n^{\setminus j},\mathfrak{B})\right)\cdot\psi_{i,j}^{(t)}(\boldsymbol{X}_{n,s})\sum_{r=1}^{L}\mathbf{1}_{|\langle \boldsymbol{w}_i,\boldsymbol{z}_{\boldsymbol{X}}^{(r)}\rangle|\geq b_i}\tilde{z}_{n,j}^{(r)}\right]\\
&= \sum_{\boldsymbol{X}_{n,s}\in\mathfrak{N}}\mathbb{E}\left[\frac{1}{\tau}\left(\int_0^1\widehat{\ell}'_{s,t}(\nu)(1-\widehat{\ell}_{s,t}(\nu))\langle f_t(\boldsymbol{X}_n)-f_t(\boldsymbol{X}_n^{\setminus j}),f_t(\boldsymbol{X}_{n,s})\rangle d\nu\right.\right.\\
&\quad -\sum_{X_{n,u}\in\mathfrak{N}\setminus\{\boldsymbol{X}_{n,s}\}}\int_0^1\widehat{\ell}'_{s,t}(\nu)\widehat{\ell}'_{u,t}(\nu)\langle f_t(\boldsymbol{X}_n)-f_t(\boldsymbol{X}_n^{\setminus j}),f_t(X')\rangle d\nu\\
&\quad \left.\left.-\int_0^1\widehat{\ell}'_{s,t}(\nu)\widehat{\ell}'_{p,t}(\nu)\langle f_t(\boldsymbol{X}_n)-f_t(\boldsymbol{X}_n^{\setminus j}),f_t(\boldsymbol{Y}_n)\rangle d\nu\right)\psi_{i,j}^{(t)}(\boldsymbol{X}_{n,s})\sum_{r=1}^{L}\mathbf{1}_{|\langle \boldsymbol{w}_i,\boldsymbol{z}_{\boldsymbol{X}}^{(r)}\rangle|\geq b_i}\tilde{z}_{n,j}^{(r)}\right]\\
&\overset{\textcircled{1}}{\leq}\frac{|\mathfrak{N}|}{\tau}\mathbb{E}\left[\left(\int_0^1\widehat{\ell}'_{s,t}(\nu)(1-\widehat{\ell}_{s,t}(\nu))d\nu+\sum_{X_{n,u}\in\mathfrak{N}\setminus\{\boldsymbol{X}_{n,s}\}}\int_0^1\widehat{\ell}'_{s,t}(\nu)\widehat{\ell}'_{u,t}(\nu)d\nu\right)\right.\\
&\quad \left.\times\max_{X_{n,u}\in\mathfrak{N}\setminus\{\boldsymbol{X}_{n,s}\}}|\langle f_t(\boldsymbol{X}_n)-f_t(\boldsymbol{X}_n^{\setminus j}),f_t(X_{n,u})\rangle|\cdot|\psi_{i,j}^{(t)}(\boldsymbol{X}_{n,s})|\cdot\sum_{r=1}^{L}\mathbf{1}_{|\langle \boldsymbol{w}_i,\boldsymbol{z}_{\boldsymbol{X}}^{(r)}\rangle|\geq b_i}|\tilde{z}_{n,j}^{(r)}|\right]\\
&\quad +\mathbb{E}\left[\frac{|\mathfrak{N}|}{\tau}\int_0^1\widehat{\ell}'_{s,t}(\nu)\widehat{\ell}'_{p,t}(\nu)d\nu\cdot|\langle f_t(\boldsymbol{X}_n)-f_t(\boldsymbol{X}_n^{\setminus j}),f_t(\boldsymbol{Y}_n)\rangle|\cdot|\psi_{i,j}^{(t)}(\boldsymbol{X}_{n,s})|\cdot\sum_{r=1}^{L}\mathbf{1}_{|\langle \boldsymbol{w}_i,\boldsymbol{z}_{\boldsymbol{X}}^{(r)}\rangle|\geq b_i}|\tilde{z}_{n,j}^{(r)}|\right]\\
&\overset{\textcircled{2}}{\leq}\frac{|\mathfrak{B}|}{\tau}\mathbb{E}\left[\int_0^1\widehat{\ell}'_{s,t}(\nu)d\nu\cdot\max_{x\in\mathfrak{B}}\left(\sum_{i\in\mathcal{M}_j}\langle \boldsymbol{w}_i^{(t)},\boldsymbol{M}_j\rangle|h_{i,t}(x)|\right)\cdot|\psi_{i,j}^{(t)}(\boldsymbol{X}_{n,s})|\cdot\sum_{r=1}^{L}(\tilde{z}_{n,j}^{(r)})^2\right]\\
&\quad +\widetilde{O}(\Xi_2)\max_{i'\notin\mathcal{M}_j}|\langle w_{i'}^{(t)},\boldsymbol{M}_j\rangle|\cdot\mathbb{E}\left[\sum_{\boldsymbol{X}_{n,s}\in\mathfrak{N}}\frac{1}{\tau}\int_0^1\widehat{\ell}'_{s,t}(\nu)d\nu\cdot|\psi_{i,j}^{(t)}(\boldsymbol{X}_{n,s})|\cdot\sum_{r=1}^{L}(\tilde{z}_{n,j}^{(r)})^2\right]\\
&= R_{1,1}+R_{1,2}+\frac{1}{d^{\Omega(\log d)}}
\end{aligned}
\tag{326}
$$

where for $\textcircled{1}$ and $\textcircled{2}$, we argue as follows:

for $\textcircled{1}$, we used the fact that the expectations over $s\in\mathfrak{N}$ in the summation can be viewed as independently and uniformly selecting from $s\in\mathfrak{N}$, which allows us to equate $\sum_{X\in\mathfrak{N}}=|\mathfrak{N}|$.

for $\textcircled{2}$, we use Fact E.1to ensure that $\sum_{i\in[m]}\mathbf{1}_{h_{i,t}(\boldsymbol{X}_n)\neq 0}\leq\widetilde{O}(\Xi_2)$ with high probability.

we have for any $x\in\mathfrak{B}$:

$$
|\langle f_t(\boldsymbol{X}_n)-f_t(\boldsymbol{X}_n^{\setminus j}),f_t(x)\rangle|\leq\sum_{i'\in\mathcal{M}_j}|\langle w_{i'}^{(t)},\boldsymbol{M}_j\rangle|\cdot|h_{i',t}(x)|+\widetilde{O}(\Xi_2)\max_{i'\in\mathcal{M}_j}|\langle w_{i'}^{(t)},\boldsymbol{M}_j\rangle| \tag{327}
$$

which gives the desired inequality.

Now we proceed to deal with $R_{1,1}$, since $i\in\mathcal{M}_j^{\star}$; we have automatically $\mathbf{1}_{h_{i,t}(\boldsymbol{X}_{n,s})\neq 0}=\mathbf{1}_{|\hat{z}_{n,s,j}|\neq 0}$ w.h.p., so we can transform $R_{1,1}$ as

$$
\begin{aligned}
R_{1,1} &= \mathbb{E}\left[\frac{|\mathfrak{N}|}{\tau}\int_0^1\widehat{\ell}'_{s,t}(\nu)d\nu\max_{x\in\mathfrak{B}}\left(\sum_{i'\in\mathcal{M}_j}\langle w_{i'}^{(t)},\boldsymbol{M}_j\rangle|h_{i',t}(x)|\right)|\psi_{i,j}^{(t)}(\boldsymbol{X}_{n,s})|\sum_{r=1}^{L}(\tilde{z}_{n,j}^{(r)})^2\right]+\frac{1}{d^{\Omega(\log d)}}\\
&= \mathbb{E}\left[\frac{|\mathfrak{B}|}{\tau}\int_0^1\widehat{\ell}'_{s,t}(\nu)d\nu\left(\sum_{i'\in\mathcal{M}_j}\langle w_{i'}^{(t)},\boldsymbol{M}_j\rangle^2+\Upsilon_j^{(t)}\right)|\psi_{i,j}^{(t)}(\boldsymbol{X}_{n,s})|\sum_{r=1}^{L}(\tilde{z}_{n,j}^{(r)})^2\right]+\frac{1}{d^{\Omega(\log d)}}
\end{aligned}
\tag{328}
$$

where $\Upsilon_j^{(t)}$ is defined as:

$$\Upsilon_j^{(t)} := \max_{x \in \mathfrak{B}} \left( \sum_{i' \in \mathcal{M}_j} |h_{i',t}(x)| \cdot |\langle w_{i'}^{(t)}, M_j \rangle| - \sum_{i' \in \mathcal{M}_j} \langle w_{i'}^{(t)}, M_j \rangle^2 \right) \tag{329}$$

We proceed to give a high probability bound for $\sum_{i \in \mathcal{M}_j} |h_{i,t}(X_{n,s})| \cdot |\langle w_i^{(t)}, M_j \rangle|$, which lies in the core of our proof. In order to apply Lemma E.7. to the pre-activation in $h_{i,t}(X_{n,s})$, one can first expand as

$$\sum_{i \in [m]} |h_{i,t}(X_{n,s})| \cdot |\langle w_i^{(t)}, M_j \rangle|$$

$$\lesssim \sum_{j' \in [d]} \sum_{i \in \mathcal{M}_{j'}} |\langle w_i^{(t)}, M_{j'} \rangle| \cdot \sum_{q=1}^{L} |\tilde{z}_{n,s,j'}^{(q)}| \cdot |\langle w_i^{(t)}, M_j \rangle| + \sum_{i \in [m]} \widetilde{O}\left( \frac{\|w_i^{(t)}\|_2}{\sqrt{d}} \right) |\langle w_i^{(t)}, M_j \rangle|$$

$$= \sum_{j' \in [d]} \sum_{i \in \mathcal{M}_{j'} \cap \mathcal{M}_j} \langle w_i^{(t)}, M_{j'} \rangle \sum_{q=1}^{L} |\tilde{z}_{n,s,j'}^{(q)}| |\langle w_i^{(t)}, M_j \rangle|$$

$$+ \sum_{j' \in [d]} \sum_{i \in \mathcal{M}_{j'} \setminus \mathcal{M}_j} |\langle w_i^{(t)}, M_{j'} \rangle| \cdot \sum_{q=1}^{L} |\tilde{z}_{n,s,j'}^{(q)}| |\langle w_i^{(t)}, M_j \rangle| + \sum_{i \in [m]} \widetilde{O}\left( \frac{\|w_i^{(t)}\|_2}{\sqrt{d}} \right) |\langle w_i^{(t)}, M_j \rangle|$$

$$\leq \sum_{i \in \mathcal{M}_j} \langle w_i^{(t)}, M_j \rangle^2 \sum_{q=1}^{L} |\tilde{z}_{n,s,j'}^{(q)}| + \sum_{\substack{j' \neq j \\ i \in \mathcal{M}_j \cap \mathcal{M}_{j'}}} |\langle w_i^{(t)}, M_{j'} \rangle| \cdot \sum_{q=1}^{L} |\tilde{z}_{n,s,j'}^{(q)}| \cdot |\langle w_i^{(t)}, M_j \rangle|$$

$$+ \sum_{j' \in [d]} \sum_{i \in \mathcal{M}_{j'} \setminus \mathcal{M}_j} |\langle w_i^{(t)}, M_{j'} \rangle| \cdot \sum_{q=1}^{L} |\tilde{z}_{n,s,j'}^{(q)}| \cdot |\langle w_i^{(t)}, M_j \rangle| + \sum_{i \in [m]} \widetilde{O}\left( \frac{\|w_i^{(t)}\|_2}{\sqrt{d}} \right) |\langle w_i^{(t)}, M_j \rangle| \tag{330}$$

And we proceed to calculate the last two terms on the RHS as follows: Firstly, from Lemma J.1. We know for the set of neurons $\Gamma_j := \{j' \neq j, \ j' \in [d] : \mathcal{M}_j \cap \mathcal{M}_{j'} \neq \emptyset\}$, we have $|\Gamma_j| \leq O(\log d)$, and

$$\left| \sum_{j' \neq j} \sum_{i' \in \mathcal{M}_j \cap \mathcal{M}_{j'}} |\langle w_{i'}^{(t)}, M_{j'} \rangle| \cdot \sum_{q=1}^{L} |\tilde{z}_{n,s,j'}^{(q)}| \cdot |\langle w_i^{(t)}, M_j \rangle| \right|$$

$$\leq O\left( \frac{\log d}{\sqrt{\Xi_2}} \right) \cdot \sum_{j' \in \Gamma_j} O(\tau \log^2 d) \sum_{q=1}^{L} |\tilde{z}_{n,s,j'}^{(q)}| \tag{331}$$

$$\leq O\left( \frac{\tau}{\log d} \right)$$

where in the last inequality we have taken into account that $\mathbb{E}[\sum_{q=1}^{L} |\tilde{z}_{n,s,j}^{(q)}|] = \widetilde{O}(1/d)$ and have used Lemma E.8. The same techniques also provide the following bound:

$$\left| \sum_{j' \in [d]} \sum_{i' \in \mathcal{M}_{j'} \setminus \mathcal{M}_j} |\langle w_{i'}^{(t)}, M_{j'} \rangle| \cdot \sum_{q=1}^{L} |\tilde{z}_{n,s,j'}^{(q)}| \cdot |\langle w_i^{(t)}, M_j \rangle| \right| \leq O\left( \frac{1}{\sqrt{d}} \right) \sum_{j' \neq j} O(\Xi_2) \sum_{q=1}^{L} |\tilde{z}_{n,s,j'}^{(q)}|$$

$$\leq O\left( \frac{\Xi_2^2}{\sqrt{d}} \right) \tag{332}$$

Therefore via a union bound, we have

$$\sum_{i' \in [m]} |h_{i',t}(X_{n,s})| \cdot |\langle w_{i'}^{(t)}, M_j \rangle| \leq \sum_{i' \in \mathcal{M}_j} \langle w_{i'}^{(t)}, M_j \rangle^2 \sum_{q=1}^{L} |\tilde{z}_{n,s,j'}^{(q)}| + O\left( \frac{\tau}{\log d} \right) \tag{333}$$

The same arguments also gives ( + further applying Lemma E.7.)

$$\max_{x \in \{\boldsymbol{X}_n, \boldsymbol{Y}_n\}} \left\{ \sum_{i' \in [m]} |h_{i',t}(x)| \cdot |\langle w_{i'}^{(t)}, \boldsymbol{M}_j \rangle| \right\} \leq \sum_{i' \in \mathcal{M}_j} \langle w_{i'}^{(t)}, \boldsymbol{M}_j \rangle^2 \sum_{r=1}^{L} |\tilde{z}_{n,j'}^{(r)}| + O\left(\frac{\tau}{\log d}\right) \quad (334)$$

which also implies that

$$\Upsilon_j^{(t)} \leq \max_{x \in \mathfrak{B}} \sum_{i' \in [m]} |h_{i,t}(x)| \cdot |\langle w_{i'}^{(t)}, \boldsymbol{M}_j \rangle| - \sum_{i' \in \mathcal{M}_j} |\langle w_{i'}^{(t)}, \boldsymbol{M}_j \rangle|^2 \sum_{r=1}^{L} |\tilde{z}_{n,j'}^{(r)}| \leq O\left(\frac{\tau}{\log d}\right) \quad (335)$$

Now we are ready to control the quantity $R_{1,1}$. The idea here is to "decorrelate" the factor $\widehat{\ell}'_{s,t}(\nu)$ from the others. Defining

$$\mathfrak{B} \backslash^s := \{\boldsymbol{Y}_n\} \cup \mathfrak{N} \setminus \{\boldsymbol{X}_{n,s}\}, \quad \text{and} \quad \mathfrak{B}'_s := \mathfrak{B} \backslash^s \cup \{\boldsymbol{X}_{n,s}^{\backslash j}\}, \quad (336)$$

there exists a constant $G'_1 > 0$ such that, if

$$\sum_{i \in \mathcal{M}_j} \langle \boldsymbol{w}_i^{(t)}, \boldsymbol{M}_j \rangle^2 (\sum_{r=1}^{L} \tilde{z}_{n,j}^{+(r)})^2 \leq (\frac{\epsilon_j}{\epsilon_{\max}})^2 \tau G'_1 \log d, \quad (337)$$

we have w.h.p.

$$
\begin{aligned}
\frac{\ell'_{s,t}(\boldsymbol{X}_n, \mathfrak{B})}{\ell'_{s,t}(\boldsymbol{X}_n, \mathfrak{B}'_s)} &= \frac{e^{\langle f_t(\boldsymbol{X}_n), f_t(\boldsymbol{X}_{n,s}) \rangle / \tau}}{e^{\langle f_t(\boldsymbol{X}_n), f_t(\boldsymbol{X}_{n,s}^{\backslash j}) \rangle / \tau}} \cdot \frac{\sum_{x \in \mathfrak{B}'_s} e^{\langle f_t(\boldsymbol{X}_n), f_t(x) \rangle / \tau}}{\sum_{x \in \mathfrak{B}} e^{\langle f_t(\boldsymbol{X}_n), f_t(x) \rangle / \tau}} \\
&\leq \frac{e^{2 \langle f_t(\boldsymbol{X}_n), f_t(\boldsymbol{X}_{n,s}) \rangle / \tau}}{e^{2 \langle f_t(\boldsymbol{X}_n), f_t(\boldsymbol{X}_{n,s}^{\backslash j}) \rangle / \tau}} \\
&\leq e^{2(\frac{\epsilon_j}{\epsilon_{\max}})^2 G'_1 \log d + O(1/\log d)} \leq o\left(\frac{d}{\Xi_2^5}\right)
\end{aligned}
\quad (338)
$$

Now we define

$$\mathfrak{N}'_s := \{X_{n,s'} \in \mathfrak{N} \setminus \{\boldsymbol{X}_{n,s}\} : \sum_{q=1}^{L} \tilde{z}_{n,u,j}^{(q)} = 0\} \cup \{X_{n,s}^{\backslash j}\}. \quad (339)$$

Note that from concentration inequality of Bernoulli variables, we know

$$|\mathfrak{N}'_s| = \Omega(|\mathfrak{N}|) \quad (340)$$

Thus we have (notice that the outer factor $|\mathfrak{N}|$ can be inserted into the expectation by sacrificing some constant factors):

$$
\begin{aligned}
R_{1,1} &\leq O\left(\frac{|\mathfrak{N}|}{\tau}\right) \cdot \mathbb{E}\left[\widehat{\ell}_{s,t}(1)\left(\sum_{i\in\mathcal{M}_j}\langle w_{i'}^{(t)}, \boldsymbol{M}_j\rangle^2 + \Upsilon_j^{(t)}\right) \cdot |\psi_{i,j}^{(t)}(\boldsymbol{X}_{n,s})| \cdot \sum_{r=1}^{L}(\tilde{z}_{n,j}^{(r)})^2\right] \\
&= O\left(\frac{|\mathfrak{N}|\log\log d}{d\tau}\right)\mathbb{E}\left[\ell_{s,t}'(\boldsymbol{X}_n, \mathfrak{B})\left(\sum_{i\in\mathcal{M}_j}\langle w_{i'}^{(t)}, \boldsymbol{M}_j\rangle^2 + \Upsilon_j^{(t)}\right) \cdot |\psi_{i,j}^{(t)}(\boldsymbol{X}_{n,s})| \,\Big|\, \sum_{r=1}^{L}|\tilde{z}_{n,j}^{(r)}| \neq 0\right] \\
&= \widetilde{O}\left(\frac{|\mathfrak{N}|}{d\tau}\right)\mathbb{E}\left[\frac{\ell_{s,t}'(\boldsymbol{X}_n, \mathfrak{B})}{\ell_{s,t}'(\boldsymbol{X}_n, \mathfrak{B}_s')} \cdot \ell_{s,t}'(\boldsymbol{X}_n, \mathfrak{B}_s')\left(\sum_{i\in\mathcal{M}_j}\langle w_{i'}^{(t)}, \boldsymbol{M}_j\rangle^2 + \Upsilon_j^{(t)}\right) \,\Big|\, |\psi_{i,j}^{(t)}(\boldsymbol{X}_{n,s})|, \sum_{r=1}^{L}|\tilde{z}_{n,j}^{(r)}| \neq 0\right] \\
&\leq O\left(\frac{1}{\tau\Xi_2^4}\right)\mathbb{E}\left[\sum_{\boldsymbol{X}_{n,s}\in\mathfrak{N}'}\ell_{s,t}'(\boldsymbol{X}_n, \mathfrak{B}_s')\left(\sum_{i\in\mathcal{M}_j}\langle w_{i'}^{(t)}, \boldsymbol{M}_j\rangle^2 + \Upsilon_j^{(t)}\right) \cdot |\psi_{i,j}^{(t)}(\boldsymbol{X}_{n,s})| \,\Big|\, \sum_{r=1}^{L}|\tilde{z}_{n,j}^{(r)}| \neq 0\right] \\
&\overset{①}{\leq} \sum_{q=1}^{L}|\langle\boldsymbol{w}_i^{(t)}, \boldsymbol{M}_j\rangle\tilde{z}_{n,s,j}^{(q)} - b_i^{(t)}| \cdot O\left(\frac{1}{\Xi_2^3}\right) \cdot \boldsymbol{Pr}(\sum_{q=1}^{L}|\tilde{z}_{n,s,j}^{(q)}| \neq 0) \\
&\leq \sum_{r=1}^{L}|\langle\boldsymbol{w}_i^{(t)}, \boldsymbol{M}_j\rangle\tilde{z}_{n,j}^{(r)}| \cdot O\left(\frac{1}{\Xi_2^3}\right) \cdot \boldsymbol{Pr}(\sum_{r=1}^{L}|\tilde{z}_{n,j}^{(r)}| \neq 0)
\end{aligned}
\tag{341}
$$

Where in inequality ①, we have used the independence of $\sum_{q=1}^{L}|\tilde{z}_{n,s,j}^{(q)}|$ with respect to $\ell_{s,t}'(\boldsymbol{X}_n, \mathfrak{B}_s')$, and the fact that

$$
\sum_{\boldsymbol{X}_{n,s}\in\mathfrak{N}}\ell_{s,t}'(\boldsymbol{X}_n, \mathfrak{B}_s') \leq 1
\tag{342}
$$

Now turn back to deal with $R_{1,2}$ (by Newton-Leibniz). Indeed, noticing that

$$
\max_{i'\notin\mathcal{M}_j}|\langle w_{i'}^{(t)}, \boldsymbol{M}_j\rangle| \leq O\left(\frac{1}{\sqrt{d}}\right) \quad \text{(from Lemma E.1)},
\tag{343}
$$

and that

$$
\sum_{x\in\mathfrak{N}}\mathbb{E}\left[\frac{1}{\tau}\int_0^1\widehat{\ell}_{s,t}(\nu)\,\mathrm{d}\nu \cdot |\psi_{i,j}^{(t)}(\boldsymbol{X}_{n,s})| \cdot \sum_{r=1}^{L}(\tilde{z}_{n,j}^{(r)})^2\right] \leq \widetilde{O}\left(\frac{1}{d}\right) \cdot \sum_{r=1}^{L}|\langle\boldsymbol{w}_i^{(t)}, \boldsymbol{M}_j\rangle\tilde{z}_{n,j}^{(r)} - b_i^{(t)}|
\tag{344}
$$

We have $R_{1,2} \leq o(R_{1,1})$. For $R_2$ (in By Newton-Leibniz), we can see from the definition of

$$
\ell_{s,t}'(\boldsymbol{X}_n^{\backslash j}, \mathfrak{B})
\tag{345}
$$

that it is independent of $\tilde{z}_{n,j}^{(r)}$. Notice further that

$$
\boldsymbol{1}_{\langle\boldsymbol{w}_i^{(t)}, \boldsymbol{z}_{\boldsymbol{X}}^{(r)}\rangle\geq b_i^{(t)}} = \boldsymbol{1}_{\tilde{z}_{n,j}^{(r)}\neq 0}
\tag{346}
$$

with high probability due to our assumption, and also the fact that $\boldsymbol{1}_{\tilde{z}_{n,j}^{(r)}\neq 0}\tilde{z}_{n,j}^{(r)}$ has mean zero and is independent of $\ell_{s,t}'(\boldsymbol{X}_n^{\backslash j}, \boldsymbol{X}_{n,s})$, we have

$$
\begin{aligned}
R_2 &\leq \mathrm{poly}(d) \cdot e^{-\Omega(\log^2 d)} \\
&= d^{O(1)} \cdot e^{-\Omega(\log^2 d)} \\
&= d^{O(1)} \cdot (e^{\log d})^{-\Omega(\log d)} \\
&= d^{O(1)} \cdot d^{-\Omega(\log d)} \\
&= d^{-\Omega(\log d)} \lesssim \frac{1}{\mathrm{poly}(d)^{\Omega(\log d)}}
\end{aligned}
\tag{347}
$$

Combining the pieces above together, we can have

$$\Psi_{i,j,2}^{(t)} \leq O\left(\frac{1}{\Xi_2^3}\mathbb{E}[(\sum_{r=1}^{L}\tilde{z}_{n,j}^{(r)2})]\right) \cdot \sum_{r=1}^{L}|\langle \boldsymbol{w}_i^{(t)}, \boldsymbol{M}_j\rangle \tilde{z}_{n,j}^{(r)} - b_i^{(t)}| \tag{348}$$

Now we turn to $\Psi_{i,j,1}^{(t)}(j)$, whose calculation is similar. Defining

$$\mathfrak{B}_j := \{\sum_{r=1}^{L}|\tilde{z}_{n,j'}^{(r)}| = 0, \ \forall j' \neq j\}, \tag{349}$$

we separately discuss the cases when events $\mathfrak{B}_j$ or $\mathfrak{B}_j^c$ hold:

When $\mathfrak{B}_j^c$ happens,

$$\text{sign}(\psi_{i,j}^{(t)}(\boldsymbol{Y}_n)) = \text{sign}(\sum_{s=1}^{L}\langle \boldsymbol{w}_i^{(t)}, \boldsymbol{M}_j\rangle \tilde{z}_{n,j}^{+(s)}) \tag{350}$$

with high probability by Fact E.1 since we assumed $i \in \mathcal{M}_j^\star$. Thus, if $\langle \boldsymbol{w}_i^{(t)}, \boldsymbol{M}_j\rangle > 0$, we have

$$\mathbb{E}\left[\left(\ell_{p,t}' - 1\right)\psi_{i,j}^{(t)}(\boldsymbol{Y}_n) \cdot \sum_{r=1}^{L}\mathbf{1}_{|\langle \boldsymbol{w}_i, \boldsymbol{z}_{\boldsymbol{X}}^{(r)}\rangle| \geq b_i}\tilde{z}_{n,j}^{(r)}\ \middle|\ \mathfrak{B}_j^c\right] \geq 0 \tag{351}$$

If $\langle \boldsymbol{w}_i^{(t)}, \boldsymbol{M}_j\rangle < 0$, the opposite inequality holds as well.

When $\mathfrak{B}_j$ happens, it is easy to derive that

$$\begin{aligned}|\langle f_t(\boldsymbol{X}_n), f_t(\boldsymbol{Y}_n)\rangle| &\leq \sum_{i' \in \mathcal{M}_j}\langle w_{i'}^{(t)}, \boldsymbol{M}_j\rangle^2(\sum_{s=1}^{L}\tilde{z}_{n,j}^{+(s)})^2 + O(\Xi_2) \cdot O\left(\frac{\|w_{i'}^{(t)}\|_2}{\sqrt{d}}\right)\\ &\leq \langle \boldsymbol{w}_i^{(t)}, \boldsymbol{M}_j\rangle^2(\sum_{s=1}^{L}\tilde{z}_{n,j}^{+(s)})^2 + O\left(\frac{\Xi_2}{\sqrt{d}}\right)\end{aligned} \tag{352}$$

and therefore

$$\begin{aligned}&|\langle f_t(\boldsymbol{X}_n), f_t(\boldsymbol{Y}_n)\rangle - \langle f_t(\boldsymbol{X}_n^{\backslash j}), f_t(\boldsymbol{Y}_n^{\backslash j})\rangle|\\ &\leq \sum_{i' \in \mathcal{M}_j}\langle w_{i'}^{(t)}, \boldsymbol{M}_j\rangle^2(\sum_{s=1}^{L}\tilde{z}_{n,j}^{+(s)})^2 + O\left(\frac{\Xi_2}{\sqrt{d}}\right)\end{aligned} \tag{353}$$

From previous analysis, we also have

$$\begin{aligned}&|\langle f_t(\boldsymbol{X}_n), f_t(\boldsymbol{X}_{n,s})\rangle - \langle f_t(\boldsymbol{X}_n^{\backslash j}), f_t(\boldsymbol{X}_{n,s})\rangle|\\ &\leq \sum_{i' \in \mathcal{M}_j}\langle w_{i'}^{(t)}, \boldsymbol{M}_j\rangle^2(\sum_{r=1}^{L}\tilde{z}_{n,j}^{(r)})^2 + O\left(\frac{\tau}{\log d}\right) + O\left(\frac{\Xi_2}{\sqrt{d}}\right)\end{aligned} \tag{354}$$

These inequalities allow us to apply the same techniques in bounding $\Phi_{1,2}^{(t)}$ as follows. We define $\mathfrak{B}_p' := \mathfrak{N} \cup \{\boldsymbol{Y}_n^{\backslash j}\}$. Then, for some $G_2' = \Theta(1)$, we can have:

$$\begin{aligned}\frac{\ell_{p,t}'(\boldsymbol{X}_n^{\backslash j}, \mathfrak{B})}{\ell_{p,t}'(\boldsymbol{X}_n^{\backslash j}, \mathfrak{B}_p')} &= \frac{e^{\langle f_t(\boldsymbol{X}_n), f_t(\boldsymbol{Y}_n)\rangle/\tau}}{e^{\langle f_t(\boldsymbol{X}_n^{\backslash j}), f_t(\boldsymbol{X}_n^{\backslash j})\rangle/\tau}} \cdot \frac{\sum_{x \in \mathfrak{B}_p'}e^{\langle f_t(\boldsymbol{X}_n^{\backslash j}), f_t(x)\rangle/\tau}}{\sum_{x \in \mathfrak{B}}e^{\langle f_t(\boldsymbol{X}_n), f_t(x)\rangle/\tau}}\\ &\leq e^{2(\frac{\epsilon_j}{\epsilon_{\max}})^2 G_2' \log d + O(1/\log d)} \leq O\left(\frac{d}{\Xi_2^5}\right)\end{aligned} \tag{355}$$

Now we can proceed to compute as follows:

$$\mathbb{E}\left[\ell'_{p,t}(\boldsymbol{X}_n, \mathfrak{B}) \cdot \psi_{i,j}^{(t)}(\boldsymbol{Y}_n) \cdot \sum_{r=1}^{L} \mathbf{1}_{|\langle \boldsymbol{w}_i, \boldsymbol{z}_{\boldsymbol{X}}^{(r)}\rangle| \geq b_i} \tilde{z}_{n,j}^{(r)} \,\middle|\, \mathfrak{B}_j\right]$$

$$=\mathbb{E}\left[\frac{\ell'_{p,t}(\boldsymbol{X}_n^{\setminus j}, \mathfrak{B})}{\ell'_{p,t}(\boldsymbol{X}_n^{\setminus j}, \mathfrak{B}'_p)} \cdot \ell'_{p,t}(\boldsymbol{X}_n^{\setminus j}, \mathfrak{B}'_p) \cdot \psi_{i,j}^{(t)}(\boldsymbol{Y}_n) \cdot \sum_{r=1}^{L} \mathbf{1}_{|\langle \boldsymbol{w}_i, \boldsymbol{z}_{\boldsymbol{X}}^{(r)}\rangle| \geq b_i} \tilde{z}_{n,j}^{(r)} \,\middle|\, \mathfrak{B}_j\right] \tag{356}$$

$$\leq \sum_{s=1}^{L} |\langle \boldsymbol{w}_i^{(t)}, \boldsymbol{M}_j\rangle \tilde{z}_{n,j}^{+(s)} - b_i^{(t)}| \cdot O\left(\frac{1}{\Xi_2^5}\right) \cdot \boldsymbol{Pr}(\sum_{s=1}^{L} |\tilde{z}_{n,j}^{+(s)}| \neq 0)$$

But from Lemma E.10, and the fact that Lemma D.1 still holds for Stage III, we have:

$$\mathbb{E}\left[\psi_{i,j}^{(t)}(\boldsymbol{Y}_n) \cdot \sum_{r=1}^{L} \mathbf{1}_{|\langle \boldsymbol{w}_i, \boldsymbol{z}_{\boldsymbol{X}}^{(r)}\rangle| \geq b_i} \tilde{z}_{n,j}^{(r)} \cdot \mathbf{1}_{\mathfrak{B}_j}\right]$$

$$=\text{sign}(\sum_{s=1}^{L}\left(\langle \boldsymbol{w}_i^{(t)}, \boldsymbol{M}_j\rangle \tilde{z}_{n,j}^{+(s)}\right)) \cdot (\sum_{s=1}^{L} |\langle \boldsymbol{w}_i^{(t)}, \boldsymbol{M}_j\rangle \tilde{z}_{n,j}^{+(s)} - b_i^{(t)}|) \cdot \frac{1}{\text{polylog}(d)} \cdot \mathbb{E}[\sum_{s=1}^{L} |\tilde{z}_{n,j}^{+(s)}|]$$

$$\tag{357}$$

Combining both cases above gives the bound of $\Psi_{i,j,1}^{(t)}$. Combining results for $\Psi_{i,j,1}^{(t)}$ and $\Psi_{i,j,2}^{(t)}$ concludes the proof. The constant $G_1$ in the statement can be defined as:

$$G_1 := \min\{G'_1, G'_2\}. \tag{358}$$

$\square$

## J.3 Proof of Lemma E.3:

*Proof of Lemma E.3:* First we deal with the case of $i \in \mathcal{M}_j^\star$, we have

$$\mathbf{1}_{|\langle \boldsymbol{w}_i^{(t)}, \boldsymbol{z}_{\boldsymbol{X}}^{(r)}\rangle| \geq b_i^{(t)}} \tilde{z}_{n,j}^{(r)} = \mathbf{1}_{\tilde{z}_{n,j}^{(r)} \neq 0} \tilde{z}_{n,j}^{(r)} \tag{359}$$

when conditions in Lemma E.1 hold. Now by denoting

$$\widetilde{\psi}_{i,j}^{(t)}(\tilde{z}_{n,j}^+) := \sum_{s=1}^{L}\left(\langle \boldsymbol{w}_i^{(t)}, \boldsymbol{M}_j\rangle \tilde{z}_{n,j}^{+(s)} - b_i^{(t)}\right)\mathbf{1}_{\langle \boldsymbol{w}_i^{(t)}, \boldsymbol{M}_j\rangle \tilde{z}_{n,j}^{+(s)} > 0}$$

$$-\left(\langle \boldsymbol{w}_i^{(t)}, \boldsymbol{M}_j\rangle \tilde{z}_{n,j}^{+(s)} + b_i^{(t)}\right)\mathbf{1}_{\langle \boldsymbol{w}_i^{(t)}, \boldsymbol{M}_j\rangle \tilde{z}_{n,j}^{+(s)} < 0} \tag{360}$$

we can then easily rewrite $\Psi_{i,j}^{(t)}$ as (by using Fact E.1)

$$\Psi_{i,j}^{(t)} = \mathbb{E}\left[\left(1 - \ell'_{p,t}\right)\widetilde{\psi}_{i,j}^{(t)}(\tilde{z}_{n,j}^+) - \sum_{\boldsymbol{X}_{n,s} \in \mathfrak{N}} \ell'_{s,t} \cdot \widetilde{\psi}_{i,j}^{(t)}(\tilde{z}_{n,s,j})\right]\sum_{r=1}^{L} \mathbf{1}_{\tilde{z}_{n,j}^{(r)} \neq 0} \tilde{z}_{n,j}^{(r)} + \frac{1}{\text{poly}(d)^{\Omega(\log d)}}$$

$$= \mathbb{E}_{\boldsymbol{X}_n, \boldsymbol{Y}_n}\left[\left(\widetilde{\psi}_{i,j}^{(t)}(\tilde{z}_{n,j}^+) - I_1 - I_2\right)\sum_{r=1}^{L} \mathbf{1}_{\tilde{z}_{n,j}^{(r)} \neq 0} \tilde{z}_{n,j}^{(r)}\right] + \frac{1}{\text{poly}(d)^{\Omega(\log d)}}$$

$$\tag{361}$$

where $I_1$ and $I_2$ are defined as follows:

$$I_1 := \mathbb{E}_{\mathfrak{N}}\left[\left(\ell'_{p,t}\right)\widetilde{\psi}_{i,j}^{(t)}(\tilde{z}_{n,j}^+) + \sum_{\boldsymbol{X}_{n,s} \in \mathfrak{N}} \ell'_{s,t} \cdot \widetilde{\psi}_{i,j}^{(t)}(\tilde{z}_{n,s,j})\mathbf{1}_{\tilde{z}_{n,s,j} = \tilde{z}_{n,j}^+}\right]$$

$$\overset{①}{=} \widetilde{\psi}_{i,j}^{(t)}(\tilde{z}_{n,j}^+)\mathbb{E}_{\mathfrak{N}}\left[\frac{|\mathfrak{N}|e^{\langle f_t(\boldsymbol{X}_n), f_t(\boldsymbol{X}_{n,s})\rangle/\tau}\mathbf{1}_{\tilde{z}_{n,s,j} = \tilde{z}_{n,j}^+} + e^{\langle f_t(\boldsymbol{X}_n), f_t(\boldsymbol{Y}_n)\rangle/\tau}}{e^{\langle f_t(\boldsymbol{X}_n), f_t(\boldsymbol{X}_{n,s})\rangle/\tau} + \sum_{x \in \mathfrak{N}\setminus\{\boldsymbol{X}_{n,s}\}} e^{\langle f_t(\boldsymbol{X}_n), f_t(x)\rangle/\tau}}\right] \tag{362}$$

$$I_2 := \mathbb{E}_{\mathfrak{N}}\left[\sum_{\boldsymbol{X}_{n,s} \in \mathfrak{N}} \ell'_{s,t} \cdot \widetilde{\psi}_{i,j}^{(t)}(\tilde{z}_{n,s,j})\mathbf{1}_{\tilde{z}_{n,s,j} \neq \tilde{z}_{n,j}^+}\right]$$

where in ① we used the identification $\tilde{z}_{n,j}^+ = \tilde{z}_{n,s,j}$. The tricky part here is since all the variables inside the expectation is non-negative, we can use Jensen's inequality to move the expectation of $e^{\langle f_t(\boldsymbol{X}_n), f_t(X_{n,u}) \rangle / \tau}$ to the denominator. We let

$$V := e^{\langle f_t(\boldsymbol{X}_n), f_t(\boldsymbol{Y}_n) \rangle / \tau} \tag{363}$$

and consider it fixed when computing $I_1$ as follows: conditioned on $\tilde{z}_{n,j}^+ \neq 0$, we have

$$
\begin{aligned}
&\frac{I_1}{\widetilde{\psi}_{i,j}^{(t)}(\tilde{z}_{n,j}^+)} \\
=&\mathbb{E}_{\mathfrak{N}} \left[ \frac{|\mathfrak{N}| \cdot e^{\langle f_t(\boldsymbol{X}_n), f_t(\boldsymbol{X}_{n,s}) \rangle / \tau} \mathbf{1}_{\tilde{z}_{n,s,j} = \tilde{z}_{n,j}^+} + V}{e^{\langle f_t(\boldsymbol{X}_n), f_t(\boldsymbol{X}_{n,s}) \rangle / \tau} + V + \sum_{x \in \mathfrak{N} \setminus \{\boldsymbol{X}_{n,s}\}} e^{\langle f_t(\boldsymbol{X}_n), f_t(X_{n,u}) \rangle / \tau}} \right] \\
\geq&\mathbb{E}_{\boldsymbol{X}_{n,s}} \left[ \frac{e^{\langle f_t(X_p^+), f_t(\boldsymbol{X}_{n,s}) \rangle / \tau} \cdot \mathbf{1}_{\tilde{z}_{n,s,j} = \tilde{z}_{n,j}^+} + \frac{1}{|\mathfrak{N}|} V}{\frac{1}{|\mathfrak{N}|} e^{\langle f_t(\boldsymbol{X}_n), f_t(\boldsymbol{X}_{n,s}) \rangle / \tau} + \frac{1}{|\mathfrak{N}|} V + \frac{|\mathfrak{N}|-1}{|\mathfrak{N}|} \mathbb{E}_{\boldsymbol{X}_n} \left[ e^{\langle f_t(\boldsymbol{X}_n), f_t(\boldsymbol{X}_n) \rangle / \tau} \right]} \right] \quad \text{(by Jensen inequality)} \\
=&\mathbb{E}_{\boldsymbol{X}_{n,s}} \left[ \frac{e^{\langle f_t(\boldsymbol{X}_n), f_t(\boldsymbol{X}_{n,s}) \rangle / \tau} \cdot \mathbf{1}_{\tilde{z}_{n,s,j} = \tilde{z}_{n,j}^+} + \frac{1}{|\mathfrak{N}|} V}{\frac{1}{|\mathfrak{N}|} \left( e^{\langle f_t(\boldsymbol{X}_n), f_t(\boldsymbol{X}_{n,s}) \rangle / \tau} + V \right) + \frac{|\mathfrak{N}|-1}{|\mathfrak{N}|} \mathbb{E}_{\boldsymbol{X}_n} \left[ e^{\langle f_t(\boldsymbol{X}_n), f_t(\boldsymbol{X}_n) \rangle / \tau} \left( \mathbf{1}_{\tilde{z}_{n,s,j} = \tilde{z}_{n,j}^+} + \mathbf{1}_{\tilde{z}_{n,s,j} \neq \tilde{z}_{n,j}^+} \right) \right]} \right] \\
\overset{①}{\geq}&\mathbb{E}_{\boldsymbol{X}_{n,s}} \left[ \frac{X + \frac{1}{|\mathfrak{N}|} V}{\frac{1}{|\mathfrak{N}|} (X + V) + \frac{|\mathfrak{N}|-1}{|\mathfrak{N}|} \left( 1 + \frac{1}{\text{poly}(d)} \right) \mathbb{E}_{\boldsymbol{X}_n}[X]} \right] \\
&\text{where} \quad X := e^{\langle f_t(\boldsymbol{X}_n), f_t(\boldsymbol{X}_n) \rangle / \tau} \cdot \mathbf{1}_{\tilde{z}_{n,s,j} = \tilde{z}_{n,j}^+} \geq 0) \\
\overset{②}{\geq}&1 - O\left( \frac{1}{\text{poly}(d)} \right)
\end{aligned}
\tag{364}
$$

- in ①, we need to go through similar analysis as in the proof of Lemma E.2 to obtain that, with high probability over $\boldsymbol{X}_n$ and $\boldsymbol{X}_n^{\setminus j}$:

$$
\begin{aligned}
&\frac{\langle f_t(\boldsymbol{X}_n), f_t(\boldsymbol{X}_n) \rangle - \langle f_t(\boldsymbol{X}_n), f_t(\boldsymbol{X}_n^{\setminus j}) \rangle}{\tau} \\
\geq&\frac{1}{\tau} \sum_{i \in \mathcal{M}_j} \langle \boldsymbol{w}_i^{(t)}, \boldsymbol{M}_j \rangle^2 (\sum_{s=1}^{L} \tilde{z}_{n,j}^{+(s)})^2 - O\left( \frac{1}{\log d} \right) \\
\geq&G_2 (\frac{\epsilon_j}{\epsilon_{\max}})^2 \log d - O\left( \frac{1}{\log d} \right)
\end{aligned}
\tag{365}
$$

for some very large constant $G_2 = \Theta(1)$, which gives (the $\frac{1}{\text{poly}(d)}$ here depends on how large $G_2$ is):

$$
\begin{aligned}
&\mathbb{E}_{\boldsymbol{X}_n} \left[ e^{\langle f_t(\boldsymbol{X}_n), f_t(\boldsymbol{X}_n) \rangle / \tau} \cdot \mathbf{1}_{\tilde{z}_{n,s,j} \neq \tilde{z}_{n,j}^+} \right] \\
\leq&\frac{1}{\text{poly}(d)} \mathbb{E}_{\boldsymbol{X}_n} \left[ e^{\langle f_t(\boldsymbol{X}_n), f_t(\boldsymbol{X}_n) \rangle / \tau} \cdot \mathbf{1}_{\tilde{z}_{n,s,j} = \tilde{z}_{n,j}^+} \right]
\end{aligned}
\tag{366}
$$

- in inequality ②, we need to argue as follows, where

$$\widetilde{\mathbb{E}}[X] \overset{\text{abbr.}}{=} \mathbb{E}_{\boldsymbol{X}_n}[X] \tag{367}$$

is only integrated over the randomness of $\boldsymbol{X}_n$:

$$
\mathbb{E}_{\boldsymbol{X}_{n,s}}\left[\frac{X + \frac{1}{|\mathfrak{N}|}V}{\frac{1}{|\mathfrak{N}|}(X + V) + \frac{|\mathfrak{N}|-1}{|\mathfrak{N}|}\left(1 + \frac{1}{\text{poly}(d)}\right)\widetilde{\mathbb{E}}[X]}\right]
$$

$$
= 1 - \frac{1}{|\mathfrak{N}|}\mathbb{E}_{\boldsymbol{X}_{n,s}}\left[\frac{\widetilde{\mathbb{E}}[X]}{\frac{1}{|\mathfrak{N}|}(X + V) + \frac{|\mathfrak{N}|-1}{|\mathfrak{N}|}\left(1 + \frac{1}{\text{poly}(d)}\right)\widetilde{\mathbb{E}}[X]}\right] \tag{368}
$$

$$
\geq 1 - \frac{1}{|\mathfrak{N}|}\cdot\frac{\widetilde{\mathbb{E}}[X]}{\left(\frac{|\mathfrak{N}|-1}{|\mathfrak{N}|}\left(1 + \frac{1}{\text{poly}(d)}\right)\widetilde{\mathbb{E}}[X]\right)} \quad (\text{since } X + V \geq 0)
$$

$$
\geq 1 - \frac{1}{\text{poly}(d)}
$$

The same analysis applies to $I_2$, which we can bound as

$$
\left|\frac{I_2}{\widetilde{\psi}_{i,j}^{(t)}(-\tilde{z}_{n,j}^+)}\right| \leq \frac{1}{\text{poly}(d)} \tag{369}
$$

Combining both $I_1$ and $I_2$, we have

$$
\Psi_{i,j}^{(t)} \leq \frac{1}{\text{poly}(d)}\sum_{s=1}\left|\langle\boldsymbol{w}_i^{(t)}, \boldsymbol{M}_j\rangle\tilde{z}_{n,j}^{+(s)}\right| \tag{370}
$$

In the case of $i \in \mathcal{M}_j$, we have with probability $\leq \widetilde{O}\left(\frac{1}{d}\right)$ that

$$
\mathbf{1}_{\langle\boldsymbol{w}_i^{(t)}, \boldsymbol{z}_{\boldsymbol{X}}^{(r)}\rangle\geq b_i^{(t)}} \neq \mathbf{1}_{\langle\boldsymbol{w}_i^{(t)}, \boldsymbol{M}_j\rangle\tilde{z}_{n,j}^{(r)}>0} \quad \text{or} \quad \mathbf{1}_{\langle\boldsymbol{w}_i^{(t)}, \boldsymbol{z}_{\boldsymbol{Y}}^{(s)}\rangle>b_i^{(t)}} \neq \mathbf{1}_{\langle\boldsymbol{w}_i^{(t)}, \boldsymbol{M}_j\rangle\tilde{z}_{n,j}^{+(s)}>0} \tag{371}
$$

When such events happen, we can obtain a bound of

$$
O\left(\frac{1}{d}\right)b_i^{(t)} \tag{372}
$$

over $\Psi_{i,j}^{(t)}$, which times the probability $\widetilde{O}\left(\frac{1}{d}\right)$ leads to our bound.

Combining the above observations and the analyses, we can complete the proof. $\qquad\square$

## J.4 PROOF OF LEMMA E.4:

*Proof of Lemma E.4:* Let $j \in [d]$, since the case of $j \in [d_1] \setminus [d]$ can be similarly dealt with. We first look at the following $\mathcal{E}_{1,i,j}^{(t)}$ term in $\mathcal{E}_{i,j}^{(t)}$:

$$
\mathcal{E}_{1,i,j}^{(t)} = \mathbb{E}\left[h_{i,t}(\boldsymbol{Y}_n)\sum_{r=1}^{L}\mathbf{1}_{|\langle\boldsymbol{w}_i, \boldsymbol{z}_{\boldsymbol{X}}^{(r)}\rangle|\geq b_i}\langle\boldsymbol{M}_j, \tilde{\xi}_n^{(r)}\rangle\right] \tag{373}
$$

we have

$$
|\mathcal{E}_{1,i,j}^{(t)}| = \mathbb{E}\left[\sum_{s=1}^{L}\sum_{r=1}^{L}|\langle w_i^{(t)}, \boldsymbol{z}_{\boldsymbol{Y}}^{(s)}\rangle|\mathbf{1}_{|\langle w_i^{(t)}, \frac{z_{\boldsymbol{X}}^{(r)}+z_{\boldsymbol{Y}}^{(s)}}{2}\rangle|\geq b_i^{(t)}+|\langle w_i^{(t)}, \boldsymbol{z}_{\boldsymbol{X}}^{(r)} - \frac{z_{\boldsymbol{X}}^{(r)}+z_{\boldsymbol{Y}}^{(s)}}{2}\rangle|}|\langle\boldsymbol{M}_j, \tilde{\xi}_n^{(r)}\rangle|\right] \tag{374}
$$

When $\left\{|\langle w_i^{(t)}, \frac{z_{\boldsymbol{X}}^{(r)}+z_{\boldsymbol{Y}}^{(s)}}{2}\rangle| \geq b_i^{(t)} + |\langle w_i^{(t)}, \boldsymbol{z}_{\boldsymbol{X}}^{(r)} - \frac{z_{\boldsymbol{X}}^{(r)}+z_{\boldsymbol{Y}}^{(s)}}{2}\rangle|\right\}$ happens (which we know from Fact E.1 has prob $\leq \widetilde{O}(\frac{1}{d})$), using Lemma E.7, we have

$$
|\mathcal{E}_{1,i,j}^{(t)}| \leq \widetilde{O}\left(\frac{1}{d^2}\right)\|\boldsymbol{w}_i^{(t)}\|_2 \tag{375}
$$

Proof of $\mathcal{E}_{2,i,j}^{(t)}$ can refer to Lemma E.7 and Lemma E.8 in(Wen & Li, 2021).

$$
\begin{aligned}
\mathcal{E}_{2,i,j}^{(t)} =& \mathbb{E}\left[ \sum_{\boldsymbol{X}_{n,s}\in\mathfrak{N}} \ell_{s,t}'(\boldsymbol{X}_n,\mathfrak{B})\cdot h_{i,t}(\boldsymbol{X}_{n,s}) \sum_{r=1}^{L} \mathbf{1}_{|\langle \boldsymbol{w}_i, \boldsymbol{z}_{\boldsymbol{X}}^{(r)}\rangle|\geq b_i}\langle \boldsymbol{M}_j, \tilde{\xi}_n^{(r)}\rangle \right] \\
\leq& O\left( \frac{\|\boldsymbol{w}_i^{(t)}\|_2 \Xi_2^2}{d^2\tau} \right)\cdot \max_{i'\in[m]}\left( \left|\langle w_{i'}^{(t)}, \boldsymbol{M}_j\rangle\right| \right)
\end{aligned}
\tag{376}
$$

Combining the results of $\mathcal{E}_{1,i,j}^{(t)}$ and $\mathcal{E}_{2,i,j}^{(t)}$, we can conclude the proof. $\qquad\square$

### J.5  PROOF OF LEMMA E.5:

*Proof of Lemma E.5:* The proof essentially relies on the condition that Lemma E.1 holds for all $t'\in[T_3, t]$. We first consider the case where $i\in\mathcal{M}_j^\star$. Similar to how $\psi_{i,j}^{(t)}(x)$ are defined for each $x$ in Definition E.1, for each $j'\neq j$, we let

$$
\rho_{i,j}^{(t)}(\boldsymbol{Y}_n) := \sum_{s=1}^{L}\left( \langle \boldsymbol{w}_i^{(t)}, z_Y^{(s)}\rangle - \langle \boldsymbol{w}_i^{(t)}, \boldsymbol{M}\tilde{z}_n^{+(s)}\rangle \right) \mathbf{1}_{\tilde{z}_{n,j}^{+(s)}\neq 0}
\tag{377}
$$

Now it is straightforward to decompose $\Phi_{i,j}^{(t)}$ as follows:

$$
\begin{aligned}
&\Phi_{i,j}^{(t)} \\
=&\mathbb{E}\left[ \left( (\ell_{p,t}'-1)\cdot\phi_{i,j}^{(t)}(\boldsymbol{Y}_n) + \sum_{\boldsymbol{X}_{n,s}\in\mathfrak{N}} \ell_{s,t}'\cdot\phi_{i,j}^{(t)}(\boldsymbol{X}_{n,s}) \right) \sum_{r=1}^{L}\mathbf{1}_{|\langle \boldsymbol{w}_i, \boldsymbol{z}_{\boldsymbol{X}}^{(r)}\rangle|\geq b_i}\tilde{z}_{n,j}^{(r)} \right] \\
=&\sum_{j'\in[d], j'\neq j} \langle \boldsymbol{w}_i^{(t)}, \boldsymbol{M}_{j'}\rangle\mathbb{E}\left[ \left( (\ell_{p,t}'-1)\cdot\sum_{s=1}^{L}\tilde{z}_{n,j'}^{+(s)} + \sum_{\boldsymbol{X}_{n,s}\in\mathfrak{N}} \ell_{s,t}'\cdot\sum_{q=1}^{L}\tilde{z}_{n,s,j'}^{(q)}\mathbf{1}_{\tilde{z}_{n,s,j'}^{(q)}\neq 0} \right) \sum_{r=1}^{L}\mathbf{1}_{|\langle \boldsymbol{w}_i, \boldsymbol{z}_{\boldsymbol{X}}^{(r)}\rangle|\geq b_i}\tilde{z}_{n,j}^{(r)} \right] \\
&+\mathbb{E}\left[ \left( (\ell_{p,t}'-1)\cdot\rho_{i,j}^{(t)}(\boldsymbol{Y}_n) + \sum_{\boldsymbol{X}_{n,s}\in\mathfrak{N}} \ell_{s,t}'\cdot\rho_{i,j}^{(t)}(\boldsymbol{X}_{n,s}) \right) \sum_{r=1}^{L}\mathbf{1}_{|\langle \boldsymbol{w}_i, \boldsymbol{z}_{\boldsymbol{X}}^{(r)}\rangle|\geq b_i}\tilde{z}_{n,j}^{(r)} \right] + \frac{1}{\mathrm{poly}(d)^{\Omega(\log d)}} \\
=&H_1 + H_2 + \frac{1}{\mathrm{poly}(d)^{\Omega(\log d)}}
\end{aligned}
\tag{378}
$$

Indeed, from similar arguments as in the proof of Lemma E.3 and Lemma E.4, we can trivially obtain

$$
|H_2|\leq O\left( \frac{\Xi_2^2}{d^2} \right)\|\boldsymbol{w}_i^{(t)}\|_2
\tag{379}
$$

Now we turn to $H_1$. Since $\max_{j'\neq j}|\langle \boldsymbol{w}_i^{(t)}, \boldsymbol{M}_{j'}\rangle|\leq O\left( \frac{\epsilon_j}{\epsilon_{\max}}\frac{\|\boldsymbol{w}_i^{(t)}\|_2}{\sqrt{d}\Xi_2^5} \right)$, we can simply get (Since w.h.p., $|\{j'\in[d]:\tilde{z}_{n,j'}^{+(s)}\neq 0\}|=\widetilde{O}(1)$, and if $\tilde{z}_{n,j'}^{+(s)}=0$, the negative terms are small from similar analysis in Lemma E.2)

$$
\begin{aligned}
&|H_1| \\
\leq& O\left( \frac{\epsilon_j}{\epsilon_{\max}}\frac{\|\boldsymbol{w}_i^{(t)}\|_2}{\sqrt{d}\Xi_2^5} \right) \sum_{j'\neq j, j'\in[d]} \mathbb{E}\left[ \left( (\ell_{p,t}'-1)\cdot\sum_{s=1}^{L}\tilde{z}_{n,j'}^{+(s)} + \sum_{\boldsymbol{X}_{n,s}\in\mathfrak{N}} \ell_{s,t}'\cdot\sum_{q=1}^{L}\tilde{z}_{n,s,j'}^{(q)}\mathbf{1}_{\tilde{z}_{n,s,j'}^{(q)}\neq 0} \right) \sum_{r=1}^{L}\mathbf{1}_{|\langle \boldsymbol{w}_i, \boldsymbol{z}_{\boldsymbol{X}}^{(r)}\rangle|\geq b_i}\tilde{z}_{n,j}^{(r)} \right] \\
\leq& O\left( \frac{\epsilon_j}{\epsilon_{\max}}\frac{\|\boldsymbol{w}_i^{(t)}\|_2}{d^{3/2}} \right)
\end{aligned}
\tag{380}
$$

Then we can obtain a crude bound for all $t \in \left[\frac{d^{1.01}}{\eta}, \frac{d^{1.99}}{\eta}\right]$ by

$$\Phi_{i,j}^{(t)} \leq (H_1 + H_2) + \frac{1}{\text{poly}(d)^{\Omega(\log d)}} \leq \widetilde{O}\left(\frac{\epsilon_j}{\epsilon_{\max}} \frac{\|\boldsymbol{w}_i^{(t)}\|_2}{d^{3/2}}\right) \tag{381}$$

The harder part is to deal with iterations $t \in \left[\frac{d^{1.495}}{\eta}, \frac{d^{1.498}}{\eta}\right]$. We first establish a connection between $\Psi^{(t)}$ and $\Phi^{(t)}$. We first assume that for all $j' \neq j, j' \in [d]$, it holds that

$$\left|\Psi_{i',j'}^{(t_1)}\right| / \left|\langle w_{i'}^{(t_1)}, \boldsymbol{M}_{j'}\rangle\right| \leq \Omega\left(\frac{\Xi_2^2}{\sqrt{dt\eta}}\right), \tag{382}$$

which is true for all iteration $t \leq \frac{d \text{ polylog}(d)}{\eta}$ from simple calculations.

Now suppose at some $t_1 \geq \frac{d \text{ polylog}(d)}{\eta}$, there exists some $j' \neq j, j' \in [d]$ and $i' \in \mathcal{M}_j^\star$ such that

$$\left|\Psi_{i',j'}^{(t_1)}\right| / \left|\langle w_{i'}^{(t_1)}, \boldsymbol{M}_{j'}\rangle\right| \geq \Omega\left(\frac{\Xi_2}{\sqrt{dt\eta}}\right) \tag{383}$$

which means we have the followings:

$$\mathbb{E}\left[\left((\ell'_{p,t_1} - 1)\sum_{s=1}^{L} \tilde{z}_{n,j'}^{+(s)} + \sum_{\boldsymbol{X}_{n,s} \in \mathfrak{N}} \ell'_{s,t_1} \sum_{q=1}^{L} \tilde{z}_{n,s,j'}^{(q)}\right)\sum_{r=1}^{L} \tilde{z}_{n,j'}^{(r)}\right] \geq \Omega\left(\frac{\Xi_2}{\sqrt{dt\eta}}\right) \tag{384}$$

Letting $\Delta > 0$ be defined as the number such that if $\mathfrak{F}_j^{(t)} = \Delta$, we can have

$$\left|\Psi_{i',j'}^{(t)}\right| / \left|\langle w_{i'}^{(t)}, \boldsymbol{M}_{j'}\rangle\right| \leq O\left(\frac{\sqrt{\Xi_2}\tau \log d}{\sqrt{dt\eta}}\right). \tag{385}$$

Then from the calculations in the proof of Lemma E.2, there must be a constant $\delta > \Omega(1)$ such that

$$\mathfrak{F}_j^{(t_1)} \geq \Delta - \delta\tau \log d. \tag{386}$$

However, such growth cannot continue since for some $t' = \Theta(t/\sqrt{\Xi_2})$, we have for each $i' \in \mathcal{M}_j'$:

$$\begin{aligned}
\left|\langle w_{i'}^{(t_1+t')}, \boldsymbol{M}_{j'}\rangle\right| &\geq \left|\langle w_{i'}^{(t_1+t')}, \boldsymbol{M}_{j'}\rangle\right|(1 - \eta\lambda) + \Psi_{i',j'}^{(t_1+t'-1)} + \Phi_{i',j'}^{(t)} + O\left(\frac{\Xi_2^2}{d^2}\right) \\
&\geq \left|\langle w_{i'}^{(t)}, \boldsymbol{M}_{j'}\rangle\right|(1 - \eta\lambda)^{t'} + \sum_{s=t}^{t+t'-1} \Psi_{i',j'}^{(s)} - O\left(\frac{t'\Xi_2^2}{d^{3/2}}\right)
\end{aligned} \tag{387}$$

where the bounds for $\Phi_{i',j'}^{(s)}$ for each $s \in [t_1, t_1 + t']$ are obtained from induction over iterations $s' \in \left[\frac{d^{1.01}}{\eta}, s\right]$. Therefore there must exist $t''' \in [t, t+t']$ such that $\left|\Psi_{i',j'}^{(t''')}\right| \leq O\left(\frac{\sqrt{\Xi_2}}{\tau\sqrt{dt\eta}}\right)$ or otherwise $\mathfrak{F}_j^{(t+t')} \geq \mathfrak{F}_j^{(t)} + t' \cdot O\left(\frac{\sqrt{\Xi_2}\tau \log d}{\sqrt{dt\eta}}\right) \geq \Delta + \delta\tau \log d$, which results in that $\left|\Psi_{i',j'}^{(t)}\right| \leq O\left(\frac{\tau \log d}{\sqrt{dt\eta}}\right)\|\boldsymbol{w}_i^{(t)}\|_2$, following the same reasoning in Lemma E.3. Above arguments actually proved that $\left|\Psi_{i',j'}^{(t)}\right| \leq \Omega\left(\frac{\Xi_2}{\sqrt{dt\eta}}\right)\|\boldsymbol{w}_i^{(t)}\|_2$ at all $t \in \left[\frac{d \text{ polylog}(d)}{\eta}, \frac{d^{1.498}}{\eta}\right]$. Therefore we can use the results of all $\Psi_{i',j'}^{(t)}$, where $j' \neq j, j' \in [d]$ to get (combined with Fact E.1):

$$|H_1| \leq \widetilde{O}\left(\max_{j' \neq j, j' \in [d]} \Psi_{i,j}^{(t)}\right) \leq \widetilde{O}\left(\frac{\Xi_2^2}{d^2}\|\boldsymbol{w}_i^{(t)}\|_2\right) \tag{388}$$

For iterations $t \geq \frac{d^{1.498}}{\eta}$, the proof is essentially the same: we only need to notice that the difference $\Psi_{i,j}^{(t)} - \lambda\langle \boldsymbol{w}_i^{(t)}, \boldsymbol{M}_j\rangle$ here will bounce around zero, while the compensation terms in $H_1$ are bounded

by $\widetilde{O}\left(\frac{\|\boldsymbol{w}_i^{(t)}\|_2}{d^{1.98}}\right)$. These observations indeed prove the case $i \in \mathcal{M}_j^\star$. When $i \in \mathcal{M}_j \setminus \mathcal{M}_j^\star$, notice that with prob $\leq \widetilde{O}\left(\frac{1}{d}\right)$ it holds $\mathbf{1}_{|\langle \boldsymbol{w}_i^{(t)}, z_Y^{(s)}\rangle| \geq b_i^{(t)}} = \mathbf{1}_{\tilde{z}_{n,j'}^{+(s)} \neq 0}$ for any $x \in \mathfrak{B}$. Now we expand

$$
\begin{aligned}
&H_1 \\
&= \sum_{j' \in \mathcal{N}_i, j' \neq j} \langle \boldsymbol{w}_i^{(t)}, \boldsymbol{M}_{j'}\rangle \mathbb{E}\left[\left((\ell'_{p,t}-1)\cdot \sum_{s=1}^{L}\tilde{z}_{n,j'}^{+(s)} + \sum_{\boldsymbol{X}_{n,s}\in\mathcal{N}} \ell'_{s,t}\cdot\sum_{q=1}^{L}\tilde{z}_{n,s,j'}^{(q)}\mathbf{1}_{\tilde{z}_{n,s,j'}^{(q)}\neq 0}\right)\sum_{r=1}^{L}\mathbf{1}_{|\langle\boldsymbol{w}_i,z_{\boldsymbol{X}}^{(r)}\rangle|\geq b_i}\tilde{z}_{n,j}^{(r)}\right] \\
&\quad + \sum_{j' \notin \mathcal{N}_i, j' \neq j} \langle \boldsymbol{w}_i^{(t)}, \boldsymbol{M}_{j'}\rangle \mathbb{E}\left[\left((\ell'_{p,t}-1)\cdot \sum_{s=1}^{L}\tilde{z}_{n,j'}^{+(s)} + \sum_{\boldsymbol{X}_{n,s}\in\mathcal{N}} \ell'_{s,t}\cdot\sum_{q=1}^{L}\tilde{z}_{n,s,j'}^{(q)}\mathbf{1}_{\tilde{z}_{n,s,j'}^{(q)}\neq 0}\right)\sum_{r=1}^{L}\mathbf{1}_{|\langle\boldsymbol{w}_i,z_{\boldsymbol{X}}^{(r)}\rangle|\geq b_i}\tilde{z}_{n,j}^{(r)}\right]
\end{aligned}
\tag{389}
$$

Indeed, the event that there are some $j' \in \mathcal{N}_i$ (which means $i \in \mathcal{M}_j$) such that $z_{p,j'} \neq 0$ has probability $\leq \widetilde{O}(1/d)$. Thus the first term on the RHS is trivially bounded by $\widetilde{O}(1/d^2)\|\boldsymbol{w}_i^{(t)}\|_2$. For the second term of $H_1$, we can again go through similar procedure as above to obtain that

$$
\begin{aligned}
&\mathbb{E}\left[(\ell'_{p,t}-1)\sum_{s=1}^{L}\tilde{z}_{n,j'}^{+(s)} + \sum_{\boldsymbol{X}_{n,s}\in\mathfrak{N}} \ell'_{s,t}\sum_{q=1}^{L}\tilde{z}_{n,s,j'}^{(q)}\mathbf{1}_{\tilde{z}_{n,s,j'}^{(q)}\neq 0}\right]\sum_{r=1}^{L}\mathbf{1}_{|\langle\boldsymbol{w}_i,z_{\boldsymbol{X}}^{(r)}\rangle|\geq b_i}\tilde{z}_{n,j}^{(r)} \\
&\leq \max\left\{\widetilde{O}\left(\frac{\sqrt{\Xi_2}}{\sqrt{dt\eta}}\right), \frac{1}{d^{1.99}}\right\}\|\boldsymbol{w}_i^{(t)}\|_2
\end{aligned}
\tag{390}
$$

Then again we have

$$
|H_1| \leq \widetilde{O}\left(\max_{j'\neq j, j'\in[d]}\Psi_{i,j}^{(t)}\right) \leq \widetilde{O}\left(\max\left\{\frac{\Xi_2^2}{d^2}, \frac{\Xi_2}{\sqrt{dt\eta}}\right\}\right)\|\boldsymbol{w}_i^{(t)}\|_2
\tag{391}
$$

which can be combined with the bound for $H_2$ to conclude the proof. $\qquad\square$

### J.6 PROOF OF LEMMA E.6:

*Proof of Lemma E.6:* By using Bernoulli concentration, we know that whenever $\sum_{j\in[d]}\mathbf{1}_{\sum_{r=1}^{L}|\tilde{z}_{n,j}^{(r)}|\neq 0} = \Omega(\log\log d)$ (which happens with constant probability), we have

$$
\sum_{j\in[d]}\mathbf{1}_{\sum_{q=1}^{L}|\tilde{z}_{n,s,j}^{(q)}|=\sum_{r=1}^{L}|\tilde{z}_{n,j}^{(r)}|} \leq C\sum_{j\in[d]}\mathbf{1}_{\sum_{r=1}^{L}|\tilde{z}_{n,j}^{(r)}|\neq 0}.
\tag{392}
$$

(with prob $\geq 1 - \frac{1}{d^{\Omega(\log\log d)}}$ for all $X_{n,s} \in \mathfrak{N}$)

And also from Definition E.2 we know that if for some $j \in [d]$, $\sum_{r=1}^{L}|\tilde{z}_{n,j}^{(r)}| = \sum_{q=1}^{L}|\tilde{z}_{n,s,j}^{(q)}|$, then

$$
\sum_{i\in\mathcal{M}_j^\star}(f_{t,\theta^\star,i}(X_n)h_{i,t}(X_{n,s}) - f_{t,\theta^\star,i}(X_n)h_{i,t}(Y_n)) \geq \kappa\tau\sum_{j\in\mathcal{M}_j^\star}|\langle\boldsymbol{w}_i^{(t)}, \mathbf{M}_j\rangle| + O\left(\frac{1}{\log d}\right)
\tag{393}
$$

which can be obtained by similar calculations in Lemma E.2.

Noticing that the event $\sum_{r=1}^{L}|\tilde{z}_n^{(r)}| \neq 0$ happens with prob $\geq 1 - \frac{1}{\text{polylog}(d)}$, we have

$$\widetilde{L}(f_{t,\theta^\star}, f_t)$$

$$\leq \left(1 - \frac{1}{\text{polylog}(d)}\right) \mathbb{E}\left[\log\left(\sum_{X \in \mathfrak{B}} e^{\langle f_{t,\theta^\star}(X_n), f_t(X)\rangle/\tau - \langle f_{t,\theta^\star}(X_n), f_t(Y_n)\rangle/\tau}\right) \middle| \sum_{r=1}^L |\tilde{z}_n^{(r)}| \neq 0\right]$$

$$+ \Pr(\sum_{r=1}^L |\tilde{z}_n^{(r)}| = 0) \cdot O(\log |\mathfrak{B}|)$$

$$\overset{(1)}{=} \left(1 - \frac{1}{\text{polylog}(d)}\right) \mathbb{E}\left[\log\left(\sum_{X \in \mathfrak{B}} e^{\sum_{j \in [d]} \sum_{i \in \mathcal{M}_j^\star}(f_{t,\theta^\star,i}(X_n)h_{i,t}(X) - f_{t,\theta^\star,i}(X_n)h_{i,t}(Y_n))/\tau}\right) \middle| \sum_{r=1}^L |\tilde{z}_n^{(r)}| \neq 0\right]$$

$$+ \Pr(\sum_{r=1}^L |\tilde{z}_n^{(r)}| = 0) \cdot O(\log |\mathfrak{B}|)$$

$$\leq O\left(\frac{1}{\log d}\right)$$

$$\tag{394}$$

$(1)$ is because $\theta^\star$ has a value only when $i \in \mathcal{M}_j^\star$ occurs. where the last inequality combines the Bernoulli concentration results of $\sum_{j \in [d]} \sum_{r=1}^L |\tilde{z}_{n,j}^{(r)}| \sum_{q=1}^L |\tilde{z}_{n,s,j}^{(q)}|$ and a union bound for all $s \in [\mathfrak{N}]$, and that $\sum_{i \in \mathcal{M}_j^\star} |\langle \boldsymbol{w}_i^{(t)}, \mathbf{M}_j\rangle| \geq \Omega\left(\frac{\sqrt{\tau}}{\mathbb{E}_2}\right)$. $\qquad\square$

### J.7 PROOF OF LEMMA E.7(A):

*Proof of Lemma E.7(a):* Since the mean of $\langle \boldsymbol{w}_i, z_X^{(r)}\rangle$ is zero, we can simply compute the variance as

$$\text{Var}\left(\langle \boldsymbol{w}_i, z_X^{(r)\backslash j}\rangle + \frac{1}{L}\langle \boldsymbol{w}_i, \boldsymbol{M}_j\rangle \tilde{z}_{n,j}^{(r)}\right)$$

$$\leq \frac{2}{L^2}\mathbb{E}\left[\left(\langle \boldsymbol{w}_i, \sum_{j' \neq j, j' \in [d]} \boldsymbol{M}_{j'}\tilde{z}_{n,j'}^{(r)} + \tilde{\xi}_n^{(r)}\rangle\right)^2\right] + \frac{2}{L^2}\mathbb{E}\left[\left(\langle \boldsymbol{w}_i, \boldsymbol{M}_j\rangle \tilde{z}_{n,j}^{(r)}\right)^2\right]$$

$$\leq \frac{4}{L^2}\sum_{s=1}^{d_1}(\boldsymbol{w}_i)_s^2(M_{j'})_s^2\mathbb{E}\left[\sum_{j' \neq j}(\tilde{z}_{n,j'}^{(r)})^2\right] + \frac{4}{L^2}\sum_{s=1}^{d_1}(\boldsymbol{w}_i)_s^2\mathbb{E}\left[(\tilde{\xi}_{n,s}^{(r)})^2\right]$$

$$+ \frac{2}{L^2}\sum_{s=1}^{d_1}(\boldsymbol{w}_i)_s^2(\boldsymbol{M}_j)_s^2\mathbb{E}\left[(\tilde{z}_{n,j'}^{(r)})^2\right]$$

$$\leq \widetilde{\mathcal{O}}\left(\frac{\|\boldsymbol{w}_i\|_2^2}{d_1}\right) + \widetilde{\mathcal{O}}\left(\frac{\|\boldsymbol{w}_i\|_2^2}{d}\right) + \widetilde{\mathcal{O}}\left(\frac{\|\boldsymbol{w}_i\|_2^2}{d_1}\right)$$

$$= \widetilde{\mathcal{O}}\left(\frac{\|\boldsymbol{w}_i\|_2^2}{d}\right).$$

$$\tag{395}$$

Now we can use Chebychev's inequality to conclude: For a random variable $X$ with mean zero, Chebyshev's inequality tells us:

$$\Pr(|X| \geq t) \leq \frac{\mathrm{Var}(X)}{t^2}$$

$$\Pr(|\langle \boldsymbol{w}_i, z_X^{(r)\backslash j}\rangle + \frac{1}{L}\langle \boldsymbol{w}_i, \boldsymbol{M}_j\rangle \tilde{z}_{n,j}^{(r)}| \geq t) \leq \frac{\mathrm{Var}(\langle \boldsymbol{w}_i, z_X^{(r)\backslash j}\rangle + \langle \boldsymbol{w}_i, \boldsymbol{M}_j\rangle \tilde{z}_{n,j}^{(r)})}{t^2}$$

$$\Pr\left(\left(\langle \boldsymbol{w}_i, z_X^{(r)\backslash j}\rangle + \frac{1}{L}\langle \boldsymbol{w}_i, \boldsymbol{M}_j\rangle \tilde{z}_{n,j}^{(r)}\right)^2 \geq t^2\right) \leq \frac{\mathrm{Var}(\langle \boldsymbol{w}_i, z_X^{(r)\backslash j}\rangle + \langle \boldsymbol{w}_i, \boldsymbol{M}_j\rangle \tilde{z}_{n,j}^{(r)})}{t^2} \tag{396}$$

$$\Pr\left(\left(\langle \boldsymbol{w}_i, z_X^{(r)\backslash j}\rangle + \frac{1}{L}\langle \boldsymbol{w}_i, \boldsymbol{M}_j\rangle \tilde{z}_{n,j}^{(r)}\right)^2 \geq \frac{\lambda\|\boldsymbol{w}_i\|_2^2\sqrt{\log d}}{d}\right) \leq \frac{\widetilde{\mathcal{O}}\left(\frac{\|\boldsymbol{w}_i\|_2^2}{d}\right)}{\frac{\lambda\|\boldsymbol{w}_i\|_2^2\sqrt{\log d}}{d}}$$

$$\Pr\left(\left(\langle \boldsymbol{w}_i, z_X^{(r)\backslash j}\rangle + \frac{1}{L}\langle \boldsymbol{w}_i, \boldsymbol{M}_j\rangle \tilde{z}_{n,j}^{(r)}\right)^2 \geq \frac{\lambda\|\boldsymbol{w}_i\|_2^2\sqrt{\log d}}{d}\right) \leq O\left(\frac{1}{\lambda}\right).$$

As to the tail bounds for other variables, it suffices to go through some similar calculations. $\qquad\square$

### J.8 PROOF OF LEMMA E.7(B):

*Proof of Lemma E.7(b):*

$$\langle \boldsymbol{w}_i, \boldsymbol{M}\tilde{z}_n^{(r)}\rangle = \langle \boldsymbol{w}_i, \sum_{j=1}^d \boldsymbol{M}_j\tilde{z}_{n,j}^{(r)}\rangle = \sum_s^{d_1}(\boldsymbol{w}_i)_s(\sum_{j=1}^d \boldsymbol{M}_j\tilde{z}_{n,j}^{(r)})_s = \sum_s^{d_1}(\boldsymbol{w}_i)_s(\sum_{j=1}^d M_{j,s}\tilde{z}_{n,j}^{(r)}). \tag{397}$$

$z_{n,j}^{(i)}$ is a bounded random variable in the interval $[-1,1]$, so $z_{n,j}^{(i)}$ is sub-Gaussian variable with variance proxy 1, then $\tilde{z}_{n,j}^{(r)} = \sum_{i=1}^L \delta_{n,i}^{(r)} z_{n,j}^{(i)}$ is also is sub-Gaussian variable with variance proxy $\Delta_n^{(r)}$. $\sum_{j=1}^d M_{j,s}\tilde{z}_{n,j}^{(r)}$ is sub-Gaussian variable with variance proxy $\max_{j\in[d]}\|\boldsymbol{M}_j\|_\infty^2$, so $\langle \boldsymbol{w}_i, \boldsymbol{M}\tilde{z}_n^{(r)}\rangle$ is sub-Gaussian variable with variance proxy $\|\boldsymbol{w}_i\|_2^2 \cdot \max_{j\in[d]}\|\boldsymbol{M}_j\|_\infty^2$.

Sub-Gaussian Tail Bound

$$\Pr[|X - \mu| \geq t] \leq 2\exp\left(-\frac{t^2}{2\sigma^2}\right)$$

$$\Pr(\left|\langle \boldsymbol{w}_i, \boldsymbol{M}\tilde{z}_n^{(r)}\rangle\right| \geq t) \leq 2\exp\left(\frac{-t^2}{\|\boldsymbol{w}_i\|_2^2 \cdot \max_{j\in[d]}\|\boldsymbol{M}_j\|_\infty^2}\right)$$

$$\Pr(\left(\langle \boldsymbol{w}_i, \boldsymbol{M}\tilde{z}_n^{(r)}\rangle\right)^2 \geq t^2) \leq 2\exp\left(\frac{-t^2}{\|\boldsymbol{w}_i\|_2^2 \cdot \max_{j\in[d]}\|\boldsymbol{M}_j\|_\infty^2}\right) \tag{398}$$

$$\Pr\left(\left(\langle \boldsymbol{w}_i, \boldsymbol{M}\tilde{z}_n^{(r)}\rangle\right)^2 \geq \|\boldsymbol{w}_i\|_2^2 \cdot \max_{j\in[d]}\|\boldsymbol{M}_j\|_\infty^2 \log^4 d\right) \lesssim e^{-\Omega(\log^2 d)}.$$

$\qquad\square$

### J.9 PROOF OF LEMMA E.7(C):

*Proof of Lemma E.7(c):* $\xi_n^{(i)}$ ($\xi_{n,k}^{(i)}$) is a Gaussian random vector (variable), so $\sum_{i=1}^L \delta_{n,i}^{(r)}\xi_n^{(i)}$ ($\sum_{i=1}^L \delta_{n,i}^{(r)}\xi_{n,k}^{(i)}$) is a Gaussian random vector (variable), and therefore $\tilde{\xi}_n^{(r)}$ ($\tilde{\xi}_{n,k}^{(r)}$) is a Gaussian random vector (variable). $\tilde{\xi}_{n,k}^{(r)}$ is a sub-Gaussian random variable and each term $w_{i,k} \cdot \tilde{\xi}_{n,k}^{(r)}$ is a sub-Gaussian random variable, and its variance is: $\Delta_n^{(r)} \cdot w_{i,k}^2 \cdot \sigma_\xi^2$ ($\Delta_n^{(r)} = \sum_{i=1}^L (\delta_{n,i}^{(r)})^2$). Therefore $\sum_{k=1}^{d_1} w_{i,k} \cdot \tilde{\xi}_{n,k}^{(r)}$ is a sub-Gaussian random variable, and its variance is: $\sum_{k=1}^{d_1} \Delta_n^{(r)} \cdot w_{i,k}^2 \cdot \sigma_\xi^2 = \Delta_n^{(r)} \cdot \|\boldsymbol{w}_i\|_2^2 \cdot \sigma_\xi^2$.

Sub-Gaussian Tail Bound

$$\Pr[|X - \mu| \geq t] \leq 2\exp\left(-\frac{t^2}{2\sigma^2}\right)$$

$$\Pr\left(\left|\sum_{k=1}^{d_1} w_{i,k} \cdot \tilde{\xi}_{n,k}^{(r)}\right| \geq t\right) \leq 2\exp\left(\frac{-t^2}{2\Delta_n^{(r)}\|\boldsymbol{w}_i\|_2^2\sigma_\xi^2}\right)$$

$$\Pr\left(\left(\sum_{k=1}^{d_1} w_{i,k} \cdot \tilde{\xi}_{n,k}^{(r)}\right)^2 \geq t^2\right) \leq 2\exp\left(\frac{-t^2}{2\Delta_n^{(r)}\|\boldsymbol{w}_i\|_2^2\sigma_\xi^2}\right)$$

$$\Pr\left(\left(\sum_{k=1}^{d_1} w_{i,k} \cdot \tilde{\xi}_{n,k}^{(r)}\right)^2 \geq \frac{\|\boldsymbol{w}_i\|_2^2 \log^4 d}{d}\right) \lesssim e^{-\Omega(\log^2 d)}. \tag{399}$$

$\square$

*Proof of Lemma E.8:* The proof is similar to the proof of (Allen-Zhu & Li, 2022; Wen & Li, 2021).
$\square$

*Proof of Lemma E.9:* The proof is similar to the proof of (Allen-Zhu & Li, 2022; Wen & Li, 2021).
$\square$

## J.10 PROOF OF LEMMA E.10(A):

*Proof of Lemma E.10(a):* In the proof we will make the following simplification of notations: we drop the time superscript $^{(t)}$, and also the subscript for neuron index $i$. We start with the case when $0 < \alpha_j < b_i^{(t)}$ and rewrite the expectation as follows:

$$\mathbb{E}\left[h_i(\boldsymbol{Y}_n)\sum_{r=1}^{L}\mathbf{1}_{|\langle\boldsymbol{w}_i,\boldsymbol{z}_{\boldsymbol{X}}^{(r)}\rangle|\geq b_i}\tilde{z}_{n,j}^{(r)}\right]$$

$$=\mathbb{E}\left[\sum_{s=1}^{L}\left(\text{ReLU}\left(\langle\boldsymbol{w}_i,\boldsymbol{z}_{\boldsymbol{Y}}^{(s)}\rangle-b_i\right)-\text{ReLU}\left(-\langle\boldsymbol{w}_i,\boldsymbol{z}_{\boldsymbol{Y}}^{(s)}\rangle-b_i\right)\right)\sum_{r=1}^{L}\mathbf{1}_{|\langle\boldsymbol{w}_i,\boldsymbol{z}_{\boldsymbol{X}}^{(r)}\rangle|\geq b_i}\tilde{z}_{n,j}^{(r)}\right]$$

$$=\mathbb{E}\left[\sum_{s=1}^{L}\left(\left(\langle\boldsymbol{w}_i,\boldsymbol{z}_{\boldsymbol{Y}}^{(s)}\rangle-b_i\right)\mathbf{1}_{\langle\boldsymbol{w}_i,\boldsymbol{z}_{\boldsymbol{Y}}^{(s)}\rangle\geq b_i}-\left(-\langle\boldsymbol{w}_i,\boldsymbol{z}_{\boldsymbol{Y}}^{(s)}\rangle-b_i\right)\mathbf{1}_{-\langle\boldsymbol{w}_i,\boldsymbol{z}_{\boldsymbol{Y}}^{(s)}\rangle\geq b_i}\right)\sum_{r=1}^{L}\mathbf{1}_{|\langle\boldsymbol{w}_i,\boldsymbol{z}_{\boldsymbol{X}}^{(r)}\rangle|\geq b_i}\tilde{z}_{n,j}^{(r)}\right]$$

$$=\mathbb{E}\left[\sum_{s=1}^{L}\left(\langle\boldsymbol{w}_i,\boldsymbol{z}_{\boldsymbol{Y}}^{(s)}\rangle-b_i\right)\mathbf{1}_{\langle\boldsymbol{w}_i,\boldsymbol{z}_{\boldsymbol{Y}}^{(s)}\rangle\geq b_i}\sum_{r=1}^{L}\mathbf{1}_{|\langle\boldsymbol{w}_i,\boldsymbol{z}_{\boldsymbol{X}}^{(r)}\rangle|\geq b_i}\tilde{z}_{n,j}^{(r)}\right.$$
$$\left.-\sum_{s=1}^{L}\left(-\langle\boldsymbol{w}_i,\boldsymbol{z}_{\boldsymbol{Y}}^{(s)}\rangle-b_i\right)\mathbf{1}_{-\langle\boldsymbol{w}_i,\boldsymbol{z}_{\boldsymbol{Y}}^{(s)}\rangle\geq b_i}\sum_{r=1}^{L}\mathbf{1}_{|\langle\boldsymbol{w}_i,\boldsymbol{z}_{\boldsymbol{X}}^{(r)}\rangle|\geq b_i}\tilde{z}_{n,j}^{(r)}\right]$$

$$=\mathbb{E}\left[\sum_{s=1}^{L}\langle\boldsymbol{w}_i,\boldsymbol{z}_{\boldsymbol{Y}}^{(s)}\rangle\mathbf{1}_{\langle\boldsymbol{w}_i,\boldsymbol{z}_{\boldsymbol{Y}}^{(s)}\rangle\geq b_i}\sum_{r=1}^{L}\mathbf{1}_{|\langle\boldsymbol{w}_i,\boldsymbol{z}_{\boldsymbol{X}}^{(r)}\rangle|\geq b_i}\tilde{z}_{n,j}^{(r)}-b_i\sum_{s=1}^{L}\mathbf{1}_{\langle\boldsymbol{w}_i,\boldsymbol{z}_{\boldsymbol{Y}}^{(s)}\rangle\geq b_i}\sum_{r=1}^{L}\mathbf{1}_{|\langle\boldsymbol{w}_i,\boldsymbol{z}_{\boldsymbol{X}}^{(r)}\rangle|\geq b_i}\tilde{z}_{n,j}^{(r)}\right.$$
$$\left.+\sum_{s=1}^{L}\langle\boldsymbol{w}_i,\boldsymbol{z}_{\boldsymbol{Y}}^{(s)}\rangle\mathbf{1}_{-\langle\boldsymbol{w}_i,\boldsymbol{z}_{\boldsymbol{Y}}^{(s)}\rangle\geq b_i}\sum_{r=1}^{L}\mathbf{1}_{|\langle\boldsymbol{w}_i,\boldsymbol{z}_{\boldsymbol{X}}^{(r)}\rangle|\geq b_i}\tilde{z}_{n,j}^{(r)}+b_i\sum_{s=1}^{L}\mathbf{1}_{-\langle\boldsymbol{w}_i,\boldsymbol{z}_{\boldsymbol{Y}}^{(s)}\rangle\geq b_i}\sum_{r=1}^{L}\mathbf{1}_{|\langle\boldsymbol{w}_i,\boldsymbol{z}_{\boldsymbol{X}}^{(r)}\rangle|\geq b_i}\tilde{z}_{n,j}^{(r)}\right]$$

$$=\mathbb{E}\left[\sum_{s=1}^{L}\langle\boldsymbol{w}_i,\boldsymbol{z}_{\boldsymbol{Y}}^{(s)}\rangle\mathbf{1}_{|\langle\boldsymbol{w}_i,\boldsymbol{z}_{\boldsymbol{Y}}^{(s)}\rangle|\geq b_i}\sum_{r=1}^{L}\mathbf{1}_{|\langle\boldsymbol{w}_i,\boldsymbol{z}_{\boldsymbol{X}}^{(r)}\rangle|\geq b_i}\tilde{z}_{n,j}^{(r)}\right]$$
$$-\mathbb{E}\left[b_i\sum_{s=1}^{L}\left(\mathbf{1}_{\langle\boldsymbol{w}_i,\boldsymbol{z}_{\boldsymbol{Y}}^{(s)}\rangle\geq b_i}-\mathbf{1}_{-\langle\boldsymbol{w}_i,\boldsymbol{z}_{\boldsymbol{Y}}^{(s)}\rangle\geq b_i}\right)\sum_{r=1}^{L}\mathbf{1}_{|\langle\boldsymbol{w}_i,\boldsymbol{z}_{\boldsymbol{X}}^{(r)}\rangle|\geq b_i}\tilde{z}_{n,j}^{(r)}\right] \tag{400}$$

Thus the expectation can be expanded as:

$$
\begin{aligned}
&\mathbb{E}\left[h_i(\boldsymbol{Y}_n)\sum_{r=1}^{L}\mathbf{1}_{|\langle \boldsymbol{w}_i, \boldsymbol{z}_{\boldsymbol{X}}^{(r)}\rangle|\geq b_i}\tilde{z}_{n,j}^{(r)}\right]\\
=&\mathbb{E}\left[\sum_{s=1}^{L}\langle \boldsymbol{w}_i, z_Y^{(s)}\rangle\mathbf{1}_{|\langle \boldsymbol{w}_i, \boldsymbol{z}_{\boldsymbol{Y}}^{(s)}\rangle|\geq b_i}\sum_{r=1}^{L}\mathbf{1}_{|\langle \boldsymbol{w}_i, \boldsymbol{z}_{\boldsymbol{X}}^{(r)}\rangle|\geq b_i}\tilde{z}_{n,j}^{(r)}\right]\\
&-\mathbb{E}\left[b_i\sum_{s=1}^{L}\left(\mathbf{1}_{\langle \boldsymbol{w}_i, \boldsymbol{z}_{\boldsymbol{Y}}^{(s)}\rangle\geq b_i}-\mathbf{1}_{-\langle \boldsymbol{w}_i, \boldsymbol{z}_{\boldsymbol{Y}}^{(s)}\rangle\geq b_i}\right)\sum_{r=1}^{L}\mathbf{1}_{|\langle \boldsymbol{w}_i, \boldsymbol{z}_{\boldsymbol{X}}^{(r)}\rangle|\geq b_i}\tilde{z}_{n,j}^{(r)}\right]\\
=&\frac{1}{L}\mathbb{E}\left[\alpha_j\sum_{s=1}^{L}\tilde{z}_{n,j}^{+(s)}\sum_{r=1}^{L}\tilde{z}_{n,j}^{(r)}\mathbf{1}_{|\langle \boldsymbol{w}_i, \frac{z_X^{(r)}+z_Y^{(s)}}{2}\rangle|\geq b_i+|\langle \boldsymbol{w}_i, z_X^{(r)}-\frac{z_X^{(r)}+z_Y^{(s)}}{2}\rangle|}\right]\\
&+\mathbb{E}\left[\sum_{s=1}^{L}\left(S^{(s)\backslash j}-b_i\right)\mathbf{1}_{\langle \boldsymbol{w}_i, \boldsymbol{z}_{\boldsymbol{Y}}^{(s)}\rangle\geq b_i}\sum_{r=1}^{L}\mathbf{1}_{\langle \boldsymbol{w}_i, \boldsymbol{z}_{\boldsymbol{X}}^{(r)}\rangle\geq b_i}\tilde{z}_{n,j}^{(r)}\right]\\
&+\mathbb{E}\left[\sum_{s=1}^{L}\left(S^{(s)\backslash j}+b_i\right)\mathbf{1}_{\langle \boldsymbol{w}_i, \boldsymbol{z}_{\boldsymbol{Y}}^{(s)}\rangle\leq -b_i}\sum_{r=1}^{L}\mathbf{1}_{\langle \boldsymbol{w}_i, \boldsymbol{z}_{\boldsymbol{X}}^{(r)}\rangle\leq -b_i}\tilde{z}_{n,j}^{(r)}\right]\\
&+\mathbb{E}\left[\sum_{s=1}^{L}\left(\frac{1}{L}\alpha_j\tilde{z}_{n,j}^{+(s)}+S^{(s)\backslash j}-b_i\right)\mathbf{1}_{\langle \boldsymbol{w}_i, \boldsymbol{z}_{\boldsymbol{Y}}^{(s)}\rangle\geq b_i}\sum_{r=1}^{L}\mathbf{1}_{\langle \boldsymbol{w}_i, \boldsymbol{z}_{\boldsymbol{X}}^{(r)}\rangle\leq -b_i}\tilde{z}_{n,j}^{(r)}\right]\\
&+\mathbb{E}\left[\sum_{s=1}^{L}\left(\frac{1}{L}\alpha_j\tilde{z}_{n,j}^{+(s)}+S^{(s)\backslash j}+b_i\right)\mathbf{1}_{\langle \boldsymbol{w}_i, \boldsymbol{z}_{\boldsymbol{Y}}^{(s)}\rangle\leq -b_i}\sum_{r=1}^{L}\mathbf{1}_{\langle \boldsymbol{w}_i, \boldsymbol{z}_{\boldsymbol{X}}^{(r)}\rangle\geq b_i}\tilde{z}_{n,j}^{(r)}\right]\\
=&J_1+J_2+J_3
\end{aligned}
\tag{401}
$$

Now we need to obtain absolute bounds for both $J_2$ and $J_3$. We start with $J_2$, where

$$
\begin{aligned}
J_2=&\mathbb{E}\left[\sum_{s=1}^{L}\left(S^{(s)\backslash j}-b_i\right)\mathbf{1}_{\langle \boldsymbol{w}_i, \boldsymbol{z}_{\boldsymbol{Y}}^{(s)}\rangle\geq b_i}\sum_{r=1}^{L}\mathbf{1}_{\langle \boldsymbol{w}_i, \boldsymbol{z}_{\boldsymbol{X}}^{(r)}\rangle\geq b_i}\tilde{z}_{n,j}^{(r)}\right]\\
&+\mathbb{E}\left[\sum_{s=1}^{L}\left(S^{(s)\backslash j}+b_i\right)\mathbf{1}_{\langle \boldsymbol{w}_i, \boldsymbol{z}_{\boldsymbol{Y}}^{(s)}\rangle\leq -b_i}\sum_{r=1}^{L}\mathbf{1}_{\langle \boldsymbol{w}_i, \boldsymbol{z}_{\boldsymbol{X}}^{(r)}\rangle\leq -b_i}\tilde{z}_{n,j}^{(r)}\right]\\
=&\mathbb{E}\left[\sum_{s=1}^{L}\left(S^{(s)\backslash j}-b_i\right)\sum_{r=1}^{L}\mathbf{1}_{\langle \boldsymbol{w}_i, \frac{z_X^{(r)}+z_Y^{(s)}}{2}\rangle\geq b_i+|\langle \boldsymbol{w}_i, z_X^{(r)}-\frac{z_X^{(r)}+z_Y^{(s)}}{2}\rangle|}\tilde{z}_{n,j}^{(r)}\right]\\
&+\mathbb{E}\left[\sum_{s=1}^{L}\left(S^{(s)\backslash j}+b_i\right)\sum_{r=1}^{L}\mathbf{1}_{\langle \boldsymbol{w}_i, \frac{z_X^{(r)}+z_Y^{(s)}}{2}\rangle\leq -b_i-|\langle \boldsymbol{w}_i, z_X^{(r)}-\frac{z_X^{(r)}+z_Y^{(s)}}{2}\rangle|}\tilde{z}_{n,j}^{(r)}\right]
\end{aligned}
\tag{402}
$$

We proceed with the first term

$$
\mathbb{E}\left[\sum_{s=1}^{L}\left(S^{(s)\backslash j}-b_i\right)\sum_{r=1}^{L}\mathbf{1}_{\langle \boldsymbol{w}_i, \frac{z_X^{(r)}+z_Y^{(s)}}{2}\rangle\geq b_i+|\langle \boldsymbol{w}_i, z_X^{(r)}-\frac{z_X^{(r)}+z_Y^{(s)}}{2}\rangle|}\tilde{z}_{n,j}^{(r)}\right]
\tag{403}
$$

First, from a trivial calculation conditioned on the randomness of $\tilde{z}_{n,j}^{(r)}$, we have:

$$
\begin{aligned}
&\mathbb{E}\left[\sum_{s=1}^{L}\left(S^{(s)\backslash j}-b_i\right)\sum_{r=1}^{L}\mathbf{1}_{\langle\boldsymbol{w}_i,\frac{z_X^{(r)}+z_Y^{(s)}}{2}\rangle\geq b_i+|\langle\boldsymbol{w}_i,\boldsymbol{z}_X^{(r)}-\frac{z_X^{(r)}+z_Y^{(s)}}{2}\rangle|}\,\tilde{z}_{n,j}^{(r)}\right]\\
=&\mathbb{E}\left[\sum_{s=1}^{L}\left(S^{(s)\backslash j}-b_i\right)\sum_{r=1}^{L}\mathbf{1}_{\langle\boldsymbol{w}_i,\frac{z_X^{(r)}+z_Y^{(s)}}{2}\rangle\geq b_i+|\bar{S}^{(r,s)\backslash j}+\bar{\alpha}_j^{(r,s)}\frac{\tilde{z}_{n,j}^{(r)}+\tilde{z}_{n,j}^{+(s)}}{2L}|}\,\tilde{z}_{n,j}^{(r)}\right]\\
=&\mathbb{E}\left[\sum_{s=1}^{L}\left(S^{(s)\backslash j}-b_i\right)\sum_{r=1}^{L}\mathbf{1}_{S^{(r,s)\backslash j}\geq b_i-\alpha_j\frac{\tilde{z}_{n,j}^{(r)}+\tilde{z}_{n,j}^{+(s)}}{2L}+|\bar{S}^{(r,s)\backslash j}+\bar{\alpha}_j^{(r,s)}\frac{\tilde{z}_{n,j}^{(r)}+\tilde{z}_{n,j}^{+(s)}}{2L}|}\,\tilde{z}_{n,j}^{(r)}\right]\\
=&\mathbb{E}\left[\sum_{s=1}^{L}\left(S^{(s)\backslash j}-b_i\right)\sum_{r=1}^{L}|\tilde{z}_{n,j}^{(r)}|\left(\mathbf{1}_{S^{(r,s)\backslash j}\geq b_i-\alpha_j\frac{\tilde{z}_{n,j}^{(r)}+\tilde{z}_{n,j}^{+(s)}}{2L}+\left|\bar{S}^{(r,s)\backslash j}+\bar{\alpha}_j^{(r,s)}\frac{\tilde{z}_{n,j}^{(r)}+\tilde{z}_{n,j}^{+(s)}}{2L}\right|}\right.\right.\\
&\qquad\left.\left.-\mathbf{1}_{S^{(r,s)\backslash j}\geq b_i+\alpha_j\frac{\tilde{z}_{n,j}^{(r)}+\tilde{z}_{n,j}^{+(s)}}{2L}+\left|\bar{S}^{(r,s)\backslash j}-\bar{\alpha}_j^{(r,s)}\frac{\tilde{z}_{n,j}^{(r)}+\tilde{z}_{n,j}^{+(s)}}{2L}\right|}\right)\right].
\end{aligned}
\tag{404}
$$

Now define:

$$
\begin{aligned}
Z &= \frac{1}{2}\left(\left|\bar{S}^{(r,s)\backslash j}+\bar{\alpha}_j^{(r,s)}\frac{\tilde{z}_{n,j}^{(r)}+\tilde{z}_{n,j}^{+(s)}}{2L}\right|+\left|\bar{S}^{(r,s)\backslash j}-\bar{\alpha}_j^{(r,s)}\frac{\tilde{z}_{n,j}^{(r)}+\tilde{z}_{n,j}^{+(s)}}{2L}\right|\right),\\
Z' &= \frac{1}{2}\left(\left|\bar{S}^{(r,s)\backslash j}+\bar{\alpha}_j^{(r,s)}\frac{\tilde{z}_{n,j}^{(r)}+\tilde{z}_{n,j}^{+(s)}}{2L}\right|-\left|\bar{S}^{(r,s)\backslash j}-\bar{\alpha}_j^{(r,s)}\frac{\tilde{z}_{n,j}^{(r)}+\tilde{z}_{n,j}^{+(s)}}{2L}\right|\right).
\end{aligned}
\tag{405}
$$

In this case, we always have $|Z'|\leq|\bar{\alpha}_j^{(r,s)}||\frac{\tilde{z}_{n,j}^{(r)}+\tilde{z}_{n,j}^{+(s)}}{2L}|$, and

$$
\begin{aligned}
&\left|\mathbf{1}_{S^{(r,s)\backslash j}\geq b_i-\alpha_j\frac{\tilde{z}_{n,j}^{(r)}+\tilde{z}_{n,j}^{+(s)}}{2L}+\left|\bar{S}^{(r,s)\backslash j}+\bar{\alpha}_j^{(r,s)}\frac{\tilde{z}_{n,j}^{(r)}+\tilde{z}_{n,j}^{+(s)}}{2L}\right|}\right.\\
&\qquad\left.-\mathbf{1}_{S^{(r,s)\backslash j}\geq b_i+\alpha_j\frac{\tilde{z}_{n,j}^{(r)}+\tilde{z}_{n,j}^{+(s)}}{2L}+\left|\bar{S}^{(r,s)\backslash j}-\bar{\alpha}_j^{(r,s)}\frac{\tilde{z}_{n,j}^{(r)}+\tilde{z}_{n,j}^{+(s)}}{2L}\right|}\right|\\
&=\mathbf{1}_{S^{(r,s)\backslash j}-b_i-Z\in[-|\alpha_j\frac{\tilde{z}_{n,j}^{(r)}+\tilde{z}_{n,j}^{+(s)}}{2L}-Z'|,|\alpha_j\frac{\tilde{z}_{n,j}^{(r)}+\tilde{z}_{n,j}^{+(s)}}{2L}-Z'|]}
\end{aligned}
\tag{406}
$$

Equality Proof: The original two threshold values are:

$$
\begin{aligned}
T_1 &= b_i-\alpha_j\frac{\tilde{z}_{n,j}^{(r)}+\tilde{z}_{n,j}^{+(s)}}{2L}+\left|\bar{S}^{(r,s)\backslash j}+\bar{\alpha}_j^{(r,s)}\frac{\tilde{z}_{n,j}^{(r)}+\tilde{z}_{n,j}^{+(s)}}{2L}\right|\\
T_2 &= b_i+\alpha_j\frac{\tilde{z}_{n,j}^{(r)}+\tilde{z}_{n,j}^{+(s)}}{2L}+\left|\bar{S}^{(r,s)\backslash j}-\bar{\alpha}_j^{(r,s)}\frac{\tilde{z}_{n,j}^{(r)}+\tilde{z}_{n,j}^{+(s)}}{2L}\right|
\end{aligned}
\tag{407}
$$

We can rewrite these two terms as:

$$
\begin{aligned}
T_1 &= b_i+\left(-\alpha_j\frac{\tilde{z}_{n,j}^{(r)}+\tilde{z}_{n,j}^{+(s)}}{2L}+\left|\bar{S}^{(r,s)\backslash j}+\bar{\alpha}_j^{(r,s)}\frac{\tilde{z}_{n,j}^{(r)}+\tilde{z}_{n,j}^{+(s)}}{2L}\right|\right)\\
T_2 &= b_i+\left(\alpha_j\frac{\tilde{z}_{n,j}^{(r)}+\tilde{z}_{n,j}^{+(s)}}{2L}+\left|\bar{S}^{(r,s)\backslash j}-\bar{\alpha}_j^{(r,s)}\frac{\tilde{z}_{n,j}^{(r)}+\tilde{z}_{n,j}^{+(s)}}{2L}\right|\right)
\end{aligned}
\tag{408}
$$

Then the midpoint between these two terms is:

$$
\text{Midpoint} = b_i+Z
\tag{409}
$$

The distance between them is:

$$
\begin{aligned}
& T_2 - T_1 \\
=& 2\alpha_j \frac{\tilde{z}_{n,j}^{(r)} + \tilde{z}_{n,j}^{+(s)}}{2L} + \left( \left| \bar{S}^{(r,s)\backslash j} - \bar{\alpha}_j^{(r,s)} \frac{\tilde{z}_{n,j}^{(r)} + \tilde{z}_{n,j}^{+(s)}}{2L} \right| - \left| \bar{S}^{(r,s)\backslash j} + \bar{\alpha}_j^{(r,s)} \frac{\tilde{z}_{n,j}^{(r)} + \tilde{z}_{n,j}^{+(s)}}{2L} \right| \right) \\
=& 2\left( \alpha_j \frac{\tilde{z}_{n,j}^{(r)} + \tilde{z}_{n,j}^{+(s)}}{2L} - Z' \right)
\end{aligned}
$$
(410)

Therefore, shifting from the midpoint $b_i + Z$ by $\left| \alpha_j \frac{\tilde{z}_{n,j}^{(r)} + \tilde{z}_{n,j}^{+(s)}}{2L} - Z' \right|$ in both directions will exactly span the distance between $T_1$ and $T_2$.

The indicator function differs by 1 if and only if:

$$
T_1 \le S^{(r,s)\backslash j} < T_2
$$
(411)

Therefore we have:

$$
\begin{aligned}
& \left| \mathbf{1}_{S^{(r,s)\backslash j} \ge b_i - \alpha_j \frac{\tilde{z}_{n,j}^{(r)} + \tilde{z}_{n,j}^{+(s)}}{2L} + \left| \bar{S}^{(r,s)\backslash j} + \bar{\alpha}_j^{(r,s)} \frac{\tilde{z}_{n,j}^{(r)} + \tilde{z}_{n,j}^{+(s)}}{2L} \right|} \right. \\
& \left. \qquad - \mathbf{1}_{S^{(r,s)\backslash j} \ge b_i + \alpha_j \frac{\tilde{z}_{n,j}^{(r)} + \tilde{z}_{n,j}^{+(s)}}{2L} + \left| \bar{S}^{(r,s)\backslash j} - \bar{\alpha}_j^{(r,s)} \frac{\tilde{z}_{n,j}^{(r)} + \tilde{z}_{n,j}^{+(s)}}{2L} \right|} \right| \\
=& \left| \mathbf{1}_{S^{(r,s)\backslash j} \ge T_1} - \mathbf{1}_{S^{(r,s)\backslash j} \ge T_2} \right| \\
=& \mathbf{1}_{S^{(r,s)\backslash j} - \frac{T_1 + T_2}{2} \in [-\frac{T_2 - T_1}{2}, \frac{T_2 - T_1}{2}]} \\
=& \mathbf{1}_{S^{(r,s)\backslash j} - b_i - Z \in [-|\alpha_j \frac{\tilde{z}_{n,j}^{(r)} + \tilde{z}_{n,j}^{+(s)}}{2L} - Z'|, |\alpha_j \frac{\tilde{z}_{n,j}^{(r)} + \tilde{z}_{n,j}^{+(s)}}{2L} - Z'|]}
\end{aligned}
$$
(412)

Inequality Proof: Let $a = \bar{S}^{(r,s)\backslash j}$, $b = \bar{\alpha}_j^{(r,s)} \frac{\tilde{z}_{n,j}^{(r)} + \tilde{z}_{n,j}^{+(s)}}{2L}$.

We know:$||a + b| - |a - b|| \le 2|b|$ and $||a + b| + |a - b|| \le 2(|a| + |b|)$

Applying this, we have:

$$
\begin{aligned}
|Z'| &= \frac{1}{2} \left| \, |a + b| - |a - b| \, \right| \le \frac{1}{2} \cdot 2|b| = |b| = \left| \bar{\alpha}_j^{(r,s)} \frac{\tilde{z}_{n,j}^{(r)} + \tilde{z}_{n,j}^{+(s)}}{2L} \right| \\
|Z| &= \frac{1}{2} \left| \, |a + b| + |a - b| \, \right| \le \frac{1}{2} \cdot 2 \cdot (|a| + |b|) = |\bar{S}^{(r,s)\backslash j}| + \left| \bar{\alpha}_j^{(r,s)} \frac{\tilde{z}_{n,j}^{(r)} + \tilde{z}_{n,j}^{+(s)}}{2L} \right|
\end{aligned}
$$
(413)

Which allows us to proceed as follows:

$$
\left| \mathbb{E}\left[ \sum_{s=1}^{L} \left( S^{(s)\backslash j} - b_i \right) \sum_{r=1}^{L} \mathbf{1}_{\langle \boldsymbol{w}_i, \frac{z_X^{(r)}+z_Y^{(s)}}{2}\rangle \geq b_i + |\langle \boldsymbol{w}_i, \boldsymbol{z}_X^{(r)} - \frac{z_X^{(r)}+z_Y^{(s)}}{2}\rangle|} \tilde{z}_{n,j}^{(r)} \right] \right|
$$

$$
= \left| \mathbb{E}\left[ \sum_{s=1}^{L} (S^{(s)\backslash j} - b_i - Z + Z) \sum_{r=1}^{L} |\tilde{z}_{n,j}^{(r)}| \left( \mathbf{1}_{\mathcal{S}^{(r,s)\backslash j} \geq b_i - \alpha_j \frac{\tilde{z}_{n,j}^{(r)}+\tilde{z}_{n,j}^{+(s)}}{2L} + |\bar{\mathcal{S}}^{(r,s)\backslash j} + \bar{\alpha}_j^{(r,s)} \frac{\tilde{z}_{n,j}^{(r)}+\tilde{z}_{n,j}^{+(s)}}{2L}|} \right. \right. \right.
$$

$$
\left. \left. \left. - \mathbf{1}_{\mathcal{S}^{(r,s)\backslash j} \geq b_i + \alpha_j \frac{\tilde{z}_{n,j}^{(r)}+\tilde{z}_{n,j}^{+(s)}}{2L} + |\bar{\mathcal{S}}^{(r,s)\backslash j} - \bar{\alpha}_j^{(r,s)} \frac{\tilde{z}_{n,j}^{(r)}+\tilde{z}_{n,j}^{+(s)}}{2L}|} \right) \mathbf{1}_{S^{\backslash j} \geq b_i - \alpha_j \frac{\tilde{z}_{n,j}^{(r)}+\tilde{z}_{n,j}^{+(s)}}{2L}} \right] \right|
$$

$$
\leq \frac{1}{L} \mathbb{E}\left[ \sum_{s=1}^{L} \sum_{r=1}^{L} (\alpha_j + 2|\bar{\alpha}_j^{(r,s)}|) |\frac{\tilde{z}_{n,j}^{(r)} + \tilde{z}_{n,j}^{+(s)}}{2}| |\tilde{z}_{n,j}^{(r)}| \mathbf{1}_{S^{(r,s)\backslash j} \geq b_i - \alpha_j \frac{\tilde{z}_{n,j}^{(r)}+\tilde{z}_{n,j}^{+(s)}}{2L}} \right]
$$

$$
+ \mathbb{E}\left[ \sum_{s=1}^{L} \sum_{r=1}^{L} |\bar{S}^{(r,s)\backslash j}| |\tilde{z}_{n,j}^{(r)}| \mathbf{1}_{S^{(r,s)\backslash j} - b_i - Z \in [-|\alpha_j \frac{\tilde{z}_{n,j}^{(r)}+\tilde{z}_{n,j}^{+(s)}}{2L} - Z'|, |\alpha_j \frac{\tilde{z}_{n,j}^{(r)}+\tilde{z}_{n,j}^{+(s)}}{2L} - Z'|]} \mathbf{1}_{S^{(r,s)\backslash j} \geq b_i - \alpha_j \frac{\tilde{z}_{n,j}^{(r)}+\tilde{z}_{n,j}^{+(s)}}{2L}} \right]
$$

$$
= \frac{1}{L} \mathbb{E}\left[ \sum_{s=1}^{L} \sum_{r=1}^{L} (\alpha_j + 2|\bar{\alpha}_j^{(r,s)}|) |\frac{\tilde{z}_{n,j}^{(r)} + \tilde{z}_{n,j}^{+(s)}}{2}| |\tilde{z}_{n,j}^{(r)}| \mathbf{1}_{S^{(r,s)\backslash j} \geq b_i - \alpha_j \frac{\tilde{z}_{n,j}^{(r)}+\tilde{z}_{n,j}^{+(s)}}{2L}} \right]
$$

$$
+ \sum_{s=1}^{L} \sum_{r=1}^{L} \sqrt{\mathbb{E}\left[ |\bar{S}^{(r,s)\backslash j}|^2 (\tilde{z}_{n,j}^{(r)})^2 \mathbf{1}_{S^{(r,s)\backslash j} \geq b_i - \alpha_j \frac{\tilde{z}_{n,j}^{(r)}+\tilde{z}_{n,j}^{+(s)}}{2L}} \right]} \cdot \sqrt{\mathbb{E}\left[ \mathbf{1}_{S^{(r,s)\backslash j} - b_i - Z \in [-|\alpha_j \frac{\tilde{z}_{n,j}^{(r)}+\tilde{z}_{n,j}^{+(s)}}{2L} - Z'|, |\alpha_j \frac{\tilde{z}_{n,j}^{(r)}+\tilde{z}_{n,j}^{+(s)}}{2L} - Z'|]} \right]}
$$

$$
\overset{\text{\textcircled{1}}}{\leq} \frac{1}{L} \mathbb{E}\left[ \sum_{s=1}^{L} \sum_{r=1}^{L} (\alpha_j + 2|\bar{\alpha}_j^{(r,s)}|) |\frac{\tilde{z}_{n,j}^{(r)} + \tilde{z}_{n,j}^{+(s)}}{2}| |\tilde{z}_{n,j}^{(r)}| \mathbf{1}_{S^{(r,s)\backslash j} \geq b_i - \alpha_j \frac{\tilde{z}_{n,j}^{(r)}+\tilde{z}_{n,j}^{+(s)}}{2L}} \right]
$$

$$
+ \sum_{s=1}^{L} \sum_{r=1}^{L} \sqrt{\mathbb{E}\left[ |\bar{S}^{(r,s)\backslash j}|^2 (\tilde{z}_{n,j}^{(r)})^2 \mathbf{1}_{S^{(r,s)\backslash j} \geq b_i - \alpha_j \frac{\tilde{z}_{n,j}^{(r)}+\tilde{z}_{n,j}^{+(s)}}{2L}} \right]} \cdot \sqrt{\frac{\mathbb{E}\left[ (\alpha_j + |\bar{\alpha}_j^{(r,s)}|)^2 (\frac{\tilde{z}_{n,j}^{(r)}+\tilde{z}_{n,j}^{+(s)}}{2L})^2 \right]}{\mathbb{E}\left[ \langle \boldsymbol{w}_i, \frac{\tilde{\xi}_n^{(r)}+\tilde{\xi}_n^{+(s)}}{2L}\rangle^2 \right]}}
$$

$$
= O(1) \left( \alpha_j + O\left( \mathbb{E}[|\bar{\alpha}_j|^2]^{1/2} \right) \right) \mathbb{E}\left[ \sum_{s=1}^{L} \sum_{r=1}^{L} |\frac{\tilde{z}_{n,j}^{(r)} + \tilde{z}_{n,j}^{+(s)}}{2}| |\tilde{z}_{n,j}^{(r)}| \right] \left( \Pr(S^{\backslash j} \geq b_i - \alpha_j C_{\tilde{z}}) + \sqrt{\frac{\mathbb{E}[|\bar{S}^{\backslash j}|^2 \mathbf{1}_{S^{\backslash j} \geq b_i - \alpha_j C_{\tilde{z}}}]}{\mathbb{E}[\langle \boldsymbol{w}_i, \frac{\tilde{\xi}_n+\tilde{\xi}_n^{+}}{2L}\rangle^2]}} \right)
$$

$$
(414)
$$

where in ① we have used the randomness of $\frac{\tilde{\xi}_n^{(r)}+\tilde{\xi}_n^{+(s)}}{2L}$ in the following manner: Fixing the randomness of $\tilde{z}_{n,j}$, we have $S^{(r,s)\backslash j} - Z$ is a random variable depending solely on the randomness of $\frac{\tilde{\xi}_n^{(r)}+\tilde{\xi}_n^{+(s)}}{2L}$, and thus we have:

$$
\mathbb{E}\left[\mathbf{1}_{S^{(r,s)\backslash j}-b_i-Z\in[-|\alpha_j\frac{\tilde{z}_{n,j}^{(r)}+\tilde{z}_{n,j}^{+(s)}}{2L}-Z'|,|\alpha_j\frac{\tilde{z}_{n,j}^{(r)}+\tilde{z}_{n,j}^{+(s)}}{2L}-Z'|]}\right]
$$

$$
\leq \mathbb{E}\left[\mathbf{1}_{\langle \boldsymbol{w}_i,\frac{\tilde{\xi}_n^{(r)}+\tilde{\xi}_n^{+(s)}}{2L}\rangle-|\langle \boldsymbol{w}_i,\tilde{\xi}_n^{+(s)}\rangle|\in[|\bar{\alpha}_j^{(r,s)}|-(\alpha_j+|\bar{\alpha}_j^{(r,s)}|),\alpha_j+2|\bar{\alpha}_j^{(r,s)}|]}\right]
$$

$$
= \mathbb{E}\left[\mathbf{1}_{\langle \boldsymbol{w}_i,\frac{\tilde{\xi}_n^{(r)}}{2L}\rangle+\langle \boldsymbol{w}_i,\frac{\tilde{\xi}_n^{+(s)}}{2L}\rangle-|\langle \boldsymbol{w}_i,\frac{\tilde{\xi}_n^{+(s)}}{L}\rangle|\in[-O(\alpha_j+|\bar{\alpha}_j^{(r,s)}|),O(\alpha_j+|\bar{\alpha}_j^{(r,s)}|)]}\right]
$$

$$
\leq \mathbb{E}\left[\frac{O((\alpha_j+|\bar{\alpha}_j^{(r,s)}|)^2(\frac{\tilde{z}_{n,j}^{(r)}+\tilde{z}_{n,j}^{+(s)}}{2L})^2)}{\langle \boldsymbol{w}_i,\frac{\tilde{\xi}_n^{(r)}+\tilde{\xi}_n^{+(s)}}{2L}\rangle^2}\right] = \frac{\mathbb{E}[O((\alpha_j+|\bar{\alpha}_j^{(r,s)}|)^2(\frac{\tilde{z}_{n,j}^{(r)}+\tilde{z}_{n,j}^{+(s)}}{2L})^2))]}{\mathbb{E}[\langle \boldsymbol{w}_i,\frac{\tilde{\xi}_n^{(r)}+\tilde{\xi}_n^{+(s)}}{2L}\rangle^2]}
\tag{415}
$$

Simultaneously, from similar analysis as above, we have for the second term in $J_2$:

$$
\left|\mathbb{E}\left[\sum_{s=1}^{L}\sum_{r=1}^{L}\left(S^{(s)\backslash j}+b_i\right)\mathbf{1}_{\langle \boldsymbol{w}_i,\frac{z_{\boldsymbol{X}}^{(r)}+z_{\boldsymbol{Y}}^{(s)}}{2}\rangle\leq -b_i-|\langle \boldsymbol{w}_i,\boldsymbol{z}_{\boldsymbol{X}}^{(r)}-\frac{z_{\boldsymbol{X}}^{(r)}+z_{\boldsymbol{Y}}^{(s)}}{2}\rangle|}\tilde{z}_{n,j}^{(r)}\right]\right|
$$

$$
= \left|\mathbb{E}\left[\sum_{s=1}^{L}\sum_{r=1}^{L}\left(S^{(s)\backslash j}+b_i\right)\mathbf{1}_{\langle \boldsymbol{w}_i,\frac{z_{\boldsymbol{X}}^{(r)}+z_{\boldsymbol{Y}}^{(s)}}{2}\rangle\leq -b_i-|\bar{S}^{(r,s)\backslash j}+\bar{\alpha}_j\frac{z_{\boldsymbol{X}}^{(r)}+z_{\boldsymbol{Y}}^{(s)}}{2}|}\tilde{z}_{n,j}^{(r)}\right]\right|
\tag{416}
$$

$$
\leq O\left(\alpha_j+\mathbb{E}[|\bar{\alpha}_j|^2]^{1/2}\right)\mathbb{E}\left[\sum_{s=1}^{L}\sum_{r=1}^{L}|\frac{\tilde{z}_{n,j}^{(r)}+\tilde{z}_{n,j}^{+(s)}}{2}||\tilde{z}_{n,j}^{(r)}|\right]\sqrt{\frac{\mathbb{E}[|\bar{S}^{\backslash j}|^2\mathbf{1}_{S^{\backslash j}\geq b_i-\alpha_j}]}{\mathbb{E}[\langle \boldsymbol{w}_i,\frac{\tilde{\xi}_n+\tilde{\xi}_n^{+}}{2L}\rangle^2]}}
$$

Now we turn to $J_3$

$$
J_3 = \mathbb{E}\left[\sum_{s=1}^{L}\sum_{r=1}^{L}\left(\frac{1}{L}\alpha_j\tilde{z}_{n,j}^{+(s)}+S^{(s)\backslash j}-b_i\right)\mathbf{1}_{\langle \boldsymbol{w}_i,\boldsymbol{z}_{\boldsymbol{Y}}^{(s)}\rangle\geq b_i}\mathbf{1}_{\langle \boldsymbol{w}_i,\boldsymbol{z}_{\boldsymbol{X}}^{(r)}\rangle\leq -b_i}\tilde{z}_{n,j}^{(r)}\right]
$$

$$
+ \mathbb{E}\left[\sum_{s=1}^{L}\sum_{r=1}^{L}\left(\frac{1}{L}\alpha_j\tilde{z}_{n,j}^{+(s)}+S^{(s)\backslash j}+b_i\right)\mathbf{1}_{\langle \boldsymbol{w}_i,\boldsymbol{z}_{\boldsymbol{Y}}^{(s)}\rangle\leq -b_i}\mathbf{1}_{\langle \boldsymbol{w}_i,\boldsymbol{z}_{\boldsymbol{X}}^{(r)}\rangle\geq b_i}\tilde{z}_{n,j}^{(r)}\right]
$$

$$
\overset{\text{①}}{=} \mathbb{E}\left[\sum_{s=1}^{L}\sum_{r=1}^{L}\left(\frac{1}{L}\alpha_j\tilde{z}_{n,j}^{+(s)}+S^{(s)\backslash j}\right)\mathbf{1}_{\langle \boldsymbol{w}_i,\boldsymbol{z}_{\boldsymbol{Y}}^{(s)}\rangle\geq b_i}\mathbf{1}_{\langle \boldsymbol{w}_i,\boldsymbol{z}_{\boldsymbol{X}}^{(r)}\rangle\leq -b_i}\tilde{z}_{n,j}^{(r)}\right]
\tag{417}
$$

$$
+ \mathbb{E}\left[\sum_{s=1}^{L}\sum_{r=1}^{L}\left(\frac{1}{L}\alpha_j\tilde{z}_{n,j}^{+(s)}+S^{(s)\backslash j}\right)\mathbf{1}_{\langle \boldsymbol{w}_i,\boldsymbol{z}_{\boldsymbol{Y}}^{(s)}\rangle\leq -b_i}\mathbf{1}_{\langle \boldsymbol{w}_i,\boldsymbol{z}_{\boldsymbol{X}}^{(r)}\rangle\geq b_i}\tilde{z}_{n,j}^{(r)}\right]
$$

① is from the symmetry of $\sum_{r=1}^{L}\boldsymbol{z}_{\boldsymbol{X}}^{(r)}$ and $\sum_{s=1}^{L}\boldsymbol{z}_{\boldsymbol{Y}}^{(s)}$ over the randomness of $z_X, z_Y$, $(\sum_{r=1}^{L}\boldsymbol{z}_{\boldsymbol{X}}^{(r)}\overset{d}{=}\sum_{s=1}^{L}\boldsymbol{z}_{\boldsymbol{Y}}^{(s)})$ we observe

$$
\sum_{s=1}^{L}\sum_{r=1}^{L}f(\boldsymbol{z}_{\boldsymbol{X}}^{(r)},\boldsymbol{z}_{\boldsymbol{Y}}^{(s)})\overset{d}{=}\sum_{s=1}^{L}\sum_{r=1}^{L}f(\boldsymbol{z}_{\boldsymbol{Y}}^{(s)},\boldsymbol{z}_{\boldsymbol{X}}^{(r)})
$$

$$
\mathbb{E}[\sum_{s=1}^{L}\sum_{r=1}^{L}f(\boldsymbol{z}_{\boldsymbol{X}}^{(r)},\boldsymbol{z}_{\boldsymbol{Y}}^{(s)})]=\mathbb{E}[\sum_{s=1}^{L}\sum_{r=1}^{L}f(\boldsymbol{z}_{\boldsymbol{Y}}^{(s)},\boldsymbol{z}_{\boldsymbol{X}}^{(r)})]
$$

$$
\mathbb{E}\left[\sum_{s=1}^{L}\sum_{r=1}^{L}b_i\mathbf{1}_{\langle \boldsymbol{w}_i,\boldsymbol{z}_{\boldsymbol{X}}^{(r)}\rangle\geq b_i}\mathbf{1}_{\langle \boldsymbol{w}_i,\boldsymbol{z}_{\boldsymbol{Y}}^{(s)}\rangle\leq -b_i}\tilde{z}_{n,j}^{(r)}\right]=\mathbb{E}\left[\sum_{s=1}^{L}\sum_{r=1}^{L}b_i\mathbf{1}_{\langle \boldsymbol{w}_i,\boldsymbol{z}_{\boldsymbol{X}}^{(r)}\rangle\leq -b_i}\mathbf{1}_{\langle \boldsymbol{w}_i,\boldsymbol{z}_{\boldsymbol{Y}}^{(s)}\rangle\geq b_i}\tilde{z}_{n,j}^{(r)}\right]
\tag{418}
$$

which allows us to drop the $b_i$ terms in $J_3$. The analysis of the rest of $J_3$ is somewhat similar.

First we observe that whenever $\mathbf{1}_{\langle \boldsymbol{w}_i, \boldsymbol{z}_{\boldsymbol{Y}}^{(s)}\rangle \geq b_i}\mathbf{1}_{\langle \boldsymbol{w}_i, \boldsymbol{z}_{\boldsymbol{X}}^{(r)}\rangle \leq -b_i} \neq 0$. we have $\langle \boldsymbol{w}_i, \boldsymbol{z}_{\boldsymbol{Y}}^{(s)}\rangle \geq b_i$ and $\langle \boldsymbol{w}_i, \boldsymbol{z}_{\boldsymbol{X}}^{(r)}\rangle \leq -b_i$, so we can get $\bar{S}^{(r,s)\backslash j} + \bar{\alpha}_j^{(r,s)}\frac{\tilde{z}_{n,j}^{(r)}+\tilde{z}_{n,j}^{+(s)}}{2L} \geq b_i + |S^{(r,s)\backslash j} + \alpha_j\frac{\tilde{z}_{n,j}^{(r)}+z_{n,j}^{+(s)}}{2L}|$

When this inequality holds, we always have $\bar{S}^{(r,s)\backslash j} \geq b_i - \bar{\alpha}_j^{(r,s)}\frac{\tilde{z}_{n,j}^{(r)}+\tilde{z}_{n,j}^{+(s)}}{2L}$.

Together with all the above observations, we proceed to compute as:

$$
\left| \mathbb{E}\left[ \sum_{s=1}^{L}\sum_{r=1}^{L}\left( \frac{1}{L}\alpha_j \tilde{z}_{n,j}^{+(s)} + S^{(s)\backslash j}\right)\mathbf{1}_{\langle \boldsymbol{w}_i, \boldsymbol{z}_{\boldsymbol{Y}}^{(s)}\rangle \geq b_i}\mathbf{1}_{\langle \boldsymbol{w}_i, \boldsymbol{z}_{\boldsymbol{X}}^{(r)}\rangle \leq -b_i}\tilde{z}_{n,j}^{(r)}\right] \right|
$$

$$
= \left| \mathbb{E}\left[ \sum_{s=1}^{L}\sum_{r=1}^{L}\left( \frac{1}{L}\alpha_j \tilde{z}_{n,j}^{+(s)} + S^{(s)\backslash j}\right)\mathbf{1}_{\bar{\alpha}_j^{(r,s)}\frac{\tilde{z}_{n,j}^{(r)}+\tilde{z}_{n,j}^{+(s)}}{2L}+\bar{S}^{(r,s)\backslash j}\geq b_i + |\alpha_j\frac{\tilde{z}_{n,j}^{(r)}+\tilde{z}_{n,j}^{+(s)}}{2L}+S^{(r,s)\backslash j}|}\tilde{z}_{n,j}^{(r)}\right] \right|
$$

$$
\leq \mathbb{E}\left[ \sum_{s=1}^{L}\sum_{r=1}^{L}\left| \bar{S}^{(r,s)\backslash j} + \bar{\alpha}_j^{(r,s)}\frac{\tilde{z}_{n,j}^{(r)}+\tilde{z}_{n,j}^{+(s)}}{2L}\right| |\tilde{z}_{n,j}^{(r)}|\mathbf{1}_{\bar{S}^{(r,s)\backslash j}\geq b_i - \bar{\alpha}_j^{(r,s)}\frac{\tilde{z}_{n,j}^{(r)}+\tilde{z}_{n,j}^{+(s)}}{2L}}\right.
$$

$$
\left. \cdot \mathbf{1}_{\bar{S}^{(r,s)\backslash j}\in\left[b_i - \alpha_j\left|\frac{\tilde{z}_{n,j}^{(r)}+\tilde{z}_{n,j}^{+(s)}}{2L}\right|+\left|\bar{S}^{(r,s)\backslash j}+\bar{\alpha}_j^{(r,s)}\frac{\tilde{z}_{n,j}^{(r)}+\tilde{z}_{n,j}^{+(s)}}{2L}\right|, b_i + \alpha_j\left|\frac{\tilde{z}_{n,j}^{(r)}+\tilde{z}_{n,j}^{+(s)}}{2L}\right|+\left|\bar{S}^{(r,s)\backslash j}-\bar{\alpha}_j^{(r,s)}\frac{\tilde{z}_{n,j}^{(r)}+\tilde{z}_{n,j}^{+(s)}}{2L}\right|\right]}\right]
$$

$$
\leq \sum_{s=1}^{L}\sum_{r=1}^{L}\sqrt{\mathbb{E}\left[|\bar{S}^{(r,s)\backslash j}|^2|\tilde{z}_{n,j}^{(r)}|^2\right]\mathbf{1}_{\bar{S}^{(r,s)\backslash j}\geq b_i - \bar{\alpha}_j^{(r,s)}\frac{\tilde{z}_{n,j}^{(r)}+\tilde{z}_{n,j}^{+(s)}}{2L}}}
$$

$$
\cdot \sqrt{\mathbb{E}\left[\mathbf{1}_{\bar{S}^{(r,s)\backslash j}\in\left[b_i - \alpha_j\left|\frac{\tilde{z}_{n,j}^{(r)}+\tilde{z}_{n,j}^{+(s)}}{2L}\right|+\left|\bar{S}^{(r,s)\backslash j}+\alpha_j\frac{\tilde{z}_{n,j}^{(r)}+\tilde{z}_{n,j}^{+(s)}}{2L}\right|, b_i + \alpha_j\left|\frac{\tilde{z}_{n,j}^{(r)}+\tilde{z}_{n,j}^{+(s)}}{2L}\right|+\left|\bar{S}^{(r,s)\backslash j}-\bar{\alpha}_j^{(r,s)}\frac{\tilde{z}_{n,j}^{(r)}+\tilde{z}_{n,j}^{+(s)}}{2L}\right|\right]}\right]}
$$

$$
+ \frac{1}{L}\mathbb{E}\left[\sum_{s=1}^{L}\sum_{r=1}^{L}\bar{\alpha}_j^{(r,s)}|\frac{\tilde{z}_{n,j}^{(r)}+\tilde{z}_{n,j}^{+(s)}}{2}||\tilde{z}_{n,j}^{(r)}|\mathbf{1}_{\bar{S}^{(r,s)\backslash j}\geq b_i - \bar{\alpha}_j^{(r,s)}\frac{\tilde{z}_{n,j}^{(r)}+\tilde{z}_{n,j}^{+(s)}}{2L}}\right]
$$

$$
\leq \sum_{s=1}^{L}\sum_{r=1}^{L}\sqrt{\mathbb{E}\left[|\bar{S}^{(r,s)\backslash j}|^2|\tilde{z}_{n,j}^{(r)}|^2\right]\mathbf{1}_{\bar{S}^{(r,s)\backslash j}\geq b_i - \bar{\alpha}_j^{(r,s)}\frac{\tilde{z}_{n,j}^{(r)}+\tilde{z}_{n,j}^{+(s)}}{2L}}}
$$

$$
\cdot \sqrt{\mathbb{E}\left[\mathbf{1}_{\langle \boldsymbol{w}_i, \frac{\tilde{\xi}_{n,j}^{(r)}-\tilde{\xi}_{n,j}^{+(s)}}{2L}\rangle\in\left[-\left|\alpha_j\frac{\tilde{z}_{n,j}^{(r)}+\tilde{z}_{n,j}^{+(s)}}{2L}+|\bar{\alpha}_j^{(r,s)}|\frac{\tilde{z}_{n,j}^{(r)}+\tilde{z}_{n,j}^{+(s)}}{2L}\right|, \left|\alpha_j\frac{\tilde{z}_{n,j}^{(r)}+\tilde{z}_{n,j}^{+(s)}}{2L}+|\bar{\alpha}_j^{(r,s)}|\frac{\tilde{z}_{n,j}^{(r)}+\tilde{z}_{n,j}^{+(s)}}{2L}\right|\right]}\right]}
$$

$$
\leq \frac{1}{L}O(\alpha_j + \mathbb{E}[|\bar{\alpha}_j|^2]^{1/2})\mathbb{E}[\sum_{s=1}^{L}\sum_{r=1}^{L}|\frac{\tilde{z}_{n,j}^{(r)}+\tilde{z}_{n,j}^{+(s)}}{2}||\tilde{z}_{n,j}^{(r)}|]\cdot\sqrt{\frac{\mathbb{E}[|\bar{S}^{\backslash j}|^2\mathbf{1}_{\bar{S}^{\backslash j}\geq b_i - \bar{\alpha}_j C_{\tilde{z}}}]}{\mathbb{E}[\langle \boldsymbol{w}_i, \frac{\tilde{\xi}_n^{(r)}+\tilde{\xi}_n^{+(s)}}{2L}\rangle^2]}}
$$

$$
+ \frac{1}{L}\left(\mathbb{E}[|\bar{\alpha}_j|^2]^{1/2}\mathbb{E}[\sum_{s=1}^{L}\sum_{r=1}^{L}|\frac{\tilde{z}_{n,j}^{(r)}+\tilde{z}_{n,j}^{+(s)}}{2}||\tilde{z}_{n,j}^{(r)}|]\right)\Pr(\bar{S}^{\backslash j}\geq b_i - \bar{\alpha}_j C_{\tilde{z}})
$$

$$
\tag{419}
$$

where in the last inequality, we have use the following reasoning: we know that $\langle \boldsymbol{w}_i, \frac{\tilde{\xi}_n^{+(s)}-\tilde{\xi}_n^{(r)}}{2L}\rangle$ has the same distribution with $\langle \boldsymbol{w}_i, \frac{\tilde{\xi}_n^{(r)}+\tilde{\xi}_n^{+(s)}}{2L}\rangle$.($\langle \boldsymbol{w}_i, (\frac{\tilde{\xi}_n^{+(s)}-\tilde{\xi}_n^{(r)}}{2L}\rangle \overset{d}{=} \langle \boldsymbol{w}_i, \frac{\tilde{\xi}_n^{(r)}+\tilde{\xi}_n^{+(s)}}{2L}\rangle)$ We use the randomness to obtain that

$$
\mathbb{E}_{\tilde{\xi}_n^{(r)}, \tilde{\xi}_n^{+(s)}}\left[\mathbf{1}_{\langle \boldsymbol{w}_i, \frac{\tilde{\xi}_n^{(r)}-\tilde{\xi}_n^{+(s)}}{2L}\rangle\in\left[-|\alpha_j\frac{\tilde{z}_{n,j}^{(r)}+\tilde{z}_{n,j}^{+(s)}}{2L}+|\bar{\alpha}_j^{(r,s)}|\frac{\tilde{z}_{n,j}^{(r)}+\tilde{z}_{n,j}^{+(s)}}{2L}|,|\alpha_j\frac{\tilde{z}_{n,j}^{(r)}+\tilde{z}_{n,j}^{+(s)}}{2L}+|\bar{\alpha}_j^{(r,s)}|\frac{\tilde{z}_{n,j}^{(r)}+\tilde{z}_{n,j}^{+(s)}}{2L}|\right]}\right]
$$

$$
\leq \frac{|\alpha_j + \bar{\alpha}_j^{(r,s)}|^2\cdot|\frac{\tilde{z}_{n,j}^{(r)}+\tilde{z}_{n,j}^{+(s)}}{2L}|^2}{\mathbb{E}_\xi[\langle \boldsymbol{w}_i, \frac{\tilde{\xi}_n^{+(s)}-\tilde{\xi}_n^{(r)}}{2L}\rangle^2]} = \frac{|\alpha_j + \bar{\alpha}_j^{(r,s)}|^2\cdot|\frac{\tilde{z}_{n,j}^{(r)}+\tilde{z}_{n,j}^{+(s)}}{2L}|^2}{\mathbb{E}_\xi[\langle \boldsymbol{w}_i, \frac{\tilde{\xi}_n^{(r)}+\tilde{\xi}_n^{+(s)}}{2L}\rangle^2]}
$$

$$
\tag{420}
$$

The second term of $J_3$ can be similarly bounded by the same quantity. Now by combining all the results of $J_1, J_2, J_3$ above, we have the desired result for (a).

$$
\mathbb{E}\left[ h_i(\boldsymbol{Y}_n) \sum_{r=1}^{L} \mathbf{1}_{|\langle \boldsymbol{w}_i, \boldsymbol{z}_X^{(r)} \rangle| \geq b_i} \tilde{z}_{n,j}^{(r)} \right]
$$

$$
\leq \frac{1}{L} \alpha_{i,j}^{(t)} \cdot \mathbb{E}\left[ \sum_{s=1}^{L} \tilde{z}_{n,j}^{+(s)} \sum_{r=1}^{L} \tilde{z}_{n,j}^{(r)} \mathbf{1}_{|\langle \boldsymbol{w}_i^{(t)}, \frac{z_X + z_Y}{2} \rangle| \geq b_i + |\langle \boldsymbol{w}_i^{(t)}, z_X - \frac{z_X + z_Y}{2} \rangle|} \right]
$$

$$
\pm \left( \alpha_{i,j}^{(t)} + O\left( \sqrt{\mathbb{E}|\bar{\alpha}_{i,j}^{(t)}|^2} \right) \right) \cdot \mathbb{E}\left[ \sum_{s=1}^{L} \sum_{r=1}^{L} |\frac{\tilde{z}_{n,j}^{(r)} + \tilde{z}_{n,j}^{+(s)}}{2}| |\tilde{z}_{n,j}^{(r)}| \right] \cdot O(L_1 + L_2)
$$

(421)

$\square$

## J.11 PROOF OF LEMMA E.10(B):

*Proof of Lemma E.10(b):* This proof is extremely similar to the above proof of Lemma E.10(a), we describe the differences here and sketch the remaining. First we need to decompose the expectation as follows:

$$
\mathbb{E}\left[ h_i(\boldsymbol{Y}_n) \sum_{r=1}^{L} \mathbf{1}_{|\langle \boldsymbol{w}_i, \boldsymbol{z}_X^{(r)} \rangle| \geq b_i} \left( \sum_{i=1}^{L} \delta_{n,i}^{(r)} z_{n,j}^{(i)} \right) \right]
$$

$$
= \mathbb{E}\left[ \frac{1}{L}(\alpha_j - b_i) \sum_{s=1}^{L} \tilde{z}_{n,j}^{+(s)} \sum_{r=1}^{L} \tilde{z}_{n,j}^{(r)} \mathbf{1}_{|\langle \boldsymbol{w}_i, \frac{z_X^{(r)} + z_Y^{(s)}}{2} \rangle| \geq b_i + |\langle \boldsymbol{w}_i, \boldsymbol{z}_X^{(r)} - \frac{z_X^{(r)} + z_Y^{(s)}}{2} \rangle|} \right]
$$

$$
+ \mathbb{E}\left[ \sum_{s=1}^{L} \left( S^{(s)\backslash j} \right) \sum_{r=1}^{L} \tilde{z}_{n,j}^{(r)} \mathbf{1}_{|\langle \boldsymbol{w}_i, \frac{z_X^{(r)} + z_Y^{(s)}}{2} \rangle| \geq b_i + |\langle \boldsymbol{w}_i, \boldsymbol{z}_X^{(r)} - \frac{z_X^{(r)} + z_Y^{(s)}}{2} \rangle|} \right]
$$

$$
+ \mathbb{E}\left[ \sum_{s=1}^{L} \left( \frac{1}{L} \alpha_j \tilde{z}_{n,j}^{+(s)} + S^{(s)\backslash j} \right) \sum_{r=1}^{L} \tilde{z}_{n,j}^{(r)} \mathbf{1}_{|\langle \boldsymbol{w}_i, {}_X^{(r)} - \frac{z_X^{(r)} + z_Y^{(s)}}{2} \rangle| \geq b_i + |\langle \boldsymbol{w}_i, \frac{z_X^{(r)} + z_Y^{(s)}}{2} \rangle|} \right]
$$

$$
= J_1 + J_2 + J_3
$$

(422)

where we have used the following facts:

$$
\sum_{s=1}^{L} \sum_{r=1}^{L} \mathbb{E}[b_i \mathbf{1}_{\langle \boldsymbol{w}_i, z_X^{(r)} \rangle \geq b_i} \mathbf{1}_{\langle \boldsymbol{w}_i, z_Y^{(s)} \rangle \geq b_i} \tilde{z}_{n,j}^{(r)}] = - \sum_{s=1}^{L} \sum_{r=1}^{L} \mathbb{E}[b_i \mathbf{1}_{\langle \boldsymbol{w}_i, z_X^{(r)} \rangle \leq -b_i} \mathbf{1}_{\langle \boldsymbol{w}_i, z_Y^{(s)} \rangle \leq -b_i} \tilde{z}_{n,j}^{(r)}];
$$

$$
\sum_{s=1}^{L} \sum_{r=1}^{L} \mathbb{E}[b_i \mathbf{1}_{\langle \boldsymbol{w}_i, z_X^{(r)} \rangle \geq b_i} \mathbf{1}_{\langle \boldsymbol{w}_i, z_Y^{(s)} \rangle \leq -b_i} \tilde{z}_{n,j}^{(r)}] = \sum_{s=1}^{L} \sum_{r=1}^{L} \mathbb{E}[b_i \mathbf{1}_{\langle \boldsymbol{w}_i, z_Y^{(s)} \rangle \geq b_i} \mathbf{1}_{\langle \boldsymbol{w}_i, z_X^{(r)} \rangle \leq -b_i} \tilde{z}_{n,j}^{(r)}];
$$

$$
\mathbf{1}_{\langle \boldsymbol{w}_i, z_X^{(r)} \rangle \geq b_i} \mathbf{1}_{\langle \boldsymbol{w}_i, z_Y^{(s)} \rangle \geq b_i} = \mathbf{1}_{\langle \boldsymbol{w}_i, \frac{z_X^{(r)} + z_Y^{(s)}}{2} \rangle \geq b_i + |\langle \boldsymbol{w}_i, z_X^{(r)} - \frac{z_X^{(r)} + z_Y^{(s)}}{2} \rangle|};
$$

$$
\mathbf{1}_{\langle \boldsymbol{w}_i, z_X^{(r)} \rangle \leq -b_i} \mathbf{1}_{\langle \boldsymbol{w}_i, z_Y^{(s)} \rangle \leq -b_i} = \mathbf{1}_{\langle \boldsymbol{w}_i, \frac{z_X^{(r)} + z_Y^{(s)}}{2} \rangle \leq -b_i - |\langle \boldsymbol{w}_i, z_X^{(r)} - \frac{z_X^{(r)} + z_Y^{(s)}}{2} \rangle|};
$$

$$
\mathbf{1}_{\langle \boldsymbol{w}_i, z_X^{(r)} \rangle \geq b_i} \mathbf{1}_{\langle \boldsymbol{w}_i, z_Y^{(s)} \rangle \leq -b_i} = \mathbf{1}_{\langle \boldsymbol{w}_i, z_X^{(r)} \rangle \leq -b_i} \mathbf{1}_{\langle \boldsymbol{w}_i, z_Y^{(s)} \rangle \geq b_i} = \mathbf{1}_{\langle \boldsymbol{w}_i, z_X^{(r)} - \frac{z_X^{(r)} + z_Y^{(s)}}{2} \rangle| \geq b_i + |\langle \boldsymbol{w}_i, \frac{z_X^{(r)} + z_Y^{(s)}}{2} \rangle|}.
$$

(423)

Now observe that $J_2$ can be dealt with as follows: define events $A_3 := \{|\bar{S}^{(r,s)\backslash j} + \bar{\alpha}_j^{(r,s)} C_{\bar{z}}| \geq (\alpha_j C_{\bar{z}} - b_i)/2\}$ and $A_4 := \{\bar{S}^{(r,s)\backslash j} \geq (\alpha_j C_{\bar{z}} - b_i)/2\}$, and notice that $\mathbf{1} \leq \mathbf{1}_{A_3} + \mathbf{1}_{A_4}$, we can compute

$$J_2 = \mathbb{E}\left[\sum_{s=1}^{L}\left(S^{(s)\setminus j}\right)\sum_{r=1}^{L}\mathbf{1}_{|\langle \boldsymbol{w}_i, \frac{z_X^{(r)}+z_Y^{(s)}}{2}\rangle|\geq b_i + |\langle \boldsymbol{w}_i, \boldsymbol{z}_X^{(r)} - \frac{z_X^{(r)}+z_Y^{(s)}}{2}\rangle|}\tilde{z}_{n,j}^{(r)}\right]$$

$$= \mathbb{E}\left[\sum_{s=1}^{L}\left(S^{(s)\setminus j}\right)\sum_{r=1}^{L}|\tilde{z}_{n,j}^{(r)}|\left(\mathbf{1}_{A_3} + \mathbf{1}_{A_4}\mathbf{1}_{A_3^c}\right)\right.$$

$$\left.\cdot\mathbf{1}_{S^{(r,s)\setminus j}\in\left[b_i-\alpha_j\left|\frac{\tilde{z}_{n,j}^{(r)}+\tilde{z}_{n,j}^{+(s)}}{2L}\right|+\left|\bar{S}^{(r,s)\setminus j}+\bar{\alpha}_j^{(r,s)}\frac{\tilde{z}_{n,j}^{(r)}+\tilde{z}_{n,j}^{+(s)}}{2L}\right|,\ b_i+\alpha_j\left|\frac{\tilde{z}_{n,j}^{(r)}+\tilde{z}_{n,j}^{+(s)}}{2L}\right|+\left|\bar{S}^{(r,s)\setminus j}-\bar{\alpha}_j^{(r,s)}\frac{\tilde{z}_{n,j}^{(r)}+\tilde{z}_{n,j}^{+(s)}}{2L}\right|\right]}\right]$$

$$\overset{(1)}{\leq}\mathbb{E}\left[\sum_{s=1}^{L}\sum_{r=1}^{L}\left(O(\alpha_j+|\bar{\alpha}_j^{(r,s)}|)|\frac{\tilde{z}_{n,j}^{(r)}+\tilde{z}_{n,j}^{+(s)}}{2L}||\tilde{z}_{n,j}^{(r)}|\mathbf{1}_{A_3} + |\left(S^{(s)\setminus j}\right)||\tilde{z}_{n,j}^{(r)}|(\mathbf{1}_{A_3}+\mathbf{1}_{A_4})\right)\right.$$

$$\left.\cdot\mathbf{1}_{S^{(r,s)\setminus j}\in[\alpha_j|\frac{\tilde{z}_{n,j}^{(r)}+\tilde{z}_{n,j}^{+(s)}}{2L}|-b_i+|\bar{S}^{(r,s)\setminus j}+\bar{\alpha}_j^{(r,s)}|\frac{\tilde{z}_{n,j}^{(r)}+\tilde{z}_{n,j}^{+(s)}}{2L}||,\ b_i+\alpha_j|\frac{\tilde{z}_{n,j}^{(r)}+\tilde{z}_{n,j}^{+(s)}}{2L}|+|\bar{S}^{(r,s)\setminus j}-\bar{\alpha}_j^{(r,s)}|\frac{\tilde{z}_{n,j}^{(r)}+\tilde{z}_{n,j}^{+(s)}}{2L}||]}\right]$$

$$\leq \frac{1}{L}\mathbb{E}\left[\sum_{s=1}^{L}\frac{\tilde{z}_{n,j}^{(r)}+\tilde{z}_{n,j}^{+(s)}}{2}\sum_{r=1}^{L}\tilde{z}_{n,j}^{(r)}\right]\cdot O(\alpha_j+\mathbb{E}[|\bar{\alpha}_j|^2]^{1/2})\cdot\left(\Pr(A_3)+\sqrt{\mathbb{E}[|\bar{S}^{\setminus j}|^2(\mathbf{1}_{A_3}+\mathbf{1}_{A_4})]/\mathbb{E}[\langle \boldsymbol{w}_i,\xi_p\rangle^2]}\right)$$

$$\tag{424}$$

where $\textcircled{1}$ relies on the fact that whenever $\mathbf{1}_{\langle \boldsymbol{w}_i, \frac{\tilde{z}_{n,j}^{(r)}+\tilde{z}_{n,j}^{+(s)}}{2L}\rangle\geq b_i+|\langle \boldsymbol{w}_i, \frac{\tilde{z}_{n,j}^{(r)}}{L}-\frac{\tilde{z}_{n,j}^{(r}+\tilde{z}_{n,j}^{+(s)}}{2L}\rangle|}\neq 0$, we have

$|S^{(s)\setminus j}|\leq b_i+(\alpha_j+|\bar{\alpha}_j^{(r,s)}|)|\frac{\tilde{z}_{n,j}^{(r)}+\tilde{z}_{n,j}^{+(s)}}{2L}|+|\bar{S}^{(r,s)\setminus j}|$, and also that we have assumed $b_i < \alpha_j C_{\tilde{z}}$. The term $J_3$ can be bounded from similar analysis as in the proof of Lemma E.10(a), but changing the factor from $O(L_1+L_2)$ to $O(L_3+L_4)$. Combining these calculations gives the desired results. $\square$

### J.12 PROOF OF LEMMA E.11:

*Proof of Lemma E.11:* Again in this proof we omit the time superscript $^{(t)}$. First we deal with the case where the features under consideration is $\boldsymbol{M}_j, j \in [d]$. $\tilde{\xi}_n^{(r)}$ and $\tilde{\xi}_n^{+(r)}$ are independent. Now we can write as follows:

$$\mathbb{E}\left[h_i(\boldsymbol{Y}_n)\sum_{r=1}^{L}\mathbf{1}_{|\langle \boldsymbol{w}_i, \boldsymbol{z}_X^{(r)}\rangle|\geq b_i}\langle \tilde{\xi}_n^{(r)}, \boldsymbol{M}_j\rangle\right]$$

$$=\mathbb{E}\left[\sum_{s=1}^{L}\left(\langle \boldsymbol{w}_i, \boldsymbol{z}_Y^{(s)}\rangle - b_i\right)\mathbf{1}_{\langle \boldsymbol{w}_i, \boldsymbol{z}_Y^{(s)}\rangle\geq b_i}\sum_{r=1}^{L}\mathbf{1}_{|\langle \boldsymbol{w}_i, \boldsymbol{z}_X^{(r)}\rangle|\geq b_i}\langle \tilde{\xi}_n^{(r)}, \boldsymbol{M}_j\rangle\right]$$

$$+\mathbb{E}\left[\sum_{s=1}^{L}\left(\langle \boldsymbol{w}_i, \boldsymbol{z}_Y^{(s)}\rangle + b_i\right)\mathbf{1}_{\langle \boldsymbol{w}_i, \boldsymbol{z}_Y^{(s)}\rangle\leq -b_i}\sum_{r=1}^{L}\mathbf{1}_{|\langle \boldsymbol{w}_i, \boldsymbol{z}_X^{(r)}\rangle|\geq b_i}\langle \tilde{\xi}_n^{(r)}, \boldsymbol{M}_j\rangle\right]$$

$$=I_1 + I_2$$

$$\tag{425}$$

For the first term on the RHS, we have

$$\mathbb{E}\left[\sum_{s=1}^{L}\left(\langle \boldsymbol{w}_i, \boldsymbol{z}_Y^{(s)}\rangle - b_i\right)\mathbf{1}_{\langle \boldsymbol{w}_i, \boldsymbol{z}_Y^{(s)}\rangle\geq b_i}\sum_{r=1}^{L}\mathbf{1}_{|\langle \boldsymbol{w}_i, \boldsymbol{z}_X^{(r)}\rangle|\geq b_i}\langle \tilde{\xi}_n^{(r)}, \boldsymbol{M}_j\rangle\right]$$

$$\tag{426}$$

where $\{M_j\}_{j\in[d]}$ is a basis for $\mathbb{R}^{d_1}$ satisfying $\|M_j\|_\infty \leq O(1/\sqrt{d_1})$.

$$
\mathbb{E}\left[\sum_{s=1}^L \left(\langle w_i, z_Y^{(s)}\rangle - b_i\right) \mathbf{1}_{\langle w_i, z_Y^{(s)}\rangle \geq b_i} \sum_{r=1}^L \mathbf{1}_{|\langle w_i, z_X^{(r)}\rangle| \geq b_i} \langle \tilde\xi_n^{(r)}, M_j\rangle\right]
$$

$$
\leq \mathbb{E}\left[\sum_{s=1}^L (\langle w_i, z_Y^{(s)}\rangle - b_i)\mathbf{1}_{\langle w_i, z_Y^{(s)}\rangle \geq b_i} \sum_{r=1}^L \Big(\mathbf{1}_{|\langle w_i, z_X^{(r)}\rangle + 2(\alpha-\alpha')|\langle\tilde\xi_n^{(r)},M_j\rangle||\geq b_i}\right.
$$

$$
\left. - \mathbf{1}_{|\langle w_i, z_X^{(r)}\rangle - 2(\alpha-\alpha')|\langle\tilde\xi_n^{(r)},M_j\rangle||\geq b_i}\Big)|\langle\tilde\xi_n^{(r)}, M_j\rangle|\right]
$$

$$
\leq \mathbb{E}_{(I-M_jM_j^\top)\tilde\xi_n^{(r)}}\left[\sum_{s=1}^L \left(\langle w_i, z_Y^{(s)}\rangle - b_i\right)\mathbf{1}_{\langle w_i, z_Y^{(s)}\rangle \geq b_i} \sum_{r=1}^L \left|\langle\tilde\xi_n^{(r)}, M_j\rangle\right|\right]
$$

$$
\times \mathbb{E}_{(I-M_jM_j^\top)\tilde\xi_n^{(r)}}\left[\left|\mathbf{1}_{|\langle w_i, z_X^{(r)}\rangle + 2(\alpha-\alpha')|\langle\tilde\xi_n^{(r)},M_j\rangle||\geq b_i} - \mathbf{1}_{|\langle w_i, z_X^{(r)}\rangle - 2(\alpha-\alpha')|\langle\tilde\xi_n^{(r)},M_j\rangle||\geq b_i}\right|\right]
$$

$$
\leq O(1)\cdot \mathbb{E}\left[(\alpha_j + |\bar\alpha_j|)^2 \sum_{s=1}^L\sum_{r=1}^L \langle\tilde\xi_n^{(r)}, M_j\rangle^3 \cdot \frac{\left[\langle w_i, z_Y^{(s)}\rangle - b_i\right]}{\mathbb{E}[\langle w_i, \tilde\xi_n^{(r)}\rangle^2]}\mathbf{1}_{\langle w_i, z_Y^{(s)}\rangle \geq b_i}\right]
$$

$$\tag{427}$$

where in the last inequality we have used the randomness of $(I - M_jM_j^\top)\tilde\xi_n^{(r)}$, which allow us to obtain the denominator $\Omega(1)\mathbb{E}|\langle w_i, \tilde\xi_n^{(r)}\rangle|^2$.

Noticing that $\sum_{j\in[d]}(|\alpha_j|^2 + |\bar\alpha_j|^2) = \mathcal{O}(\|w_i\|_2^2)$ coupled with Cauchy–Schwarz inequality, we can similarly obtain:

$$
\mathbb{E}\left[\sum_{s=1}^L \left(\langle w_i, z_Y^{(s)}\rangle - b_i\right)\mathbf{1}_{\langle w_i, z_Y^{(s)}\rangle \geq b_i} \sum_{r=1}^L \mathbf{1}_{|\langle w_i, z_X^{(r)}\rangle| \geq b_i}\langle\tilde\xi_n^{(r)}, M_j\rangle\right]
$$

$$
\leq O(1)\cdot \mathbb{E}\left[(\alpha_j + |\bar\alpha_j|)^2 \cdot \sum_{s=1}^L\sum_{r=1}^L \langle\tilde\xi_n^{(r)}, M_j\rangle^3 \cdot \frac{\left[\langle w_i, z_Y^{(s)}\rangle - b_i\right]}{\mathbb{E}[\langle w_i, \tilde\xi_n^{(r)}\rangle^2]}\mathbf{1}_{\langle w_i, z_Y^{(s)}\rangle \geq b_i}\right]
$$

$$
\leq O(1)\cdot \mathbb{E}\left[(\alpha_j + |\bar\alpha_j|)^2 \cdot \sum_{r=1}^L \langle\tilde\xi_n^{(r)}, M_j\rangle^3\right] \cdot \frac{1}{\|w_i\|_2^2/d} \times \|w_i\|_2^2 \cdot \Pr(h_{i,t}(Y_n) \neq 0)
$$

$$
\leq \widetilde{\mathcal{O}}\left(\frac{\|w_i\|_2}{d^2}\right) \cdot \Pr(h_{i,t}(Y_n) \neq 0)
$$

$$\tag{428}$$

where in the second inequality we have used the following arguments: first we can compute

$$
(\langle w_i, z_Y^{(s)}\rangle - b_i) \leq \sum_{j\in\mathcal{N}_i} |\langle w_i, M_j\rangle \tilde z_{n,j}^{+(s)}| + \left|\sum_{j\notin\mathcal{N}_i}\langle w_i, M_j\rangle \tilde z_{n,j}^{+(s)}\right| + |\langle w_i, \tilde\xi_n^{+(s)}\rangle|
$$

$$
\leq \clubsuit + \spadesuit + \heartsuit \tag{429}
$$

And from Lemma D.1, Lemma E.7. and Lemma E.8, we have

$$
|\clubsuit| \geq \Omega(\|w_i^{(t)}\|_2) \quad \text{with prob} \leq \widetilde{\mathcal{O}}\left(\frac{1}{d}\right) \tag{430}
$$

$$
|\spadesuit|, |\heartsuit| \leq \widetilde{\mathcal{O}}\left(\frac{\|w_i^{(t)}\|_2}{\sqrt{d}}\right) \quad \text{w.h.p.} \tag{431}
$$

For the second term on the RHS, we can prove it in a similar way. Summing up over $I_1$, $I_2$, we have the desired bound. For the dense feature $M_j^\perp$, the analysis is similar and we omit for brevity. $\qquad\square$

## K PROOF OF ADDITIONAL LEMMAS

### K.1 PROOF OF LEMMA H.1:

*Proof of Lemma H.1:* For the logit $\ell_{s,t}(\boldsymbol{X}_n, \mathfrak{B})$ of negative sample $\boldsymbol{X}_{n,s}$, we can simply calculate:

$$
\begin{aligned}
&\left| \ell'_{s,t}(\boldsymbol{X}_n, \mathfrak{B}) - \frac{1}{|\mathfrak{B}|} \right| \\
&= \left| \frac{e^{\mathrm{Sim}_{f_t}(\boldsymbol{X}_n, \boldsymbol{X}_{n,s})/\tau}}{\sum_{\boldsymbol{X} \in \mathfrak{B}} e^{\mathrm{Sim}_{f_t}(\boldsymbol{X}_n, \boldsymbol{X})/\tau}} - \frac{1}{|\mathfrak{B}|} \right| \\
&= \left( \sum_{\boldsymbol{X} \in \mathfrak{B}} e^{\langle f_t(\boldsymbol{X}_n), f_t(\boldsymbol{X}) - f_t(\boldsymbol{X}_{n,s}) \rangle/\tau} \right)^{-1} - \frac{1}{|\mathfrak{B}|} \\
&= \left| |\mathfrak{B}| - \sum_{\boldsymbol{X} \in \mathfrak{B}} e^{\langle f_t(\boldsymbol{X}_n), f_t(\boldsymbol{X}) - f_t(\boldsymbol{X}_{n,s}) \rangle/\tau} \right| \cdot \left( |\mathfrak{B}| \cdot \sum_{\boldsymbol{X} \in \mathfrak{B}} e^{\langle f_t(\boldsymbol{X}_n), f_t(\boldsymbol{X}) - f_t(\boldsymbol{X}_{n,s}) \rangle/\tau} \right)^{-1} \\
&\leq \sum_{\boldsymbol{X} \in \mathfrak{B}} \left| 1 - e^{\langle f_t(\boldsymbol{X}_n), f_t(\boldsymbol{X}) - f_t(\boldsymbol{X}_{n,s}) \rangle/\tau} \right| \cdot \left( |\mathfrak{B}| \cdot \sum_{\boldsymbol{X} \in \mathfrak{B}} e^{\langle f_t(\boldsymbol{X}_n), f_t(\boldsymbol{X}) - f_t(\boldsymbol{X}_{n,s}) \rangle/\tau} \right)^{-1} \\
&\overset{①}{\leq} \max_{\boldsymbol{X} \in \mathfrak{B}} |\langle f_t(\boldsymbol{X}_n), f_t(\boldsymbol{X}) - f_t(\boldsymbol{X}_{n,s}) \rangle| \cdot \left( \tau \sum_{\boldsymbol{X} \in \mathfrak{B}} e^{\langle f_t(\boldsymbol{X}_n), f_t(\boldsymbol{X}) - f_t(\boldsymbol{X}_{n,s}) \rangle/\tau} \right)^{-1} \\
&\overset{②}{\leq} \widetilde{\mathcal{O}} \left( \frac{\sum_{i \in [m]} \|\boldsymbol{w}_i^{(t)}\|_2^2}{\tau |\mathfrak{B}|} \cdot \exp \left( \frac{\sum_{i \in [m]} \|\boldsymbol{w}_i^{(t)}\|_2^2}{\tau} \right) \right) \\
&\overset{③}{\leq} \widetilde{\mathcal{O}} \left( \frac{\sum_{i \in [m]} \|\boldsymbol{w}_i^{(t)}\|_2^2}{\tau |\mathfrak{B}|} \right) = \widetilde{\mathcal{O}} \left( \frac{1}{\mathrm{polylog}(d)} \right).
\end{aligned}
\tag{432}
$$

① is because we have $|1 - e^a| \leq |a|$ for $a \leq 0.1$.

② is because we have $|\langle f_t(\boldsymbol{X}_n), f_t(\boldsymbol{X}) - f_t(\boldsymbol{X}_{n,s}) \rangle| \leq \tilde{\mathcal{O}} \left( \sum_{i \in [m]} \|\boldsymbol{w}_i^{(t)}\|_2^2 \right) \cdot L^2$

③ is because we have $e^x \leq O(1)$ for $x \leq \frac{1}{2}$ $\qquad \square$

### K.2 PROOF OF LEMMA J.1:

*Proof of Lemma J.1:* Recalling the definition of $\mathcal{M}_j$. According to Definition B.1 and Lemma B.2(b), we know:

$$
\langle \boldsymbol{w}_i^{(0)}, \boldsymbol{M}_j \rangle^2 \geq c_2 \sigma_0^2 \log d.
\tag{433}
$$

We can write:

$$
\mathbb{P}[i \in \mathcal{M}_j \cap \mathcal{M}_{j'}] = \mathbb{P}[i \in \mathcal{M}_j] \cdot \mathbb{P}[i \in \mathcal{M}_{j'}].
\tag{434}
$$

Now,

$$
\mathbb{P}[i \in \mathcal{M}_j] = \mathbb{P}[\langle \boldsymbol{w}_i^{(0)}, \boldsymbol{M}_j \rangle^2 \geq c_2 \sigma_0^2 \log d].
\tag{435}
$$

Since $\langle \boldsymbol{w}_i^{(0)}, \boldsymbol{M}_j \rangle \sim \mathcal{N}(0, \sigma_0^2)$, it follows that:

$$
\langle \boldsymbol{w}_i^{(0)}, \boldsymbol{M}_j \rangle^2 / \sigma_0^2 \sim \chi^2(1).
\tag{436}
$$

Using the tail bound of $\chi^2(1)$, we have:

$$
\mathbb{P}\left[ \langle \boldsymbol{w}_i^{(0)}, \boldsymbol{M}_j \rangle^2 \geq c_2 \sigma_0^2 \log d \right] = \mathbb{P}\left[ \chi^2(1) \geq c_2 \cdot \log d \right] \leq \exp\left( -\Omega(\log d) \right) = \frac{1}{d^{\Omega(1)}}.
\tag{437}
$$

Therefore:

$$\mathbb{P}[i \in \mathcal{M}_j \cap \mathcal{M}_{j'}] \leq \frac{1}{d^{\Omega(1)}}. \tag{438}$$

There are $m$ neurons in total, and each one falls into the intersection with probability $\frac{1}{d^{\Omega(1)}}$, so the total expectation is:

$$\mathbb{E}[|i \in \mathcal{M}_j \cap \mathcal{M}_{j'}|] \leq m \cdot \frac{1}{d^{\Omega(1)}}. \tag{439}$$

It follows that $\mu = \mathbb{E}[|i \in \mathcal{M}_j \cap \mathcal{M}_{j'}|] \leq O(\log d)$. We can use Chernoff Bound to derive high probability upper bound. First, we treat $X_i = \mathbf{1}_{i \in \mathcal{M}_j \cap \mathcal{M}_{j'}}$ as independent 0-1 Bernoulli variables and define:

$$X = \sum_{i=1}^{m} X_i = |i \in \mathcal{M}_j \cap \mathcal{M}_{j'}|, \quad \text{and} \quad \mathbb{E}[X] = \mu \leq O(\log d). \tag{440}$$

Apply Chernoff bound we have:

$$\Pr[|i \in \mathcal{M}_j \cap \mathcal{M}_{j'}| \leq O(\log d)] \geq 1 - o(1/d^4). \tag{441}$$

$\square$