# OpenReview forum: "Theoretical Analysis of Contrastive Learning under Imbalanced Data: From Training Dynamics to a Pruning Solution"
_ICLR.cc/2026/Conference — ICLR 2026 Poster_

### Official Review · Reviewer_Zaui · 2025-10-19

**Soundness:** 3
**Presentation:** 3
**Contribution:** 3
**Rating:** 6
**Confidence:** 4

**Summary:**

This paper develops a theoretical framework to analyze how contrastive learning behaves under imbalanced data distributions, with a specific focus on Transformer-based encoders. It identifies how feature frequency imbalance (i.e. majority vs. minority features) affects neuron-level learning dynamics and the quality of representation learning.

The paper shows that the training progresses through three distinct stages—initial feature growth, specialization of “lucky” neurons, and final convergence. It also shows minority features are learned more weakly, leading to fewer neurons specializing in them and resulting in degraded representations. It further shows introducing magnitude-based pruning during training mitigates these effects by amplifying updates for neurons aligned with minority features, thus improving representation balance.

Theoretical results are corroborated by empirical experiments on various dataset (CIFAR10-LT, CIFAR100-LT, and ImageNet-LT), demonstrating consistent performance gains and reduced accuracy gaps between majority class and minority classes.
In addition, the paper provides theoretical proofs for convergence, establishes feature alignment properties, and rigorously derives the effects of pruning on the learning dynamics.

**Strengths:**

**Originality** The paper seems quite novel. Given my knowledge I am not aware of many existing works exploring the statistical generalization theory of contrastive learning and data imbalance. While prior works have addressed imbalance heuristically or empirically, this paper seems the first formal theoretical treatment of the phenomenon.

**Quality**
The math of the paper seems rigorous, supported by clearly stated lemmas and theorems. Empirical experimental results seem to align with statistical theory. The synthetic data experiments (Appendix A.2) further validate theoretical predictions in controlled settings.

**Clarity**
The clarity of the paper is good. Although this paper is theoretically heavy and thus not easy to read, its clear presentation of results has made the task easier. The three training stages are clearly delineated, and key insights are summarized upfront (Section 3.1). The figures and tables are easy to interpret, and a table of notation is provided. The proof sketch is helpful too.

**Significance**
This work has strong implications for self-supervised learning on real-world, imbalanced datasets. It offers theoretical understanding in the study of contrastive learning, showing how imbalance alters neuron specialization and model complexity requirements.

**Weaknesses:**

There is no major technical flaw detected. However, I do have the following concern.

Assumptions might be a bit strong.

(1) The entire analysis seems to focus on the sparse coding model with orthogonal features and independent Gaussian noise. I understand the technical challenge in analyzing a more general setting. But the current setting feels a bit simple, because in reality features are usually correlated. This orthogonality assumption prevents the theory from describing how contrastive learning handles overlapping or dependent semantic features.

(2) The paper’s assumption that self-attention remains identity-fixed fundamentally limits the scope of its theory. In other words, $W_K, W_Q, W_V$​ are not learnable.

**Questions:**

(1) In Transformers, neurons often encode superposed features that are only linearly separable after training. The analysis assumes near-orthogonal features and pure specialization at convergence. Can the authors comment on whether their theory predicts or forbids feature superposition (neurons simultaneously encoding multiple correlated features)?

(2) Could the authors maybe clarify how the temperature parameter ττ influences imbalance sensitivity in their theory?
Specifically, does a smaller ττ (sharper similarity weighting) amplify majority-feature dominance by concentrating gradients, while a larger ττ mitigates it by smoothing updates?

(3) If the attention mechanism were trainable, pruning neurons downstream of attention would alter gradient flow through $W_K, W_Q, W_V$​. Could the authors speculate on whether such coupling might reinforce or dampen the minority-feature amplification effect?

(4) Could the authors comment on how neuron specialization (Theorem 3.2) is connected to the generalization performance?
In particular, does specialization provably enhance linear separability or any other preferred properties of the learned representations?

---

> ### Author Response · Authors · 2025-11-21
> **Response to Reviewer Zaui - Part 1**
>
> **Response to Reviewer Zaui - Part 1**
>
> Thank you very much for your thoughtful and encouraging assessment of our work. We truly appreciate your recognition across all aspects of the paper. The scarcity of theoretical studies on contrastive learning under data imbalance motivated us to pursue this direction, and we are glad that the novelty is acknowledged. We are also grateful that you found the mathematical analysis rigorous, this was the most technically demanding part of our work. We appreciate your positive comments on clarity as well because we made considerable effort to present the theory intuitively through remarks, figures, and structured notation, and we will further improve readability based on reviewers’ feedback. Finally, we are pleased that you view the work as significant. We hope that our analysis can deepen the understanding of contrastive learning on imbalanced datasets and inspire future developments in the area.
>
> >*W1: The entire analysis seems to focus on the sparse coding model with orthogonal features and independent Gaussian noise. I understand the technical challenge in analyzing a more general setting. But the current setting feels a bit simple, because in reality, features are usually correlated. This orthogonality assumption prevents the theory from describing how contrastive learning handles overlapping or dependent semantic features.*
>
> **Response 1:** Thanks for raising this point. To clarify, the assumption of a sparse coding model with orthogonal features and independent Gaussian noise is common in the feature-learning framework and is used here mainly for technical convenience in the analysis, and the key theoretical insights remain valid even when these assumptions are only approximately satisfied. While extending the theory to more general settings is certainly an interesting direction, doing so would require a separate investigation beyond the scope of this work. Our focus in this paper is instead on providing the first theoretical analysis of contrastive learning in Transformer architectures under data imbalance, which already introduces several non-trivial challenges.
>
> **First, the assumption of a sparse coding model with orthogonal features and independent Gaussian noise is common in theoretical work and can reflect the real data distributions.** Our data assumption is adopted from the widely used sparse coding model, which constitutes a common foundation for theoretical analyses of deep learning (Allen-Zhu \& Li, 2022; Wen \& Li, 2021). Moreover, sparse coding provides a conceptual framework for modeling real-world data across diverse domains, including CV (Protter \& Elad, 2008; Yang et al., 2009; Mairal et al., 2014), NLP (Arora et al., 2018), compressed sensing (Candes \& Recht, 2012; Candes \& Tao, 2010), and neuroscience (Vinje \& Gallant, 2000; Olshausen \& Field, 1997; 2004; Foldiak, 2003). The references can be found in the data model subsection under the assumptions of our paper.
>
> **Second, relaxing these assumptions would involve several tedious and intractable derivations, and addressing them properly would require a separate study beyond the scope of this work.** In particular, moving to correlated or overlapping features would require modeling how neurons learn mixtures of features and how feature co-occurrence affects contrastive gradients. For example, this would affect the proof of Lemma F.3 in Appendix F, because when features are no longer orthogonal or the noise is no longer independent, a large number of additional terms would appear, altering the gradient expression and requiring the entire proof to be rewritten from the beginning.
>
> **Third, our setting already presents several non-trivial challenges compared with prior theoretical work,** in particular, analyzing contrastive learning in Transformer architectures under data imbalance and understanding how pruning can lead to performance improvements. (1) Introducing data imbalance not only makes the gradient analysis substantially more complex, but also induces intricate interactions between neurons and the features they learn. For example, in Appendix C (the proof of Theorem C.1) and Appendix G (the proof of Lemma B.2). (2) As shown in Appendix F in Equations (186) and (211), establishing pruning-related results requires carefully tracking how magnitude-based pruning alters neuron–feature alignment, which already demands more technical effort than prior work.

---

> ### Author Response · Authors · 2025-11-21
> **Response to Reviewer Zaui - Part 2**
>
> **Response to Reviewer Zaui - Part 2**
>
> >*W2\&Q3: The paper’s assumption that self-attention remains identity-fixed fundamentally limits the scope of its theory. In other words, $W_K$, $W_Q$, $W_V$ are not learnable. If the attention mechanism were trainable, pruning neurons downstream of attention would alter gradient flow through $W_K$, $W_Q$, $W_V$. Could the authors speculate on whether such coupling might reinforce or dampen the minority-feature amplification effect?*
>
> **Response 2:** Thank you for this insightful question. We will provide a detailed response to both W2 and Q3 together. Specifically, we will discuss: (1) why the self-attention mechanism is assumed to remain identity-fixed, and (2) what happens if $W_K$, $W_Q$, and $W_V$ become learnable, including how they evolve and how this affects our results.
>
> Before answering your question, we clarify the motivation for assuming the self-attention layer remains fixed. We begin by simplifying the attention notation. The output of a neuron receiving input from an attention head is typically given by: $\text{Relu}(w_i^\top W_V X( \text{softmax}(W_Q X)^\top W_K X ) )$. By defining a combined attention weight $W_{KQ} = W_Q^\top W_K$ and absorbing the value projection into the MLP weights such that $w_i = W_V^\top w_i$, we can rewrite this as $\text{Relu}(w_{i}^\top X( \text{softmax}(X^\top W_{KQ} X ) ).$ Thus, even if $W_V$ is not an identity matrix, it can be linearly absorbed into the MLP weight vectors. Consequently, this does not qualitatively affect our results, and the analysis remains invariant. This simplification is a common practice in theoretical analyses of attention (e.g., see Ref.[zhang2024trained]). Regarding $W_K$ and $W_Q$, prior research on ViT [li2023theoretical] operates under a related assumption: that $W_Q$ and $W_K$ are initialized near an optimal feature-space mapping. It demonstrates that these weights converge to feature subspaces (i.e., $MM^\top$) during training, effectively denoising the signal while preserving important input features.
>
> Based on this discussion above, we intuitively believe that the coupling between a learnable attention module and the pruning applied to downstream MLP neurons would not affect (or may even reinforce) the amplification effect observed in our analysis. As the attention matrices progressively align with the feature space during training, neurons focus more strongly on meaningful feature directions, effectively denoising the representations and triggering an earlier onset of stage 2. Under this intuition, the following analysis remains largely unchanged. Although trainable attention may introduce secondary effects, we do not expect them to qualitatively alter the amplification mechanism or our conclusions.  However, allowing $W_K$ and $W_Q$ to be fully trainable within the proof significantly increases the technical difficulty under the current theoretical framework, and doing so would provide limited additional value to the main contribution of our paper.
>
> [zhang2024trained] Zhang, Ruiqi, Spencer Frei, and Peter L. Bartlett. "Trained transformers learn linear models in-context." Journal of Machine Learning Research 25.49 (2024): 1-55.
>
> [li2023theoretical] Li, Hongkang, et al. "A Theoretical Understanding of Shallow Vision Transformers: Learning, Generalization, and Sample complexity." International Conference on Learning Representations (ICLR 2023). 2023.
>
> >*Q1: In Transformers, neurons often encode superposed features that are only linearly separable after training. The analysis assumes near-orthogonal features and pure specialization at convergence. Can the authors comment on whether their theory predicts or forbids feature superposition (neurons simultaneously encoding multiple correlated features)?*
>
> **Response 3:** Thank you for the question. We believe our theory is aligned with this phenomenon.
>
> In our theoretical framework, not all neurons become lucky neurons that specialize in a single feature. Many ordinary neurons still learn mixed or combined features. Because of this, the encoder in Transformers does not produce perfectly clean or orthogonal representations along each dimension. So we still need a downstream linear classifier to disentangle the embedding dimensions associated with the lucky neurons. Or, when the task depends on combinations of features, the dimensions encoding mixed components can also support the separation. That said, if the model already contains lucky neurons corresponding to each relevant feature, then these lucky neurons alone would be sufficient to ensure linear separability.

---

> ### Author Response · Authors · 2025-11-21
> **Response to Reviewer Zaui - Part 3**
>
> **Response to Reviewer Zaui - Part 3**
>
> >*Q2: Could the authors maybe clarify how the temperature parameter $\tau$ influences imbalance sensitivity in their theory? Specifically, does a smaller $\tau$ (sharper similarity weighting) amplify majority-feature dominance by concentrating gradients, while a larger $\tau$ mitigates it by smoothing updates?*
>
> **Response 4:** Thank you for this thoughtful question. Our work focuses on discussing imbalanced features and pruning ratios, and it does not talk about the temperature parameter $\tau$. We now provide a discussion. First, we appreciate your intuition regarding the temperature $\tau$, but our proof does not contain related information. Next, we will explain the influence of $\tau$ on the results within our proof.
>
> In our theory (see Theorem 3.2), the temperature indeed affects the alignment of neurons with minority features: a larger $\tau$ increases the degree of alignment for neurons learning minority features, while a smaller $\tau$ decreases it. However, according to Equation (89) in Section E.1 of Appendix E, as $\tau$ increases, the time needed to reach $T_2$ also increases. This reflects a dynamic balance between the magnitude of alignment and the time required for learning.
>
> >*Q4: Could the authors comment on how neuron specialization (Theorem 3.2) is connected to the generalization performance? In particular, does specialization provably enhance linear separability or any other preferred properties of the learned representations?*
>
> **Response 5:** Thank you for the thoughtful question. We believe that neuron specialization can enhance linear separability and thereby improve downstream performance.
>
> Our theory shows that each underlying semantic feature is captured cleanly by a subset of lucky neurons. When upstream contrastive learning produces a representation in which all semantic features are encoded in pure and separable directions, the resulting feature space becomes highly structured: it contains explicit axes corresponding to every true feature. If a downstream task relies on any subset of these features, a linear probe (or any simple classifier) can easily extract them because the corresponding feature directions are directly represented by the lucky neurons. In this sense, stronger neuron specialization leads to better linear separability and, consequently, improved downstream generalization.
>
> We have added our response to this point in Remark 3 of Theorem 3.1.

---

> ### Comment · Reviewer_Zaui · 2025-11-26
> **Re: rebuttal**
>
> I would like to thank the authors for the detailed rebuttal. The clarifications addressed my major concerns.
>
> For W1, the explanation of why the sparse-coding and orthogonality assumptions are used—and why extending to correlated features would require a fundamentally different and far more complex analysis—was clear and convincing.
>
> For W2/Q3, the justification for treating attention as fixed and your discussion of how learnable attention would likely reinforce rather than weaken the amplification effect resolved my question.
>
> The responses to Q1 (on feature superposition), Q2 (on the role of temperature), and Q4 (on how specialization supports linear separability and generalization) were each helpful and consistent with the framing of the paper.
>
> Overall, I am satisfied with the rebuttal, and I will maintain my positive assessment and recommend the paper for acceptance.

---

> > ### Author Response · Authors · 2025-11-26
> > **Appreciation for Your Positive Assessment and Recommendation for Acceptance**
> >
> > Thank you for your detailed response. We are glad to see that your major concerns have been addressed. We sincerely appreciate your positive assessment and your recommendation for accepting our paper.

---

### Official Review · Reviewer_yCvM · 2025-10-26

**Soundness:** 3
**Presentation:** 2
**Contribution:** 4
**Rating:** 6
**Confidence:** 3

**Summary:**

This paper provides a robust theoretical analysis of the training dynamics of Transformer-MLP models in learning feature representations through contrastive learning in imbalanced data scenarios. Specifically, the analysis of neuron weight evolution reveals how a minority of features undermines overall model performance. Building upon this, they revisit amplitude-based pruning methods, theoretically demonstrating that pruning yields more robust and balanced representations.

**Strengths:**

This paper explores how imbalanced data degrades representation quality in contrastive learning from a novel perspective of neural weight evolution. Through extensive theoretical analysis, the authors demonstrate that a minority of features weakens representational power while increasing the demand for complex architectures. Building upon this, they further prove that pruning techniques enhance gradient updates along these dominant features, thereby mitigating performance degradation caused by imbalance. This forward-looking approach holds significant promise for advancing the field.

**Weaknesses:**

1：This paper uses numerous notations, some of which lack clear definitions upon their first appearance. Additionally, maintaining consistent notation throughout the text would improve readability.

2：Could the authors please clarify the meaning of "feature frequency"? Specifically, how are the majority and minority features identified within the unsupervised learning framework?

3: The study is confined to the Transformer-MLP model. Could the authors discuss the generalizability of their approach to other architectures or tasks, such as long-tailed visual recognition?

4: While the authors provide substantial theoretical proofs, the effectiveness of the pruning strategy for imbalanced scenarios remains unverified by strong experimental evidence. Additionally, could its potential as a plug-and-play module for existing methods be discussed?

5:The authors are advised to provide precise citations to the appendix for their key conclusions, and to thoroughly review the manuscript to correct various notational errors and inconsistencies.

**Questions:**

As described in “Weaknesses”.

---

> ### Author Response · Authors · 2025-11-21
> **Response to Reviewer yCvM - Part 1**
>
> **Response to Reviewer yCvM - Part 1**
>
> Thank you for your positive assessment of our work. We appreciate your recognition of the theoretical contributions and the insights on feature imbalance. Below, we address the concerns and questions you raised.
>
> >*W\&Q1: This paper uses numerous notations, some of which lack clear definitions upon their first appearance. Additionally, maintaining consistent notation throughout the text would improve readability.*
>
> **Response 1**: Thank you for pointing out the readability issue. Following your suggestion, we have made several revisions to improve clarity and consistency. Please refer to the General Response for details.
>
> >*W\&Q2: Could the authors please clarify the meaning of "feature frequency"? Specifically, how are the majority and minority features identified within the unsupervised learning framework?*
>
> **Response 2:** Thank you for the question. In our paper, feature frequency refers to how often a feature appears across the data distribution. A majority feature is one that appears in many samples, while a minority feature appears only rarely. Importantly, this notion is defined at the feature level rather than the class level, and therefore applies naturally in unsupervised settings.  For example, in an imbalanced dataset that contains imbalanced features, animals are the majority class, and features such as eyes, mouths, or fur textures are common majority features. In contrast, if vehicles are the minority class, then features such as headlights, logos, or wings (for airplanes) would be considered minority features.
>
> We also emphasize that the definition of feature frequency is introduced primarily to derive the theoretical bound and to characterize the different behaviors of neurons when learning features of varying frequencies. This information is not required in practice, since the algorithm does not rely on access to the true feature frequencies.

---

> ### Author Response · Authors · 2025-11-21
> **Response to Reviewer yCvM - Part 2**
>
> **Response to Reviewer yCvM - Part 2**
>
> >*W\&Q3: The study is confined to the Transformer-MLP model. Could the authors discuss the generalizability of their approach to other architectures or tasks, such as long-tailed visual recognition?*
>
> **Response 3**: Thank you for the insightful question. Although our theoretical analysis is developed within the Transformer–MLP framework, we believe that the main insights are likely to extend to other architectures, though doing so requires additional derivations and potentially different technical tools. From our point of view, within the context of training-dynamics analysis, the focus on the Transformer–MLP framework should not be regarded as a strong weakness. Deriving a fully generic bound across arbitrary architectures and training dynamics is inherently challenging, and existing theoretical works within this feature learning framework concentrate on one specific, simplified architectural setting.
>
> Specifically, our intuition for the generalizability of the theoretical insights is rooted in the fact that the neural network backbone ultimately serves as a feature extractor, regardless of the specific architecture. While different architectures may vary in how efficiently or quickly they learn features, neurons eventually align with the underlying features. Consequently, neurons display distinct behaviors when learning majority versus minority features, with those associated with majority features typically exhibiting larger magnitudes. Leveraging this property, our theoretical insights and the pruning approach are expected to transfer across architectures.
>
> However, that said, deriving a generic bound across all architectures is not only technically challenging but also beyond the scope of most existing theoretical works, since architectural differences induce distinct optimization paths, feature interaction patterns, and data representations that would require separate derivations and analytical tools.
>
> The reason is that neuron dynamics are highly dependent on the model structure and the specific loss function. Extending the analysis to each case would require substantial technical changes, in many instances, the proof would need to be entirely reconstructed rather than adapted by simple substitutions. For example, if the network structure or loss function changes, we would even need to modify the gradients and lemmas in Appendix B before the first stage, which would lead to an entirely reconstructed proof. Existing feature learning frameworks even in the supervised learning setting also focus on only one model architecture [Feature Purification] [VIT]. Therefore, deriving a fully general theory that holds across all models and all tasks is not feasible within a single paper on feature learning theory.
>
> [Feature Purification] Allen-Zhu, Zeyuan, and Yuanzhi Li. "Feature purification: How adversarial training performs robust deep learning." 2021 IEEE 62nd annual symposium on foundations of computer science (FOCS). IEEE, 2022.
>
> [VIT] Li, Hongkang, et al. "A theoretical understanding of shallow vision transformers: Learning, generalization, and sample complexity." arXiv preprint arXiv:2302.06015 (2023).

---

> ### Author Response · Authors · 2025-11-21
> **Response to Reviewer yCvM - Part 3**
>
> **Response to Reviewer yCvM - Part 3**
>
> >*W\&Q4: While the authors provide substantial theoretical proofs, the effectiveness of the pruning strategy for imbalanced scenarios remains unverified by strong experimental evidence. Additionally, could its potential as a plug-and-play module for existing methods be discussed?*
>
> **Response 4:** Thank you very much for these inspiring questions. If plug-and-play modules refer to the generalizability of the methodology across different models or tasks, then we suspect that the approach aligns with this interpretation: the algorithm can be applied to different models or tasks without modifying the overall training pipeline, under mild conditions. The main requirement is that the loss function should, in some way, measure the similarity between two outputs, such as cross-entropy loss in supervised learning or contrastive loss in self-supervised learning.
>
> Although a more detailed analysis would be required before drawing a definitive conclusion, we would like to share our intuition. As outlined in the proof sketch, the effectiveness of the approach may come from the different sensitivities of neurons to imbalanced features. Neurons that learn minority features typically have smaller magnitudes compared with those that learn majority features. Magnitude-based pruning then helps identify and retain the neurons that are more sensitive to minority features, which increases the contribution of samples containing minority features during optimization. Prior works have observed alignment between features and neuron activations, which provides empirical support for the idea that pruning can selectively highlight feature-specific neurons. Consistent with this, our real-world experiments using both ResNet and ViT backbones demonstrate effectiveness across different architectures.
>
> If the reviewers have a different interpretation of plug-and-play modules, please let us know, and we are happy to provide further clarification and discussion.
>
> >*W\&Q5: The authors are advised to provide precise citations to the appendix for their key conclusions, and to thoroughly review the manuscript to correct various notational errors and inconsistencies.*
>
> **Response 5:** Thank you for your helpful suggestion. We have added precise Appendix citations after all key lemmas and theorems to make it easier for readers to locate the corresponding proofs. We also appreciate your comments regarding notation errors and inconsistencies. Specifically, The proof of Lemma 3.1 is provided in Appendix C, Section C.4 Proof of Lemma 3.1; the proof of Lemma 3.2 is provided in Appendix D, Section D.4 Proof of Lemma 3.2; the proof of Theorem 3.1 is provided in Appendix E, Section E.4 Proof of Theorem 3.1; and the proof of Theorem 3.2 is provided in Appendix F, Section F.4 Proof of Theorem 3.2.

---

> ### Author Response · Authors · 2025-11-26
> **Looking forward to the discussion with you**
>
> Thank you again for the valuable comments. We have provided detailed responses in the rebuttal and believe we have addressed the concerns, particularly the major issues related to (i) the clarification of feature frequency and (ii) the discussion of generalizability. We would greatly appreciate your feedback on whether our clarifications resolve these issues. Please let us know if any concerns remain.
>
> For W\&Q2, we clarified that feature frequency refers to how often a feature appears across the data distribution, and we emphasized that this definition is introduced primarily to derive the theoretical bound and to characterize the different behaviors of neurons when learning features of varying frequencies. For W\&Q3, we added additional discussion indicating that our theoretical insights and the pruning approach are likely to extend to other architectures.
>
> For the remaining issues, such as the notation issues (W\&Q1) and providing precise citations (W\&Q5), we have made revisions to improve readability and added the corresponding citations to the appendix. For the discussion of the plug-and-play module (W\&Q4), we suspect that the algorithm can be applied to different models or tasks, under mild conditions, without modifying the overall training pipeline.

---

### Official Review · Reviewer_xbUv · 2025-11-01

**Soundness:** 3
**Presentation:** 4
**Contribution:** 4
**Rating:** 6
**Confidence:** 4

**Summary:**

The paper analyzes why contrastive learning deteriorates under imbalanced data. Using a sparse latent-feature model and a Transformer-MLP encoder with InfoNCE, it shows that rare (minority) features grow more slowly, fewer neurons specialize in them, and neurons are forced to represent multiple features, effectively increasing the capacity needed to cover all features. To mitigate this, the authors propose a magnitude-based, forward-masked but backward-unmasked pruning scheme: small-magnitude parameters are masked out only in the forward pass, but all parameters are still updated. This selectively amplifies gradients along minority-feature directions and restores more balanced representations. Experiments on CIFAR-LT and ImageNet-LT confirm better linear-probe accuracy and smaller head–tail gaps.

**Strengths:**

- Provides a rare, neuron-level theoretical analysis of contrastive learning under data imbalance, clearly explaining how minority features are under-learned.
- Connects the analysis to a simple, practical fix (magnitude-based forward-masked, backward-unmasked pruning), making the work actionable.
- Writing and structure are generally clear, making a dense theoretical contribution reasonably accessible.

**Weaknesses:**

- Experiments mainly compare “with vs. without pruning” and lack baselines from other long-tailed methods.
- Sensitivity to pruning ratio/schedule is not deeply analyzed.
- Paper could more explicitly discuss limitations and when the proposed analysis may not apply.

**Questions:**

- The paper sometimes reasons at the neuron level but prunes at the parameter level; please clarify the exact relationship between “parameter-level masking” and the claimed neuron-level effects.
- Which parts of the analysis rely essentially on (i) identity/self attention, (ii) sparse independent features, or (iii) single-layer MLP, and which parts you believe could be relaxed or extended? A short discussion of “what breaks / what survives” under more realistic assumptions would help.
- Can you provide empirical evidence of “lucky” vs. mixed-feature neurons (e.g., alignment plots)?
- In the experiments, how does the method behave when the imbalance ratio = 1 (i.e., no explicit long tail)? Since the proposed pruning is fairly general, do you observe gains in “implicitly imbalanced” settings (e.g., feature-frequency skew induced by augmentations or views) even without a constructed LT distribution?

---

> ### Author Response · Authors · 2025-11-21
> **Response to Reviewer xbUv - Part 1**
>
> **Response to Reviewer xbUv - Part 1**
>
> We sincerely thank the reviewer xbUv for the positive evaluation and the recognition of our contributions. We are grateful that the reviewer highlights our neuron level theoretical analysis of contrastive learning under data imbalance, as well as the practical value of the proposed magnitude-based forward-masked, backward-unmasked pruning scheme. We also appreciate the reviewer’s comments on the clarity of the writing and structure. These insights are very encouraging, and we address the reviewer’s questions and suggestions in detail below.
>
> >*W1: Experiments mainly compare “with vs. without pruning” and lack baselines from other long-tailed methods.*
>
> **Response 1:** We appreciate the reviewers’ suggestions. We would like to clarify that the main focus of this paper is to provide a theoretical understanding of why training on imbalanced data can be harmful, and to explain why pruning-based strategies can improve performance relative to vanilla contrastive learning. For this reason, we intentionally use vanilla contrastive learning (without pruning) as our baseline.
>
> **Moreover, to the best of our knowledge, there is no widely accepted standard baseline for contrastive learning in this setting.** The closest related works fall under supervised contrastive learning [PaCo, BCL], which assume access to label information and apply re-weighting or re-balancing strategies within a supervised learning framework. Furthermore, the loss formulation and the definition of positive/negative pairs in these approaches differ fundamentally from our setting, which prevents us from making a direct comparison. Consequently, we consider such comparisons to fall outside the intended scope of this work. We have also added these missing citations to the revised manuscript. If the reviewers believe additional relevant references should be included, we would be happy to discuss and incorporate them.
>
> **Nevertheless, to address the reviewer’s concern, we have conducted a conceptually simple experiment that may serve as a competitive baseline.** Specifically, we consider a scenario where label information is hypothetically available and apply a straightforward re-weighting strategy based on class imbalance; that is, the overall loss is computed as a weighted sum of per-sample contrastive losses, where each weight is inversely related to the frequency of the corresponding class. For the Cifar10-LT dataset with $\rho=100$, the accuracy is **79.97\%**, which is worse than our pruning approach **(81.31\%)**. We suspect that the reason is twofold: (i) class imbalance does not necessarily imply feature imbalance, and (ii) the re-weighting strategy is overly simplistic and requires substantial fine-tuning to be effective. Nonetheless, this experiment suggests that the pruning approach remains a robust and competitively effective solution.
>
> [PaCo] Cui, Jiequan, et al. "Parametric contrastive learning." Proceedings of the IEEE/CVF international conference on computer vision. 2021.
>
> [BCL] Zhu, Jianggang, et al. "Balanced contrastive learning for long-tailed visual recognition." Proceedings of the IEEE/CVF conference on computer vision and pattern recognition. 2022.

---

> ### Author Response · Authors · 2025-11-21
> **Response to Reviewer xbUv - Part 2**
>
> **Response to Reviewer xbUv - Part 2**
>
> >*W2: Sensitivity to pruning ratio/schedule is not deeply analyzed.*
>
> **Response 2:** Thank you for the insightful question. While this is indeed an important direction, addressing it is challenging and would require additional assumptions about the relationship between minority and majority features, which is beyond the scope of this paper. Moreover, pursuing it theoretically may not be worthwhile, because numerical results show that performance is not sensitive to the pruning ratio. Across a wide range of ratios, one can consistently observe performance improvements.
>
> **First, prior work, e.g., Figure 3 in [Self-Damaging], shows that the pruning algorithm is effective over a substantially large range (approximately 0 to 0.9).** Specifically, increasing the pruning ratio within this range can improve linear separability over the vanilla approach, albeit not in a monotonically increasing manner. However, pruning too aggressively (beyond 0.9) leads to a sharp performance drop.
>
> **However, providing a fully precise characterization of how the pruning ratio varies over the entire interval $[0,1]$ is highly nontrivial in the current theoretical framework, as it would require modeling pruning effects across multiple neuron types and making more precise assumptions about the data distribution.** Moreover, because this is not the primary focus of our work, and empirical evidence from the reference paper already suggests that moderate pruning improves linear separability, we believe that pursuing a more detailed theoretical analysis in this direction is not particularly meaningful for the scope of this paper.
>
> [Self-Damaging] Ziyu Jiang, Tianlong Chen, Bobak J Mortazavi, and Zhangyang Wang. Self-damaging contrastive learning. In International conference on machine learning, pp. 4927–4939. PMLR, 2021.
>
> >*W3: Paper could more explicitly discuss limitations and when the proposed analysis may not apply.*
>
> **Response 3:** Thank you to the reviewer for the suggestion regarding discussing the limitations. Our work has two main limitations. The first concerns studying the pruning ratio and pruning scheme in magnitude-based pruning. Providing a fully precise characterization of how performance varies across different ratios and schemes is highly nontrivial, and doing so would require making more precise assumptions about the data distribution. This will be part of our future work. Furthermore, existing theoretical results in our feature learning framework focus on a single, simplified architectural setting. Extending the analysis to more complex or realistic models will be another direction for future work, and may require fundamentally different derivations and analytical tools.
>
> We have added a limitations section to the main text of our paper.
>
> >*Q1: The paper sometimes reasons at the neuron level but prunes at the parameter level; please clarify the exact relationship between “parameter-level masking” and the claimed neuron-level effects.*
>
> **Response 4:** Thank you for raising this important question. We would first like to clarify that both our theorem and the proposed algorithm are developed at the neuron level. While our work is formulated in this setting, we believe that the underlying insights are generalizable to parameter-level pruning.
>
> First, in our proposed algorithm, each neuron corresponds to a weight vector $w_i$, and the full parameter matrix $\theta$ is composed of these neuron-level row vectors. **We would like to apologize for the misunderstanding caused by the typo in Algorithm 1.** In the textual description of the algorithm, we stated that the pruning is at the neuron level, but in Algorithm 1, we wrote it as parameter-level pruning. This was a typo that caused the confusion, the correct version should be that we prune $\alpha$ fraction of the neurons with the smallest magnitude.
>
> **Second, we expect that magnitude-based pruning at the parameter level would yield an improvement in generalization analogous to that established in Theorem 3.2.** As shown in Theorem 3.1, neurons that successfully learn minority features have their learning dynamics dominated by $\alpha_{i,j}\,M_{j}$. Since $M_{j}$ is a dense vector (satisfying $||M_j||\_\infty \le \widetilde{O}(\frac{1}{\sqrt{d_1}})$ for all $j \in [d]$), and since $\alpha_{i,j}$ is primarily determined by the minority feature frequency $\epsilon_{\min}$, applying magnitude-based pruning is likely to remove the parameters within these neurons.

---

> ### Author Response · Authors · 2025-11-21
> **Response to Reviewer xbUv - Part 3**
>
> **Response to Reviewer xbUv - Part 3**
>
> >*Q2: Which parts of the analysis rely essentially on (i) identity/self attention, (ii) sparse independent features, or (iii) single-layer MLP, and which parts you believe could be relaxed or extended? A short discussion of “what breaks / what survives” under more realistic assumptions would help.*
>
> **Response 5:** Thank you for the constructive question. We are glad to clarify the extent to which each assumption is essential. We believe that assumptions (i) identity/self-attention and (iii) single-layer MLP can be relaxed or extended to more realistic settings without breaking the key conclusions, while assumption (ii), the sparse independent features assumption, is also derived from real-world data distribution, but it corresponds to the result after applying data augmentation to real data. This also explains why data augmentation helps the model learn better features.
>
> **(i) Identity/self-attention:** Our theory relies on the attention mechanism performing a weighted averaging, and since this core structure remains unchanged regardless of whether identity or multi-head attention is used, the theoretical phenomena we describe continue to hold under more general attention settings.
>
> **(iii) Single-layer MLP:** Our model relies on a one-layer MLP whose bias term acts as a threshold, determining whether a neuron becomes activated. In deeper networks, additional nonlinearities and bias terms would raise the threshold for a neuron to consistently activate, which would reduce the number of neurons that successfully learn a minority feature but increase their activation magnitudes. Thus, the qualitative behavior should still hold, although the scale may change.
>
> **(ii) Sparse independent features:** This assumption is also based on real-world data, but it corresponds to an intermediate state of the data. In practice, features can be correlated in the raw data. Data augmentation methods or decorrelation techniques help to mitigate such effects, and data augmentation can break these correlations so that correlated features become irrelevant ones. Therefore, our assumption corresponds to the setting after data augmentation.
>
> >*Q3: Can you provide empirical evidence of “lucky” vs. mixed-feature neurons (e.g., alignment plots)?*
>
> **Response 6:** Thank you for the suggestion. We have now generated empirical evidence for distinguishing lucky neurons and mixed-feature neurons. Here, we provide an illustrative example [Figure 2](https://imgur.com/a/s9KktOR) . The following heatmap visualizes the alignment values of 24 neurons (indexed 0-23) across 9 features (indexed 0-8), where the first five features are majority features and the last four are minority features.
>
> **Example of a lucky neuron:**
> In our experiment, $d_1 = 500$, so the expected alignment from random initialization is approximately $0.002$. After training, Neuron 0 exhibits a strong alignment with Feature 4 (around $0.3$), while its alignment with the remaining features is negligible. Thus, Neuron 0 can be viewed as a lucky neuron for Feature 4.
>
> **Example of a mixed-feature neuron:**
> Neuron 10, in contrast, does not exhibit a dominant alignment with any single feature. Instead, it shows moderate alignment with multiple features, specifically, its values on Features 0, 2, and 7 are 0.13, 0.18, and 0.15, respectively, while remaining negligible on all other features. This behavior corresponds to a mixed-feature neuron.
>
> **Additional examples:**
> Further instances observed in our experiments include, but are not limited to. Lucky neurons: Neuron 6 and Neuron 17 for Feature 0; Neuron 3 and Neuron 18 for Feature 1; Neuron 4 for Feature 2; Neuron 13 for Feature 3. Mixed-feature neurons: Neuron 1 (Feature 1 and 2), Neuron 2 (Features 1 and 3), Neuron 21 (Features 3 and 8).
>
> We have included these results in Appendix A.2 (Experiment 5) of the revised manuscript. These empirical patterns closely reflect the specialization and superposition behaviors predicted by our theoretical analysis.

---

> ### Author Response · Authors · 2025-11-21
> **Response to Reviewer xbUv - Part 4**
>
> **Response to Reviewer xbUv - Part 4**
>
> >*Q4: In the experiments, how does the method behave when the imbalance ratio = 1 (i.e., no explicit long tail)? Since the proposed pruning is fairly general, do you observe gains in “implicitly imbalanced” settings (e.g., feature-frequency skew induced by augmentations or views) even without a constructed LT distribution?*
>
> **Response 7:** Thank you for this inspiring question, and we believe your hypothesis is correct! According to our theory, pruning can still improve performance even when the dataset is balanced. The key reason is that the improvement does not rely on class imbalance but rather on the imbalance of feature frequencies within the data. Even in a balanced dataset, some features appear much more frequently than others across samples, which aligns with your observation of non-common features within each class.
> To further justify this insight, we conducted an additional experiment on the full CIFAR-10 dataset and observed an accuracy improvement when training a linear classifier, increasing from 90.93% to 91.52% after pruning.
>
> Intuitively, class balance does not guarantee feature balance. For example, in an animal-image dataset with equal numbers of samples per class, common features like eyes or mouths appear in almost all animals, while minority features such as wings or rare textures occur much less frequently. Our pruning method strengthens the neurons responsible for these minority features, allowing them to receive larger gradients and better capture rare but informative patterns. As a result, performance can improve even without an explicitly long-tailed class distribution. We therefore expect gains to arise in implicitly imbalanced settings, such as feature-frequency skew induced by augmentations, although the effect is smaller than in strongly long-tailed scenarios from numerical observations.

---

> ### Author Response · Authors · 2025-11-26
> **Looking forward to the discussion with you**
>
> Thank you again for the valuable comments. We have provided detailed responses in the rebuttal and believe we have addressed the concerns, particularly the major issues related to (i) the experiments regarding baselines from other long-tailed methods, (ii) the analysis of the pruning ratio/schedule, and (iii) the additional experiment on lucky vs. mixed-feature neurons. We would greatly appreciate your feedback on whether our clarifications resolve these issues, and please let us know if any concerns remain.
>
> For W1: We have conducted a conceptual experiment that serves as a competitive baseline, which suggests that the pruning approach remains a robust and competitively effective solution. For W2: we would like to clarify that the performance is not sensitive to the pruning ratio/schedule. For Q3: we conducted new experiments on lucky vs. mixed-feature neurons. These empirical patterns closely reflect the specialization and superposition behaviors predicted by our theoretical analysis.
>
> For the remaining issues, such as discussing limitations (W3), the discussion of more realistic assumptions (Q2), and the discussion of implicitly imbalanced settings (Q4), we have provided clarifications on these points and added a limitations section in the revised version. For the misunderstanding caused by a typo (Q1), we clarified that our work is based on neuron-level pruning.

---

### Official Review · Reviewer_8inz · 2025-11-01

**Soundness:** 3
**Presentation:** 2
**Contribution:** 3
**Rating:** 6
**Confidence:** 3

**Summary:**

The paper deals with contrastive learning, a self-supervised framework in which input data are paired into positive or negative pairs based on their similarity in semantic meaning. In particular, data are generated according to the Sparse Coding Model: they are linear combinations of feature vectors plus noise, feature $j$ has an activation probability controlled by a parameter $\epsilon_j$, and features are in general imbalanced (majority features have $\epsilon_j = \epsilon_{\rm max}$, minority ones $\epsilon_j = \epsilon_{\rm min}$). Positive pairs are formed with inputs sharing the same token-aggregate features. The model is a Transformer-MLP built with a single head attention layer followed by an MLP. Training is performed by minimizing the InfoNCE loss, suitable for contrastive learning. A pruning mask filtering temporarily small magnitude neurons is applied when computing the gradients at each training step, and released before updating the neurons.

The paper provides formal results on the dynamics of training of this model. In particular, Lemmas 3.1, 3.2 and Theorem 3.1 show the existence of 3 temporal regimes during training with no pruning: in the first, neuron weights grow in feature directions but are suppressed in non-feature directions, with the growth rate in a feature direction depending on its frequency; in the second, few (lucky) neurons align significantly with single features, while ordinary neurons align in composite directions; in the last (at convergence), the training error is small and neurons are strongly aligned with a subset of features, weakly aligned with the remaining features, and small in the non-feature directions (still, only a limited number of neurons specialize in learning a single feature). Theorem 3.2 shows that pruning amplifies the learning of minority features.

**Strengths:**

Both the architecture and the training protocol are relevant. Theoretical predictions for the training dynamics of Transformer-based encoders are of utmost interest. Narratives on the assumptions behind the data model and on the formal results are provided. Numerical results on real data support the theoretical claims on the advantage of pruning.

**Weaknesses:**

The formal results are hard to read, as the main text is not really self-contained (see Questions below), despite the commendable effort of Table 1. Numerical illustrations in the vanilla setting, even with synthetic data, could help explaining the practical relevance of the bounds provided (for example, by tracking the inner products of lucky/non-lucky neurons with features during training in the 3 regimes, and comparing with theoretical bounds). Considering that reviewing proofs in Appendix is out of question due to time limitations, numerical checks should help strengthening the claims provided by the main text.

**Questions:**

- Can the authors make the main text more self-contained? For example:
    - I could find the hypothesis on the batch size $K$ (such that empirical gradients approximate population ones) only in Lemma B.1;
    - I am assuming $C_m$, $C_z$ to be positive constants, is that the case?
    - The scaling of $T_1$, $T_2$ is not given in the main text;
    - $\Xi_2$ is not defined in the main text, etc.
- In appendix A.2, the authors provide numerical experiments with synthetic data. Can they report in the main text numerical evidence for the 3 training regimes they are able to identify theoretically, for better illustration and check?
- How are negative pairs chosen for the case of real data of section 4?

Minor:

Sometimes * is used instead of $\star$ to denote lucky neurons.

---

> ### Author Response · Authors · 2025-11-21
> **Response to Reviewer 8inz**
>
> **Response to Reviewer 8inz**
>
> We sincerely thank the reviewer 8inz for the time and effort spent reviewing our paper. We are grateful for the positive assessment that both the architecture and the training protocol are relevant, and for highlighting that theoretical predictions for the training dynamics of Transformer-based encoders are of interest. We address the reviewer’s concerns and questions in detail below.
>
> We now proceed to address the reviewer’s questions in detail.
>
> >*Q1: Can the authors make the main text more self-contained?*
>
> **Response 1:** Thank you for pointing this out. Following your suggestion, we have made several revisions to improve clarity and ensure the main text is more self-contained. Please refer to the General Response for details.
>
> >*Q2: In appendix A.2, the authors provide numerical experiments with synthetic data. Can they report in the main text numerical evidence for the 3 training regimes they are able to identify theoretically, for better illustration and check?*
>
> **Response 2:** We thank the reviewer for the helpful suggestion, and we are pleased to report that it indeed helps readers better understand our insights. **The numerical experiment illustrating the three training regimes are shown in [Figure 1](https://imgur.com/a/9Z7fEA5).** In the figure, the blue curve represents the growth of a neuron’s projection onto its dominant feature, the orange curve represents its projection onto any other non-dominant feature, and the green curve corresponds to its projection onto the noise space direction, which remains larger than the projection magnitude on any other feature. As predicted by Lemma 1, during the first stage the neuron primarily grows along feature directions while suppressing noise. In the second stage, consistent with Lemma 2, the projection onto the dominant feature grows at a faster rate than any other feature, resulting in a clear separation. In the third stage, as training approaches $T_3$, the neuron converges, and its final representation is determined mainly by the dominant feature, with negligible components along other directions and noise space. These numerical results align closely with our theoretical characterization of the three-stage learning dynamics.
>
> We have added this numerical experiment to the Introduction section of the main text.
>
> >*Q3: How are negative pairs chosen for the case of real data of section 4?*
>
> **Response 3:** We apologize for missing the details in this part. We follow the standard setting in [SimCLR].
>
> Specifically, for each original image, we apply two independent random data augmentations to obtain two views of the same image. These two augmented views are treated as a positive pair. In a minibatch containing $N$ original images, the augmentation process generates $2N$ views in total. For any data, its corresponding augmented view from the same image is the positive, and all the remaining $2N - 2$ views in the minibatch are treated as negative samples. No class labels are used when forming positive or negative pairs.
>
> [SimCLR] Chen, Ting, et al. "A simple framework for contrastive learning of visual representations." International conference on machine learning, 2020.
>
> >*Q (Minor): Sometimes * is used instead of $\star$ to denote lucky neurons.*
>
> **Response 4:** We thank the reviewer for pointing this out. We have replaced all occurrences of * with $\star$ to denote lucky neurons in Theorem 3.2.

---

> ### Author Response · Authors · 2025-11-26
> **Looking forward to the discussion with you**
>
> Thank you again for the valuable comments. We have provided detailed responses in the rebuttal and believe we have addressed the concerns, particularly the major issues related to (i) the additional experiments and (ii) the selection of negative pairs in our real-data experiment. We would greatly appreciate your feedback on whether our clarifications resolve these issues, and please let us know if any concerns remain.
>
> For Q2, we have completed the 3-stage training experiment and incorporated it into the Introduction of the revised version. This experiment validates our 3-stage theory and gives readers a clear intuitive understanding. For Q3, our selection of negative pairs in our real-data experiment follows the standard setting in [SimCLR]. Other issues, such as the notation issues (Q1 and Q(minor)), have been addressed by adding definitions or orders at each occurrence and revising the relevant parts.

---

### Author Response · Authors · 2025-11-21
**General Response on Notation Issues for @Reviewer 8inz and @Reviewer yCvM**

**General Response on Notation Issues for @Reviewer 8inz and @Reviewer yCvM**

>*Q1: Can the authors make the main text more self-contained? (8inz)*

>*W\&Q1: This paper introduces many notations, some of which are not clearly defined at first use, and several inconsistencies also affect readability. (yCvM)*

**Response 1:** Thank you for the helpful suggestions regarding self-containment and readability. Yes, the theoretical analysis introduces many notations. Following your advice, we have added clear definitions in the main text when each notation first appears. We have added the order of the batch size $K = \operatorname{poly}(d_1)$ after equation (3). For $C_m$ and $C_z$, your assumption is correct: they are positive constants. We have added the definition of $C_m$ in Theorem 3.1 and the definition of $C_z$ in Lemma 3.1. The expression $T_1 = \Theta\ \left( \frac{d_1 \log d}{\eta \log \log d} \right)$ has been added in Lemma 3.1, and $T_2 = T_1 + \Theta\ \left( \frac{d \tau \log d}{\epsilon_{\max} \eta \log \log d} \right)$ has been added in Lemma 3.2. We have also added the definition of $\Xi_2 = d^{C_m - \left( \tfrac{\epsilon_{\min}}{\epsilon_{\max}} \right)^2}$ in Theorem 3.1. The pruning rate $\alpha$ has also been introduced in Key Insights 3 when it first appears.

---

### Meta-Review · Area_Chair_52s5 · 2026-01-07

**Summary:**

This work received four reviews, all aligned to borderline acceptance.

Despite the average positive evaluation, multiple points have been raised, including difficulty in paper reading due to non self-containedness (8inz,yCvM) lack of figures to check for theoretical bounds (8inz), unclear choice of negative pairs (8inz), baseline lackness (xbUv), limitations (xbUv), unclear pruning at parameter/neuron level (xbUv), differences in imbalanced settings (xbUv,yCvM), non generalizability to other architectures and tasks (yCvM) and limiting orthogonality assumptions (Zaui).

**Reviewer Concerns:**

Through their rebuttal, the authors provided some clarifications to the raised points;

Still the comparison on long-tailed methods was not provided. Besides, the authors acknowledge as a limitation that deriving a generic bound is hard and not feasible within the scope of this work. Finally, heavily imbalances setups are not explored, and that still remains as a weakness.

**Reviewer Scores:**

It is realistic to assume that the evaluation would still remain aligned to be marginally above acceptance threshold, despite the work falling short in empirical evaluation and very limiting theoretical assumptions.

---

### Decision · Program_Chairs · 2026-01-26

Accept (Poster)